# Causal Bayesian Optimization: Foundations, Methods, and Applications

## Abstract

Causal Bayesian Optimization (CBO) integrates causal inference with Bayesian optimization to enable sample-efficient intervention selection in systems governed by causal structure. This survey provides a comprehensive and systematic review of the CBO landscape, organizing the growing literature through a unified BO-loop perspective that reveals how causal assumptions shape four core components: intervention search spaces, surrogate construction, acquisition design, and decision policies. We organize methods along recurring design axes, including graph and system-knowledge assumptions, environmental assumptions, intervention representation, surrogate architecture, and decision rules, and we clarify conceptual and notational connections between CBO and adjacent fields, including causal bandits, Bayesian experimental design, safe optimization, policy search, and causal abstraction. To address the lack of standardized evaluation in the field, we introduce a reproducibility-oriented benchmark that covers hard- and soft-intervention settings, implements both the standard GAP metric and a new trajectory-aware Path-Aware GAP (PA-GAP) metric, and evaluates seven CBO methods alongside a non-causal BO baseline under a common scoring protocol. Within this benchmark-specified regime, where the benchmark SCM, intervention domains, and reference optima are fixed and known-graph methods receive the benchmark graph unless explicitly specified otherwise, our empirical study across thirteen datasets, three budget levels, and two metrics reveals that no single method dominates uniformly: rankings depend critically on the dataset, budget, metric, and method-specific use of causal information, and strong non-causal baselines remain competitive in several settings. We further add controlled graph-misspecification and omitted-variable stress tests, which show that rankings can change substantially when the learner-side causal information is perturbed. We therefore identify robustness to causal-assumption violations, scalable unknown-graph optimization, mixed intervention types, realistic cost models, tighter theoretical guarantees, and integration with modern representation learning and causal abstractions as open challenges for moving CBO from controlled benchmarks toward reliable deployment.

## 1 Introduction

Many scientific and engineering problems require the selection of interventions in systems whose variables are causally coupled. A clinician chooses a drug dosage that propagates through metabolic pathways to affect a clinical endpoint; an engineer tunes process parameters that cascade through a manufacturing pipeline; a policymaker adjusts economic levers whose effects depend on the causal structure of the economy. In each case, the key challenge is the same: the system's response to an intervention is not governed by statistical association but by a causal mechanism, and purely correlational optimization can recommend actions that appear promising in observational data, yet fail or even cause harm when deployed (Pearl, 2009; Peters et al., 2017).

Bayesian Optimization (BO) offers a basic framework for sample-efficient optimization when evaluations are expensive, noisy, or constrained (Jones et al., 1998; Shahriari et al., 2016). By iteratively fitting a probabilistic surrogate and selecting queries via an acquisition function that balances exploitation against exploration, BO can identify near-optimal solutions with far fewer evaluations than grid or random search.

However, standard BO treats the objective as a black box: it is agnostic to the causal relationships among the variables it manipulates. Causal Bayesian Optimization (CBO) closes this gap by embedding structural causal knowledge into the BO loop, enabling the optimizer to reason about *which* variables to intervene on, *how* observational data relate to interventional effects, and *where* the experimental budget is best spent (Aglietti et al., 2020).

Since its introduction, CBO has rapidly expanded into a diverse family of methods. These developments can be organized around recurring design choices: (i) what is assumed about the causal graph, causal sufficiency, hidden confounding, and identifiability; (ii) what environment is being optimized, such as static, dynamic, contextual, constrained, adversarial, or multi-objective settings; (iii) what form interventions take, including hard clamps, soft mechanism changes, scopes, and policies; (iv) whether the surrogate models direct intervention effects or individual mechanisms; and (v) how the acquisition or decision rule allocates budget across outcome improvement, structure learning, feasibility, and robustness. Representative extensions include dynamic environments (Aglietti et al., 2021), safety constraints (Aglietti et al., 2023), mechanism-level surrogates with regret guaranties (Sussex et al., 2023), functional and policy interventions (Gultchin et al., 2023), contextual scope selection (Arsenyan et al., 2026), adversarial non-stationarity (Sussex et al., 2024), high-dimensional scaling (Wu et al., 2024), unknown-graph optimization via information-theoretic exploration (Branchini et al., 2023) or optimistic model selection (Mukherjee et al., 2024), noise-robust acquisition with learned priors (Li et al., 2023), and abstraction-based causal decision making, including causally abstracted bandits and multi-level causal optimization (Zennaro et al., 2024; Dyer et al., 2025; Zeitler, 2025).

Despite this progress, the field lacks two critical ingredients for maturation. First, a **unified conceptual framework** that reveals the shared structure across CBO variants and connects them to adjacent fields, such as causal bandits, Bayesian experimental design, active causal discovery, and safe reinforcement learning, is missing. Individual papers typically position their contributions relative to the original CBO formulation, but rarely articulate how different notions of uncertainty (effect, mechanism, and graph uncertainty) lead to fundamentally different acquisition principles, or how contextual and functional intervention methods are instances of a common policy-search abstraction. Second, **standardized evaluation infrastructure** is absent. Existing papers use different datasets, incompatible codebases, inconsistent intervention conventions, and different metrics, making cross-paper comparisons unreliable. A method that excels when the graph is known and interventions are low-dimensional may behave very differently under soft interventions, unknown graphs, or context-dependent policies, yet these distinctions are obscured when each paper evaluates in isolation.

This survey addresses both gaps. We provide a systematic, technically detailed review of the CBO landscape organized through a unified BO-loop perspective, and we introduce a reproducibility-oriented benchmark that standardizes and repackages inherited CBO tasks together with newly added scenarios under a common execution and scoring protocol. In terms of concreteness, our contributions are as follows.

- **A unified design-space perspective.** We organize CBO methods by graph and system-knowledge assumptions, environmental assumptions, intervention representation, surrogate architecture, and decision rule (Section 3). This taxonomy is intended as a survey-level organization of recurring design patterns and couplings, rather than as an ablation study proving that components are interchangeable.

- **Connections to adjacent fields.** We clarify conceptual and notational links between CBO and causal bandits, Bayesian experimental design, active causal discovery, safe optimization, policy search, and causal abstraction / multi-scale decision making, while distinguishing such connections from formal equivalences or reductions (Section 3.7).

- **A reproducibility-oriented benchmark.** We release a standardized benchmark harness covering eight hard-intervention and five soft-intervention scenarios, combining inherited tasks from prior CBO/function-network work with newly added or reconstructed scenarios. The benchmark provides standardized GAP, a new trajectory-aware PA-GAP metric, and a common best-so-far trajectory interface that enables fair re-scoring of outputs from heterogeneous implementations (Section 4).

- **A unified empirical comparison.** We evaluate seven CBO methods and a non-causal BO baseline across thirteen datasets, three budget levels, and two metrics. The aggregate rank plots are retained as descriptive overview summaries, but we add statistical reliability checks and task-track labels to distinguish matched comparisons, stress-test comparisons, and descriptive single-method or single-dataset settings. The results show that no method is statistically separable as uniformly best, rankings depend strongly on dataset–budget-metric interactions, and strong non-causal baselines remain competitive in several settings (Section 4.4–4.5).

- **Structured open problems.** We identify six concrete research directions (robustness to causal assumptions, scalability, richer intervention models, realistic evaluation, theoretical foundations, and integration with representation learning) that must be addressed for CBO to achieve reliable real-world deployment (Section 5).

**Paper outline.** Section 1 places this survey relative to existing reviews. Section 2 establishes the theoretical foundations in causal inference and Bayesian optimization. Section 3 presents the methodological frameworks organized by design axes. Section 4 describes the benchmark, metrics, datasets, and empirical results. Section 5 discusses open problems, and Section 6 concludes.

### Related Surveys and Scope

Several surveys cover topics adjacent to CBO, but none provides the unified treatment of causal optimization methods and standardized empirical evaluation that we pursue here.

**Bayesian optimization surveys.** Comprehensive reviews of BO cover surrogate modeling, acquisition functions, and scalability (Shahriari et al., 2016; Frazier, 2018; Garnett, 2023), but treat the objective as a black box and do not address causal structure, intervention design, or the observation-intervention trade-off central to CBO.

**Causal inference and causal discovery.** Textbooks and surveys on causal inference (Pearl, 2009; Peters et al., 2017; Hernán & Robins, 2020; Glymour et al., 2019) provide the framework of the structural causal model that underlies CBO, but focus on the identification and estimation of causal effects rather than on sequential optimization of interventions under budget constraints. Causal discovery surveys (Spirtes et al., 2000; Chickering, 2002; Glymour et al., 2019) address the learning of the graph from data, which is a component of unknown-graph CBO but not its main goal.

**Causal bandits.** The causal bandit literature (Lattimore et al., 2016; Lu et al., 2021; Nair et al., 2021) shares CBO's goal of selecting interventions using causal structure, but operates under different modeling assumptions: bandits typically assume discrete action sets with parametric reward models, whereas CBO handles continuous intervention domains with nonparametric (GP-based) surrogates. We discuss the relationship between CBO and causal bandits in detail in Section 3.7.

**Bayesian experimental design.** Bayesian optimal experimental design (Chaloner & Verdinelli, 1995; Foster et al., 2021) selects experiments to maximize information about model parameters, which overlaps with the structure-learning component of unknown-graph CBO. However, experimental design typically does not include an optimization objective beyond parameter estimation.

**Scope of this survey.** We focus on methods that explicitly combine (i) a structural causal model or causal graph with (ii) a sequential Bayesian optimization loop to select interventions. We exclude pure causal effect estimation without an optimization loop, causal bandits without GP-based surrogates (except when contrasting with CBO), and reinforcement learning methods that learn policies through environment interaction without an explicit causal model. We include both known- and unknown-graph settings, hard and soft interventions, and single- and multi-objective formulations.

**Survey protocol and inclusion criteria.** To make the construction of the review corpus explicit, we followed a targeted search-and-screening protocol with a literature cutoff date of June 28, 2026. We searched

Table 1: Summary of notation used throughout this survey.

| Symbol | Meaning |
|---|---|
| $M = \langle U, V, F, P(U) \rangle$ | Structural Causal Model |
| $V = \{V_1, \ldots, V_n\}$ | Endogenous (observed) variables |
| $U$ | Exogenous variables (determined outside the system) |
| $\mathrm{Pa}_i$ | Parents of $V_i$ in the causal graph $\mathcal{G}$ |
| $\mathcal{G}$ | Causal directed acyclic graph (DAG) |
| $do(X{=}\mathbf{x})$ | Hard intervention setting $X$ to value $\mathbf{x}$ |
| $X, Y$ | Manipulable variables, target outcome |
| $C$ | Context variables (used only by contextual extensions, Section 3.3) |
| $X_s \subseteq X$ | Intervention scope (subset of manipulable variables) |
| $\mathbf{x}_s \in \mathcal{D}(X_s)$ | Intervention value in the feasible domain |
| $f(X_s, \mathbf{x}_s)$ | Interventional objective: $\mathbb{E}[Y \mid do(X_s{=}\mathbf{x}_s)]$ ($C$ enters only in contextual extensions) |
| $\mathcal{GP}(m, k)$ | Gaussian process with mean $m$ and kernel $k$ |
| $\mu_t(\cdot),\ \sigma_t^2(\cdot)$ | GP posterior mean and variance after $t$ observations |
| $\alpha(\cdot)$ | Acquisition function |
| $T$ | Total number of interventional trials (budget) |
| $y^*$ | Reference optimum (for metric computation) |
| $R_t$ | Best-so-far improvement ratio at trial $t$ |

OpenReview, PMLR, arXiv, Google Scholar, and the proceedings of major machine learning and causality venues using keyword combinations such as "causal Bayesian optimization," "causal global optimization," "Bayesian optimization with interventions," "causal optimization," and "optimization under causal structure." Candidate papers were first screened by title and abstract, and then by their methodological formulation. A work was included if its central contribution couples an explicit causal model, such as an SCM, causal graph, or distribution over graphs, with a sequential Bayesian-optimization-style procedure for intervention selection. We further used backward and forward citation tracing from the original CBO paper and subsequent representative extensions to identify closely related work. Papers whose primary objective is only causal effect estimation, graph recovery, bandit learning, experimental design, or reinforcement learning were excluded from the main corpus unless they directly clarify the boundary between CBO and adjacent fields. This protocol yields a focused corpus of methods in which causal assumptions actively modify the Bayesian optimization loop, rather than a broad survey of causal decision-making methods.

## 2 Theoretical Foundations

This section establishes the conceptual and mathematical foundations underlying Causal Bayesian Optimization. We first review structural causal models, interventions, and identifiability (Section 2.1). We then summarize classical Bayesian optimization with an emphasis on components that CBO modifies (Section 2.2). Finally, we formulate the CBO problem and highlight the key ways in which causality reshapes the BO loop (Section 2.3). Table 1 consolidates the notation used throughout the paper.

### 2.1 Causal Inference

Most CBO methods adopt the framework of Structural Causal Models (SCMs) (Pearl, 2009), which provide a compact, modular representation of how variables are generated and how they respond to interventions.

**Definition 1** (Structural Causal Model (SCM)). *An SCM is a tuple $M = \langle U, V, F, P(U) \rangle$ where $U$ is a set of exogenous variables, $V = \{V_1, \ldots, V_n\}$ is a set of endogenous variables, and $F = \{f_1, \ldots, f_n\}$ is a collection of structural assignments*

$$V_i := f_i(\mathrm{Pa}_i, U_i), \qquad i = 1, \ldots, n, \tag{1}$$

*where $\mathrm{Pa}_i \subseteq V \setminus \{V_i\}$ denotes the direct causes (parents) of $V_i$ in the causal graph and $U_i \subseteq U$ is the exogenous noise that affects $V_i$. The distribution $P(U)$ completes the model and, together with $F$, induces a unique joint*

*distribution on $V$.* *The exogenous variables $U$ are variables determined outside the modeled causal system; their probability distribution $P(U)$ is explicitly part of the SCM specification, so they are not "unmodeled" but rather externally determined.*

**Causal graphs, assumptions, and the causal Markov property.** An SCM induces a causal graph $\mathcal{G}$, commonly represented as a directed acyclic graph (DAG), with a node for each endogenous variable and an edge $V_j \to V_i$ whenever $V_j \in \mathrm{Pa}_i$. The graph encodes a qualitative causal structure in which variables are direct causes of others but does not, by itself, specify functional forms or effect magnitudes. That quantitative information resides in the structural equations $F$.

Two standard assumptions link the graph to the distribution. The *causal Markov condition* states that each variable is conditionally independent of its non-descendants given its parents (Pearl, 2009; Spirtes et al., 2000). Under the additional assumption that exogenous variables are mutually independent (*causal sufficiency*), the observational distribution factorizes according to the graph.

$$P(V_1, \ldots, V_n) \;=\; \prod_{i=1}^{n} P(V_i \mid \mathrm{Pa}_i). \tag{2}$$

The *faithfulness* assumption further requires that all conditional independencies in $P$ are entailed by the Markov condition applied to $\mathcal{G}$, ruling out cancelations that could hide edges. Together, these assumptions enable structure learning from observational data and underpin the identifiability results that CBO exploits.

**Hard interventions.** In Pearl's framework, a *hard* (or *perfect*) intervention $do(X{=}\mathbf{x})$ replaces the structural equation for $X$ with the constant $\mathbf{x}$, severing all incoming edges to $X$ while leaving other mechanisms unchanged. This produces a modified SCM $M_{do(X{=}\mathbf{x})}$ and an interventional distribution that admits a *truncated factorization*:

$$P(V_1, \ldots, V_n \mid do(X{=}\mathbf{x})) \;=\; \prod_{V_i \notin X} P(V_i \mid \mathrm{Pa}_i) \bigg|_{X=\mathbf{x}}. \tag{3}$$

Hard interventions are the default in the original CBO formulation and in most Hard intervention benchmarks.

**Soft interventions.** A *soft* (or *imperfect*) intervention modifies the structural mechanism of a variable without completely severing its incoming edges. Formally, a soft intervention on $V_i$ replaces $f_i$ with a new mechanism $\tilde{f}_i(\mathrm{Pa}_i, U_i; \theta)$ parameterized by an intervention parameter $\theta$. *In the broader literature, soft interventions can also encompass structural changes such as dropping or adding parent dependencies, or policy-based functional interventions where $\theta$ parameterizes a function mapping contexts to mechanism parameters. For each soft-intervention method surveyed below, we specify which type of soft intervention is considered and what $\theta$ represents in that context (e.g., MCBO uses $\theta$ as a shift or rescaling of the mechanism input, whereas fCBO treats $\theta$ as a policy function in an RKHS).* This is important in several CBO variants, notably MCBO (Sussex et al., 2023) and fCBO (Gultchin et al., 2023), that model mechanism modifications rather than variable clamping. Soft interventions yield a richer action space but require modeling how the intervention parameter interacts with the existing mechanism.

**Counterfactuals.** SCMs also define counterfactual quantities that describe outcomes under hypothetical actions in a specific latent world (indexed by $u \in U$), typically written as $Y_{\mathbf{x}}(u)$ (Pearl, 2009). *Most CBO methods surveyed in this paper optimize interventional expectations rather than counterfactual quantities. We include counterfactuals here to clarify the SCM hierarchy and to mark a possible future direction: counterfactual CBO would be relevant for personalized optimization after observing unit-specific histories, retrospective evaluation of alternative actions for the same latent unit, or counterfactual safety constraints. We do not treat counterfactual BO as an established component of the current CBO methods unless explicitly stated.*

**Identifiability and the *do*-calculus.** In practice, the full SCM is rarely known, so CBO relies on *identifiability*: whether an interventional quantity such as $\mathbb{E}[Y \mid do(X = \mathbf{x})]$ can be expressed purely in terms of

the observational distribution given assumptions about the graph. Pearl's *do*-calculus provides three general inference rules for transforming interventional expressions into observational quantities when identification is possible (Pearl, 2009; 2012). A common special case is the *backdoor adjustment*: if a set $Z$ satisfies the backdoor criterion relative to $(X, Y)$, then

$$\mathbb{E}[Y \mid do(X=\mathbf{x})] = \sum_z \mathbb{E}[Y \mid X=\mathbf{x}, Z=z] \, P(Z=z). \tag{4}$$

When identification holds but closed-form evaluation is difficult, one can approximate the required expectations via Monte Carlo integration of the identified estimand.

**Identifiability as an algorithmic assumption in CBO.** In CBO, identifiability is not merely a causal inference condition; it directly shapes the optimizer's behavior. CBO benefits from identifiability to construct observationally informed priors, which accelerates sample efficiency, but the framework does not strictly rely on it to function. If a causal effect is not identifiable under the given graph, a sound algorithm can recognize this and fall back to a standard uninformative GP prior while still leveraging the causal graph for search-space reduction via POMIS. When an intervention effect is identifiable from observational data, the observational sample can be used to construct informative GP priors, initialize surrogates, or eliminate dominated intervention scopes *before* any costly intervention is performed. Two failure modes should be distinguished. First, if identifiability fails on a known, correctly specified graph (e.g., an unblockable backdoor path), a sound adjustment procedure will recognize this and default to an uninformative prior, avoiding bias. Second, overconfident and biased priors arise when the graph or adjustment assumptions are wrong— for example, when the algorithm mistakenly treats an effect as identifiable due to graph misspecification. The latter is a failure of the causal graph assumption, not a failure of identifiability given the correct graph. This asymmetry means that empirical comparisons should carefully distinguish methods that assume correct identifiable priors from those that rely primarily on interventional data or explicitly model graph and identification uncertainty. Throughout this survey, we follow the common CBO assumption of causal sufficiency and perfect interventions unless otherwise stated.

## 2.2 Classical Bayesian Optimization

**Bayesian Optimization (BO)** is a sequential strategy for optimizing an expensive black-box objective $f : \mathcal{X} \to \mathbb{R}$ on a bounded domain $\mathcal{X} \subset \mathbb{R}^d$ (Kushner, 1964; Zhilinskas, 1975; Mockus et al., 1978; Mockus, 1989; Jones et al., 1998; Shahriari et al., 2016). At each iteration $t$, BO selects a query $\mathbf{x}_t$, observes $y_t = f(\mathbf{x}_t) + \epsilon_t$ with noise $\epsilon_t$, and updates a probabilistic model to guide future queries. BO is characterized by two interacting components.

**Probabilistic surrogate.** The most common surrogate is a Gaussian Process (GP) (Rasmussen & Williams, 2006), specified by a prior mean function $m_0(\cdot)$ and a kernel (covariance function) $k(\cdot, \cdot)$. Given $t$ observations $\mathcal{D}_t = \{(\mathbf{x}_i, y_i)\}_{i=1}^t$, the GP posterior of any candidate input $\mathbf{x}$ is available in closed form:

$$\mu_t(\mathbf{x}) = m_0(\mathbf{x}) + \mathbf{k}_t(\mathbf{x})^\top (\mathbf{K}_t + \sigma_\epsilon^2 \mathbf{I})^{-1} (\mathbf{y}_t - \mathbf{m}_0), \tag{5}$$

$$\sigma_t^2(\mathbf{x}) = k(\mathbf{x}, \mathbf{x}) - \mathbf{k}_t(\mathbf{x})^\top (\mathbf{K}_t + \sigma_\epsilon^2 \mathbf{I})^{-1} \mathbf{k}_t(\mathbf{x}), \tag{6}$$

where $\mathbf{k}_t(\mathbf{x})$ is the vector of covariances between $\mathbf{x}$ and the observed inputs, $\mathbf{K}_t$ is the kernel matrix over the observed inputs, and $\sigma_\epsilon^2$ is the variance of the observation noise. The posterior mean $\mu_t$ provides a point estimate, while $\sigma_t^2$ quantifies the predictive uncertainty, enabling the acquisition function to distinguish well-explored from poorly-understood regions.

**Acquisition functions.** The acquisition function $\alpha(\mathbf{x}; \mathcal{D}_t)$ scores candidate inputs by balancing exploitation (questioning where $\mu_t$ is favorable) against exploration (questioning where predictive uncertainty or information gain is large). Two widely used choices are Expected Improvement (EI) and the Upper Confidence Bound (UCB):

$$\alpha_{\mathrm{EI}}(\mathbf{x}) = \mathbb{E}\big[\max(f(\mathbf{x}) - f^+, 0) \mid \mathcal{D}_t\big], \tag{7}$$

$$\alpha_{\mathrm{UCB}}(\mathbf{x}) = \mu_t(\mathbf{x}) + \beta_t^{1/2} \, \sigma_t(\mathbf{x}), \tag{8}$$

where $f^+$ is the incumbent value. In a noiseless setting, $f^+ = \max_i f(\mathbf{x}_i)$ is the best latent function value observed; in a noisy setting, the latent function value $f$ is unobserved and $f^+$ is typically replaced by the best noisy observation $y^+ = \max_i y_i$ or by a posterior estimate of the latent incumbent. $\beta_t > 0$ is a schedule parameter that controls the exploration–exploitation trade-off (Jones et al., 1998; Srinivas et al., 2010). The BO loop iterates: maximize $\alpha$ to select $\mathbf{x}_{t+1}$, evaluate $f$, and update the GP posterior.

**Regret and sample efficiency.** Under regularity conditions in the kernel and the appropriate choice of $\beta_t$, GP-UCB achieves a sublinear cumulative regret $\sum_{t=1}^{T}[f(\mathbf{x}^*) - f(\mathbf{x}_t)] = \mathcal{O}(\sqrt{T\gamma_T})$, where $\gamma_T$ is the maximum information gain of the kernel (Srinivas et al., 2010; Chowdhury & Gopalan, 2017). This theoretical grounding is relevant to CBO because MCBO (Sussex et al., 2023) and ACBO (Sussex et al., 2024) extend GP-UCB-style regret analysis to causal settings, obtaining bounds that can improve over the non-causal case when the causal graph induces a factored reward structure.

**Beyond standard GPs.** Although GPs remain the dominant surrogate in CBO, the broader BO literature also uses random forests, Bayesian neural networks, neural processes, and deep kernel learning (Shahriari et al., 2016; Garnett, 2023). These alternatives are not causal by themselves. They become CBO components only when the modeled target is a causal estimand, when observational data enter through an identified causal prior, or when the architecture respects graph or mechanism structure. Thus, surrogate choice encodes both a statistical model class and a causal modeling target: effect-level surrogates model $\theta \mapsto \mathbb{E}[Y \mid do(X_s = \theta)]$, whereas mechanism-level surrogates model structural equations and propagate uncertainty through the graph.

### 2.3 Causal Bayesian Optimization

The foundational premise of Causal Bayesian Optimization (CBO) is that optimization takes place within a system governed by causal relationships: the manipulable variables are linked to the target through structural mechanisms rather than through a black-box input–output map. It is precisely this causal structure that requires and enables the optimizer to make algorithmic choices that standard BO does not face: which variables to intervene on (the intervention scope), at what level, and how abundant observational data relate to interventional effects through causal identification. This coupling of combinatorial scope selection with continuous value optimization, guided by a causal graph, is the defining feature that distinguishes CBO from black-box BO.

CBO introduces three innovations relative to standard BO. First, *graph-based scope reduction* uses the causal graph to prune intervention scopes that are structurally redundant or observationally dominated, concentrating the budget on scopes that the graph predicts can meaningfully move the outcome. Second, *observationally informed priors* convert abundant observational data into prior beliefs about interventional effects when identifiability holds, giving the surrogate a reasonable global shape before costly interventions begin. Third, CBO explicitly models an *observation–intervention trade-off*, allowing the algorithm to decide whether additional observational or interventional data is more valuable at each step.

Section 3.1 formalizes the CBO problem, defines the key concepts of minimal and possibly-optimal intervention sets, presents the surrogate and acquisition design, and identifies the limitations that motivate the extensions reviewed in the remainder of Section 3.

## 3 Methodological Frameworks

Causal Bayesian Optimization (CBO) is best viewed as a design space rather than a single algorithm. All CBO methods instantiate a common loop: maintain a probabilistic model of candidate interventional effects, score candidate interventions with an acquisition or decision rule, select the next intervention or observation action, and update beliefs using newly obtained observational and/or interventional data. What differentiates methods is where the causal structure is injected into this loop and which uncertainty sources are modeled.

**A unifying lens: design axes and assumptions.** In the literature, most variants can be understood through five interacting axes (Figure 1):

1. **Graph and system-knowledge assumptions.** Known graph, unknown graph, graph uncertainty, causal sufficiency, hidden confounding, and identifiability assumptions.

2. **Environmental assumptions.** Static, dynamic, contextual, constrained, adversarial or non-stationary, and multi-objective settings.

3. **Intervention representation.** Hard interventions, soft mechanism modifications, intervention scopes, functional policies, context-dependent actions, and mixed discrete–continuous actions.

4. **Surrogate architecture.** Direct effect-level surrogates, mechanism-level surrogates, graph-uncertain surrogates, multi-output models, and neural or deep-kernel variants.

5. **Decision rule and budget allocation.** EI-/UCB-style acquisition, information-gain search, optimism over plausible models, bandit allocation across scopes, and online-learning rules.

We use these axes to highlight what assumptions each method relaxes or introduces, what mathematical object changes relative to baseline CBO, and what statistical efficiency or computational cost follows.

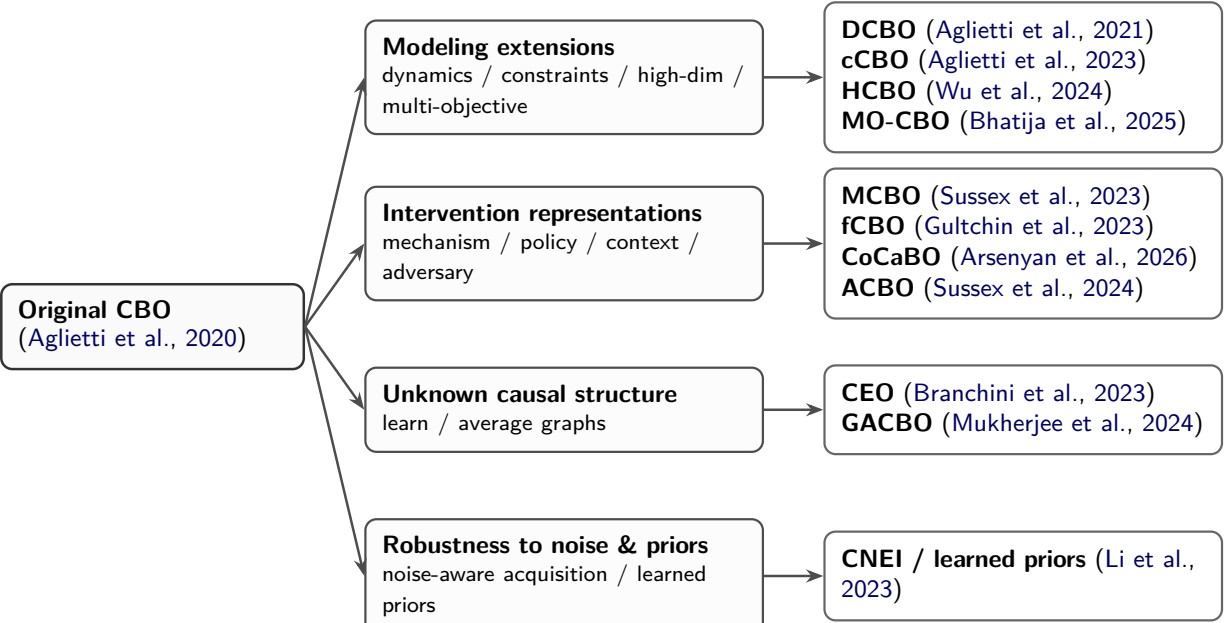

Figure 1: Causal Bayesian Optimization development map organized by the primary bottleneck addressed.

**How we summarize methods.** For each approach, we emphasize: (i) assumptions (graph knowledge, intervention type, observed variables); (ii) the main algorithmic lever (search-space reduction, surrogate design, acquisition/decision policy); and (iii) failure modes and scaling constraints. This avoids re-stating paper narratives verbatim and instead clarifies how the methods relate at the level of design choices in the CBO loop. Unless a statement is explicitly tied to benchmark evidence in Section 4, the "main challenge" and limitation remarks in this section are *analytical*: they follow from each method's stated assumptions, surrogate level, and decision rule (see also the identification-strategy audit in Table 4), and should not be read as new empirical findings of this survey.

**Common notation for method summaries.** To make the method summaries precise and comparable, we use a single notation throughout this section: $\mathcal{G}$ denotes the causal graph, $X_s \subseteq X$ an intervention scope, $\boldsymbol{\theta}$ an intervention value or policy parameter, $Y$ the target outcome, and $\mathcal{D}_{\text{obs}}$ and $\mathcal{D}_{\text{int}}$ the observational and interventional datasets. Effect-level surrogates model the map $g_s(\boldsymbol{\theta}) = \mathbb{E}[Y \mid do(X_s{=}\boldsymbol{\theta})]$ directly, whereas

mechanism-level surrogates place a model $\hat{f}_j$ on each structural equation $f_j(\mathrm{Pa}_j, U_j)$ and obtain the effect on $Y$ by propagating samples or uncertainty through $\mathcal{G}$. For each method family we state, in this notation, the core mathematical object that distinguishes it—the constrained objective for cCBO (Equation 16), the coverage objective for HCBO (Equation 17), mechanism-level propagation for MCBO (Equation 18), the contextual policy objective for CoCaBO (Equation 19), and the optimistic model-set acquisition for GACBO (Equation 20)—before the prose discussion.

### 3.1 Causal Bayesian Optimization: Core Framework

The original CBO framework (Aglietti et al., 2020) formalizes causal optimization as a search over interventions in a known causal graph. Let $X$ denote manipulable variables, $C$ non-manipulable context variables, and $Y$ the target outcome. An action consists of an intervention scope $X_s \subseteq X$ and an assignment $\mathbf{x}_s$ in the feasible intervention domain $\mathcal{D}(X_s)$.

**Definition 2** (CBO objective and action space)**.** *A CBO action is a pair $(X_s, \mathbf{x}_s)$ where $X_s \subseteq X$ is an intervention scope and $\mathbf{x}_s \in \mathcal{D}(X_s)$ is an assignment in the feasible intervention domain. The interventional objective is*

$$f(X_s, \mathbf{x}_s) \;=\; \mathbb{E}[\, Y \mid do(X_s = \mathbf{x}_s)\,]. \tag{9}$$

*The global causal optimization problem is*

$$(X_s^{\star}, \mathbf{x}_s^{\star}) \in \underset{X_s \subseteq X, \; \mathbf{x}_s \in \mathcal{D}(X_s)}{\arg\min} f(X_s, \mathbf{x}_s), \tag{10}$$

*with* $\arg\min$ *replaced by* $\arg\max$ *for maximization tasks.*

Definition 2 states the baseline CBO problem strictly according to its original formulation (Aglietti et al., 2020), which does not involve context variables. Context variables $C$ are introduced only in Section 3.3 when discussing contextual extensions such as CoCaBO, where the objective becomes $\max_\pi \; \mathbb{E}_C[\mathbb{E}[Y \mid do(X_s = \pi(C)), C]]$ (Equation 19). The objective in Definition 2 is an interventional expectation: none of the methods surveyed in this section optimizes counterfactual quantities $Y_{\mathbf{x}}(u)$, which we treat only as a future direction (Section 2.1).

The key point is that the optimizer is free to choose the scope of the intervention, not only the level. This yields a mixed discrete–continuous search problem and motivates two core ideas in CBO: graph-based pruning of candidate scopes and observationally informed priors over interventional effects.

**Why causality changes the BO problem.** A naive approach would flatten $(X_s, \mathbf{x}_s)$ into a single input vector and run the standard BO on the interventional objective. This ignores the structural information encoded in the causal graph. Causal knowledge enables two advantages that standard BO cannot exploit: (i) *scope reduction*: identifying which variables need to be manipulated at all, thereby shrinking the combinatorial search; and (ii) *informative priors*: converting plentiful observational data into prior beliefs about interventional effects, which is possible only when identifiability holds (Section 2.1).

**Scope reduction: minimal intervention sets.** A central insight of CBO is that the causal graph can prune intervention scopes that are structurally redundant. Two key notions formalize this:

**Definition 3** (Minimal Intervention Set (MIS) (Lee & Bareinboim, 2018; 2019))**.** *An intervention scope $X_s \subseteq X$ is a* minimal intervention set *for the target $Y$ if (i) there exists a directed path from at least one variable in $X_s$ to $Y$ in the causal graph $\mathcal{G}$ that is not blocked by the intervention on the remaining variables in $X_s$, and (ii) no proper subset $X_s' \subsetneq X_s$ satisfies condition (i). Equivalently, $X_s$ is a minimal set of manipulable variables whose joint intervention can affect $Y$ through the graph.*

**Definition 4** (Possibly-Optimal MIS (POMIS) (Lee & Bareinboim, 2018; 2019))**.** *A MIS $X_s$ is* possibly optimal *if, given the observational distribution $P(V)$ and the causal graph $\mathcal{G}$, one cannot determine from*

*observational evidence alone that* $\max_{\mathbf{x}_s} \mathbb{E}[Y \mid do(X_s = \mathbf{x}_s)]$ *is dominated by the optimal objective achievable under some other MIS $X_{s'}$. Formally, the set of POMIS is the subset of MISs that remains after eliminating scopes whose optimal interventional value is provably dominated under the available graph and distributional information.*

By restricting the search to POMIS, CBO avoids wasting budget on scopes that are causally irrelevant (no path to $Y$) or observationally dominated (another scope provably achieves a better objective). This reduction is not merely computational; it changes the statistical problem by concentrating evaluations on scopes that the graph predicts can meaningfully move the outcome. This graph-based scope reduction is one of the main reasons CBO can be statistically more efficient than standard BO: it avoids spending interventional budget on scopes that are causally irrelevant to $Y$ or dominated under the available graph and observational information. This within-graph scope reduction is complementary to causal abstraction: POMIS prunes intervention sets inside a fixed causal representation, whereas causal abstraction methods relate decisions across graphs at different levels of granularity through abstraction maps (Zennaro et al., 2024; Dyer et al., 2025). The latter becomes important when the optimizer must choose between fine-grained mechanistic variables and coarser latent or aggregate variables.

**Surrogate modeling: effect-level GPs with causal priors.** The CBO models the interventional objective for each candidate scope with an effect-level GP. The key innovation is the prior construction: when the interventional effect is identifiable, observational data are used to build an initial estimate of the interventional mean and its uncertainty:

$$f_s(\mathbf{x}_s) := \mathbb{E}[Y \mid do(X_s = \mathbf{x}_s)], \tag{11}$$

$$f_s(\cdot) \sim \mathcal{GP}(m_s(\cdot), \, k_s(\cdot, \cdot)), \tag{12}$$

$$m_s(\mathbf{x}_s) \approx \widehat{\mathbb{E}}[Y \mid do(X_s = \mathbf{x}_s)], \tag{13}$$

$$k_s(\mathbf{x}_s, \mathbf{x}'_s) = k_{\mathrm{RBF}}(\mathbf{x}_s, \mathbf{x}'_s) + \widehat{\sigma}_s(\mathbf{x}_s)\,\widehat{\sigma}_s(\mathbf{x}'_s), \tag{14}$$

where $k_{\mathrm{RBF}}(\mathbf{x}, \mathbf{x}') = \exp\big(-\|\mathbf{x} - \mathbf{x}'\|^2/(2\ell^2)\big)$ and $\widehat{\sigma}_s(\mathbf{x}_s)$ is the uncertainty in the observationally-derived effect estimate. The observational data thus serve as prior information, not as additional labels, giving the surrogate a reasonable global shape before costly interventions begin.

**Acquisition and the observation–intervention trade-off.** CBO uses a cost-aware expected improvement to choose between pairs (scope, value):

$$\mathrm{CEI}_s(\mathbf{x}_s) = \frac{\mathbb{E}\big[(y^\star - f_s(\mathbf{x}_s))_+\big]}{\mathrm{Co}(X_s, \mathbf{x}_s)}, \tag{15}$$

where $y^\star$ is the best value observed so far and $\mathrm{Co}(\cdot)$ is an intervention cost. Beyond the standard exploration–exploitation trade-off, CBO introduces an *observation–intervention trade-off*: the algorithm can decide whether to allocate budget to additional observational samples (cheap but potentially biased for interventional decisions) or to new interventions (expensive but causally informative). This framing is foundational: many later variants can be interpreted as changing *what is uncertain* and therefore changing *which data is worth acquiring.*

**The generic CBO loop.** At each iteration, the CBO loop proceeds in three stages: (i) the surrogate predicts the causal effect for each candidate intervention (scope, value), (ii) the acquisition function scores candidates using the posterior predictions, and (iii) the highest-scoring intervention is selected and evaluated. Algorithm 1 summarizes the generic CBO procedure, highlighting where causal knowledge enters. Steps 1–2 exploit the graph *before* the optimization loop; Steps 4–5 use a causal structure *within* each iteration. This template applies, with variations, to all methods reviewed in the remainder of this section.

**Limitations that motivate extensions.** The modularity of CBO also exposes its bottlenecks: (i) reliance on a correctly known graph and identifiable observational priors; (ii) limited robustness to noise and prior misspecification; and (iii) scaling challenges due to exponentially many candidate scopes and separate surrogates per scope. The remainder of this section can be read as a sequence of responses to these bottlenecks, organized by the design axis that each method primarily addresses.

---

**Algorithm 1** Generic Causal Bayesian Optimization Loop

---

**Require:** Causal graph $\mathcal{G}$ (or prior over graphs), observational data $\mathcal{D}_{\text{obs}}$, budget $T$
**Ensure:** Best intervention $(X_s^\star, \mathbf{x}_s^\star)$
 1: **Scope reduction** [causal]: Use $\mathcal{G}$ to identify candidate scopes (MIS/POMIS)
 2: **Prior construction** [causal]: Use $\mathcal{D}_{\text{obs}}$ and identifiability to set GP prior means $\{m_s\}$
 3: **for** $t = 1, \ldots, T$ **do**
 4:     **Surrogate update**: Update GP posteriors $\{\mu_t^{(s)}, \sigma_t^{(s)}\}$ for each scope $s$
 5:     **Acquisition** [causal]: Select $(X_s, \mathbf{x}_s) = \arg\max_{s,\mathbf{x}} \alpha_s(\mathbf{x}; \mathcal{D}_t)$ across scopes
 6:     **Evaluate**: Observe $y_t = f(X_s, \mathbf{x}_s) + \epsilon_t$ via intervention
 7:     **Update**: Add $(X_s, \mathbf{x}_s, y_t)$ to interventional dataset $\mathcal{D}_t$
 8: **end for**
 9: **return** $(X_s^\star, \mathbf{x}_s^\star)$ with best observed objective

---

## 3.2 Modeling Extensions

The original CBO formulation assumes a static causal system, a single objective, and an intervention space that remains tractable after graph-based pruning. Modeling extensions relax these assumptions by changing what the optimizer must represent: time variation that renders the objective non-stationary, feasibility that restricts attention to a safe set, dimensionality that enlarges the scope of the space, or multiple competing goals that shift the target from a single optimum to a Pareto set, in each case keeping the same surrogate–acquisition loop while adding structure that decomposes the problem into smaller subproblems that share statistical strength.

**Time-evolving causal systems (DCBO).** Dynamic Causal Bayesian Optimization (DCBO) addresses environments where the best intervention is non-stationary, typically because mechanisms drift or the system state evolves (Aglietti et al., 2021). The main contribution is not a new acquisition rule, but a transfer mechanism. Under invariance assumptions such as a fixed graph and additive-noise structure, interventional knowledge from earlier time steps can be mapped into an informative prior for the current step. This treats time as a source of reusable evidence so that exploration can be amortized rather than restarted. The trade-off is the sensitivity of the assumption. If the graph or key mechanisms shift in ways not captured by the transfer recursion, the prior can become overconfident and bias the search, so DCBO is strongest when temporal variation is structured and not adversarial.

**Constraints and safe causal optimization (cCBO).** Constrained Causal Bayesian Optimization (cCBO) targets settings where interventions must satisfy safety or feasibility constraints (Aglietti et al., 2023). The key insight from the modeling is that constraints change the definition of a useful intervention scope. A scope that can move the objective but cannot maintain feasibility is effectively irrelevant; cCBO therefore extends graph-based pruning to constraint-relevant pathways through constrained minimal intervention sets, and it couples objective and constraints through multi-output surrogates so evidence about feasibility shapes where the objective surrogate should be trusted. Formally, cCBO solves

$$(X_s^\star, \mathbf{x}_s^\star) \in \underset{X_s, \mathbf{x}_s}{\arg\max} \ \mathbb{E}[Y \mid do(X_s = \mathbf{x}_s)] \quad \text{s.t.} \quad \Pr\big(c_j(X_s, \mathbf{x}_s) \leq 0 \ \forall j\big) \geq 1 - \delta, \tag{16}$$

where each $c_j$ is a constraint function modeled jointly with the objective via multi-output GPs, and $\delta$ controls the tolerated feasibility violation. In practice, this often acts as a regularizer that prevents spending the budget in regions that look promising under the objective surrogate but are unlikely to be deployable. The main limitation is scaling with the number of constraints and candidate scopes, although structural pruning mitigates this in graphs with sparse causal connectivity.

**High-dimensional intervention spaces (HCBO).** High-Dimensional Causal Bayesian Optimization (HCBO) addresses the combinatorial bottleneck of scope selection in large graphs (Wu et al., 2024). Its central move is to replace exact enumeration of minimal or possibly-optimal scopes with an efficient coverage-based approximation. Specifically, HCBO selects a scope family $\mathcal{S}^\star \subseteq 2^X$ by solving a coverage problem

that approximately maximizes the fraction of causal pathways to $Y$ that are "covered" by at least one scope in $\mathcal{S}^\star$. Let $\mathrm{Paths}(\mathcal{G}, Y)$ denote the set of directed paths from manipulable variables to $Y$. HCBO seeks

$$\mathcal{S}^\star \in \underset{\mathcal{S} \subseteq 2^X,\, |\mathcal{S}| \leq K}{\arg\max} \; \big| \{ p \in \mathrm{Paths}(\mathcal{G}, Y) : \exists\, X_s \in \mathcal{S},\; X_s \cap p \neq \varnothing \} \big|, \tag{17}$$

where $K$ is a user-specified scope budget. "Intrinsic causal dimension" refers to the minimal $K$ such that $\mathcal{S}^\star$ covers all causal pathways to $Y$. This reframes high-dimensional CBO as a hierarchical problem: select a promising scope family, then run continuous BO within the chosen scopes. HCBO also introduces normalized cross-scope scoring so that surrogates trained on different scopes can be compared in a meaningful way despite scale and data imbalance. The cost of scalability is that any coverage proxy can miss narrow, high-impact pathways, so HCBO is most reliable when the system exhibits causal sparsity or a low intrinsic causal dimension.

**Multiple outcomes and Pareto structure (MO-CBO).** Multi-Objective Causal Bayesian Optimization (MO-CBO) generalizes CBO to settings with multiple targets, where the goal is to approximate a Pareto front rather than a single optimum (Bhatija et al., 2025). The key observation is that causal structure is especially valuable when objectives compete. Intervening on everything can obscure trade-offs by coupling variables that need not be coupled. MO-CBO uses the graph to reduce the search to a minimal collection of local multi-objective subproblems and then allocates evaluation budget across them using a cross-scope criterion based on relative hypervolume improvement. Within each local problem, standard multi-objective BO components can be used, for example, DGEMO (Konakovic Lukovic et al., 2020). The main challenge remaining is combinatorial. The number of local subproblems can still grow quickly with dense graphs or many objectives, motivating stronger cross-scope sharing and joint surrogates that exploit causal factorization across objectives.

### 3.3 Intervention Strategy Developments

Beyond modeling extensions, another major line of work changes what an intervention looks like and how decisions are made during optimization. The common motivation is that treating each intervention as a single black-box query can waste information. These methods modify the BO loop so it can reuse causal structure inside the system, express richer actions such as policies, and stay meaningful under contextual variation or non-stationarity.

Two orthogonal design choices appear in this section and should not be conflated: (i) expanding the *intervention representation*, from static values to mechanism parameters, policies, or context-dependent actions (fCBO, CoCaBO), and (ii) changing the *surrogate architecture*, from effect-level to mechanism-level models (MCBO, ACBO). These are independent axes of the design space introduced at the start of Section 3: one can pair mechanism-level surrogates with only hard interventions, or effect-level surrogates with soft interventions. Relatedly, which uncertainty dominates a method is not itself a design choice but a property *induced* by the designer's system-knowledge assumptions (known versus unknown graph, causal sufficiency, known feasible set); the taxonomy therefore lists system-knowledge assumptions, rather than uncertainty sources, as the primary axis, and the discussion below identifies the induced uncertainty for each method.

Two further clarifications help place the methods of this section in the taxonomy. First, CoCaBO and ACBO differ not only in intervention representation but also on the *environmental* axis: CoCaBO assumes a contextual environment in which the best action depends on observed exogenous context, whereas ACBO assumes an adversarial or non-stationary environment; grouping them here reflects shared machinery for richer decision-making, not identical assumptions. Second, a closely related formulation outside the SCM tradition is Bayesian optimization of function networks (Astudillo & Frazier, 2021), which optimizes over a known computational DAG whose intermediate node values are observed. Function networks anticipate the mechanism-level, soft-intervention view adopted by MCBO and ACBO, and the function-network benchmarks introduced there are inherited by our soft-intervention benchmark (Section 4.5); the key difference from SCM-based CBO is that function-network inputs are causally independent decision variables rather than variables embedded in a causal system.

**Mechanism-level modeling with guarantees (MCBO).** Model-based Causal Bayesian Optimization (MCBO) models the causal system at the level of structural mechanisms rather than only the intervention-to-outcome effect (Sussex et al., 2023). It places Gaussian process priors on individual structural equations and propagates uncertainty through the known causal graph to obtain uncertainty over the target outcome. This changes data reuse. Observations of intermediate variables update upstream mechanisms and therefore tighten uncertainty about downstream outcomes, which can reduce the number of costly interventions needed to identify good actions. MCBO then uses an optimism-based decision rule in the spirit of UCB and provides regret guaranties, connecting CBO to theoretical results from Gaussian process bandits. The main limitations are practical. Mechanism posteriors must be maintained for multiple equations, uncertainty propagation increases computational cost, and acquisition optimization can be challenging. MCBO also assumes a known graph and a mechanism model class that is expressive enough for the system at hand.

Formally, MCBO places independent GP priors on the structural mechanisms, $\hat{f}_j \sim \mathcal{GP}(m_j, k_j)$ for each node $V_j$, and models the outcome of an intervention $\boldsymbol{\theta}$ (a hard clamp or soft mechanism shift) by propagating through the graph in topological order:

$$V_j = \hat{f}_j\big(\mathrm{Pa}_j(\boldsymbol{\theta}), U_j\big) \text{ for each non-intervened node,} \qquad \hat{Y}(\boldsymbol{\theta}) = \hat{f}_Y\big(\mathrm{Pa}_Y(\boldsymbol{\theta}), U_Y\big), \qquad (18)$$

where each parent value $\mathrm{Pa}_j(\boldsymbol{\theta})$ is itself generated by upstream mechanism models under the intervention. The decision rule is optimism-based: MCBO selects $\boldsymbol{\theta}_t \in \arg\max_{\boldsymbol{\theta}} \mathrm{ucb}_t(\boldsymbol{\theta})$, where $\mathrm{ucb}_t(\boldsymbol{\theta})$ upper-bounds the propagated posterior of $\mathbb{E}[Y \mid \boldsymbol{\theta}]$ using calibrated confidence bounds on each mechanism, which yields the cumulative-regret guarantees established in Sussex et al. (2023).

Compared to effect-level CBO, MCBO has two additional trade-offs that deserve explicit discussion. First, maintaining $|V|$ separate mechanism-level GPs and propagating posterior uncertainty through the full graph at every acquisition step is computationally far more expensive than fitting a single effect-level GP. The per-iteration cost scales with the number of mechanism models and the depth of the graph, limiting current MCBO implementations to graphs with tens of nodes. Second, mechanism-level models are highly vulnerable to unobserved confounders between intermediate nodes: if a hidden confounder links two internal variables, the GP for the downstream mechanism will be fitted on systematically biased inputs, and this bias propagates through the graph. Effect-level CBO can mitigate the same issue more simply by discarding the biased observational prior and proceeding with an uninformative GP prior.

A related nuance concerns intermediate observations under effect-level models. The claim that effect-level models "cannot reuse intermediate observations" is only strictly true for the GP posterior itself: a direct $X_s \to Y$ surrogate does not condition on intermediate node values. However, intermediate observations collected during interventional trials can be appended to the observational dataset and used to re-estimate or refine the observationally-derived prior mean $m_s(\cdot)$ at each step. This indirect reuse does not provide the same per-mechanism uncertainty reduction as MCBO, but it is a practical pathway that existing effect-level implementations could exploit.

**Policies as interventions (fCBO).** Functional Causal Bayesian Optimization (fCBO) expands the action space from selecting a value to selecting a policy that maps the observed context to an action (Gultchin et al., 2023). This is motivated by domains such as personalized treatment, where a single fixed intervention is rarely optimal for all individuals. fCBO makes policy search tractable by defining a kernel over policies using RKHS tools and running Bayesian optimization in this function space with a functional expected improvement rule. In the soft-intervention terminology of Section 2.1, fCBO's intervention parameter is the policy itself: $\theta = \pi \in \mathcal{H}_k$, an element of an RKHS of functions, and the functional intervention $do(X_s = \pi(Z))$ replaces the mechanism of $X_s$ with the policy applied to the values of its input variables $Z$, rather than with a fixed constant. The key contribution is that it unifies causal optimization and policy learning within a single sequential procedure, where the surrogate models policy-to-outcome effects and the acquisition balances exploration across qualitatively different decision rules. The main challenge is representation. Because the policy space is large, empirical performance depends strongly on the chosen parameterization and on whether the kernel captures a meaningful similarity between policies.

**Context-dependent scope selection (CoCaBO).** Contextual Causal Bayesian Optimization (Co-CaBO) targets settings where the exogenous context affects outcomes and where the best intervention

depends on the realized context (Arsenyan et al., 2026). The additional difficulty is that one must decide not only which action to take, but also which contextual variables should be used to condition decisions. CoCaBO addresses this by separating the problem into two levels. At the outer level, it allocates the budget across candidate mixed policy scopes using a bandit-style UCB rule. At the inner level, it optimizes interventions within the selected scope using a mixed-variate Bayesian optimization routine such as HEBO (Cowen-Rivers et al., 2022). Formally, a *mixed policy scope* $\mathcal{M}_k = (X_{s_k}, C_{s_k})$ pairs an intervention scope $X_{s_k}$ with a context subset $C_{s_k}$. CoCaBO optimizes a context-dependent objective of the form

$$\max_{\pi:\mathcal{C}\to\mathcal{X}} \mathbb{E}_C[\mathbb{E}[Y \mid do(X_{s_k} = \pi(C_{s_k})), C]], \tag{19}$$

where $\pi$ maps observed context values to intervention values within the chosen scope, and the outer expectation is over the context distribution. This architecture makes explicit that context introduces a discrete model selection problem on top of continuous optimization. The main limitations are derived from the hierarchy. The outer allocation must spend trials to distinguish scopes, and improvements learned within one scope do not necessarily transfer to others.

**Non-stationarity and adversarial environments (ACBO).** Adversarial Causal Bayesian Optimization (ACBO) addresses cases where the system response changes over time or is influenced by an adversary that can also intervene in the system (Sussex et al., 2024). This shifts the objective from identifying a single best intervention to achieving low regret over a series of rounds. ACBO combines multiplicative weight update from adversarial online learning (Littlestone & Warmuth, 1994; Freund & Schapire, 1997) with causal modeling to construct calibrated optimistic reward estimates via a causal UCB oracle. ACBO inherits MCBO's soft-intervention model: the learner's action $\boldsymbol{\theta}$ enters the structural mechanisms as an additional input exactly as in Equation 18, with the adversary's action entering the same mechanisms, so interventions shift mechanism inputs rather than clamping node values; the multiplicative-weights layer then operates over a discretized set of such actions. A central benefit of using the causal graph is that it can reduce regret dependence on the nominal action dimension when rewards factor through causal compositions, improving over non-causal analogs such as GP-MW (Sessa et al., 2019). The main challenges are computational and statistical. The oracle can be expensive to optimize, and in non-stationary regimes, the method relies on uncertainty sets that must track drift accurately to avoid misleading optimism.

### 3.4 Unknown causal structure

A major practical barrier for Causal Bayesian Optimization is that many applications do not provide a trusted causal graph in advance. In this regime, the optimization loop must allocate a limited intervention budget across two coupled goals: improving the outcome and reducing the ambiguity about which causal explanations remain consistent with the data. This introduces an additional uncertainty source, structure uncertainty, on top of uncertainty about causal mechanisms and observation noise.

To keep terminology consistent throughout the paper, we use *unknown graph* for the assumption class in which the learner is not given the causal graph; *graph uncertainty* (used interchangeably with *structure uncertainty*) for the learner's remaining uncertainty over graph structure, however it is represented (a posterior, a confidence set, or a candidate list); and *graph misspecification* for the distinct situation in which the learner is given, and trusts, a single graph that differs from the true one. The methods in this subsection address the first two; graph misspecification is instead probed by the stress tests in Appendix B.1. Existing approaches mainly differ in how they represent uncertainty over candidate graphs and in whether they explicitly target the decision-relevant structure, that is, the parts of the graph that materially change which intervention should be selected.

**Information-seeking joint learning and optimization (CEO).** Causal Entropy Optimization, CEO, couples structure learning and optimization using an explicitly information-seeking decision rule (Branchini et al., 2023). The method maintains a posterior distribution over the candidate graphs and predicts the interventional outcomes by averaging across this posterior, producing a surrogate that reflects both the mechanism uncertainty and the graph uncertainty. The next intervention is then chosen to maximize the expected information gain about the identity of the best intervention, rather than to maximize the predicted

reward alone. This focus has an important practical consequence. CEO is not optimized to recover a globally accurate graph. It spends budget on experiments that resolve the structural ambiguities that matter to decide how to act so that it can focus on a decision-relevant subgraph for the target $Y$.

The main limitation is computational overhead. Approximating a posterior over graphs typically requires sampling in a large discrete space, and estimating information gain must be repeated across many candidate interventions. As a result, CEO is most practical when the graph is small to medium or when strong structural priors substantially narrow the set of plausible graphs.

**Optimism over plausible models (GACBO).** Graph-Agnostic Causal Bayesian Optimization, GACBO, follows a more algorithmic route that emphasizes scalability (Mukherjee et al., 2024). Rather than choosing interventions by entropy reduction, it maintains a confidence set of causal models that remain statistically plausible given the data and selects interventions that could be optimal under at least one model in this set. Formally, let $\mathcal{C}_t$ denote the confidence set of plausible causal models (graph–mechanism pairs) at round $t$. GACBO selects the next intervention by solving

$$(X_s, \mathbf{x}_s) \in \arg\max_{X_s, \mathbf{x}_s} \ \max_{M \in \mathcal{C}_t} \ \hat{f}_M(X_s, \mathbf{x}_s), \tag{20}$$

where $\hat{f}_M$ is the predicted interventional objective under model $M$. This "optimism in the face of uncertainty" principle selects interventions that look promising under *some* plausible causal explanation, which drives exploration toward experiments that can eliminate incorrect models. This implements optimism in the face of uncertainty and replaces posterior averaging with best-case evaluation over models consistent with the observations. The benefit is that GACBO avoids explicit entropy computations and aligns closely with optimistic bandit and Bayesian optimization principles, which can make it easier to implement and scale.

The main risk is over-optimism early in the learning process. When the confidence set is wide, the method may conduct trials on interventions that look good only under structural hypotheses that have not yet been ruled out. Performance, therefore, depends on the calibration of the confidence set and on how quickly interventional evidence eliminates incorrect graphs and mechanisms. Recent work extends this direction by exploring alternative uncertainty decompositions, including approaches that emphasize learning exogenous distributions and broader formulations of unknown-graph problems (Ren et al., 2026; Durand et al., 2025).

**Summary.** Unknown-graph CBO turns intervention design into an adaptive experimental design problem in which structure learning is not a preprocessing step but part of the optimization loop. The CEO and GACBO represent two complementary ways to couple these goals. The CEO prioritizes information gain about the best intervention under a Bayesian posterior, while GACBO prioritizes optimistic improvement under a frequentist-style set of plausible models. The choice between them largely reflects an information-versus-optimism trade-off and the computational budget available for representing and updating uncertainty over graphs.

## 3.5 Robustness to noise and priors

Even with a known graph, practical CBO pipelines face two recurring robustness issues. First, interventional outcomes can be noisy, heteroscedastic, or affected by unmodeled disturbances, which can destabilize improvement-based acquisitions. Second, observationally derived priors may be misspecified, either because identifiability assumptions fail, or because the estimand is hard to estimate accurately from finite data, or because observational correlations leak confounding into a prior mean.

Counter-noise Expected Improvement (CNEI) (Li et al., 2023) targets both issues through a noise-aware acquisition and a learned prior construction. CNEI modifies EI-style selection to account for the noise level so that the acquisition does not reward an apparent improvement dominated by measurement variance. In parallel, it estimates a surrogate prior mean by fitting predictive models to observational data and using the resulting predictor as a warm-start mean for the GP. This can improve the early-stage search by giving the surrogate a reasonable global shape before sufficient interventional data accumulate.

Table 2: Classification of major CBO methods by key assumptions and modeling choices. Each row lists the graph/system-knowledge assumption, surrogate modeling target, decision strategy, and intervention or environmental feature introduced by the method. Superscripts mark causal-sufficiency assumptions: ‡ indicates that the method's formulation can accommodate hidden confounders in the graph (e.g., via identification-based adjustment or averaging over structures), whereas † indicates that the method models individual mechanisms or rewards directly and therefore assumes causal sufficiency (fully observed parent–child relations). Unmarked methods follow the common known-graph setting in which causal sufficiency is typically assumed but is not central to the method's design.

| Method | Graph/system assumptions | Surrogate target | Acquisition / strategy | Environment / intervention feature |
|---|---|---|---|---|
| CBO (2020) | Known DAG$^{\ddagger}$ | Effect-level GP for $Y$ with causal prior | Causal EI; cost-aware; observe vs. intervene | Scope + value, hard interventions |
| DCBO (2021) | Known DAG$^{\ddagger}$ | Dynamic effect-level GP | Time-aware causal EI | Intervention sequences over time |
| MCBO (2023) | Known DAG$^{\dagger}$ | Mechanism-level GPs, full SCM | UCB / optimism, regret bounds | Hard/soft interventions; mechanism modeling |
| cCBO (2023) | Known DAG$^{\ddagger}$ | Multi-output GP, objective and constraints | Constrained EI, feasibility-weighted | Safety and feasibility constraints |
| fCBO (2023) | Known DAG | GP over policies, RKHS function space | Functional EI | Functional and policy interventions |
| CoCaBO (2026) | Known DAG | BO within scope plus bandit over scopes | Two-layer, MAB-UCB then BO within scope | Contextual mixed interventions |
| ACBO (2024) | Known DAG$^{\dagger}$ | Mechanism-level modeling, full SCM | MW-style online learning plus causal UCB oracle | Adversarial and non-stationary environments |
| MO-CBO (2025) | Known DAG | GP per objective, local MOBO subproblems | Relative hypervolume improvement | Multi-objective Pareto optimization |
| HCBO (2024) | Known DAG | Multiple effect-level GPs, selected scopes | Normalized UCB-style scope scoring | High-dimensional scope selection |
| CEO (2023) | Unknown DAG$^{\ddagger}$ | Bayesian model average over graphs, mixture | Entropy and information gain | Learns graph structure during optimization |
| GACBO (2024) | Unknown DAG$^{\dagger}$ | Confidence set over graphs and mechanisms | Optimism / UCB over plausible models | Structure discovery via optimistic exploration |
| CNEI (2023) | Known DAG | GP for $Y$ with learned supervised prior | Counter-noise EI | Noise-robust acquisition; learned priors |

A critical distinction is that CNEI's learned prior is a supervised, purely correlational predictor $\hat{\mathbb{E}}[Y \mid X]$ rather than a causally adjusted estimand $\hat{\mathbb{E}}[Y \mid do(X)]$. This makes CNEI a fundamental departure from standard CBO prior construction: when confounding is present, the correlational prior can encode spurious associations that a causal adjustment would have removed. CNEI therefore trades causal soundness for practical robustness, relying on the noise-aware acquisition to prevent the potentially biased prior from dominating the search as interventional evidence accumulates.

Table 3: Datasets used in empirical evaluations across CBO variants (as reported in the corresponding papers). We add a category column to make the dataset's family structure explicit and use light grid lines to improve readability. The dataset-usage table is restricted to single-objective CBO evaluations.

| Category | Dataset | CBO | cCBO | fCBO | MCBO | HCBO | CEO | GACBO | CNEI | ACBO | CoCaBO |
|---|---|---|---|---|---|---|---|---|---|---|---|
| Synthetic (hard) | ToyGraph | ✓ | ✓ | ✓ | ✓ | | ✓ | ✓ | | | |
| | Synthetic | ✓ | | | | | | | ✓ | | |
| | Synthetic-2 | | ✓ | | | | | | | | |
| | Chain-hard | | | ✓ | | | | | | | |
| Synthetic (soft) | Ackley | | | | ✓ | | | ✓ | | ✓ | |
| | Rosenbrock | | | | ✓ | | | ✓ | | ✓ | |
| | Dropwave | | | | ✓ | | | ✓ | | ✓ | |
| | Alpine2 | | | | ✓ | | | ✓ | | ✓ | |
| | Chain-soft | | | ✓ | | | | | | | |
| Real / fitted SCMs (hard) | Ecology | ✓ | | | | | ✓ | | | | |
| | Healthcare | ✓ | ✓ | ✓ | ✓ | ✓ | ✓ | | | | ✓ |
| | Protein-reconstructed | | ✓ | | | | | | | | |
| | Epidemiology | | | | | | ✓ | ✓ | | | |

A key caveat is that observationally learned priors are not inherently causal. They can encode spurious associations when confounding is present, so their role should be weak guidance with appropriately inflated uncertainty rather than a substitute for interventional evidence. This motivates a broader open direction: robust CBO procedures that explicitly control how strongly observational information can influence acquisition decisions under noise, misspecification, and possible hidden confounding.

## 3.6 Comparison tables and unifying observations

This subsection consolidates the cross-method synthesis in one place: rather than closing each preceding subsection with its own summary, we compare the method families once, here, along the design axes introduced at the start of Section 3. Tables 2 and 3 summarize the field from two complementary angles. Table 2 organizes methods by key design dimensions: assumptions about the causal graph, the surrogate modeling target (effect-level, mechanism-level, or graph-uncertainty) and the acquisition or decision strategy. Table 3 maps empirical coverage and reveals substantial fragmentation: beyond a small core of shared benchmarks (ToyGraph, Healthcare), many datasets appear in only one or two papers, which complicates cross-paper performance comparison.

We additionally include an identification-strategy audit table. In particular, none of the benchmarked released pipelines exposes a general front-door or full-ID implementation in the benchmark path used here; where observational adjustment is used, it is method-specific and should not be interpreted as a complete ID-algorithm implementation.

Together, these tables highlight two orthogonal axes of variation. The *modeling axis* ranges from effect-level surrogates (CBO, DCBO, cCBO, HCBO, CNEI) through mechanism-level models with uncertainty propagation (MCBO, ACBO) to explicit graph-uncertainty methods (CEO, GACBO). The *action axis* ranges from scope-and-value selection (CBO, DCBO, HCBO) through policy- or context-dependent decisions (fCBO, CoCaBO) to adversarial interaction (ACBO). We additionally include an identification-strategy audit table that separates each pipeline's surrogate level, method-level identification assumption, and benchmark effect source. This audit clarifies that the benchmark compares released pipelines as complete systems, not isolated acquisition functions, and that none of the benchmarked released pipelines exposes a general front-door or full-ID implementation in the benchmark path used here. Recognizing these interactions is important for practitioners choosing a method and for researchers identifying gaps in the design space.

Table 4: Identification-strategy audit for benchmarked pipelines. "Method-level identification" describes the causal-effect identification strategy assumed or implemented by the released method. "Benchmark effect source" describes how the benchmark wrapper supplies or approximates effects when re-running the released pipeline.

| Method | Surrogate level | Method-level identification | Benchmark effect source |
|---|---|---|---|
| BO | Target-level GP | None; non-causal baseline | Direct black-box evaluations |
| CoCaBO | Mixed-input BO | No explicit ID estimand in our wrapper; graph/scope-based policy selection | Benchmark SCM / task oracle evaluations |
| CBO | Effect-level GP | Observational adjustment when identifiable; not full ID in released code | Benchmark wrapper uses SCM/oracle effect estimates where available |
| cCBO | Effect-level / constrained GP | Observational adjustment with constraints; not full ID in released code | Benchmark wrapper uses SCM/oracle effect estimates where available |
| DCBO | Dynamic effect-level / temporal GP | Temporal adjustment under known dynamic-graph assumptions; not full ID | Benchmark dynamic SCM / oracle evaluations |
| HCBO | Scope-level / high-dimensional CBO | Graph-based scope selection; not full ID | Benchmark SCM / SEM evaluations |
| CEO | Graph-uncertain / oracle SEM search | Structure-learning over candidate graphs; not full ID over arbitrary latent structures | Candidate graph / SEM evaluations |
| MCBO | Mechanism-level GP | None required for effect identification; mechanisms are modeled directly | Mechanism-level simulation / forward propagation |

## 3.7 Connections to Adjacent Fields

CBO does not exist in isolation. Several established research areas share overlapping goals, tools, or assumptions. Understanding these connections clarifies what CBO inherits, where it innovates, and what cross-pollination opportunities remain.

**Causal abstractions and multi-scale decision making.** Causal abstraction studies how high-level causal models relate to lower-level mechanistic models through abstraction maps. This perspective is directly relevant to CBO whenever interventions can be chosen at multiple resolutions. Causally Abstracted Multi-Armed Bandits (CAMAB) (Zennaro et al., 2024) formalizes decision making across causally linked abstraction levels, while AT-UCB (Dyer et al., 2025) shows that exploiting abstraction hierarchies can reduce cumulative regret in complex simulators, including epidemiological models—precisely the kind of complex simulation setting where CBO's aspiration for real-world deployment meets computational tractability limits. For CBO, this suggests a multi-scale analogue of scope reduction: rather than only pruning intervention scopes within one graph, the optimizer may transfer or prune decisions across graphs at different granularities. A first proposal in this direction for CBO itself is MFACBO (Zeitler, 2025), which connects causal abstraction levels to multi-fidelity Bayesian optimization; we note that this proposal is a workshop contribution and not yet peer-reviewed. We therefore treat causal abstraction learning as distinct from causal representation learning from raw observations, and as a promising route for CBO over latent, coarse-grained, or hierarchical causal variables.

**Causal bandits.** The causal bandit literature (Lattimore et al., 2016; Lu et al., 2021; Nair et al., 2021) also selects interventions using causal structure, but typically assumes discrete action sets, parametric reward models, and focuses on cumulative regret in stationary environments. CBO extends this setting in three directions: (i) continuous intervention domains requiring nonparametric surrogates, (ii) mixed discrete–continuous action spaces (scope value + and (iii) more robust uncertainty modeling through GP. Conversely, causal bandits offer tighter finite-sample regret bounds and principled treatments of partial graph knowledge that CBO methods have only begun to adopt (e.g., MCBO's regret analysis draws on GP-bandit theory). A promising direction is to unify the two frameworks by developing CBO methods with bandit-style theoretical guaranties for continuous domains.

**Bayesian experimental design.** The Bayesian optimal experimental design (BOED) (Chaloner & Verdinelli, 1995; Foster et al., 2021) selects experiments to maximize information about the model pa-

rameters. CEO (Branchini et al., 2023) can be viewed as a BOED method applied to intervention selection: choose experiments to maximize the information gained about the identity of the best intervention. The key distinction is that standard BOED targets parameter estimation without an optimization objective, whereas CEO jointly optimizes information and reward. This connection suggests that advances in amortized BOED (Foster et al., 2021) could accelerate information-seeking CBO methods.

**Active causal discovery.** Active structure learning (Spirtes et al., 2000; Chickering, 2002; Eberhardt & Scheines, 2007) selects interventions to learn the causal graph itself, while CBO selects interventions to optimize an outcome. Unknown-graph CBO methods (CEO, GACBO) bridge these goals by learning graph structure as a means to better optimization. The distinction is that active discovery aims for global graph accuracy, whereas CBO needs only *decision-relevant* structure, the parts of the graph that change which intervention is optimal. This focus on decision-relevant learning is a truly novel aspect of unknown-graph CBO.

**Safe and constrained optimization.** Safe Bayesian optimization (Sui et al., 2015; Berkenkamp et al., 2023) maintains feasibility during optimization, which is closely related to cCBO's (Aglietti et al., 2023) constrained causal optimization. The causal contribution is that safety constraints can be analyzed through the graph: a scope that cannot affect a constraint variable need not model constraint feasibility, reducing the multi-output modeling burden. Future work could combine safe BO's theoretical safety guaranties with CBO's causal constraint analysis.

**Policy search and reinforcement learning.** Functional CBO (fCBO) (Gultchin et al., 2023) and contextual CBO (CoCaBO) (Arsenyan et al., 2026) can be viewed as policy search methods operating within an explicit causal model. Unlike model-free reinforcement learning, these methods exploit the known graph to decompose the policy-to-outcome mapping and to construct informative priors from observational data. The main limitation relative to RL is the assumption of a single-step (or few-step) decision problem; extending CBO to sequential multi-stage intervention policies remains an open challenge.

## 4 Experiments

Although this paper is primarily a survey, we include a benchmark study to support two of the survey's main claims. First, existing CBO papers are often evaluated in heterogeneous codebases and on only partially overlapping datasets, making cross-paper comparison difficult. Second, commonly reported efficiency metrics can be sensitive to how optimization trajectories are summarized, especially when improvements occur late in the evaluation budget. Our benchmark is therefore designed to separate algorithm execution from evaluation: methods can run in their native implementations, but their outputs are exported into a common best-so-far trajectory format and re-scored with identical metric code.

### 4.1 CBO benchmark and evaluation protocol

We release a CBO benchmark at https://anonymous.4open.science/r/CausalBO_Benchmark. The benchmark is a standardized repackaging and extension of inherited CBO/function-network tasks, rather than an entirely new collection of datasets. It includes eight curated hard-intervention scenarios and a separate set of soft-intervention benchmarks, with dataset provenance and code-authoritative domains documented in Appendix A.4. It provides standardized implementations of GAP and PA-GAP. Because existing CBO methods often rely on incompatible software environments, the benchmark does not force all algorithms into a single execution stack. Instead, each method is run in its native or vendored implementation, and its trajectory is exported and re-scored using the same evaluation pipeline. This design enables reproducible comparisons across heterogeneous implementations and makes it easy to add new methods, datasets, or metrics.

Because the implemented methods target different problem formulations, we interpret the aggregate rank plots as descriptive overview figures rather than as a single homogeneous leaderboard. The benchmark uses native or vendored implementations with released defaults, common trajectory export, and common scoring; it is therefore a reproducibility harness rather than a fully controlled reimplementation study. Appendix B.3

reports rank-reliability analyses for the retained rank summaries, and Appendix A.2 documents method settings and wrapper choices.

To make the formulation labels of Table 5 usable as fixed reference points rather than user-configured settings, each task in the repository ships with a frozen configuration: the intervention domains, optimization direction, observational sample sizes, constraint functions and thresholds for the constrained tasks, the policy space for Chain-soft, and the reference optimum used for scoring. A new method can therefore be evaluated on a given formulation without any task-design decisions being left to the user.

**Execution protocol.** For each data set, method, and trial limit, we run 20 random seeds. A seed fixes the observational sample, the initial interventional design, and the stochastic components of the optimizer. All methods are evaluated under the same trial limits, intervention domains, optimization direction, and scoring code. An interventional evaluation counts as one trial. Observational data used to construct priors, learn graphs, or initialize surrogates are fixed before the optimization loop and are not counted as interventional trials unless otherwise stated. Each method is run with its released default hyperparameters unless explicitly stated in Appendix A.2; we do not tune hyperparameters separately for each data set.

**Trajectory export and scoring.** The output of each method is converted to a common trajectory format with two required fields: trial number and best-so-far objective value in the natural optimization direction of the task. For minimization tasks, this is the minimum value observed up to the trial $t$; for maximization tasks, it is the maximum value observed up to the trial $t$. The trajectory includes an initial value followed by interventional trials. Thus, if the budget is $B$ interventions, the exported trajectory contains $B + 1$ entries: one initial best value and $B$ post-intervention best-so-far values. GAP and PA-GAP are then computed from these exported trajectories using the same evaluation script for all methods.

**Reference optimum.** Both GAP and PA-GAP require a reference optimum $y^*$. In our benchmark, $y^*$ is computed offline with oracle access to the benchmark SCM, function generator, or precomputed interventional data. This value is used only for scoring and is never provided to any optimizer. For low-dimensional or discrete intervention domains, we enumerate or densely evaluate admissible intervention scopes and values. For continuous domains, we use high-budget offline search with multi-start optimization or stored oracle intervention sets. The exact offline budgets, grids, restarts, and dataset-specific procedures are reported in the Appendix A.3.

**Failure handling.** When a method proposes an infeasible or out-of-domain intervention, we follow the behavior of the released implementation. If the implementation projects the proposals back to the admissible domain, the projected intervention is evaluated. Otherwise, the trial is marked as a failed proposal, and the trajectory records the current best-so-far value. Runs with numerical errors, missing outputs, or non-finite objective values are logged and are not silently removed. Runtime is not used as a primary metric because implementations differ substantially in language, hardware support, and dependency stacks.

**Cost model.** Unless otherwise stated, each intervention has a unit cost. This choice isolates the efficiency of the sample but does not evaluate the full cost-aware behavior of the CBO acquisition functions. In real applications, the cost of the intervention can depend on the target variable, the number of variables manipulated, the magnitude of the intervention, safety constraints, or whether the action is observational or experimental. We therefore report the current results as unit-cost benchmarks and treat realistic intervention-cost modeling as a separate evaluation dimension.

### 4.2 Performance metrics

We evaluated each run using two numerical efficiency metrics, GAP and PA-GAP, and one visualization metric based on the best-so-far objective trajectory. All scalar metrics are computed from the same exported trajectories described above.

We emphasize that GAP and PA-GAP are auxiliary normalized efficiency summaries that complement, rather than replace, standard BO metrics. Best-so-far trajectories are simple-regret-equivalent (see the trajectory-

visualization paragraph below), and normalized average-reward trajectories, the analogue of cumulative-regret reporting, are provided in Appendix B.4. The reason for additionally reporting GAP and PA-GAP is aggregation: raw simple regret is not comparable across datasets with different objective scales, initializations, and optimization directions, whereas GAP and PA-GAP normalize improvement relative to the initial value and reference optimum, enabling the cross-dataset rank summaries used in this benchmark. Readers interested only in per-dataset behavior can rely on the regret-equivalent trajectory plots.

**GAP.** GAP was introduced in DCBO to quantify the efficiency of optimization by combining the final improvement with the speed with which the best value is discovered (Aglietti et al., 2021). Let $R_t$ denote the best improvement ratio so far in the trial $t$, computed in the natural optimization direction of the task. For maximization tasks,

$$R_t = \min\left\{ \frac{y(\mathbf{x}_t^*) - y(\mathbf{x}_{\text{init}})}{y^* - y(\mathbf{x}_{\text{init}})}, 1 \right\}, \tag{21}$$

and for minimization tasks,

$$R_t = \min\left\{ \frac{y(\mathbf{x}_{\text{init}}) - y(\mathbf{x}_t^*)}{y(\mathbf{x}_{\text{init}}) - y^*}, 1 \right\}. \tag{22}$$

Here, $\mathbf{x}_t^*$ denotes the best point so far in the trial $t$. The clipping prevents numerical artifacts when an observed value exceeds the offline reference optimum. Let $R_T$ be the final improvement ratio and let $t^*$ be the first trial in which $R_T$ is achieved. If no improvement occurs, we set $t^* = T$. GAP is then

$$\text{GAP} = \left[ \underbrace{R_T}_{\text{Final improvement}} + \underbrace{\frac{T - t^*}{T}}_{\text{Discovery efficiency}} \right] \bigg/ \underbrace{\left( 1 + \frac{T-1}{T} \right)}_{\text{Normalization}}. \tag{23}$$

Under this convention, GAP lies in $[0, 1]$, with larger values indicating larger final improvement and earlier discovery.

**Path-Aware GAP (PA-GAP).** GAP depends only on the final best value and the first trial at which it is reached. This can under-value runs that improve late or make steady progress over the full budget. To better reflect the full optimization trajectory, we use Path-Aware GAP:

$$\text{PA-GAP} = \frac{1}{T} \sum_{t=1}^{T} \left( \underbrace{R_t}_{\text{Improvement ratio}} \cdot \underbrace{\frac{T - (t-1)}{T}}_{\text{Efficiency weight}} \right). \tag{24}$$

PA-GAP multiplies the best-so-far improvement by a time-dependent efficiency weight, so later improvements are down-weighted but not discarded, and averages these weighted improvements over the full trajectory. Since $R_t \in [0, 1]$, raw PA-GAP lies in

$$\left[ 0, \frac{T+1}{2T} \right].$$

The upper bound is attained when the reference optimum is found in the first trial and remains the best value so far thereafter. Thus, PA-GAP is a raw trajectory-weighted efficiency score rather than a unit-normalized metric. Larger values indicate faster and stronger trajectory-level improvement, and raw PA-GAP values should be compared directly only under the same trial budget. Appendix A.1.2 gives a counterexample where GAP ranks a worse final trajectory above a better late-improving one, while PA-GAP resolves the ordering.

**Trajectory visualization.** In addition to scalar metrics, we plot best-so-far objective trajectories over trials. For a fixed dataset and optimization direction, the best-so-far trajectory is equivalent to a simple-regret trajectory up to an additive constant and, for maximization tasks, a sign reversal. We therefore use best-so-far plots as simple-regret-equivalent visualizations in the original objective scale. We also report normalized average-reward trajectories in Appendix B.4 as a diagnostic complement to GAP and PA-GAP.

### 4.3 Datasets and task conventions

Table 3 summarizes the datasets used in the CBO literature. Most evaluations rely on SCM-generated data, where observational samples are drawn from the unintervened SCM and interventional samples are generated by applying hard or soft interventions to the SCM mechanisms. The optimization direction is data-dependent. For hard-intervention SCM benchmarks, we follow the standard CGO/CBO convention and treat the target as a minimization objective unless explicitly stated otherwise. ToyGraph, Synthetic, Synthetic-2, Chain-hard, Healthcare, Protein-reconstructed, and Epidemiology are minimization tasks, while Ecology is a maximization task. For the soft-intervention benchmarks for the function-network, Ackley, Rosenbrock, Dropwave, and Alpine2 are treated as reward-maximization tasks following the convention used in the BO, MCBO, and ACBO function-network. Chain-soft is a minimization task because the objective is to reduce the target variable $Y$.

#### 4.3.1 Hard-intervention SCM benchmarks

Hard-intervention benchmarks evaluate optimization when intervened variables are clamped to fixed values. They include small synthetic SCMs, a reconstructed protein-signaling SCM, and real-world-inspired SCMs from ecology, healthcare, and epidemiology. The benchmark also provides optional high-dimensional generators, but the main cross-method comparison focuses on the eight curated scenarios for which multiple implementations can be evaluated under the same protocol.

**ToyGraph.** ToyGraph was introduced in the original CBO work (Aglietti et al., 2020). It is a minimal three-node chain-structured SCM with variables $X \to Z \to Y$, where $X$ and $Z$ are manipulable and $Y$ is the target. The task is to minimize $Y$. The benchmark uses a simple nonlinear mechanism in which $Z$ depends exponentially on $X$ and $Y$ is generated from a nonlinear transformation of $Z$. Its small size makes it a useful sanity-check benchmark for hard-intervention methods. Detailed equations and visualization are provided in the Appendix A.5.1.

**Synthetic.** Synthetic was also introduced in the original CBO work (Aglietti et al., 2020). It contains seven observed variables, $A, B, C, D, E, F, Y$ and two latent variables, $U_1$ and $U_2$. The variables $B$, $D$, and $E$ are manipulable, while $Y$ is the target. The task is to minimize $Y$. The SCM combines nonlinear dependencies, latent variables, and multiple intervention options, making it a richer testbed than ToyGraph for structured causal optimization. Detailed equations and visualization are provided in the Appendix A.5.2.

**Synthetic-2.** Synthetic-2 was introduced into the cCBO line of work (Aglietti et al., 2023). It is a lightweight three-node SCM with variables $X \to Z \to Y$, where $X$ and $Z$ are manipulable and $Y$ is the target. The task is to minimize $Y$. It follows a nonlinear chain structure similar to ToyGraph but uses explicitly Gaussian exogenous noise and is generated directly from the specified SCM. Detailed equations and visualization are provided in the Appendix A.5.3.

**Chain-hard.** Chain-hard is the hard-intervention version of the Chain benchmark introduced in fCBO (Gultchin et al., 2023). It consists of four observed variables $X, W, Z, Y$, where $W$ and $Z$ are manipulable, $X$ is a non-manipulable context variable and $Y$ is the target. The task is to minimize $Y$. In the setting of hard-interventions, interventions in $W$ and $Z$ replace the corresponding structural equations with fixed constants in $[-1, 1]$. Because the result depends on the interaction between $Z$ and $X$, this benchmark is more expressive than the simple chain benchmarks while remaining low-dimensional. Detailed equations and visualization are provided in the Appendix A.5.4.

**Ecology.** The Ecology benchmark was introduced in the CBO (Aglietti et al., 2020) and is based on a Bermuda reef ecosystem model (Courtney et al., 2017). The task is to maximize net ecosystem calcification under environmental interventions. The SCM includes variables such as chlorophyll on the surface of the sea $a$, salinity, total alkalinity, dissolved inorganic carbon, seawater $p\mathrm{CO}_2$, bottom temperature, bottom light, availability of nutrients, seawater pH, saturation state of aragonite, and the target variable NEC. We follow the CBO setup and use the available SCM and the data released in the public CBO repository. Detailed equations and visualization are provided in the Appendix A.6.1.

**Protein-reconstructed.** The Protein benchmark is based on the protein-signaling data of Sachs et al. (2005) and the graph used in cCBO (Aglietti et al., 2023). The target is the extracellular signal-regulated kinases 1 and 2, Erk, and the task is to minimize Erk by perturbing Mek, PKC, PKA. and Akt while respecting biological constraints on PKC and PKA. Because the fitted SCM used in the original cCBO Protein experiments is not released, our benchmark is a reconstruction rather than an exact reproduction. We retain the connected subgraph used in cCBO, consisting of Raf, Erk, P38, Jnk, Akt, Mek, PKA, and PKC, and fit structural mechanisms using 852 observational samples from Sachs et al. (2005). We label this data set as Protein-reconstructed in the benchmark files to make it clear that it tests transparent SCM reconstruction rather than bitwise replication of cCBO. Appendix A.6.2 reports the retained variables, parent sets, preprocessing, functional model class, fitting procedure, constraints, intervention domains, equations, and visualization.

**Healthcare.** Healthcare was introduced in CBO (Aglietti et al., 2020). It is based on a causal graph over age, BMI, cancer status, statin use, aspirin use, and PSA level (Thompson & Yung, 2019; Ferro et al., 2015). The objective is to minimize PSA through interventions with statin and aspirin. Although the original data template specifies several variables as binary, existing CBO implementations commonly relax aspirin and statin to continuous variables in $(0, 1)$ for BO-style optimization. We follow this convention to remain comparable to the prior work. Detailed equations and visualization are provided in the Appendix A.6.3.

**Epidemiology.** Epidemiology was introduced in CEO (Branchini et al., 2023). The task is to minimize the viral load of HIV in a modified epidemiology SCM based on Havercroft & Didelez (2012). In the code-authoritative benchmark configuration, interventions are applied to $L$ and $B$, with $L \in [0.479, 801.787]$ and $B \in [-0.994, 0.999]$; $T$ and $R$ are non-intervention SCM variables. Since CEO uses a modified HIV SCM, more specific clinical interpretations of $B$ and $L$ are not provided in the original article. We follow CEO and use the given SCM to generate both observational and interventional datasets. Detailed equations and visualization are provided in the Appendix A.6.4.

### 4.3.2 Soft-intervention function-network benchmarks

Soft-intervention benchmarks evaluate settings where interventions modify structural mechanisms rather than simply clamping variables to fixed values. Ackley, Rosenbrock, Dropwave, and Alpine2 are classical global-optimization functions represented as function-network BO benchmarks introduced by Astudillo & Frazier (2021) and adopted in MCBO and ACBO (Sussex et al., 2023; 2024). We preserve the inherited task domains after any internal normalization used by wrappers: Ackley and Rosenbrock use $[-2, 2]$, Dropwave uses $[-5.12, 5.12]$, and Alpine2 uses $[0, 10]$. Thus, the optimizer may operate on normalized coordinates internally, but the benchmark task domains are not changed. In these tasks, the output node is treated as a reward to be maximized, even when this corresponds to a sign-flipped version of a standard minimization benchmark. Chain-soft is instead a minimization task because the objective is to reduce the target variable $Y$.

We explicitly distinguish these *computational-graph benchmarks* from *SCM benchmarks*. In the function networks, the manipulable input variables are mutually independent, orthogonal search dimensions: there are no causal relationships among the inputs themselves, so an intervention on one input cannot sever incoming edges of, or cascade into, another manipulable variable. These tasks therefore probe optimization over a known computational composition, but they do not test confounding or cascading effects among manipulable variables, which are core challenges of causal optimization tested by SCM benchmarks such as Healthcare or Ecology. Chain-soft is the only soft-intervention SCM task in the current repository, and results on the two families are labeled and interpreted separately (Table 5).

**Ackley.** Ackley is a continuous, differentiable, non-separable, scalable, and multimodal benchmark. In its function-network form, the task is to maximize the output node. The network decomposes the objective into two intermediate components: one for the average squared magnitude of the inputs and one for the average cosine term. These components are combined at the output node to recover the Ackley objective in reward-maximization form. We use the domain $[-2, 2]^D$ with $D = 6$. Detailed equations and visualization are provided in the Appendix A.7.1.

**Rosenbrock.** Rosenbrock is a continuous, differentiable, non-separable, scalable, and unimodal benchmark known for its narrow curved valley. In the function-network formulation, the task is to maximize the output node. The network is chain-structured: the first node computes the first Rosenbrock term, and each subsequent node recursively adds the contribution of an adjacent variable pair to the accumulated output. We use the domain $[-2, 2]^D$ and consider $D \in \{3, 5, 7\}$. Detailed equations and visualization are provided in the Appendix A.7.2.

**Dropwave.** Dropwave is a highly multimodal two-dimensional benchmark. In function-network form, the task is to maximize the output node. An intermediate node first computes the radial quantity $\sqrt{x_1^2 + x_2^2}$, and then the output node applies the oscillatory Dropwave transformation. We use the standard domain $[-5.12, 5.12]^2$. Detailed equations and visualization are provided in the Appendix A.7.3.

**Alpine2.** Alpine2 is a continuous, differentiable, separable, scalable, and multimodal benchmark. In function-network form, the task is to maximize the output node. It is represented as a chain graph in which each node combines its own decision variable with the previous node through the multiplicative structure of Alpine2. We use the domain $[0, 10]^K$ with $K = 6$. Detailed equations and visualization are provided in the Appendix A.7.4.

**Chain-soft.** Chain-soft extends Chain-hard by allowing the mechanism of $Z$ to be modified through a context-dependent policy, rather than restricting interventions to fixed clamped values (Gultchin et al., 2023). The task is to minimize $Y$. In this setting, $Z$ can depend on the upstream context $X$ through a policy $\pi_0(X)$, while $W$ remains subject to standard hard interventions on $[-1, 1]$. The target depends on the interaction between $Z$ and $X$, so the optimal intervention in $Z$ generally varies with $X$. This makes Chain-soft a richer benchmark for context-adaptive intervention strategies. Detailed equations, policy specification, and visualization are provided in the Appendix A.7.5.

### 4.4 Hard-intervention benchmark

To contextualize the benchmark results, Table 5 defines the formulation labels used throughout this section. These labels clarify which methods are solving matched optimization problems and which are included as stress tests or descriptive comparisons.

**Compared methods.** We compare a non-causal baseline, BO, with seven CBO-family methods: CBO, cCBO, DCBO, MCBO, HCBO, CoCaBO, and CEO (Aglietti et al., 2020; 2023; 2021; Sussex et al., 2023; Wu et al., 2024; Arsenyan et al., 2026; Branchini et al., 2023). We label comparisons by how closely the benchmark matches the method's intended setting. CBO and CoCaBO are matched known-graph hard-intervention methods for the static SCM tasks. cCBO is included as a controlled constrained-CBO variant; on unconstrained tasks it should be interpreted as a modeling comparison rather than a pure constraint comparison. In particular, cCBO does not reduce to CBO when constraints are inactive: the released cCBO pipeline uses a different surrogate construction and exploration-set handling than the released CBO pipeline, so cCBO-versus-CBO differences on unconstrained tasks reflect these modeling and implementation differences rather than an effect of constraint handling. DCBO is designed for time-evolving SCMs, so its use on static hard-intervention tasks is a transfer/stress-test comparison. MCBO is a mechanism-level method and is most directly matched to function-network/mechanism-modeling settings. CEO is an unknown-graph reference method and spends part of its budget resolving structure uncertainty, whereas known-graph methods receive the benchmark graph. BO is a graph-free GP-EI baseline and is used as a robustness reference rather than a causal competitor. We exclude fCBO, CNEI, and GACBO because we did not obtain compatible public implementations of the benchmark protocol. We exclude ACBO and MO-CBO from this hard-intervention benchmark because their released implementations or task settings are not aligned with the single-objective hard-intervention protocol considered here.

**Selected datasets.** We evaluated eight hard-intervention datasets: ToyGraph, Synthetic, Synthetic-2, Chain-hard, Ecology, Protein-reconstructed, Healthcare, and Epidemiology. These datasets cover small syn-

Table 5: Formulation labels used to contextualize benchmark results. Several labels are not independently rankable because they contain only one method or one dataset; therefore the statistical rank analysis is reported by budget rather than as separate formulation-track leaderboards.

| Label | What it identifies | How it is used in the revision |
|---|---|---|
| Unconstrained hard SCM | Static hard-intervention SCM tasks without explicit feasibility constraints | Descriptive rank summaries by budget; comparisons are qualified because some methods are stress-test or graph-uncertain variants |
| Constrained hard SCM | Tasks where cCBO-style feasibility constraints are part of the intended formulation | Descriptive only: no fully matched constraint-adapted baseline is available across all datasets |
| Contextual / policy | Settings where scope or intervention choice is policy/context dependent | Descriptive only: CoCaBO is the only currently integrated method in this formulation |
| Soft computational graph | Ackley, Rosenbrock, Dropwave, and Alpine2 function-network tasks | Descriptive rank summaries by budget; explicitly not treated as SCM benchmarks with confounding or cascading manipulable variables |
| Soft SCM / policy | Chain-soft | Descriptive only: single soft-SCM dataset in the current repository |

thetic SCMs, real or fitted-SCM benchmarks, and settings that are unevenly shared across prior CBO papers. Protein-reconstructed is included as a transparent reconstruction of the cCBO Protein benchmark, rather than as an exact reproduction of the unreleased fitted SCM used in the original cCBO experiments. Ecology is treated as a maximization task; the remaining hard-intervention datasets are treated as minimization tasks.

**Results summary.** The hard-intervention benchmark shows that no method dominates uniformly across datasets, budgets, and metrics within the benchmark-specified regime. In the main tables, the environment SCM, intervention domains, and reference optima are fixed by the benchmark, and graph-using methods are evaluated with the benchmark graph unless they are explicitly unknown-graph methods. Under this protocol, the descriptive rankings vary across datasets, budgets, and metrics, and the revised rank-reliability analysis in Appendix B.3 does not identify any statistically separable uniformly best method. On the published GAP, $T = 100$ hard-intervention table, CBO has the best point-estimate average rank, but the Nemenyi critical difference exceeds the full rank spread. We therefore interpret CoCaBO's strong cells as conditional patterns, mainly on datasets where mixed scope selection and contextualized search align with the benchmark structure, rather than as evidence that CoCaBO is intrinsically strongest across CBO settings. GAP emphasizes the final best value and the time at which that final value is first found, so methods that quickly reach a strong final solution are rewarded. In this regard, CoCaBO performs especially well in Synthetics, Healthcare, and Epidemiology, DCBO performs well in ToyGraph, and MCBO performs well in Protein-reconstructed. In contrast, PA-GAP rewards the entire best-so-far trajectory, so methods that make early or steady progress can rank well even when they are not the best final performer. This explains why CBO has a strong aggregate rank despite not being the dominant GAP winner, and why BO remains competitive: on datasets such as Chain-hard, Ecology, and Healthcare, it often finds useful interventions early enough to receive favorable trajectory-level scores. These results show that causal structure can improve optimization, but its benefit depends on how the method's assumptions match the dataset and on whether performance is measured by final discovery or by the full optimization path.

**Trajectory-level behavior.** Figures 2 and 3 complement the scalar table by showing how the methods reach their final scores. In synthetic datasets, many methods improve through a few large drops followed by long plateaus, suggesting that much of the gain comes from identifying a useful intervention scope rather than from gradual local refinement. This also explains why GAP and PA-GAP can disagree. A method that reaches a moderate value early can receive a high GAP if that value is its final best, while PA-GAP

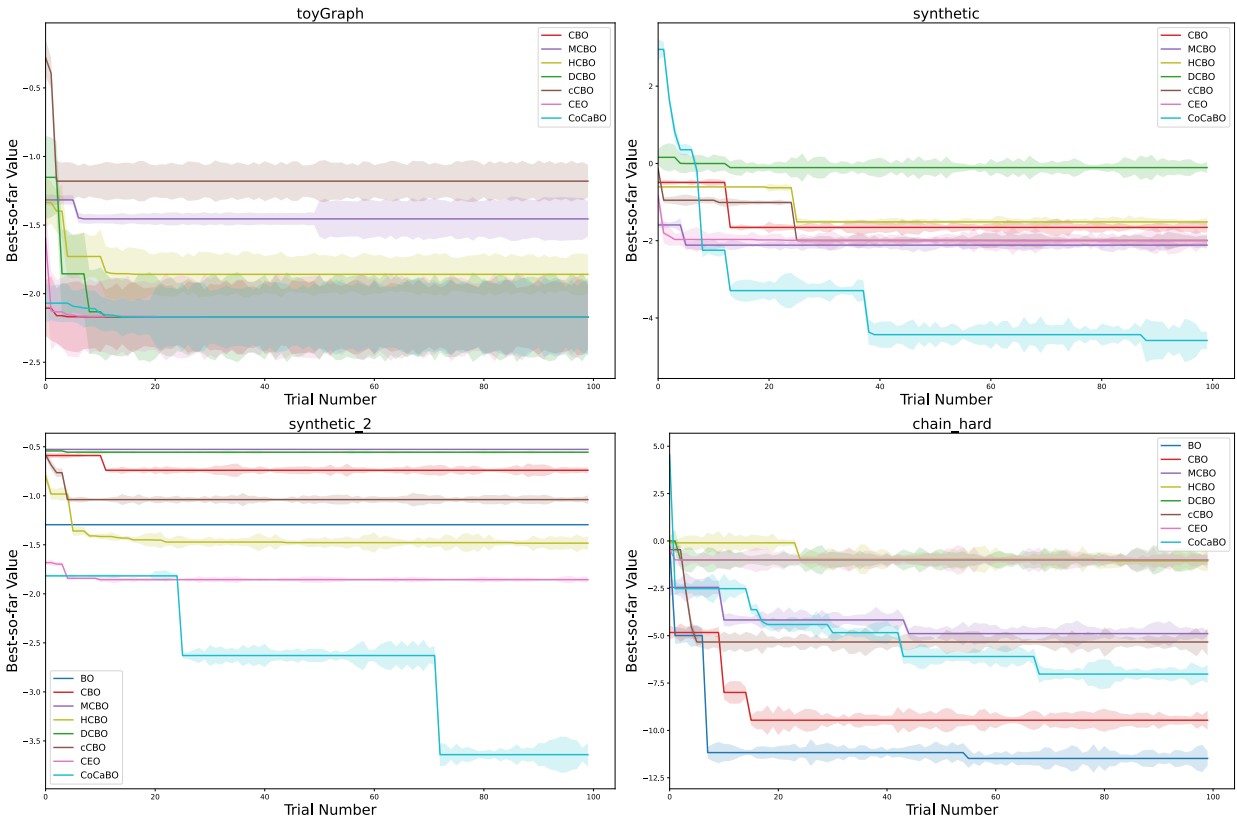

Figure 2: Best-so-far trajectories on four synthetic hard-intervention datasets: ToyGraph, Synthetic, Synthetic-2, and Chain-hard. Solid lines show the mean across 20 seeds and shaded regions show the standard deviation. Lower values indicate better objective values for these minimization tasks.

gives more credit to trajectories that continue moving toward the reference optimum. In Synthetic, CoCaBO shows strong path-level progress, consistent with its high scores under both metrics. On ToyGraph, DCBO is strong under GAP, while PA-GAP shows that early progress with CBO and HCBO-style can also matter. In Chain-hard, BO is highly competitive, indicating that a flexible black-box optimizer can perform well when the effective intervention landscape is simple or when the causal prior does not provide a decisive advantage.

The real or fitted-SCM datasets show even stronger metric dependence. Some methods remain nearly flat on particular datasets, which explains the zero or near-zero metric values in Table 6. A zero or near-zero GAP or PA-GAP value means that the method did not improve over initialization under the shared scoring protocol; it does not mean that the run was silently removed. Genuine failures such as missing outputs or non-finite trajectories are logged separately under the failure-handling protocol. In principle, a zero score could arise from (a) a poorly calibrated observational prior, (b) a loose or unreliable reference optimum, or (c) graph-induced scope selection that excludes useful interventions. In the current logs, the dominant cause is failure to improve over the initial design; in a smaller number of cases, the zero score reflects method–task incompatibility, unsupported or empty candidate scopes (cause c), or poor calibration of the observational prior (cause a). We found no case attributable to an unreliable reference optimum (cause b): the protocol in Appendix A.3 computes references offline on the ground-truth SCM, and reference error would rescale all methods' scores on a dataset rather than zero out a single method. Two named cases illustrate the distinction. cCBO on Protein-reconstructed completes all 20 seeds and returns valid, finite trajectories, but its best-so-far value remains flat over the entire budget under the reconstructed SCM and its wide Sachs-derived intervention domains, so GAP and PA-GAP are zero by definition; the runs do not crash, and the discrepancy with the original cCBO Protein results is consistent with the fact that our task is a transparent

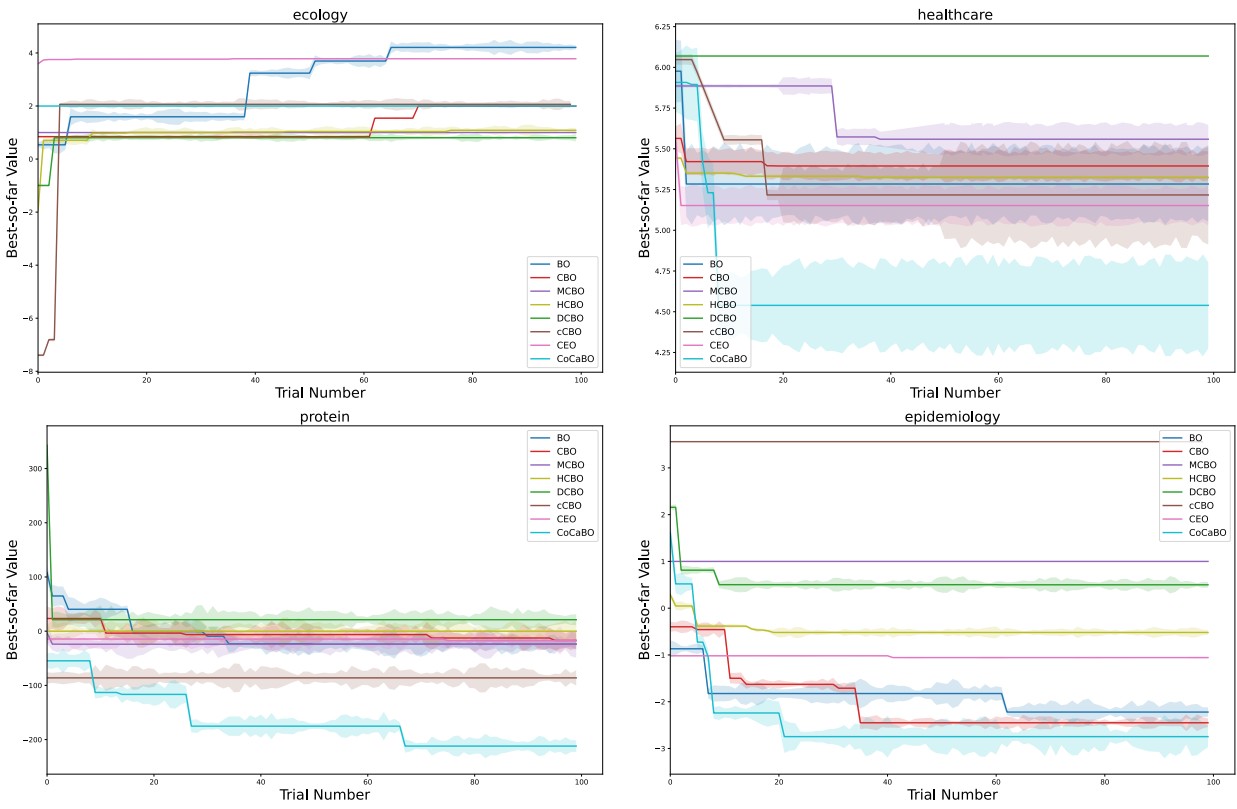

Figure 3: Best-so-far trajectories on four real or fitted-SCM hard-intervention datasets: Ecology, Protein-reconstructed, Healthcare, and Epidemiology. Solid lines show the mean across 20 seeds and shaded regions show the standard deviation. Ecology is a maximization task, while Protein-reconstructed, Healthcare, and Epidemiology are minimization tasks.

reconstruction rather than the unreleased fitted SCM used in the original experiments (Appendix A.6.2). MCBO's zero entries at $T = 20$ and $T = 50$ on several datasets reflect delayed improvement: the same runs reach non-zero scores at $T = 100$, consistent with a method that requires many trials before improving on the initial design. Ecology and Healthcare also show that strong performance can arise through different paths: some methods improve early and then plateau, while others discover a better intervention only after many trials. Reporting both GAP and PA-GAP is therefore important. GAP captures the final improvement and discovery time, while PA-GAP captures whether the optimizer makes meaningful progress throughout the budget.

**Aggregate ranking across settings.** Figure 4.4 provides a descriptive rank summary of the hard-intervention benchmark, shown separately for each trial budget $T \in \{100, 50, 20\}$. We retain this aggregate view because it is useful for visualizing broad benchmark-level patterns, but it should not be read as a homogeneous leaderboard across incompatible formulations: cCBO solves a constrained formulation, CEO spends budget on graph uncertainty, DCBO is evaluated here as a static-task stress test, and BO is a graph-free reference baseline. Within each budget a method is ranked inside every (dataset, metric) setting and its ranks are averaged over the 16 settings (eight datasets × two metrics, GAP and PA-GAP); CoCaBO and cCBO sit near the top and HCBO and MCBO at the bottom at all three budgets, while the middle of the ordering shifts only mildly with the horizon. Appendix B.3 reports, per budget, bootstrap confidence intervals over settings, Friedman/Nemenyi tests, and a conservative per-dataset test: no pair of methods is statistically separable by average rank at any budget—the rank spread stays below the Nemenyi critical difference throughout and the conservative per-dataset Friedman test never rejects, and even where the om-

Table 6: Hard-intervention performance across different trial limits and datasets. Each value is the average ± standard error across 20 random seeds and different initializations of the observational and interventional data. Higher is better for both GAP and PA-GAP. Best mean values in each row are in **bold**.

| Trial limit | Metric | Dataset | BO | CBO | cCBO | DCBO | MCBO | HCBO | CoCaBO | CEO |
|---|---|---|---|---|---|---|---|---|---|---|
| 100 | GAP | ToyGraph | 0.346±.046 | 0.810±.044 | 0.730±.023 | **0.949±.048** | 0.547±.032 | 0.727±.030 | 0.625±.054 | 0.654±.040 |
| | | Synthetic | 0.401±.038 | 0.658±.023 | 0.691±.028 | 0.482±.007 | 0.652±.030 | 0.559±.024 | **0.757±.049** | 0.565±.038 |
| | | Synthetic-2 | 0.000±.000 | 0.544±.021 | 0.549±.017 | 0.485±.003 | 0.000±.000 | 0.174±.016 | **0.562±.045** | 0.378±.038 |
| | | Chain-hard | 0.721±.032 | 0.688±.014 | **0.734±.025** | 0.542±.027 | 0.502±.046 | 0.135±.024 | 0.205±.018 | 0.502±.034 |
| | | Ecology | **0.799±.041** | 0.511±.017 | 0.697±.027 | 0.637±.026 | 0.000±.000 | 0.334±.030 | 0.000±.000 | 0.336±.032 |
| | | Protein-reconstructed | 0.751±.030 | 0.449±.043 | 0.000±.000 | 0.000±.000 | **0.878±.064** | 0.412±.032 | 0.664±.052 | 0.275±.021 |
| | | Healthcare | 0.310±.020 | 0.577±.024 | 0.842±.030 | 0.487±.001 | 0.510±.023 | 0.335±.007 | **0.964±.044** | 0.696±.066 |
| | | Epidemiology | 0.634±.049 | 0.844±.029 | 0.000±.000 | 0.375±.028 | 0.000±.000 | 0.557±.024 | **0.898±.059** | 0.308±.021 |
| | PA-GAP | ToyGraph | 0.224±.031 | 0.069±.009 | 0.235±.030 | **0.337±.043** | 0.071±.010 | 0.285±.036 | 0.087±.012 | 0.295±.037 |
| | | Synthetic | 0.198±.035 | 0.172±.028 | 0.184±.027 | 0.036±.004 | 0.162±.023 | 0.107±.020 | **0.411±.003** | 0.358±.045 |
| | | Synthetic-2 | 0.000±.000 | 0.055±.008 | 0.065±.008 | 0.002±.000 | 0.000±.000 | 0.099±.012 | **0.129±.021** | 0.035±.005 |
| | | Chain-hard | **0.495±.002** | 0.280±.042 | 0.244±.032 | 0.049±.006 | 0.111±.015 | 0.026±.005 | 0.038±.004 | 0.002±.000 |
| | | Ecology | **0.303±.039** | 0.117±.006 | 0.204±.027 | 0.144±.019 | 0.000±.000 | 0.210±.026 | 0.000±.000 | 0.016±.002 |
| | | Protein-reconstructed | **0.407±.044** | 0.254±.036 | 0.000±.000 | 0.000±.000 | 0.045±.006 | 0.000±.000 | 0.240±.030 | 0.014±.004 |
| | | Healthcare | 0.125±.014 | 0.175±.020 | 0.170±.022 | 0.000±.000 | 0.100±.021 | 0.010±.001 | **0.329±.045** | 0.200±.025 |
| | | Epidemiology | 0.299±.038 | 0.346±.021 | 0.000±.000 | 0.175±.022 | 0.000±.000 | 0.138±.017 | **0.392±.047** | 0.005±.001 |
| 50 | GAP | ToyGraph | 0.296±.043 | 0.676±.045 | 0.723±.022 | **0.896±.041** | 0.513±.024 | 0.643±.030 | 0.629±.048 | 0.690±.036 |
| | | Synthetic | 0.531±.018 | 0.597±.024 | 0.565±.023 | 0.422±.003 | 0.630±.022 | 0.432±.019 | **0.793±.056** | 0.372±.034 |
| | | Synthetic-2 | 0.000±.000 | 0.459±.021 | **0.531±.014** | 0.467±.007 | 0.000±.000 | 0.160±.020 | 0.438±.023 | 0.213±.043 |
| | | Chain-hard | **0.919±.018** | 0.462±.009 | 0.712±.023 | 0.535±.024 | 0.215±.027 | 0.302±.016 | 0.106±.016 | 0.505±.032 |
| | | Ecology | **0.798±.037** | 0.363±.015 | 0.680±.023 | 0.624±.021 | 0.000±.000 | 0.246±.016 | 0.000±.000 | 0.151±.018 |
| | | Protein-reconstructed | 0.656±.027 | 0.553±.038 | 0.000±.000 | 0.000±.000 | **0.752±.041** | 0.320±.017 | 0.731±.041 | 0.377±.025 |
| | | Healthcare | 0.366±.019 | 0.598±.019 | 0.758±.027 | 0.474±.000 | 0.315±.019 | 0.155±.002 | **0.927±.036** | 0.694±.046 |
| | | Epidemiology | 0.750±.043 | 0.644±.024 | 0.000±.000 | 0.596±.021 | 0.000±.000 | 0.461±.018 | **0.793±.041** | 0.096±.001 |
| | PA-GAP | ToyGraph | 0.177±.029 | 0.068±.008 | 0.233±.030 | 0.316±.040 | 0.063±.010 | 0.262±.033 | **0.357±.050** | 0.296±.037 |
| | | Synthetic | 0.130±.021 | 0.131±.025 | 0.191±.017 | 0.033±.004 | 0.150±.022 | 0.050±.013 | 0.349±.040 | **0.357±.032** |
| | | Synthetic-2 | 0.000±.000 | 0.043±.008 | 0.063±.008 | 0.002±.000 | 0.000±.000 | **0.091±.011** | 0.051±.014 | 0.033±.004 |
| | | Chain-hard | **0.500±.002** | 0.178±.030 | 0.233±.031 | 0.049±.006 | 0.078±.013 | 0.012±.003 | 0.034±.004 | 0.002±.000 |
| | | Ecology | **0.306±.039** | 0.085±.013 | 0.194±.027 | 0.140±.019 | 0.000±.000 | 0.208±.025 | 0.000±.000 | 0.015±.002 |
| | | Protein-reconstructed | **0.339±.031** | 0.192±.034 | 0.000±.000 | 0.000±.000 | 0.045±.006 | 0.000±.000 | 0.157±.022 | 0.003±.001 |
| | | Healthcare | 0.120±.014 | 0.131±.018 | 0.263±.038 | 0.000±.000 | 0.034±.011 | 0.009±.001 | **0.295±.042** | 0.202±.025 |
| | | Epidemiology | 0.247±.039 | 0.222±.020 | 0.000±.000 | 0.169±.021 | 0.000±.000 | 0.127±.015 | **0.354±.041** | 0.000±.000 |
| 20 | GAP | ToyGraph | 0.383±.022 | 0.510±.044 | 0.702±.023 | **0.729±.038** | 0.406±.020 | 0.372±.022 | 0.539±.043 | 0.508±.028 |
| | | Synthetic | 0.239±.014 | 0.392±.020 | 0.367±.022 | 0.231±.002 | 0.558±.016 | 0.166±.002 | **0.756±.052** | 0.390±.028 |
| | | Synthetic-2 | 0.000±.000 | 0.285±.017 | **0.474±.018** | 0.408±.002 | 0.000±.000 | 0.186±.017 | 0.000±.000 | 0.282±.023 |
| | | Chain-hard | **0.819±.014** | 0.136±.003 | 0.641±.022 | 0.510±.018 | 0.360±.023 | 0.000±.000 | 0.065±.011 | 0.513±.028 |
| | | Ecology | **0.795±.039** | 0.208±.003 | 0.625±.020 | 0.585±.016 | 0.000±.000 | 0.271±.012 | 0.000±.000 | 0.340±.043 |
| | | Protein-reconstructed | 0.503±.022 | 0.508±.034 | 0.000±.000 | 0.000±.000 | 0.540±.036 | 0.028±.016 | **0.648±.041** | 0.168±.016 |
| | | Healthcare | 0.604±.015 | 0.392±.021 | 0.490±.019 | 0.432±.000 | 0.000±.000 | 0.172±.001 | **0.810±.017** | 0.689±.036 |
| | | Epidemiology | 0.646±.030 | 0.452±.022 | 0.000±.000 | 0.457±.017 | 0.000±.000 | 0.154±.015 | **0.791±.028** | 0.000±.000 |
| | PA-GAP | ToyGraph | 0.088±.021 | **0.392±.051** | 0.227±.029 | 0.260±.034 | 0.040±.008 | 0.200±.040 | 0.190±.029 | 0.300±.036 |
| | | Synthetic | 0.026±.008 | 0.034±.012 | 0.147±.017 | 0.024±.003 | 0.116±.021 | 0.000±.000 | 0.334±.035 | **0.361±.042** |
| | | Synthetic-2 | 0.000±.000 | 0.016±.005 | 0.057±.007 | 0.002±.000 | 0.000±.000 | **0.074±.008** | 0.000±.000 | 0.027±.004 |
| | | Chain-hard | **0.357±.035** | 0.032±.007 | 0.200±.029 | 0.047±.006 | 0.034±.009 | 0.000±.000 | 0.031±.004 | 0.002±.000 |
| | | Ecology | **0.316±.039** | 0.030±.008 | 0.166±.025 | 0.126±.019 | 0.000±.000 | 0.205±.024 | 0.000±.000 | 0.015±.002 |
| | | Protein-reconstructed | 0.250±.024 | 0.070±.020 | 0.000±.000 | **0.290±.036** | 0.047±.006 | 0.000±.000 | 0.182±.045 | 0.001±.000 |
| | | Healthcare | 0.121±.014 | 0.034±.018 | 0.143±.007 | 0.000±.000 | 0.000±.000 | 0.008±.001 | **0.268±.044** | 0.208±.025 |
| | | Epidemiology | 0.158±.033 | 0.078±.021 | 0.000±.000 | 0.152±.019 | 0.000±.000 | 0.102±.011 | **0.331±.034** | 0.000±.000 |

nibus Friedman test detects mild rank heterogeneity the Nemenyi post-hoc localizes no separable pair. The same conclusion holds on the published 20-seed GAP@100 table, where the Nemenyi critical difference of 3.71 exceeds the 2.62 rank spread.

### 4.5 Soft-intervention benchmark

For soft interventions, we compare BO, MCBO, and ACBO in Ackley, Rosenbrock, Dropwave, Alpine2 and Chain-soft (Sussex et al., 2023; 2024). BO serves as a non-causal baseline, while MCBO and ACBO represent mechanism-level and adversarial CBO-style approaches. Because ACBO was designed for adversarial or non-stationary settings, its use on these stationary function-network tasks is interpreted as a stress-test/reference result rather than a claim of intrinsic dominance in its native setting. fCBO also supports policy interventions, but it is excluded from the comparison because we did not obtain a compatible reference implementation (Gultchin et al., 2023). Ackley, Rosenbrock, Dropwave, and Alpine2 are treated as reward-maximization tasks; Chain-soft is a minimization task.

**Results summary.** The soft-intervention benchmark shows a descriptive aggregate ordering in which ACBO has the strongest average rank, followed by BO and then MCBO, but this ordering should be interpreted cautiously because the soft benchmark combines two formulations: four computational-

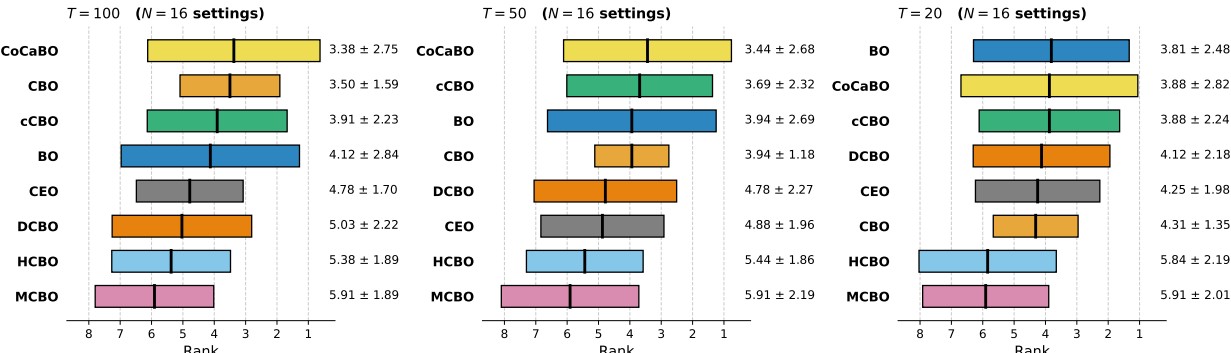

Figure 4: Descriptive average rank on the hard-intervention benchmark, shown separately for each trial budget ($T = 100, 50, 20$). Within each budget a method is ranked inside every (dataset, metric) setting and averaged over the 16 settings (eight datasets × two metrics, GAP and PA-GAP); each bar spans the mean ± one standard deviation of rank across settings and the vertical line marks the mean. Lower rank is better (rank 1, best, at the right). The spread reflects variation across settings, not statistical separation between methods; significance checks are reported in Appendix B.3.1.

graph/function-network tasks and one SCM-based policy task. ACBO is especially strong on Ackley and Chain-soft, where it obtains the best GAP across budgets and often achieves strong PA-GAP values, indicating fast and sustained progress. A plausible explanation for why an adversarially designed method performs well on these stationary tasks is that, when the environment does not change, ACBO's CBO-MW procedure effectively reduces to optimistic causal exploration: the causal UCB oracle exploits the known function-network structure much as MCBO does, while the multiplicative-weights layer over a discretized action set concentrates quickly on high-reward actions once rewards are stable, since no adversary forces re-exploration. The discretized action set may additionally act as a coarse global search grid that is well suited to highly multimodal landscapes such as Ackley, where continuous acquisition optimization can stall in local optima. We present this as a mechanism-level hypothesis consistent with the trajectories in Figure 5, not as a measured causal explanation. BO remains a highly competitive baseline on Rosenbrock and Alpine2. In these landscapes, standard black-box optimization often reaches strong final values and also receives favorable PA-GAP scores when it improves early and then maintains its best-so-far value. MCBO is most competitive on Dropwave, where it achieves the best GAP at 50 and 20 trials and the best PA-GAP at 50 and 20 trials. This pattern highlights the role of metric choice: GAP rewards the final best value and its discovery time, while PA-GAP rewards the whole best-so-far trajectory. As a result, methods such as BO can rank well when they make reliable early progress, even if they are not always the best final performer. Overall, the soft-intervention results reinforce the main conclusion from the hard-intervention benchmark: causal modeling can improve optimization, but its advantage depends on how well the method's assumptions match the intervention type, objective landscape, and evaluation metric.

**Trajectory-level behavior.** Figure 5 shows that the three methods behave differently in objective landscapes. ACBO improves quickly on Ackley and Chain-soft, which explains its strong scalar scores on these datasets. BO remains highly competitive in Rosenbrock and Alpine2, where its best-so-far trajectory often improves early enough to receive strong PA-GAP values as well as strong GAP values. Dropwave shows a complementary pattern: MCBO and ACBO both improve substantially, but MCBO is stronger under tighter budgets after applying the same PA-GAP clipping rule used throughout the benchmark. This correction caps the improvement ratio at one when a noisy observed best value exceeds the recorded expected-value reference optimum, ensuring that GAP and PA-GAP remain within their stated ranges. These trajectory

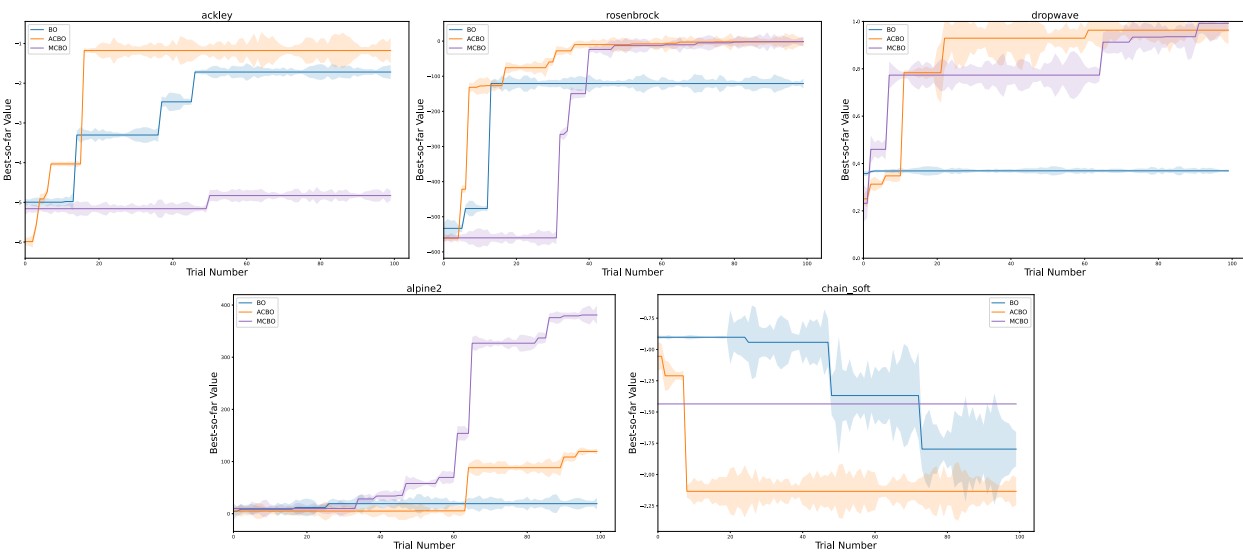

Figure 5: Best-so-far trajectories on five soft-intervention datasets: Ackley, Rosenbrock, Dropwave, Alpine2, and Chain-soft. Solid lines show the mean over 20 random seeds and shaded regions denote the standard deviation across runs.

Table 7: Soft-intervention performance across different trial limits and datasets. Each value is the average ± standard error across 20 random seeds and initializations of the observational and interventional data. Higher is better; best mean values per row are in **bold**.

| Trial limit | Metric | Dataset | BO | MCBO | ACBO |
|---|---|---|---|---|---|
| 100 | GAP | Ackley | $0.328\pm.022$ | $0.310\pm.003$ | $\mathbf{0.922\pm.082}$ |
| | | Rosenbrock | $\mathbf{0.825\pm.055}$ | $0.507\pm.041$ | $0.623\pm.047$ |
| | | Dropwave | $0.024\pm.005$ | $0.558\pm.065$ | $\mathbf{0.690\pm.094}$ |
| | | Alpine2 | $\mathbf{0.536\pm.046}$ | $0.522\pm.052$ | $0.178\pm.014$ |
| | | Chain-soft | $0.634\pm.028$ | $0.512\pm.022$ | $\mathbf{0.693\pm.052}$ |
| | PA-GAP | Ackley | $0.033\pm.004$ | $0.012\pm.003$ | $\mathbf{0.403\pm.006}$ |
| | | Rosenbrock | $0.308\pm.046$ | $0.217\pm.004$ | $\mathbf{0.412\pm.007}$ |
| | | Dropwave | $0.008\pm.001$ | $0.355\pm.041$ | $\mathbf{0.379\pm.005}$ |
| | | Alpine2 | $\mathbf{0.120\pm.014}$ | $0.076\pm.019$ | $0.015\pm.005$ |
| | | Chain-soft | $\mathbf{0.290\pm.011}$ | $0.000\pm.001$ | $0.205\pm.028$ |
| 50 | GAP | Ackley | $0.090\pm.005$ | $0.072\pm.002$ | $\mathbf{0.843\pm.071}$ |
| | | Rosenbrock | $\mathbf{0.761\pm.042}$ | $0.515\pm.049$ | $0.631\pm.052$ |
| | | Dropwave | $0.235\pm.024$ | $\mathbf{0.803\pm.085}$ | $0.318\pm.040$ |
| | | Alpine2 | $\mathbf{0.403\pm.034}$ | $0.085\pm.007$ | $0.175\pm.009$ |
| | | Chain-soft | $0.515\pm.023$ | $0.000\pm.000$ | $\mathbf{0.655\pm.033}$ |
| | PA-GAP | Ackley | $0.018\pm.003$ | $0.000\pm.000$ | $\mathbf{0.320\pm.042}$ |
| | | Rosenbrock | $0.239\pm.041$ | $0.053\pm.017$ | $\mathbf{0.344\pm.047}$ |
| | | Dropwave | $0.008\pm.001$ | $\mathbf{0.318\pm.040}$ | $0.171\pm.024$ |
| | | Alpine2 | $\mathbf{0.085\pm.008}$ | $0.003\pm.001$ | $0.000\pm.000$ |
| | | Chain-soft | $0.135\pm.038$ | $0.000\pm.000$ | $\mathbf{0.181\pm.026}$ |
| 20 | GAP | Ackley | $0.166\pm.013$ | $0.000\pm.000$ | $\mathbf{0.593\pm.049}$ |
| | | Rosenbrock | $\mathbf{0.559\pm.035}$ | $0.000\pm.000$ | $0.499\pm.047$ |
| | | Dropwave | $0.252\pm.023$ | $\mathbf{0.701\pm.067}$ | $0.293\pm.041$ |
| | | Alpine2 | $\mathbf{0.142\pm.013}$ | $0.000\pm.000$ | $0.000\pm.000$ |
| | | Chain-soft | $0.000\pm.000$ | $0.000\pm.000$ | $\mathbf{0.534\pm.024}$ |
| | PA-GAP | Ackley | $0.003\pm.001$ | $0.000\pm.000$ | $\mathbf{0.151\pm.017}$ |
| | | Rosenbrock | $0.082\pm.018$ | $0.000\pm.000$ | $\mathbf{0.216\pm.038}$ |
| | | Dropwave | $0.007\pm.001$ | $\mathbf{0.253\pm.030}$ | $0.130\pm.021$ |
| | | Alpine2 | $\mathbf{0.049\pm.006}$ | $0.000\pm.000$ | $0.000\pm.000$ |
| | | Chain-soft | $0.000\pm.000$ | $0.000\pm.000$ | $\mathbf{0.116\pm.019}$ |

patterns explain why the aggregate rank is not determined only by the number of final GAP wins. PA-GAP

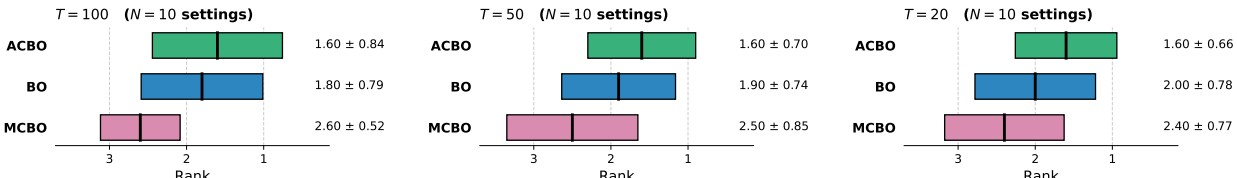

Figure 6: Descriptive average rank on the soft-intervention benchmark, shown separately for each trial budget ($T = 100, 50, 20$). Within each budget, ranks are computed inside every (dataset, metric) setting and averaged over the 10 settings (five datasets × two metrics); each bar spans the mean ± one standard deviation across settings. The five datasets combine four soft computational-graph tasks (Ackley, Rosenbrock, Dropwave, Alpine2) with one SCM-based policy task (Chain-soft), so the plot is intended as an overview rather than a definitive cross-formulation leaderboard. Lower rank is better (rank 1 at the right); the spread reflects variation across settings. Significance checks are reported in Appendix B.3.2.

gives additional credit to methods that improve early and maintain progress, which is why BO remains close to ACBO overall despite being a non-causal baseline.

**Aggregate ranking across settings.** Figure 6 summarizes performance across the soft-intervention settings as a descriptive overview, separately for each budget. ACBO has the best aggregate rank at every budget, followed by BO and then MCBO, but this should not be interpreted as a statistically established universal ordering across all soft-intervention formulations: the tests in Appendix B.3 find no separable pair of methods at any budget (here even the omnibus Friedman test does not reject). The four function-network tasks form a soft computational-graph track, while Chain-soft is a single SCM-based policy task and therefore supports only within-dataset ordering. BO obtains a strong rank because it is competitive on Rosenbrock and Alpine2 and because PA-GAP rewards its early trajectory quality in several settings. MCBO is more dataset dependent: it is strong on Dropwave, where the mechanism-level structure appears useful, but weaker on Ackley, Alpine2, and Chain-soft. As in the hard-intervention benchmark, scalar tables, trajectory plots, rank summaries, and statistical checks should be read together rather than as a single definitive leaderboard.

### 4.6 Cross-cutting analysis

The hard and soft-intervention benchmarks together span 78 dataset–budget-metric settings. Rather than summarizing each setting individually, we distill the main patterns into four cross-sectional observations.

**When does causal structure help?** Causal methods provide the clearest advantage when the graph enables effective scope reduction or when observational priors are well-calibrated. In the hard-intervention setting, CoCaBO and cCBO benefit from structured scope selection on datasets with multiple intervenable variables and clear causal pathways (Synthetic, Healthcare, Epidemiology). In the soft-intervention setting, ACBO benefits from mechanism-level modeling on datasets with complex intermediate structure (Ackley, Chain-soft). Conversely, causal structure provides little or no advantage on datasets where the effective intervention landscape is simple (Chain-hard) or where causal priors are poorly calibrated (Ecology under some methods). BO remains competitive precisely in these settings, suggesting that the value of causal modeling is conditional on the alignment between the method's structural assumptions and the problem's actual causal complexity.

**Budget sensitivity.** Rankings are moderately sensitive to budget. At tight budgets ($T = 20$), methods that rely on observational priors or scope reduction (CBO, cCBO, DCBO) tend to perform relatively better because their prior information is most valuable when interventional data are scarce. At larger budgets ($T =$

100), methods with richer exploration mechanisms (CoCaBO, ACBO) can overtake prior-driven methods by accumulating sufficient interventional evidence to overcome initial advantages from prior information.

**Metric sensitivity.**  GAP and PA-GAP can produce different rankings for the same dataset and budget. GAP rewards the final best value and its discovery time, so methods that make a single large improvement early are favored.  PA-GAP rewards steady trajectory-level progress, so methods that gradually improve throughout the budget can rank well even without the best final value. This discrepancy is most pronounced when comparing methods that plateau early (e.g., DCBO on Synthetic-2) versus methods that improve late (e.g., HCBO on Synthetic-2).  We recommend that future CBO evaluations report both metrics, as well as trajectory plots, to avoid misleading conclusions from any single scalar summary.

**Threats to validity.**  Several factors limit the generalizability of our benchmark findings. First, the benchmark is a native-wrapper reproducibility harness rather than a fully controlled reimplementation of every method; implementation details, released defaults, and dependency choices may affect performance. Second, the benchmark uses unit-cost interventions, which does not capture realistic cost heterogeneity. Third, the main hard-intervention tables assume benchmark-specified graphs and causal sufficiency for most graph-using methods. To probe this limitation rather than only state it, Appendix B.1 reports controlled learner-side graph-misspecification stress tests under edge addition, edge deletion, and edge reversal, and a companion omitted-variable stress test for hidden confounding. In all of these diagnostics, the true environment SCM and reference optimum remain fixed, while only the learner's causal information is perturbed.  Fourth, we use released default hyperparameters for all methods; per-dataset tuning could change relative rankings, as illustrated by the MCBO $\beta$ sensitivity analysis. Relatedly, the benchmark does not control for differences in identification strategy across pipelines (Table 4): the released methods differ in whether and how they construct observational priors, none exposes a general front-door or full-ID implementation in the benchmark path used here, and Monte Carlo approximation settings for identified estimands are not uniformly exposed. On datasets with latent variables—most notably Synthetic, whose confounders $U_1, U_2$ make the backdoor criterion inapplicable for some effects—performance differences therefore cannot be cleanly attributed to acquisition or decision design alone; they may partly reflect differences in identification completeness and prior-construction quality.  Fifth, GAP and PA-GAP are useful efficiency summaries but should be interpreted alongside best-so-far trajectories, simple-regret-equivalent summaries, average-reward diagnostics, and rank-reliability tests.  Finally, the set of compared methods is limited by the availability of compatible public implementations; fCBO, CNEI, GACBO, and ACBO for hard interventions are not included in the hard-intervention comparison.

## 5   Open Problems and Future Directions

The benchmark results and methodological analysis presented in this survey reveal that CBO is a maturing but incomplete field. No single method dominates across settings, and the value of causal structure depends critically on the alignment between a method's assumptions and the problem's actual causal complexity. To make the deployment gap concrete, consider using CBO to tune a treatment policy, industrial controller, or ecological intervention from observational data and a partially trusted causal graph. In such a setting, hidden confounding, graph errors, measurement noise, heterogeneous intervention costs, and safety constraints are not secondary details; any one of them can make an apparently promising intervention unreliable or harmful. In the following, we organize the most important open challenges into six concrete research directions. Each direction is stated relative to this deployment scenario: robustness to causal assumptions addresses the partially trusted graph and possible hidden confounding; scalability addresses the fact that realistic systems have far more variables and candidate scopes than current benchmarks; richer intervention models address mixed hard/soft/stochastic actions and heterogeneous intervention costs; realistic evaluation addresses the gap between benchmark-specified conditions and deployment conditions such as distribution shift; theoretical foundations address the guarantees a practitioner would need before acting on a recommended intervention; and integration with representation learning addresses settings where the causal variables themselves are not directly measured.

### 5.1 Robustness to causal assumptions

Most CBO methods assume a correctly known graph, causal sufficiency (no hidden confounders), and identifiable interventional effects. In practice, each of these assumptions can fail.

**Hidden confounding.** When unobserved confounders are present, observational priors constructed via backdoor adjustment or the *g*-formula can be systematically biased, leading the acquisition function to favor interventions that appear promising under confounded associations but fail under true causal effects. Developing CBO methods that explicitly model sensitivity to hidden confounding, for example, by propagating bounds on the causal effect under varying degrees of unmeasured confounding (Pearl, 2009), is an important open direction.

**Graph misspecification.** Even when a graph is provided, edges may be missing, spurious, or incorrectly oriented. The impact of such errors on CBO performance has been largely unexplored. Future work should characterize how graph misspecification degrades scope reduction and prior quality and develop methods that are robust to bounded graph errors.

**Prior misspecification.** CNEI (Li et al., 2023) addresses noise robustness, but the broader problem of controlling how strongly observational information influences acquisition decisions under potential misspecification remains open. Methods that adaptively downweight observational priors as interventional data accumulate, analogous to Bayesian robustness techniques in other domains, would be valuable.

### 5.2 Scalability

Current CBO methods are mainly demonstrated on graphs with tens of variables and a handful of intervention scopes. Scaling to realistic systems requires advances on multiple fronts.

**High-dimensional graphs.** HCBO (Wu et al., 2024) addresses the selection of the scope in larger graphs, but the combinatorial explosion of candidate scopes remains a fundamental bottleneck. Future methods should take advantage of causal sparsity, hierarchical decomposition, or learned embeddings of the intervention scopes to avoid exhaustive enumeration. Notably, hierarchical decomposition across levels of causal granularity is not merely a speculative direction: the causal abstraction literature has already formalized it in bandit settings. CAMAB (Zennaro et al., 2024) uses a coarse causal model to prune the intervention space of a finer model—a multi-scale analogue of POMIS-based scope reduction—and AT-UCB (Dyer et al., 2025) demonstrates substantial cumulative-regret reductions by exploiting abstraction hierarchies in complex simulators, including epidemiological models. Transferring these abstraction-based pruning and transfer mechanisms from bandit settings to GP-based CBO with continuous intervention domains is a concrete and promising path toward scalability.

**Unknown-graph scalability.** CEO (Branchini et al., 2023) and GACBO (Mukherjee et al., 2024) demonstrate unknown-graph CBO on small graphs, but maintaining a posterior or confidence set over graphs becomes intractable as the number of variables grows. Scalable alternatives might include local structure learning focused on the decision-relevant subgraph, amortized inference over graph posteriors, or graph neural network-based surrogates that learn causal structure implicitly.

**Computational cost.** Mechanism-level methods (MCBO, ACBO) require maintaining and updating multiple GPs and propagating uncertainty through the graph at each iteration. The per-iteration cost scales with the number of mechanism models and the depth of the graph. Sparse GP approximations, variational inference, or neural process surrogates could reduce this computational burden.

### 5.3 Richer intervention models

The current CBO literature mainly addresses either pure hard interventions or pure soft interventions. Real-world intervention settings are often more complex.

**Mixed intervention types.** Many practical systems involve a mix of hard interventions (e.g. turning a device on or off), soft interventions (e.g., adjusting a controller's parameters) and stochastic interventions (e.g., randomizing a treatment assignment probability). No existing CBO method handles arbitrary mixtures of these types within a single optimization loop.

**Multi-agent and sequential interventions.** ACBO (Sussex et al., 2024) considers adversarial environments, but the more general setting of multi-agent intervention, where multiple decision-makers intervene in the same system, possibly with conflicting objectives, remains unexplored. Similarly, extending CBO to sequential multi-stage intervention policies, where the intervention at each stage depends on the outcomes of previous stages, would bridge CBO and causal reinforcement learning.

**Intervention cost heterogeneity.** Our benchmark uses unit-cost interventions to isolate sample efficiency, but real interventions differ substantially in cost. The cost may depend on the target variable, the magnitude of the intervention, the safety requirements, or the logistics of implementation. Future CBO benchmarks should incorporate explicit cost models and future methods should optimize cost-adjusted efficiency rather than pure sample efficiency.

## 5.4 Realistic evaluation

Our benchmark study reveals that the evaluation design significantly affects conclusions about the quality of the method. Several aspects of the CBO evaluation require further development.

**Standardized benchmarks.** The fragmentation of the data set documented in Table 3 makes the comparison between papers unreliable. The community would benefit from a shared and versioned benchmark suite with standardized SCMs, intervention domains, observational datasets, and evaluation code, analogous to established benchmarks in reinforcement learning or supervised learning. In this spirit, the benchmark released with this survey should be read as a unified repository of existing environments together with an execution and scoring harness, not as a definitive benchmark suite. Establishing the latter would additionally require high-dimensional, complex, nonlinear simulators with rich causal structure among manipulable variables (e.g., realistic epidemiological or climate models) to stress-test scalability, combinatorial scope reduction, and robustness; building such simulators is an important direction for future work.

**Evaluation under distribution shift.** Most benchmarks assume that the test-time SCM matches the training-time SCM. In practice, the system can change between the observational data collection phase and the intervention deployment phase. Evaluating CBO under a controlled distribution shift would test the robustness of observational priors and structure assumptions.

**Trajectory-aware metrics.** Our analysis shows that GAP and PA-GAP can produce different rankings, highlighting the need for trajectory-aware evaluation. Future work should develop principled metrics that capture the full optimization trajectory, potentially incorporating cost, risk, and constraint satisfaction alongside objective improvement.

## 5.5 Theoretical foundations

The theoretical understanding of CBO lags behind its empirical development. Several foundational questions remain open.

**Finite-sample regret bounds.** MCBO (Sussex et al., 2023) provides regret bounds under mechanism-level GP models, and ACBO (Sussex et al., 2024) extends these to adversarial settings. However, regret bounds for effect-level CBO with observational priors, for unknown-graph CBO, and for constrained CBO are largely missing. Developing such bounds would clarify when causal structure provably improves over black-box BO and under what conditions.

**Identifiability-aware acquisition.** Current methods either assume full identifiability or ignore identifiability entirely. An intermediate approach would design acquisition functions that explicitly account for the degree of identifiability: using observational information aggressively when identification is strong and falling back to interventional exploration when it is weak. This would connect CBO to the literature on partial identification and sensitivity analysis in causal inference.

**Sample complexity of scope reduction.** The POMIS-based scope reduction in CBO eliminates dominated scopes using observational evidence, but the sample complexity of this elimination, i.e., how many observational samples are needed to reliably identify the correct POMIS, has not been analyzed. Such an analysis would clarify when scope reduction is reliable and when it may prematurely discard useful scopes.

### 5.6 Integration with modern representation learning

Recent advances in foundation models and causal representation learning suggest new directions for CBO.

**Causal representation learning.** Methods for learning causal variables and structure from high-dimensional observations (Schölkopf et al., 2021) could enable CBO in settings where the causal graph operates over latent variables rather than directly observed quantities. This would extend CBO to image- or text-based intervention settings. A complementary and arguably more direct route is causal abstraction learning, which operates at the level of causal graphs and mechanisms rather than raw observations: abstraction maps relate a fine-grained causal model to a coarser one whose variables are aggregates or latent summaries of the fine-grained variables. CBO over such abstracted variables is exactly what CAMAB-style multi-level decision making (Zennaro et al., 2024; Dyer et al., 2025) and the (not yet peer-reviewed) MFACBO proposal (Zeitler, 2025) begin to provide, and integrating learned abstraction maps into the CBO loop is a concrete instance of the latent-variable optimization advocated here.

**Large language models for causal reasoning.** Large language models (LLMs) have shown preliminary ability to reason about causal relationships from text descriptions. An intriguing direction is to use LLMs to elicit or refine causal graphs from domain experts, providing CBO with better structural priors without requiring formal causal discovery from data.

**Neural surrogates.** Replacing GP-based surrogates with neural network-based models (e.g., neural processes, transformers) could improve scalability and flexibility, particularly for high-dimensional mechanism functions or non-stationary environments. The challenge is maintaining the well-calibrated uncertainty quantification that makes GP-based CBO effective.

## 6 Conclusion

This survey has provided a comprehensive review of Causal Bayesian Optimization, organizing the growing literature through a unified design-space perspective that separates graph and system-knowledge assumptions, environmental assumptions, intervention representation, surrogate architecture, and decision rules. By analyzing each major CBO variant through this lens, we have shown that the field's diversity reflects systematic responses to specific bottlenecks of the original CBO framework (temporal dynamics, safety constraints, high dimensionality, unknown graphs, soft interventions, context dependence, and adversarial non-stationarity) rather than ad hoc extensions.

Our benchmark study reinforces a central finding: the value of causal structure in optimization is conditional. Causal methods provide clear advantages when their structural assumptions align with the problem, when scope reduction eliminates irrelevant interventions, when observational priors are well-calibrated, or when mechanism-level modeling captures the system's compositional structure. However, when these conditions are not met, strong non-causal BO baselines remain competitive, and method rankings depend substantially on the choice of dataset, budget, and evaluation metric. The introduction of PA-GAP as a trajectory-aware complement to GAP highlights that even the definition of "good performance" in CBO is not straightforward: final-value metrics and trajectory-level metrics can favor different methods.

Looking ahead, the most pressing challenges are not purely algorithmic. Our benchmark does not establish that CBO methods are robust in high-stakes deployment settings; rather, it shows that causal methods can be useful under benchmark-specified assumptions and that performance can change substantially when learner-side causal assumptions are perturbed. Robustness to graph misspecification, hidden confounding, imperfect interventions, measurement error, heterogeneous intervention costs, and safety constraints remains a prerequisite for CBO to move from controlled benchmarks to real-world deployment. We hope that the unified perspective, benchmark, stress-test protocols, and structured open problems presented in this survey will help researchers navigate the CBO landscape and focus their efforts on directions with practical impact.

**Broader impact and deployment caution.** CBO is increasingly positioned for deployment in sensitive domains such as healthcare dosing, resource allocation, and public policy. Deploying CBO with misspecified graphs, hidden confounding, or miscalibrated causal priors can select harmful physical interventions in real-world systems. A practitioner reading this survey should not conclude that CBO methods offer reliable advantages in such settings; the evidence applies only to the correctly specified, benchmark-controlled regime. Causal assumptions and failure logging should be treated as part of deployment risk, not merely as technical details. The graph-misspecification and omitted-variable stress tests in Appendix B.1 are a first step toward the robustness evaluation needed for deployment, but comprehensive robustness to unobserved confounding, imperfect interventions, measurement error, heterogeneous costs, and safety constraints remains an open evaluation dimension.

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

## Table of Contents

# A Technical Appendix

## A.1 Metrics analysis

DCBO introduced the GAP metric as a scalar that combines how much an optimizer improves on its initial value with how early it reaches its best value (Aglietti et al., 2021). For a minimization problem, let $y(\mathbf{x}) = \mathbb{E}[Y \mid do(X = \mathbf{x})]$, let $y^*$ denote the reference optimum (see Appendix A.3), let $\mathbf{x}_{\text{init}}$ be the initial best point, and let $\mathbf{x}_t^*$ be the best point found up to trial $t$. Let $T$ be the total number of trials, and let $t^*$ be the first trial at which the final best value $y(\mathbf{x}_T^*)$ is reached. GAP is

$$
\text{GAP} = \left[ \underbrace{\frac{y(\mathbf{x}_T^*) - y(\mathbf{x}_{\text{init}})}{y^* - y(\mathbf{x}_{\text{init}})}}_{\text{Improvement term}} + \underbrace{\frac{T - t^*}{T}}_{\text{Efficiency term}} \right] \Big/ \underbrace{\left( 1 + \frac{T-1}{T} \right)}_{\text{Normalization}}. \tag{25}
$$

### A.1.1 GAP value range

When the global optimum is found in the first trial ($t^* = 1$), the efficiency term is equal to $\frac{T-1}{T}$, and normalization makes the maximum GAP equal to 1. However, DCBO sets $t^* = 0$ when no improvement is observed. In that case, the improvement term is 0, but the efficiency term becomes 1, which produces a nonzero GAP even without improvement. Specifically, GAP is then lower-bounded by

$$
\frac{1}{1 + \frac{T-1}{T}} = \frac{T}{2T - 1}. \tag{26}
$$

To correct for this, we set $t^* = T$ when no improvement occurs. Then both the improvement term and the efficiency term are 0, so the GAP ranges over $[0, 1]$ as intended.

### A.1.2 Why PA-GAP can rank trajectories more consistently

GAP depends only on the final best value and the trial in which that final best value is first reached. This can bias the rankings when two optimization runs share the same early progress but differ in later improvements. Figure 7 illustrates this issue with a simple two-model example. Both models start from the same initial value $y(\mathbf{x}_{\text{init}}) = 10$ and reach an improved value $y(\mathbf{x}_t^*) = 5$ at the trial $t = 10$. Model 1 does not make further progress, while Model 2 later reaches the global optimum $y^* = 1$ at trial $t = 30$. With $T = 40$, GAP assigns Model 1 a higher score than Model 2, although Model 2 achieves the better final solution. This occurs because GAP adds an improvement term and an efficiency term and then evaluates only the final best value and its discovery time, rather than the full optimization path.

To better reflect the full optimization process, PA-GAP computes a weighted improvement value at each trial and then averages these values over the trajectory. For minimization tasks, the best-so-far improvement ratio in the trial $t$ is

$$
R_t = \min \left\{ \frac{y(\mathbf{x}_{\text{init}}) - y(\mathbf{x}_t^*)}{y(\mathbf{x}_{\text{init}}) - y^*}, 1 \right\}, \tag{27}
$$

and for maximization tasks, the numerator and denominator are reversed:

$$
R_t = \min \left\{ \frac{y(\mathbf{x}_t^*) - y(\mathbf{x}_{\text{init}})}{y^* - y(\mathbf{x}_{\text{init}})}, 1 \right\}. \tag{28}
$$

The clipping prevents numerical artifacts when an observed best value exceeds the offline reference optimum. The PA-GAP score used in our benchmark is then

$$
\text{PA-GAP} = \frac{1}{T} \sum_{t=1}^{T} \left( \underbrace{R_t}_{\text{Improvement ratio}} \cdot \underbrace{\frac{T - (t-1)}{T}}_{\text{Efficiency weight}} \right). \tag{29}
$$

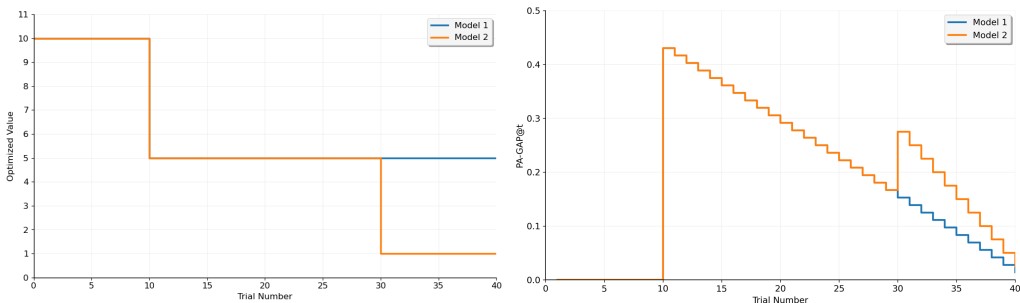

Figure 7: Demo comparison illustrating the ranking bias of GAP. The left panel shows best-so-far trajectories, while the right panel shows the corresponding per-trial PA-GAP (PA-GAP@t) contributions. The two models share the same early improvement, but Model 2 later reaches the global optimum. GAP can favor the earlier but worse final trajectory, while PA-GAP assigns credit to later improvement by averaging weighted best-so-far progress across the full path.

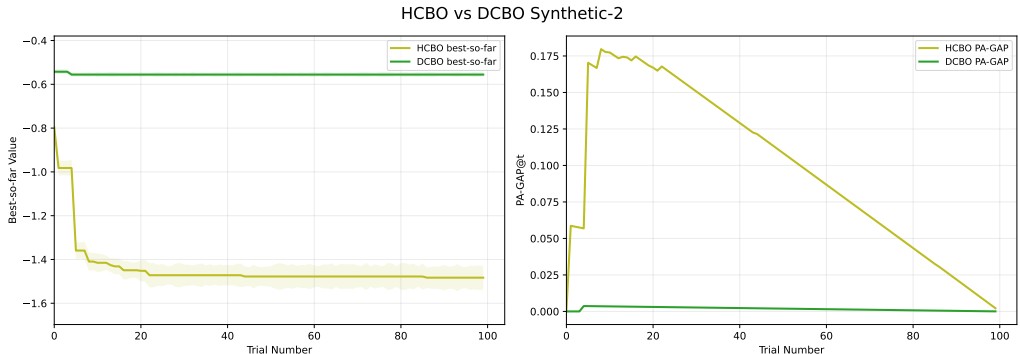

Figure 8: HCBO and DCBO on Synthetic-2. The left panel shows best-so-far trajectories, while the right panel shows the corresponding per-trial PA-GAP (PA-GAP@t) contributions. DCBO reaches its own best-so-far value early, which gives it a higher GAP score, but HCBO reaches better objective values over the trajectory and receives the higher PA-GAP score.

Here $T$ denotes the number of interventional trials, excluding the initial value in the best-so far trajectory.

This definition is not normalized to have a maximum value 1. Since $R_t \in [0, 1]$ and the efficiency weight decreases linearly from 1 at the first trial to $1/T$ at the last trial, the raw PA-GAP range is

$$0 \le \text{PA-GAP} \le \frac{1}{T} \sum_{t=1}^{T} \frac{T - (t-1)}{T} = \frac{T+1}{2T}. \tag{30}$$

The upper bound is attained when the global optimum is found in the first trial and remains the best value so far for the rest of the budget. Thus, PA-GAP should be interpreted as a raw trajectory-weighted efficiency score, not as a unit-normalized score. Larger values indicate faster and stronger best-so-far progress, but raw PA-GAP values should be compared directly only within the same trial budget.

Figure 7 shows why this trajectory-aware averaging can produce a more intuitive ranking than GAP: Model 2 receives additional credit for its late improvement instead of being judged only by the delayed discovery time of its final best value. Figure 8 shows the same issue in the Synthetic-2 benchmark by comparing HCBO and DCBO. DCBO quickly reaches a stable best-so-far value, so GAP rewards its early discovery and assigns it a higher score $(0.485 \pm .012)$ than HCBO $(0.174 \pm .071)$. However, this early DCBO value is much worse than the values eventually reached by HCBO. PA-GAP reverses the ranking: DCBO receives only $0.002 \pm .001$, while HCBO receives $0.099 \pm .054$. This better reflects the fact that HCBO makes more

Table 8: Formal comparison of GAP and PA-GAP evaluation properties.

| Property | GAP | PA-GAP |
|---|---|---|
| **What is evaluated** | Final best value + discovery time | Full best-so-far trajectory |
| **Value range** | $[0, 1]$ (after correction) | $[0, \frac{T+1}{2T}]$ |
| **Normalization** | Unit-normalized | Raw; budget-dependent upper bound |
| **Sensitivity to late improvement** | Low: only the time of the *final* best matters | High: all improvements contribute |
| **Sensitivity to budget $T$** | Moderate: efficiency term scales with $T$ | Strong: values not comparable across different $T$ |
| **Plateau handling** | Rewards early plateau at good value | Penalizes early plateau if further improvement is possible |
| **Computational cost** | $\mathcal{O}(T)$ | $\mathcal{O}(T)$ |

meaningful progress toward the reference optimum over the optimization path, even if its improvement is not captured well by GAP's final-discovery-time summary.

**Summary and formal properties.** GAP is simple and interpretable, but can produce counterintuitive rankings when methods differ primarily in their late-stage behavior or trajectory shape. PA-GAP complements GAP by rewarding sustained progress throughout the budget. Table 8 contrasts the two metrics along key evaluation dimensions.

We state two key properties of PA-GAP that justify its use as a trajectory-aware complement to GAP.

**Lemma 1** (Trajectory monotonicity of PA-GAP). *Let $\{R_t\}_{t=1}^T$ and $\{\tilde{R}_t\}_{t=1}^T$ be two sequences of the best improvement ratios so far satisfying $R_t \geq \tilde{R}_t$ for all $t \in \{1, \dots, T\}$. Then $\mathrm{PA\text{-}GAP}(\{R_t\}) \geq \mathrm{PA\text{-}GAP}(\{\tilde{R}_t\})$, with equality if and only if $R_t = \tilde{R}_t$ for all $t$.*

*Proof.* Since the efficiency weights $w_t = \frac{T-(t-1)}{T} > 0$ for all $t$, the PA-GAP functional $\frac{1}{T} \sum_{t=1}^T R_t \cdot w_t$ is a positively weighted average of the improvement ratios. The conclusion follows from the strict positivity of the weights. $\square$

This property ensures that if one trajectory Pareto-dominates another at every trial, PA-GAP reflects this dominance. GAP does not satisfy this property in general: a trajectory with uniformly higher $R_t$ can receive a lower GAP score if its final best is reached later.

**Lemma 2** (Budget dependence of PA-GAP). *The maximum attainable PA-GAP value is $\frac{T+1}{2T}$, which is strictly decreasing in $T$ and approaches $\frac{1}{2}$ as $T \to \infty$. Consequently, raw PA-GAP values are comparable only within the same budget $T$.*

*Proof.* The maximum is achieved when $R_t = 1$ for all $t$, giving PA-GAP $= \frac{1}{T} \sum_{t=1}^T \frac{T-(t-1)}{T} = \frac{T+1}{2T}$. Since $\frac{d}{dT}\left(\frac{T+1}{2T}\right) = -\frac{1}{2T^2} < 0$, the upper bound is strictly decreasing in $T$. $\square$

**Practical recommendation.** We recommend reporting both metrics alongside trajectory plots to obtain a balanced view of optimizer performance. GAP is appropriate for identifying methods that quickly find a strong final solution, while PA-GAP is appropriate for evaluating whether a method makes sustained, meaningful progress throughout the available budget.

## A.2 Implementation protocol

**Reproducibility philosophy.** All methods run with released default hyperparameters unless noted in Table 9. These are wrapper-default reproducibility results, not per-dataset tuned head-to-head comparisons. This choice prioritizes reproducibility and isolates the behavior of public implementations, but it can understate what a carefully tuned method might achieve. Appendix B.2 illustrates this sensitivity for MCBO:

Table 9: Implementation settings used in the benchmark. Method-specific defaults are preserved unless a wrapper override is needed for shared evaluation.

| Method | Benchmark-controlled settings | Method-specific notes |
|---|---|---|
| BO | Trial budget, seed, initial design, objective direction, and intervention domain fixed by wrapper; no graph input used. | Graph-free GP-EI baseline on the joint intervention vector: GPy GPRegression with RBF kernel (lengthscale = 1, variance = 1, ARD disabled), fixed observation noise $10^{-10}$, emukit Expected Improvement, and GradientAcquisitionOptimizer. All methods, including BO, use the same fixed initial interventional design per seed and the same intervention parameterization and optimization direction. |
| MCBO | Trial budget and seed fixed by wrapper; noise scale set to 0; $\beta = 10$. | Uses GP networks with MC acquisition. Acquisition optimization follows the released implementation defaults, including restart and raw-sample rules. |
| ACBO | Trial budget and seed fixed by wrapper; noise scale set to 0; $\beta = 10$; default discrete action cardinality 4. | Used only for soft-intervention function-network tasks. Runs CBO-MW with multiplicative-weights updates. |
| cCBO | Trial budget fixed by wrapper; dataset configs determine observational samples, intervention sets, and constraints. | Uses the single-task cCBO setting in our benchmark. Protein uses an internal offset so that the exported trajectory contains the intended number of evaluated trials. |
| DCBO | Trial budget and seed fixed by wrapper; dataset YAML files specify the SCM and intervention domain. | Benchmark uses a single-time setting to make DCBO comparable with static hard-intervention tasks. |
| CEO | Trial budget and seed fixed by wrapper; small candidate graph and anchor-point settings follow the benchmark runner. | Uses the benchmark-native CEO runner by default. The original vendored CEO implementation can be enabled separately. |
| CoCaBO | Trial budget, seed, and objective direction fixed by wrapper. | Uses UCB over active policy scopes and HEBO-style inner optimization. Benchmark configs do not include additional contextual variables unless specified. |
| HCBO | Seed and trial settings fixed by wrapper when supported. | Used for compatible hard-intervention settings. Acquisition and initialization constants follow the released HCBO-style implementation. |

the best UCB $\beta$ varies across datasets, budgets, and metrics, so the main table keeps the wrapper setting $\beta = 10$ and reports the sweep separately.

**Software environment.** All experiments are conducted in Python 3.10 using PyTorch 2.0 and GPyTorch 1.11 as the primary GP backend for methods that depend on Gaussian process surrogates. Each CBO method is run in its own isolated virtual environment to prevent dependency conflicts. The benchmark evaluation pipeline–including trajectory export, GAP/PA-GAP computation, and figure generation–is implemented in a shared codebase that is independent of any individual method's software stack. This separation ensures that scoring differences reflect algorithmic behavior rather than library version effects. The exact package versions for each method are documented in the benchmark repository.

**Computational resources.** All experiments were conducted on a single-GPU CUDA workstation equipped with one NVIDIA A100 GPU with 40 GiB of VRAM, Intel Xeon Platinum 8481C CPUs, and 754 GiB of system memory. Most methods complete 100 trials on a single dataset within 5–30 minutes. The main exceptions are CEO, which incurs additional overhead from graph posterior sampling, and MCBO, which propagates uncertainty through multiple mechanism-level GPs. GPU acceleration is used when supported by the corresponding implementation; otherwise, experiments are executed on the CPU.

**Wrapper design.** Wrapper-level overrides are limited to shared experimental controls: trial budget, random seed, dataset mapping, objective direction (minimization or maximization), and trajectory export format. The Method-internal choices—including the selection of the GP kernel, the acquisition optimization strategy, the restart schedules, the scope enumeration, and the internal noise models—are left at their defaults released. Each wrapper converts the method's internal output into the common trajectory format described below. Wrappers do not modify the optimization loop itself; they only control initialization and output extraction.

**Trajectory normalization.** To make results comparable across heterogeneous codebases, every method's output is converted to a common trajectory format with two fields: trial number and best-so-far objective value. For minimization tasks, this value is the running minimum observed up to the trial $t$; for maximization tasks, the running maximum. The trajectory includes an initial best-of-so-far value (before any interventional trial), followed by one entry per interventional trial. Thus, a budget of $B$ interventions produces a trajectory of $B + 1$ entries. GAP and PA-GAP are then computed from these exported trajectories using a single evaluation script shared by all methods, ensuring that scoring differences reflect algorithmic behavior rather than implementation artifacts. This design also makes it easy to add new methods or metrics without modifying existing implementations.

### A.3  Reference optima

The reference optimum $y^*$ used by GAP and PA-GAP is computed offline from the SCM benchmark, function generator or oracle interventional data, as detailed in Table 10. It is used *only* for scoring and is never provided to any optimizer during a run. At evaluation time, all methods on a given data set are compared against the same fixed reference value using the same task direction.

**Reliability categories.** The reference optima fall into two reliability classes. For datasets with analytically known optima (Dropwave, Alpine2, Ackley, Rosenbrock), the reference is exact and introduces no scoring uncertainty. For SCM-based datasets, the reference is obtained by dense grid search or large-sample uniform random search over the admissible intervention domain. In these cases, $y^*$ represents the *best known achievable outcome*, not necessarily the true global optimum. This distinction is important for datasets with complex, potentially multimodal objective landscapes: if the reference underestimates the true optimum, both GAP and PA-GAP will overestimate the improvement ratios $R_t$, potentially making all methods appear closer to optimal than they actually are. In the opposite direction, an overestimated reference (which cannot occur by construction in our protocol, since we use oracle SCM access) would inflate the denominator and deflate all scores. To mitigate this risk, we use high-budget offline search (10,000 samples for Ecology and Epidemiology, $200 \times 200$ grids for Protein-reconstructed, and 1,000-point grids for Synthetic-2) and cross-validate against known interventional datasets where available.

**Impact on metric comparisons.** Because both GAP and PA-GAP normalize the improvement relative to $y^* - y(\mathbf{x}_{\mathrm{init}})$, the quality of $y^*$ affects the absolute metric values, but does not affect *the relative rankings* between methods in the same dataset, provided that all methods are scored against the same reference. Cross data set comparisons of absolute GAP or PA-GAP values should therefore be made with caution, as differences in reference quality across datasets can introduce systematic biases.

### A.4  Consolidated dataset summary

Table 11 records benchmark provenance, including whether a task is inherited, reconstructed, or newly added, and Table 12 provides the code-authoritative structural and domain summary. Together these tables clarify that the benchmark is a standardized harness over heterogeneous prior tasks rather than a claim that every dataset is newly introduced here.

Table 12 provides a unified overview of all 13 benchmark datasets, consolidating key structural and experimental properties in a single reference. This table complements the SCM specifications per-dataset in Appendices A.5–A.7 and the usage-coverage table (Table 3) in the main text. The column "#Nodes" counts

Table 10: Reference optima used for GAP and PA-GAP scoring. Analytic optima are exact; grid/random-search references represent the best known achievable value.

| Dataset | Task | $y^*$ | Offline computation protocol |
|---|---|---|---|
| Dropwave | Max | 1.0000 | Analytic optimum from the known function generator. |
| Alpine2 | Max | 400.0000 | Analytic optimum from the known function generator. |
| Ackley | Max | 0.0000 | Analytic optimum of the negated Ackley objective. |
| Rosenbrock | Max | 0.0000 | Analytic optimum of the negated Rosenbrock objective. |
| Chain-soft | Min | $-3.3936$ | Reference value from the Chain-soft function generator. |
| Protein-recon. | Min | $-240.0234$ | Reconstructed linear SEM; $200 \times 200$ grid over PKC and PKA; minimum Erk. |
| Synthetic-2 | Min | $-3.9690$ | Best offline oracle outcome; 1000-point grids on $X$ and $Z$. |
| Synthetic | Min | $-4.4474$ | Best value across BO and CBO oracle interventional datasets. |
| ToyGraph | Min | $-2.5718$ | Best value across BO and CBO oracle interventional datasets. |
| Chain-hard | Min | $-15.8060$ | Best value across BO and CBO oracle interventional datasets. |
| Ecology | Max | 9.2652 | 10,000-sample uniform search over the manipulative-variable domain. |
| Epidemiology | Min | $-3.3906$ | 10,000-sample uniform search over the manipulative-variable domain. |
| Healthcare | Min | 4.0401 | Reference value from the distributed real-data benchmark artifact. |

endogenous (observed) variables only; latent variables are listed separately where applicable. The column "Intervention domain" specifies the admissible range per manipulable variable as used in the benchmark.

Table 11: Benchmark provenance and domain reconciliation. "Current domain" refers to the code-authoritative domain used in our benchmark.

| Dataset | Source / provenance | Current domain note | Modification status |
|---|---|---|---|
| ToyGraph | Aglietti et al. (2020) | $X \in [-5, 5]$, $Z \in [-5, 20]$ | Inherited and standardized |
| Synthetic | CBO-style synthetic SCM | $B, D, E$ code domains | Newly added SCM |
| Synthetic-2 | cCBO-style synthetic SCM | $X \in [-3, 2]$, $Z \in [-1, 1]$ | Newly added / reconstructed SCM |
| Chain-hard | Chain benchmark from fCBO-style SCM | $W, Z \in [-1, 1]$ | Hard-intervention variant |
| Ecology | Aglietti et al. (2020) / ecological SCM | Code feature-parameter ranges | Inherited and repackaged |
| Protein-reconstructed | Sachs protein data / cCBO graph | Sachs-derived intervention ranges | Reconstructed fitted SCM |
| Healthcare | Aglietti et al. (2020) PSA example | Aspirin, Statin in $[0, 1]$ | Inherited and repackaged |
| Epidemiology | CEO/DCBO epidemiology SCM | $L \in [0.479, 801.787]$, $B \in [-0.994, 0.999]$ | Reconciled to code-authoritative intervention variables |
| Ackley / Rosenbrock / Dropwave / Alpine2 | Function-network BO (Astudillo & Frazier, 2021); used by MCBO/ACBO | Standard function-network domains after wrapper normalization | Inherited function-network tasks |
| Chain-soft | Gultchin et al. (2023) | Policy coefficients in $[-0.27, 0.27]$, $W \in [-1, 1]$ | Inherited policy-intervention task |

Table 12: Consolidated summary of all thirteen benchmark datasets. For each dataset, we list the category, number of observed nodes, manipulable variables, intervention type, admissible domain per variable, optimization direction, and source reference. Latent variables (if any) are noted in parentheses.

| Dataset | Category | #Nodes | Manipulable | Type | Domain | Task | Source |
|---|---|---|---|---|---|---|---|
| ToyGraph | Synthetic (hard) | 3 | $X, Z$ | Hard | $X \in [-5, 5]$; $Z \in [-5, 20]$ | Min $Y$ | Aglietti et al. (2020) |
| Synthetic | Synthetic (hard) | 7 (+2 latent) | $B, D, E$ | Hard | $B \in [-5, 4]$; $D \in [-5, 5]$; $E \in [-6, 3]$ | Min $Y$ | Aglietti et al. (2020) |
| Synthetic-2 | Synthetic (hard) | 3 | $X, Z$ | Hard | $X \in [-3, 2]$; $Z \in [-1, 1]$ | Min $Y$ | Aglietti et al. (2023) |
| Chain-hard | Synthetic (hard) | 4 | $W, Z$ | Hard | $[-1, 1]$ each | Min $Y$ | Gultchin et al. (2023) |
| Ecology | Real/fitted (hard) | 11 | $N, O, C, T, D$ | Hard | $N \in [-2, 5]$; $O \in [2, 4]$; $C \in [0, 1]$; $T \in [2200, 2500]$; $D \in [1950, 2100]$ | Max $Y$ | Aglietti et al. (2020) |
| Protein-recon. | Real/fitted (hard) | 8 | PKC, PKA, Mek, Akt | Hard | PKC $\in [0.5, 106.5]$; PKA $\in [1.45, 4491.5]$; Mek $\in [0.5, 389.5]$; Akt $\in [1.2, 3555.5]$ | Min Erk | Aglietti et al. (2023) |
| Healthcare | Real/fitted (hard) | 6 | Aspirin, Statin | Hard | $[0, 1]$ each | Min PSA | Aglietti et al. (2020) |
| Epidemiology | Real/fitted (hard) | 5 | $L, B$ | Hard | $L \in [0.479, 801.787]$; $B \in [-0.994, 0.999]$ | Min $Y$ | Branchini et al. (2023) |
| Ackley | Synthetic (soft) | 9 | $a_0, \ldots, a_5$ | Soft (function network) | $[-2, 2]$ each | Max $Y$ | Astudillo & Frazier (2021); Sussex et al. (2023) |
| Rosenbrock | Synthetic (soft) | $D+D-1$ | $a_0, \ldots, a_{D-1}$ | Soft (function network) | $[-2, 2]$ each | Max $Y$ | Astudillo & Frazier (2021); Sussex et al. (2023) |
| Dropwave | Synthetic (soft) | 4 | $a_0, a_1$ | Soft (function network) | $[-5.12, 5.12]$ each | Max $Y$ | Astudillo & Frazier (2021); Sussex et al. (2023) |
| Alpine2 | Synthetic (soft) | 12 | $a_0, \ldots, a_5$ | Soft (function network) | $[0, 10]$ each | Max $Y$ | Astudillo & Frazier (2021); Sussex et al. (2023) |
| Chain-soft | Synthetic (soft) | 4 | $W$ (hard), $Z$ (policy) | Mixed hard/soft | $W \in [-1, 1]$; $Z = \pi(X)$ with coeffs. $\in [-0.27, 0.27]$ | Min $Y$ | Gultchin et al. (2023) |

## A.5 Synthetic dataset SCMs

This section provides the full structural equations, noise distributions, intervention domains, and causal graph diagrams for the four synthetic hard-intervention datasets used in our benchmark. For each data set, we follow a standardized format: (i) a brief description including the source reference, the number of nodes, and the direction of the task; (ii) the exogenous variables and their distributions; (iii) the endogenous structural equations; (iv) the noise model with explicit distributional parameters; (v) the intervention domain; and

(vi) a figure of the causal graph with a consistent color scheme (green = manipulable, red = target, light-gray = non-manipulable, dark-gray = latent).

### A.5.1 ToyGraph

A minimal three-node chain-structured SCM introduced by Aglietti et al. (2020), consisting of variables $X$, $Z$, and $Y$. Both $X$ and $Z$ are manipulable (intervenable); $Y$ is the target. The task is to minimize $Y$. Figure 9 shows the causal graph.

**Exogenous variable.**

$$X \sim P_X \quad \text{(input variable, exogenous)}$$

**Endogenous variables.**

$$Z = e^{-X} + \varepsilon_Z,$$
$$Y = \cos Z - e^{-Z/20} + \varepsilon_Y.$$

**Noise model.** All noise terms $\varepsilon_Z, \varepsilon_Y$ are mutually independent. In the original CBO implementation, these are drawn from $\mathcal{N}(0, \sigma^2)$ with a small variance $\sigma^2$.

**Intervention domain.** Hard interventions on $X$ and $Z$ use dataset-specific domains:

$$X \in [-5, 5], \qquad Z \in [-5, 20].$$

The admissible scopes are $\{X\}$, $\{Z\}$, and $\{X, Z\}$.

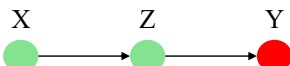

Figure 9: ToyGraph SCM. Green nodes represent manipulable variables; the red node is the target variable $Y$.

### A.5.2 Synthetic

A seven-node SCM with two latent variables, introduced by Aglietti et al. (2020). The observed nodes are $A, B, C, D, E, F, Y$, with latent confounders $U_1$ and $U_2$. Among the observed nodes, $B$, $D$, and $E$ are manipulable; $A$, $C$, and $F$ are not manipulable; and $Y$ is the target. The task is to minimize $Y$. Figure 10 shows the causal graph.

**Latent variables.**

$$U_1 = \varepsilon_{YA} \sim \mathcal{N}(0, 1),$$
$$U_2 = \varepsilon_{YB} \sim \mathcal{N}(0, 1).$$

**Exogenous variable.**

$$F = \varepsilon_F \sim \mathcal{N}(0, 1).$$

**Endogenous variables.**

$$A = F^2 + U_1 + \varepsilon_A,$$

$$B = U_2 + \varepsilon_B,$$

$$C = e^{-B} + \varepsilon_C,$$

$$D = \frac{e^{-C}}{10} + \varepsilon_D,$$

$$E = \cos A + \frac{C}{10} + \varepsilon_E,$$

$$Y = \cos D + \sin E + U_1 + U_2 \, \varepsilon_Y.$$

**Noise model.** Each $\varepsilon_\bullet$ is an independent noise term. In the original CBO implementation, the endogenous noise terms are drawn from $\mathcal{N}(0, \sigma^2)$ with small variance $\sigma^2$.

**Intervention domain.** Hard interventions on $B$, $D$, and $E$ use dataset-specific domains:

$$B \in [-5, 4], \qquad D \in [-5, 5], \qquad E \in [-6, 3].$$

The admissible scopes include all non-empty subsets of $\{B, D, E\}$ considered by the benchmark intervention-set construction.

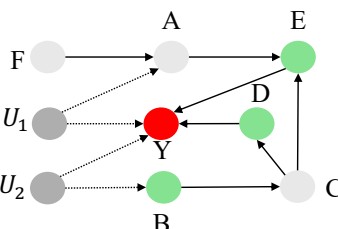

Figure 10: Synthetic SCM. Light-gray nodes are non-manipulable variables; dark-gray nodes are latent confounders; green nodes are manipulable variables; the red node is the target variable $Y$.

### A.5.3 Synthetic-2

A three-node SCM introduced in the cCBO line of work (Aglietti et al., 2023), consisting of variables $X$, $Z$, and $Y$. Both $X$ and $Z$ are manipulable; $Y$ is the target. The task is to minimize $Y$. The structure is similar to ToyGraph but uses explicitly specified Gaussian noise. Figure 11 shows the causal graph.

**Exogenous variable.**

$$X = \varepsilon_X, \qquad \varepsilon_X \sim \mathcal{N}(0, 1).$$

**Endogenous variables.**

$$Z = e^{-X} + \varepsilon_Z,$$

$$Y = \cos Z - e^{-Z/20} + \varepsilon_Y.$$

**Noise model.** $\varepsilon_X \sim \mathcal{N}(0, 1)$, $\varepsilon_Z \sim \mathcal{N}(0, 1)$, and $\varepsilon_Y \sim \mathcal{N}(0, 1)$. All noise terms are mutually independent.

**Intervention domain.** Hard interventions on $X$ and $Z$ use $X \in [-3, 2]$ and $Z \in [-1, 1]$. The admissible scopes are $\{X\}$, $\{Z\}$, and $\{X, Z\}$.

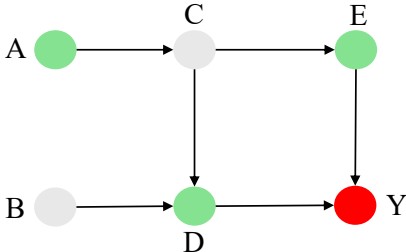

Figure 11: Synthetic-2 SCM. Green nodes represent manipulable variables; the red node is the target variable $Y$.

### A.5.4 Chain-hard

A four-node SCM introduced by Gultchin et al. (2023) as the hard-intervention version of the Chain benchmark. The observed variables are $X$, $W$, $Z$, and $Y$. Among them, $W$ and $Z$ are manipulable; $X$ is a non-manipulable context variable; and $Y$ is the target. The task is to minimize $Y$. Figure 12 illustrates the causal graph.

**Structural equations.**

$$
\begin{aligned}
X &= U_X, \\
W &= U_W, \\
Z &= -0.5X + U_Z, \\
Y &= -W - 3ZX + U_Y.
\end{aligned}
$$

**Noise model.** $U_X, U_W, U_Z, U_Y \sim \mathcal{N}(0,1)$, mutually independent.

**Intervention domain.** Hard interventions in $W$ and $Z$ are each restricted to $[-1, 1]$. A hard intervention replaces the corresponding structural equation with a fixed constant. The admissible scopes are $\{W\}$, $\{Z\}$, and $\{W, Z\}$.

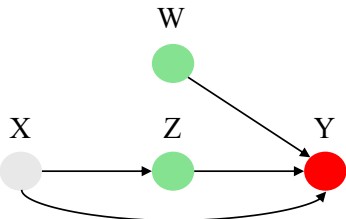

Figure 12: Chain-hard SCM. Light-gray nodes are non-manipulable variables; green nodes are manipulable variables; the red node is the target variable $Y$.

### A.6 Real dataset SCMs

This section provides the full structural equations, noise distributions, fitting details, and intervention domains for the four real or fitted-SCM hard-intervention datasets. Unlike the synthetic datasets above, these SCMs are either fitted to real observational data or derived from domain-specific causal models. We follow the same standardized format as in the Appendix A.5. For protein reconstruction, we additionally document the fitting procedure to ensure transparent reproducibility.

### A.6.1 Ecology

An 11-node SCM based on the Bermuda reef calcification model of Courtney et al. (2017), introduced in CBO (Aglietti et al., 2020). The manipulable variables are Nut, Chl$\alpha$, TA, DIC, and $\Omega_A$. The non-manipulable variables are Tem, Sal, $P_{co_2}$, Light, and pHsw. The target is NEC (net ecosystem calcification). The task is to *maximize* NEC. Figure 13 shows the causal graph.

**Exogenous variables.**

$$\text{Tem} = U_{\text{Tem}} \sim \mathcal{N}(24.184130,\ 3.220405^2),$$

$$\text{Sal} = U_{\text{Sal}} \sim \mathcal{N}(36.591624,\ 0.149197^2),$$

$$\text{Nut} = U_{\text{Nut}} \sim \mathcal{N}(0.492065,\ 1.592408^2),$$

$$\text{TA} = U_{\text{TA}} \sim \mathcal{N}(2357.893696,\ 27.609355^2).$$

**Endogenous variables.**

$$P_{co_2} = 18.798174 + 15.797384\,\text{Tem} + U_{P_{co_2}},$$

$$\text{Chl}\alpha = 0.373420 - 0.002400\,\text{Nut} + U_{\text{Chl}\alpha},$$

$$\text{Light} = 6665.081996 - 10737.462582\,\text{Chl}\alpha + U_{\text{Light}},$$

$$\text{pHsw} = 8.427706 - 0.000966\,P_{co_2} + U_{\text{pHsw}},$$

$$\text{DIC} = 2131.672107 - 0.216560\,P_{co_2} + U_{\text{DIC}},$$

$$\Omega_A = 3.245248 + 0.094332\,\text{Tem} + 0.006754\,\text{Sal} - 0.005737\,P_{co_2} + U_{\Omega_A},$$

$$\text{NEC} = 211.422555 - 0.000030\,\text{Light} + 0.016680\,\text{Nut} - 25.719277\,\text{pHsw} - 0.500403\,\Omega_A + U_{\text{NEC}}.$$

**Noise model.** $U_{P_{co_2}} \sim \mathcal{N}(0, 28.896639^2)$, $U_{\text{Chl}\alpha} \sim \mathcal{N}(0, 0.039295^2)$, $U_{\text{Light}} \sim \mathcal{N}(0, 1546.913034^2)$, $U_{\text{pHsw}} \sim \mathcal{N}(0, 0.005258^2)$, $U_{\text{DIC}} \sim \mathcal{N}(0, 18.412100^2)$, $U_{\Omega_A} \sim \mathcal{N}(0, 0.036958^2)$, and $U_{\text{NEC}} \sim \mathcal{N}(0, 1.073340^2)$. All noise terms are mutually independent.

**Intervention domain.** Hard interventions in Nut, Chl$\alpha$, TA, DIC, and $\Omega_A$ are applied in domain-specific ranges corresponding to the natural variability reported in Courtney et al. (2017). The benchmark uses the intervention domains specified in the public CBO repository.

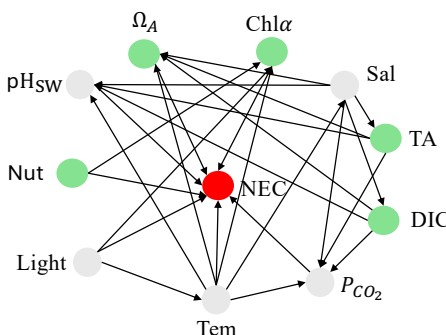

Figure 13: Ecology SCM. Light-gray nodes are non-manipulable variables; green nodes are manipulable variables; the red node is the target variable NEC.

### A.6.2 Protein-reconstructed

An eight-node SCM reconstructed from protein-signaling data from Sachs et al. (2005), following the DAG used in cCBO (Aglietti et al., 2023). The manipulable variables are PKC, PKA, Mek, and Akt. The non-manipulable variables are Raf, P38, and Jnk. The target is Erk (extracellular signal-regulated kinase). The task is to minimize Erk. Figure 14 shows the causal graph.

**Fitting procedure.** This is a transparent reconstruction, not an exact reproduction of the unreleased fitted SCM from the original cCBO experiments. We retain the connected subgraph used in cCBO and fit *linear* structural mechanisms using ordinary least squares on 852 observational samples from Sachs et al. (2005). The parent sets are determined by the cCBO DAG. The linear model class was chosen for reproducibility and interpretability; nonlinear alternatives (e.g. kernel regression) were not explored.

**Exogenous variable.**

$$\text{PKC} = U_{\text{PKC}} \sim \text{Uniform}(1, 106).$$

**Endogenous variables.**

$$\text{PKA} = 554.390731 + 0.841153\,\text{PKC} + U_{\text{PKA}},$$

$$\text{Raf} = 62.199046 - 0.177745\,\text{PKC} - 0.000379\,\text{PKA} + U_{\text{Raf}},$$

$$\text{Mek} = -1.090275 + 0.039662\,\text{PKC} - 0.000652\,\text{PKA} + 0.520845\,\text{Raf} + U_{\text{Mek}},$$

$$\text{P38} = 15.144328 + 1.234783\,\text{PKC} + 0.000591\,\text{PKA} + U_{\text{P38}},$$

$$\text{Jnk} = 52.953603 - 0.764801\,\text{PKC} - 0.005306\,\text{PKA} + U_{\text{Jnk}},$$

$$\text{Akt} = -31.110747 + 0.128905\,\text{PKA} + U_{\text{Akt}},$$

$$\text{Erk} = -23.248743 + 0.081707\,\text{PKA} - 0.029886\,\text{Mek} + U_{\text{Erk}}.$$

**Noise model.** $U_{\text{PKA}} \sim \mathcal{N}(0, 427.437996^2)$, $U_{\text{Raf}} \sim \mathcal{N}(0, 41.768782^2)$, $U_{\text{Mek}} \sim \mathcal{N}(0, 16.695865^2)$, $U_{\text{P38}} \sim \mathcal{N}(0, 13.111164^2)$, $U_{\text{Jnk}} \sim \mathcal{N}(0, 42.074715^2)$, $U_{\text{Akt}} \sim \mathcal{N}(0, 113.958504^2)$, and $U_{\text{Erk}} \sim \mathcal{N}(0, 82.760198^2)$. All noise terms are mutually independent.

**Intervention domain and constraints.** Hard interventions in PKC and PKA are drawn from the range $[0.5, 106.5]$ and $[1.45, 4491.5]$, respectively. The interventions in Mek and Akt use the domains observed in the Sachs data, which are $[0.5, 389.5]$ and $[1.2, 3555.5]$. Biological plausibility constraints on PKC and PKA are enforced by clamping the proposed values to their observed ranges.

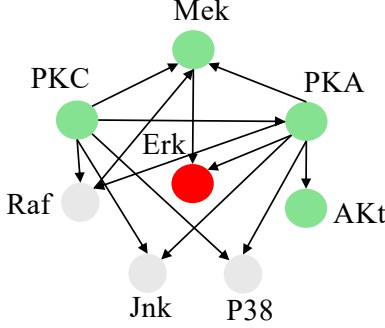

Figure 14: Protein-signaling SCM (reconstructed). Light-gray nodes are non-manipulable variables; green nodes are manipulable variables; the red node is the target variable Erk.

### A.6.3 Healthcare

A six-node SCM derived from a real-world clinical setting (Thompson & Yung, 2019), involving Age, BMI, Aspirin, Statin, Cancer, and PSA. The manipulable variables are Aspirin and Statin. Non-manipulable variables are Age, BMI, and Cancer. The target is PSA (prostate-specific antigen). The task is to minimize PSA. Figure 15 shows the causal graph.

**Exogenous variable.**

$$\text{Age} \sim \text{Uniform}(55,\, 75).$$

**Endogenous variables.**

$$\text{BMI} = 27.0 - 0.01\,\text{Age} + \varepsilon_{\text{BMI}},$$

$$\text{Aspirin} = \sigma\big(-8.0 + 0.10\,\text{Age} + 0.03\,\text{BMI}\big) + \varepsilon_{\text{Aspirin}},$$

$$\text{Statin} = \sigma\big(-13.0 + 0.10\,\text{Age} + 0.20\,\text{BMI}\big) + \varepsilon_{\text{Statin}},$$

$$\text{Cancer} = \sigma\big(2.2 - 0.05\,\text{Age} + 0.01\,\text{BMI} - 0.04\,\text{Statin} + 0.02\,\text{Aspirin}\big) + \varepsilon_{\text{Cancer}},$$

$$\text{PSA} = 6.8 + 0.04\,\text{Age} - 0.15\,\text{BMI} - 0.60\,\text{Statin} + 0.55\,\text{Aspirin} + 1.00\,\text{Cancer} + \varepsilon_{\text{PSA}},$$

where $\sigma(x) = \frac{1}{1+e^{-x}}$ denotes the sigmoid function.

**Noise model.** $\varepsilon_{\text{BMI}} \sim \mathcal{N}(0,\, 0.7^2)$, $\varepsilon_{\text{PSA}} \sim \mathcal{N}(0,\, 0.4^2)$. Each remaining $\varepsilon_{\bullet}$ is an independent noise term with small variance.

**Intervention domain.** Although the original data template specifies Aspirin and Statin as binary variables, existing CBO implementations commonly relax them to continuous values in $(0, 1)$ for BO-style optimization. We follow this convention to remain comparable to the prior work. The admissible scopes are {Aspirin}, {Statin}, and {Aspirin, Statin}.

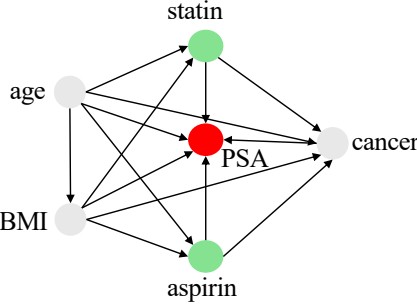

Figure 15: Healthcare SCM. Light-gray nodes are non-manipulable variables; green nodes are manipulable variables; the red node is the target variable PSA.

### A.6.4 Epidemiology

A five-node SCM introduced in CEO (Branchini et al., 2023), modeling an HIV treatment scenario based on Havercroft & Didelez (2012). The observed variables are $B$, $T$, $L$, $R$, and $Y$. In the code-authoritative benchmark configuration, the manipulable variables are $L$ and $B$; $T$ and $R$ are non-intervention SCM variables. The target is $Y$ (viral load). The task is to minimize $Y$. Figure 16 shows the causal graph.

**Exogenous variables.**

$$B \sim \text{Uniform}(-1, 1),$$

$$T \sim \text{Uniform}(4, 8).$$

**Endogenous variables.**

$$L = \exp(0.5\,T + U), \qquad U \sim \mathcal{N}(0, 1),$$

$$R = 4 + L\,T,$$

$$Y = 0.5 + \cos(4T) + \sin(-L + 2R) + B + \varepsilon, \qquad \varepsilon \sim \mathcal{N}(0, 1).$$

**Noise model.** $U \sim \mathcal{N}(0, 1)$ and $\varepsilon \sim \mathcal{N}(0, 1)$. All noise terms are mutually independent.

**Intervention domain.** Hard interventions in $L$ are drawn from $[0.479, 801.787]$ and in $B$ from $[-0.994, 0.999]$. The admissible scopes are $\{L\}$, $\{B\}$, and $\{L, B\}$. Since CEO uses a modified HIV SCM, more specific clinical interpretations of $B$ and $L$ are not provided in the original article.

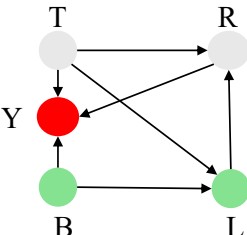

Figure 16: Epidemiology SCM. Light-gray nodes are non-manipulable variables; green nodes are manipulable variables; the red node is the target variable $Y$ (viral load).

### A.7  Soft-intervention datasets

This section details the five soft-intervention datasets used in the benchmark. These datasets fall into two categories:

1. **Function network benchmarks** (Ackley, Rosenbrock, Dropwave, Alpine2): Classical global optimization benchmarks reformulated as deterministic function networks, where each action variable propagates through intermediate nodes to the target. Interventions correspond to setting action variables, and the network structure defines soft causal relationships. These are *maximization* tasks.

2. **SCM-based policy benchmark** (Chain-soft): A full structural causal model in which a soft intervention replaces the structural equation of one variable with a context-dependent policy. This is a *minimization* task.

For each data set, we specify the network architecture, all structural equations, the action domains, and the target computation. Figures use a consistent color scheme: blue = intermediate variables, orange squares = action (intervention) variables, red = target.

### A.7.1  Ackley

A multimodal, non-separable function-network benchmark with six action variables $a_0, \ldots, a_5$, two intermediate nodes $X_0$ and $X_1$, and target node $Y$. The task is to *maximize $Y$*. Following the standard configuration in the BO function-network (Sussex et al., 2023; 2024), we use the domain $a_i \in [-2, 2]$ for $i = 0, \ldots, 5$, corresponding to $D = 6$. Figure 17 shows the function network.

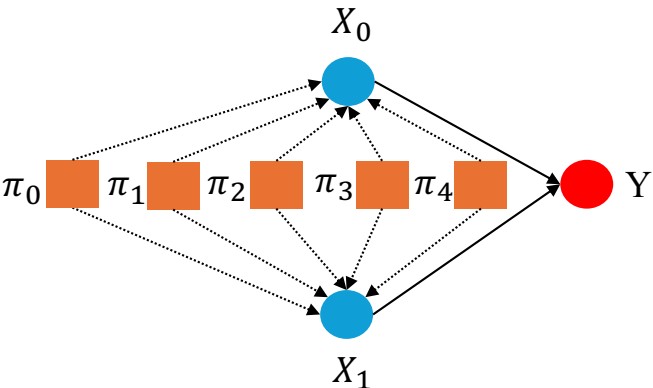

Figure 17: Ackley function network ($D = 6$). Blue nodes indicate intermediate variables; orange squares represent action (intervention) variables; the red node is the target variable $Y$.

**Action variables.** $a_i \in [-2, 2], \quad i = 0, \ldots, 5.$

**Intermediate variables.**

$$X_0 = \frac{1}{D} \sum_{i=0}^{5} a_i^2,$$

$$X_1 = \frac{1}{D} \sum_{i=0}^{5} \cos(2\pi a_i),$$

where $D = 6$.

**Target variable.**

$$Y = 20 \exp\left(-0.2\sqrt{X_0}\right) + \exp(X_1) - 20 - e.$$

**Interpretation.** $X_0$ aggregates the average squared magnitude of the action variables, while $X_1$ aggregates the average cosine term. The target $Y$ combines these two intermediate quantities to recover the Ackley objective in the form of maximum reward. The observational (unintervened) setting corresponds to $a_i = 0$ for all $i$.

### A.7.2 Rosenbrock

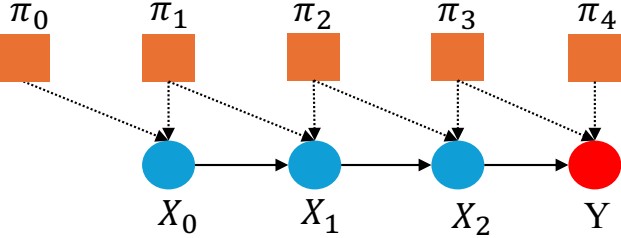

Figure 18: Rosenbrock function network (shown for $D = 3$). Blue nodes indicate intermediate variables; orange squares represent action (intervention) variables; the red node is the target variable $Y$.

A unimodal, non-separable function-network benchmark with a narrow curved valley. The network has action variables $a_0, \ldots, a_{D-1}$, intermediate nodes $X_0, \ldots, X_{D-2}$, and target node $Y$. The task is to *maximize $Y$*.

We use the domain $a_i \in [-2, 2]$ and consider $D \in \{3, 5, 7\}$. Figure 18 shows the chain-structured function network.

**Action variables.** $a_i \in [-2, 2], \quad i = 0, \ldots, D - 1.$

**Intermediate variables.** The first intermediate node computes the first Rosenbrock term:

$$X_0 = -100(a_1 - a_0^2)^2 - (1 - a_0)^2.$$

Each subsequent node recursively accumulates the next pairwise contribution:

$$X_k = -100(a_{k+1} - a_k^2)^2 - (1 - a_k)^2 + X_{k-1}, \qquad k = 1, \ldots, D - 2.$$

**Target variable.**

$$Y = X_{D-2}.$$

**Interpretation.** The Rosenbrock function network forms a causal chain in which the first node computes the initial Rosenbrock component, and each later node adds one additional pairwise contribution to the accumulated objective. The target $Y$ recovers the Rosenbrock objective in the form of maximum reward. The observational setting corresponds to $a_i = 0$ for all $i$.

### A.7.3 Dropwave

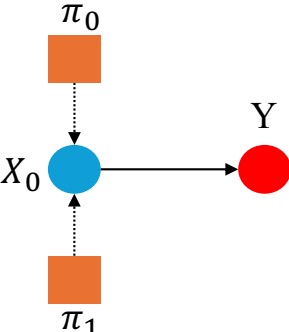

Figure 19: Dropwave function network. Blue nodes indicate intermediate variables; orange squares represent action (intervention) variables; the red node is the target variable $Y$.

A highly multimodal two-dimensional function-network benchmark with two action variables $a_0$ and $a_1$, an intermediate node $X_0$ and a target node $Y$. The task is to *maximize* $Y$. Following the BO setup of the standard function-network (Sussex et al., 2023; 2024), we use the domain $a_0, a_1 \in [-5.12, 5.12]$. Figure 19 shows the function network.

**Action variables.** $a_0, a_1 \in [-5.12, 5.12].$

**Intermediate variable.**

$$X_0 = \sqrt{a_0^2 + a_1^2} + \epsilon_0.$$

**Target variable.**

$$Y = \frac{1 + \cos(12X_0)}{2 + 0.5X_0^2} + \epsilon_Y.$$

**Interpretation.** The intermediate node $X_0$ computes the radial distance from the origin induced by the two action variables. The target $Y$ applies the oscillatory Dropwave transformation to this radial value. Following MCBO and related function-network BO settings, this benchmark can also be evaluated in a noisy setting by choosing non-zero $\epsilon_0$ and $\epsilon_Y$.

### A.7.4 Alpine2

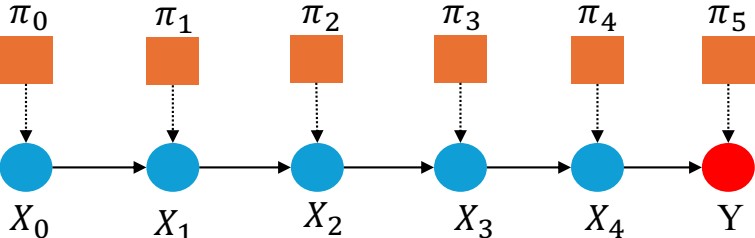

Figure 20: Alpine2 function network ($K = 6$). Blue nodes indicate intermediate variables; orange squares represent action (intervention) variables; the red node is the target variable $Y$.

A multimodal, separable chain-structured function-network benchmark with action variables $a_0, \ldots, a_5$, intermediate nodes $X_0, \ldots, X_4$, and target node $Y$. The task is to *maximize $Y$*. We use the setting $K = 6$ with the action domain $a_i \in [0, 10]$ for $i = 0, \ldots, 5$. Figure 20 shows the function network.

**Action variables.** $a_i \in [0, 10], \quad i = 0, \ldots, 5.$

**Intermediate variables.** The first intermediate node is defined as

$$X_0 = -\sqrt{a_0} \sin(a_0).$$

Each subsequent node combines its own action variable with the output of the previous node through the Alpine2 multiplicative structure:

$$X_k = \sqrt{a_k} \sin(a_k) X_{k-1}, \qquad k = 1, \ldots, 4.$$

**Target variable.**

$$Y = \sqrt{a_5} \sin(a_5) X_4.$$

**Interpretation.** Alpine2 forms a causal chain in which each stage combines its own decision variable with the output of the previous stage through a multiplicative structure. The analytical global maximum is $\prod_{i=0}^{5} \sqrt{a_i^*} \sin(a_i^*) = 2.808^6 \approx 400$ at $a_i^* \approx 7.917$ for all $i$. This subsection corresponds to the $K = 6$ configuration used in our experiments.

### A.7.5 Chain-soft

The soft-intervention Chain benchmark uses the same base SCM as Chain-hard (Appendix A.5.4), with observed nodes $X, W, Z, Y$. Here, $X$ is a non-manipulable context variable, $W$ is hard-intervenable, $Z$ is *soft*-intervenable via a context-dependent policy, and $Y$ is the target. The task is to *minimize $Y$*. Figure 21 illustrates the SCM with the intervention policy.

**Exogenous variables.**

$$X = U_X \sim \mathcal{N}(0, 1),$$
$$W = U_W \sim \mathcal{N}(0, 1).$$

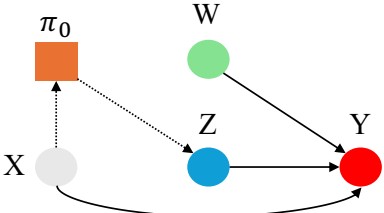

Figure 21: Chain SCM under soft intervention. Light-gray nodes are non-manipulable variables; green nodes indicate hard-intervenable variables; blue nodes indicate intermediate variables; the orange rectangle denotes the intervention policy $\pi_0$; the red node is the target variable $Y$.

**Endogenous variables (unintervened).**

$$Z = -0.5X + U_Z,$$

$$Y = -W - 3ZX + U_Y,$$

where $U_Z \sim \mathcal{N}(0,1)$ and $U_Y \sim \mathcal{N}(0,1)$, all mutually independent.

**Intervention policy.** Under soft intervention, the structural mechanism of $Z$ is replaced by a context-dependent policy $\pi_0$:

$$Z := \pi_0(X),$$

so that the intervened SCM becomes:

$$X = U_X,$$

$$W = U_W \quad \text{or} \quad W := w, \ w \in [-1, 1],$$

$$Z = \pi_0(X),$$

$$Y = -W - 3ZX + U_Y.$$

**Intervention domain and policy parameterization.** The admissible scopes include a purely functional intervention in $Z$ (through the policy $\pi_0$) and a mixed intervention combining the functional intervention in $Z$ with a hard intervention in $W$ over $[-1, 1]$. Following Gultchin et al. (2023), each functional intervention is represented using samples `GridSize = 10` and $N_\alpha = N_\beta = 10$ for the context variable, with coefficients $\alpha_i$ and $\beta_j$ uniformly sampled from $[-0.27, 0.27]$. This parameterization keeps the induced intervention values comparable to the hard-intervention range $[-1, 1]$.

**Interpretation.** Because the target $Y = -W - 3ZX + U_Y$ depends on the interaction between $Z$ and the context $X$, the optimal intervention in $Z$ generally varies with $X$. This makes Chain-soft a key benchmark for evaluating context-adaptive intervention strategies. In the figure, the dashed arrows $X \dashrightarrow \pi_0 \dashrightarrow Z$ indicate that the policy takes $X$ as input and determines the intervened value of $Z$.

## B  Additional Results

### B.1  Robustness Stress Tests

#### B.1.1  Graph-misspecification stress test

To go beyond a limitation statement, we added a graph-misspecification stress test. In this experiment, the true environment is unchanged: interventional outcomes and reference optima are always computed from the true benchmark SCM. Only the graph supplied to graph-using optimizers is perturbed. Thus, the experiment tests sensitivity to incorrect causal information without changing the true intervention task. We

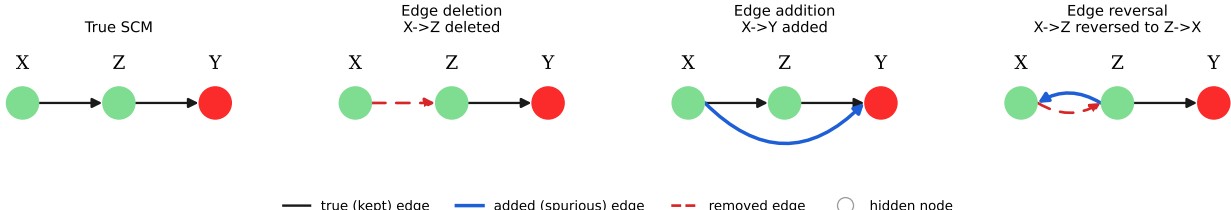

Figure 22: Perturbation taxonomy on the ToyGraph SCM ($X \to Z \to Y$). From left to right: the true benchmark graph, edge deletion ($X \to Z$ removed), edge addition (a spurious $X \to Y$ inserted), and edge reversal ($X \to Z$ flipped to $Z \to X$). Green nodes are manipulable and the red node is the target $Y$; kept edges are solid black, the spurious added edge is solid blue, and the deleted edge is dashed red. Interventional outcomes and the reference optimum $y^\star$ are always computed from the true SCM (leftmost panel); only the graph handed to the graph-using optimizers is perturbed.

consider three perturbation types: edge addition, edge deletion, and edge reversal. The results are shown in Tables 14, 15, and 16. Each perturbation setting is evaluated over 20 random seeds, matching the seed count used in the main benchmark. We present these stress tests as robustness diagnostics under controlled graph misspecification, not as a complete deployment-robustness benchmark.

To make the perturbations concrete and reproducible, we visualize them directly on the benchmark graphs. Figure 22 illustrates the three perturbation types on the canonical ToyGraph chain $X \to Z \to Y$, and Figures 23 and 24 show, for every other benchmark graph, the specific edge that is added, deleted, or reversed and—where a hidden-confounder variant is defined—the node that is removed from the learner's view. Across all panels we use a common visual encoding: green nodes are manipulable, the red node is the target $Y$, gray nodes are non-manipulable, kept true edges are solid black, spurious added edges are solid blue, deleted edges are dashed red, and a hidden node is drawn as an open dashed circle. In every panel exactly one edge (or, for the hidden-confounder variant, one node) is perturbed relative to the true SCM, so the schematics also document the minimal, single-perturbation nature of each stress test.

The stress test confirms the concern that motivated it: method rankings change substantially under graph perturbation. For example, under edge addition at 100 trials, the GAP winner varies across datasets: CoCaBO is best on ToyGraph, BO is best on Synthetic, DCBO is best on Synthetic-2, Ecology, and Epidemiology, CBO is best on Chain-hard and Healthcare, and MCBO is best on Protein-reconstructed (Table 14). Under edge deletion and edge reversal, the winners change again (Tables 15 and 16). PA-GAP also produces different rankings from GAP because it rewards full-trajectory progress rather than only final discovery time. These results support a more cautious conclusion: causal structure can help, but current CBO pipelines are sensitive to how that structure is specified and operationalized. This particular stress test addresses learner-side graph misspecification; hidden confounding is a separate dimension, which we probe with a companion omitted-variable experiment (Table 17, below). We treat comprehensive robustness to unobserved confounding as still open rather than resolved by these pilots.

Some misspecified-graph runs outperform the corresponding correct-graph run. We do not interpret this as evidence that the misspecified graph is causally preferable. Since outcomes and $y^*$ are still computed from the true SCM, such improvements reflect finite-budget search effects. A perturbed graph can weaken an overconfident observational prior, change candidate intervention scopes, alter exploration, or accidentally induce a more favorable acquisition trajectory. We therefore report these cases as evidence that CBO is sensitive to causal inductive bias, not as evidence that misspecification is beneficial.

To make the comparison interpretable, we compare each perturbed-graph score against the corresponding correct-graph score obtained with the same dataset, method, metric, and budget. Table 13 reports the resulting compact degradation summary: for each perturbation type, the fraction of graph-using method cells (6 methods × 8 datasets × 2 metrics × 3 budgets, $N{=}288$) whose mean score degrades, improves, or is unchanged relative to the correct-graph score in the main benchmark tables. Across all three perturbation

Table 13: Degradation summary for the graph-misspecification stress tests. Each cell compares a graph-using method's mean score under the perturbed graph against the corresponding correct-graph mean score in the main benchmark tables, over 6 methods $\times$ 8 datasets $\times$ 2 metrics $\times$ 3 budgets ($N$=288 per perturbation type).

| Perturbation | Degrade | Improve | Unchanged |
|---|---|---|---|
| Edge addition | 150 (52%) | 78 (27%) | 60 (21%) |
| Edge deletion | 153 (53%) | 76 (26%) | 59 (20%) |
| Edge reversal | 156 (54%) | 75 (26%) | 57 (20%) |

types, roughly half of the cells degrade (52–54%), about a quarter improve (26–27%), and about a fifth are unchanged (20–21%); CBO is the most consistently affected pipeline (34–37 of 48 cells degrade per perturbation), while HCBO improves nearly as often as it degrades. Because the stress-test pilots were executed as separate runs (visible in the shared BO control, whose scores differ slightly from the main-table BO scores), these fractions should be read as coarse sensitivity indicators rather than exact paired effect sizes. This summary prevents individual improvements under misspecification from being misread as evidence that the perturbed graph is causally better: improvements are a minority pattern and are consistent with the finite-budget search effects described above.

For methods not directly affected by the supplied fixed graph in this pilot, the perturbation does not change the optimizer. The non-causal BO baseline never reads the causal graph, and the CEO runner used in this pilot does not consume a single fixed supplied graph in the same way as the known-graph methods. We therefore treat BO and CEO as isolation controls and report a single shared baseline for each dataset–budget–metric setting, identical across the edge-addition, edge-deletion, and edge-reversal tables. Thus, cross-table differences are confined to the graph-using methods (CBO, cCBO, DCBO, MCBO, HCBO, and CoCaBO).

Table 14: Graph-misspecification stress test under edge addition (mean ± standard error over 20 seeds; illustrative pilot stress test; higher is better; best mean among methods under the same perturbation setting is in bold).

| Trial limit | Metric | Dataset | BO | CBO | cCBO | DCBO | MCBO | HCBO | CoCaBO | CEO |
|---|---|---|---|---|---|---|---|---|---|---|
| 100 | GAP | ToyGraph | 0.232±.079 | 0.288±.071 | 0.567±.001 | 0.569±.055 | 0.558±.286 | 0.509±.049 | **0.682±.109** | 0.459±.084 |
| | | Synthetic | **0.651±.065** | 0.454±.072 | 0.642±.056 | 0.337±.036 | 0.456±.082 | 0.391±.088 | 0.617±.093 | 0.396±.042 |
| | | Synthetic-2 | 0.001±.004 | 0.281±.125 | 0.289±.128 | **0.578±.025** | 0.000±.000 | 0.472±.200 | 0.122±.076 | 0.526±.183 |
| | | Chain-hard | 0.787±.103 | **0.814±.128** | 0.618±.151 | 0.333±.333 | 0.284±.041 | 0.095±.062 | 0.786±.107 | 0.347±.046 |
| | | Ecology | 0.160±.128 | 0.018±.018 | 0.426±.068 | **0.519±.001** | 0.000±.000 | 0.234±.102 | 0.000±.000 | 0.228±.051 |
| | | Protein-reconstructed | 0.331±.125 | 0.363±.139 | 0.000±.000 | 0.000±.000 | **0.615±.097** | 0.288±.091 | 0.474±.045 | 0.193±.048 |
| | | Healthcare | 0.315±.054 | **0.510±.035** | 0.509±.083 | 0.341±.093 | 0.357±.029 | 0.234±.054 | 0.257±.129 | 0.483±.086 |
| | | Epidemiology | 0.506±.056 | 0.563±.038 | 0.000±.000 | **0.599±.008** | 0.000±.000 | 0.390±.077 | 0.300±.048 | 0.218±.081 |
| | PA-GAP | ToyGraph | 0.051±.039 | 0.089±.058 | 0.073±.001 | 0.088±.061 | 0.172±.122 | 0.199±.082 | **0.320±.032** | 0.203±.099 |
| | | Synthetic | 0.214±.051 | 0.041±.096 | 0.286±.027 | 0.025±.056 | 0.113±.082 | 0.075±.093 | **0.397±.033** | 0.253±.054 |
| | | Synthetic-2 | 0.005±.002 | 0.003±.002 | 0.139±.069 | 0.089±.029 | 0.000±.000 | 0.145±.077 | 0.090±.054 | **0.155±.057** |
| | | Chain-hard | **0.497±.015** | 0.437±.005 | 0.345±.129 | 0.168±.169 | 0.077±.014 | 0.066±.051 | 0.424±.069 | 0.007±.033 |
| | | Ecology | 0.031±.011 | 0.002±.002 | 0.174±.038 | 0.178±.001 | 0.000±.000 | **0.210±.120** | 0.000±.000 | 0.007±.069 |
| | | Protein-reconstructed | **0.153±.061** | 0.057±.018 | 0.000±.000 | 0.000±.000 | 0.031±.036 | 0.000±.000 | 0.051±.024 | 0.010±.072 |
| | | Healthcare | 0.120±.021 | 0.077±.015 | **0.202±.014** | 0.000±.000 | 0.071±.051 | 0.010±.006 | 0.116±.058 | 0.138±.049 |
| | | Epidemiology | 0.242±.021 | 0.185±.059 | 0.000±.000 | **0.341±.002** | 0.000±.000 | 0.138±.080 | 0.122±.078 | 0.001±.042 |
| 50 | GAP | ToyGraph | 0.202±.099 | 0.235±.111 | 0.560±.009 | 0.545±.046 | 0.296±.296 | 0.450±.093 | **0.796±.041** | 0.489±.073 |
| | | Synthetic | 0.567±.091 | 0.000±.097 | 0.440±.048 | 0.295±.050 | 0.456±.099 | 0.391±.088 | **0.651±.028** | 0.254±.057 |
| | | Synthetic-2 | 0.007±.005 | 0.310±.033 | 0.377±.192 | **0.562±.017** | 0.000±.000 | 0.428±.217 | 0.122±.076 | 0.484±.222 |
| | | Chain-hard | 0.686±.086 | **0.889±.016** | 0.645±.198 | 0.333±.333 | 0.360±.107 | 0.095±.062 | 0.725±.143 | 0.356±.101 |
| | | Ecology | 0.259±.148 | 0.134±.134 | **0.502±.105** | 0.357±.001 | 0.000±.000 | 0.234±.102 | 0.000±.000 | 0.101±.031 |
| | | Protein-reconstructed | 0.365±.061 | 0.413±.046 | 0.000±.000 | 0.000±.000 | 0.526±.097 | **0.591±.040** | 0.392±.120 | 0.264±.028 |
| | | Healthcare | 0.439±.121 | 0.432±.073 | **0.604±.007** | 0.332±.093 | 0.357±.029 | 0.234±.054 | 0.244±.132 | 0.489±.076 |
| | | Epidemiology | 0.358±.052 | 0.355±.021 | 0.000±.000 | **0.671±.005** | 0.000±.000 | 0.390±.077 | 0.246±.079 | 0.060±.050 |
| | PA-GAP | ToyGraph | 0.024±.016 | 0.070±.053 | 0.073±.006 | 0.083±.057 | 0.108±.109 | 0.199±.082 | **0.321±.034** | 0.213±.099 |
| | | Synthetic | 0.194±.054 | 0.000±.036 | 0.201±.034 | 0.023±.050 | 0.113±.082 | 0.075±.093 | **0.344±.043** | 0.246±.077 |
| | | Synthetic-2 | 0.003±.003 | 0.003±.002 | 0.110±.056 | 0.085±.026 | 0.000±.000 | 0.081±.041 | 0.090±.054 | **0.146±.056** |
| | | Chain-hard | **0.481±.029** | 0.379±.010 | 0.327±.124 | 0.170±.170 | 0.050±.025 | 0.066±.051 | 0.391±.098 | 0.001±.054 |
| | | Ecology | 0.017±.010 | 0.002±.002 | 0.166±.040 | 0.175±.001 | 0.000±.000 | **0.208±.120** | 0.000±.000 | 0.006±.075 |
| | | Protein-reconstructed | **0.137±.054** | 0.047±.017 | 0.000±.000 | 0.000±.000 | 0.031±.036 | 0.000±.000 | 0.045±.019 | 0.009±.075 |
| | | Healthcare | 0.105±.017 | 0.069±.014 | **0.178±.010** | 0.000±.000 | 0.071±.051 | 0.009±.005 | 0.100±.051 | 0.147±.087 |
| | | Epidemiology | 0.232±.025 | 0.129±.061 | 0.000±.000 | **0.315±.003** | 0.000±.000 | 0.127±.073 | 0.085±.069 | 0.008±.002 |
| 20 | GAP | ToyGraph | 0.272±.097 | 0.167±.064 | 0.538±.000 | 0.472±.034 | 0.239±.239 | 0.450±.093 | **0.763±.057** | 0.489±.073 |
| | | Synthetic | 0.390±.138 | 0.000±.000 | 0.475±.057 | 0.295±.050 | 0.456±.099 | 0.391±.088 | **0.590±.095** | 0.254±.057 |
| | | Synthetic-2 | 0.003±.002 | 0.185±.125 | **0.515±.009** | 0.129±.129 | 0.000±.000 | 0.428±.217 | 0.000±.000 | 0.484±.222 |
| | | Chain-hard | **0.761±.124** | 0.718±.039 | 0.486±.119 | 0.333±.333 | 0.380±.129 | 0.000±.000 | 0.731±.228 | 0.356±.101 |
| | | Ecology | 0.145±.145 | 0.080±.080 | 0.422±.159 | **0.442±.000** | 0.000±.000 | 0.234±.102 | 0.000±.000 | 0.101±.031 |
| | | Protein-reconstructed | 0.370±.073 | 0.194±.091 | 0.000±.000 | 0.000±.000 | 0.526±.097 | **0.556±.044** | 0.516±.027 | 0.264±.028 |
| | | Healthcare | 0.348±.119 | 0.268±.142 | 0.413±.036 | 0.332±.093 | 0.000±.000 | 0.234±.054 | 0.198±.100 | **0.489±.076** |
| | | Epidemiology | **0.694±.042** | 0.244±.122 | 0.000±.000 | 0.373±.005 | 0.000±.000 | 0.390±.077 | 0.259±.165 | 0.001±.005 |
| | PA-GAP | ToyGraph | 0.006±.004 | 0.042±.040 | 0.070±.000 | 0.069±.047 | 0.044±.044 | 0.199±.082 | **0.324±.039** | 0.213±.099 |
| | | Synthetic | 0.156±.059 | 0.000±.000 | 0.154±.036 | 0.023±.050 | 0.113±.082 | 0.000±.000 | 0.244±.054 | **0.246±.077** |
| | | Synthetic-2 | 0.004±.001 | 0.057±.029 | 0.075±.017 | 0.001±.001 | 0.000±.000 | 0.081±.041 | 0.000±.000 | **0.146±.056** |
| | | Chain-hard | **0.463±.062** | 0.228±.019 | 0.275±.112 | 0.175±.175 | 0.036±.022 | 0.000±.000 | 0.378±.108 | 0.001±.054 |
| | | Ecology | 0.011±.011 | 0.001±.001 | 0.150±.047 | 0.168±.001 | 0.000±.000 | **0.208±.120** | 0.000±.000 | 0.006±.075 |
| | | Protein-reconstructed | 0.113±.049 | 0.025±.013 | 0.000±.000 | **0.193±.097** | 0.031±.036 | 0.000±.000 | 0.045±.020 | 0.009±.075 |
| | | Healthcare | 0.091±.015 | 0.056±.011 | 0.119±.005 | 0.000±.000 | 0.000±.000 | 0.009±.005 | 0.074±.040 | **0.147±.087** |
| | | Epidemiology | 0.228±.028 | 0.087±.045 | 0.000±.000 | **0.279±.005** | 0.000±.000 | 0.127±.073 | 0.063±.060 | 0.007±.001 |

Table 15: Graph-misspecification stress test under edge deletion (mean ± standard error over 20 seeds; illustrative pilot stress test; higher is better; best mean among methods under the same perturbation setting is in bold).

| Trial limit | Metric | Dataset | BO | CBO | cCBO | DCBO | MCBO | HCBO | CoCaBO | CEO |
|---|---|---|---|---|---|---|---|---|---|---|
| 100 | GAP | ToyGraph | 0.232±.079 | 0.392±.009 | **0.826±.001** | 0.564±.060 | 0.394±.200 | 0.234±.054 | 0.335±.096 | 0.459±.084 |
| | | Synthetic | 0.651±.065 | 0.355±.184 | 0.642±.056 | 0.337±.036 | 0.456±.082 | 0.391±.088 | **0.663±.020** | 0.396±.042 |
| | | Synthetic-2 | 0.001±.004 | 0.281±.125 | 0.306±.145 | 0.000±.004 | 0.000±.000 | **0.657±.139** | 0.122±.076 | 0.526±.183 |
| | | Chain-hard | **0.787±.103** | 0.700±.128 | 0.767±.060 | 0.448±.300 | 0.428±.085 | 0.095±.062 | 0.688±.136 | 0.347±.046 |
| | | Ecology | 0.160±.128 | 0.012±.012 | **0.426±.068** | 0.255±.008 | 0.000±.000 | 0.234±.102 | 0.000±.000 | 0.228±.051 |
| | | Protein-reconstructed | 0.331±.125 | 0.363±.139 | 0.000±.000 | 0.000±.000 | **0.615±.097** | 0.288±.091 | 0.578±.082 | 0.193±.048 |
| | | Healthcare | 0.315±.054 | **0.510±.035** | 0.445±.100 | 0.341±.093 | 0.357±.029 | 0.234±.054 | 0.431±.216 | 0.483±.086 |
| | | Epidemiology | 0.506±.056 | 0.563±.038 | 0.000±.000 | **0.661±.092** | 0.000±.000 | 0.390±.077 | 0.572±.134 | 0.218±.081 |
| | PA-GAP | ToyGraph | 0.051±.039 | 0.104±.054 | **0.355±.001** | 0.095±.056 | 0.248±.129 | 0.199±.082 | 0.103±.051 | 0.203±.099 |
| | | Synthetic | 0.214±.051 | 0.087±.044 | 0.286±.027 | 0.025±.056 | 0.113±.082 | 0.075±.093 | **0.396±.039** | 0.253±.054 |
| | | Synthetic-2 | 0.005±.002 | 0.003±.002 | 0.133±.069 | 0.000±.006 | 0.000±.000 | **0.178±.131** | 0.090±.054 | 0.155±.057 |
| | | Chain-hard | **0.497±.015** | 0.437±.005 | 0.414±.030 | 0.221±.147 | 0.085±.029 | 0.066±.051 | 0.388±.098 | 0.007±.033 |
| | | Ecology | 0.031±.011 | 0.000±.005 | 0.174±.038 | 0.041±.008 | 0.000±.000 | **0.210±.120** | 0.000±.000 | 0.007±.069 |
| | | Protein-reconstructed | **0.153±.061** | 0.057±.018 | 0.000±.000 | 0.000±.000 | 0.031±.036 | 0.000±.000 | 0.094±.077 | 0.010±.072 |
| | | Healthcare | 0.120±.021 | 0.077±.015 | **0.158±.052** | 0.000±.000 | 0.071±.051 | 0.010±.006 | 0.134±.068 | 0.138±.049 |
| | | Epidemiology | 0.242±.021 | 0.185±.059 | 0.000±.000 | **0.273±.038** | 0.000±.000 | 0.138±.080 | 0.227±.087 | 0.001±.042 |
| 50 | GAP | ToyGraph | 0.202±.099 | 0.363±.141 | **0.780±.009** | 0.525±.062 | 0.569±.288 | 0.450±.093 | 0.195±.059 | 0.489±.073 |
| | | Synthetic | 0.567±.091 | 0.271±.177 | 0.440±.048 | 0.295±.050 | 0.456±.099 | 0.391±.088 | **0.688±.082** | 0.254±.057 |
| | | Synthetic-2 | 0.007±.005 | 0.310±.033 | 0.229±.118 | 0.000±.002 | 0.000±.000 | 0.467±.251 | 0.122±.076 | **0.484±.222** |
| | | Chain-hard | 0.686±.086 | **0.889±.016** | 0.741±.151 | 0.448±.300 | 0.346±.109 | 0.095±.062 | 0.601±.165 | 0.356±.101 |
| | | Ecology | 0.259±.148 | 0.000±.005 | **0.502±.105** | 0.158±.008 | 0.000±.000 | 0.234±.102 | 0.000±.000 | 0.101±.031 |
| | | Protein-reconstructed | 0.365±.061 | 0.413±.046 | 0.000±.000 | 0.000±.000 | 0.526±.097 | **0.591±.040** | 0.562±.088 | 0.264±.028 |
| | | Healthcare | 0.439±.121 | 0.432±.073 | **0.516±.102** | 0.332±.093 | 0.357±.029 | 0.234±.054 | 0.380±.197 | 0.489±.076 |
| | | Epidemiology | 0.358±.052 | 0.355±.021 | 0.000±.000 | **0.687±.032** | 0.000±.000 | 0.390±.077 | 0.482±.140 | 0.060±.050 |
| | PA-GAP | ToyGraph | 0.024±.016 | 0.093±.045 | **0.338±.006** | 0.088±.054 | 0.180±.110 | 0.199±.082 | 0.074±.036 | 0.213±.099 |
| | | Synthetic | 0.194±.054 | 0.076±.038 | 0.201±.034 | 0.023±.050 | 0.113±.082 | 0.075±.093 | **0.346±.055** | 0.246±.077 |
| | | Synthetic-2 | 0.003±.003 | 0.003±.002 | 0.104±.058 | 0.000±.005 | 0.000±.000 | **0.158±.137** | 0.090±.054 | 0.146±.056 |
| | | Chain-hard | **0.481±.029** | 0.379±.010 | 0.387±.030 | 0.222±.148 | 0.062±.030 | 0.066±.051 | 0.380±.099 | 0.001±.054 |
| | | Ecology | 0.017±.010 | 0.000±.002 | 0.166±.040 | 0.031±.010 | 0.000±.000 | **0.210±.120** | 0.000±.000 | 0.006±.075 |
| | | Protein-reconstructed | **0.137±.054** | 0.047±.017 | 0.000±.000 | 0.000±.000 | 0.031±.036 | 0.000±.000 | 0.095±.078 | 0.009±.075 |
| | | Healthcare | 0.105±.017 | 0.069±.014 | 0.141±.044 | 0.000±.000 | 0.071±.051 | 0.009±.005 | 0.122±.061 | **0.147±.087** |
| | | Epidemiology | 0.232±.025 | 0.129±.061 | 0.000±.000 | **0.258±.033** | 0.000±.000 | 0.127±.073 | 0.196±.085 | 0.008±.002 |
| 20 | GAP | ToyGraph | 0.272±.097 | 0.375±.145 | **0.639±.000** | 0.405±.074 | 0.274±.274 | 0.450±.093 | 0.097±.039 | 0.489±.073 |
| | | Synthetic | 0.390±.138 | 0.276±.163 | 0.475±.057 | 0.295±.050 | 0.456±.099 | 0.391±.088 | **0.511±.058** | 0.254±.057 |
| | | Synthetic-2 | 0.003±.002 | 0.245±.125 | 0.000±.000 | **0.507±.026** | 0.000±.000 | 0.467±.251 | 0.000±.000 | 0.484±.222 |
| | | Chain-hard | 0.761±.124 | 0.718±.039 | **0.778±.019** | 0.318±.318 | 0.250±.077 | 0.000±.000 | 0.671±.202 | 0.356±.101 |
| | | Ecology | 0.145±.145 | 0.000±.000 | **0.422±.159** | 0.221±.147 | 0.000±.000 | 0.234±.102 | 0.000±.000 | 0.101±.031 |
| | | Protein-reconstructed | 0.370±.073 | 0.194±.091 | 0.000±.000 | 0.000±.000 | 0.526±.097 | **0.556±.044** | 0.515±.109 | 0.264±.028 |
| | | Healthcare | 0.348±.119 | 0.268±.142 | 0.413±.036 | 0.332±.093 | 0.000±.000 | 0.234±.054 | 0.317±.180 | **0.489±.076** |
| | | Epidemiology | **0.694±.042** | 0.244±.122 | 0.000±.000 | 0.514±.079 | 0.000±.000 | 0.390±.077 | 0.483±.102 | 0.001±.005 |
| | PA-GAP | ToyGraph | 0.006±.004 | 0.066±.025 | **0.296±.000** | 0.068±.047 | 0.076±.076 | 0.199±.082 | 0.047±.034 | 0.213±.099 |
| | | Synthetic | 0.156±.059 | 0.063±.034 | 0.154±.036 | 0.023±.050 | 0.113±.082 | 0.000±.000 | **0.286±.062** | 0.246±.077 |
| | | Synthetic-2 | 0.004±.001 | 0.068±.037 | 0.000±.000 | **0.158±.017** | 0.000±.000 | **0.158±.137** | 0.000±.000 | 0.146±.056 |
| | | Chain-hard | **0.463±.062** | 0.228±.019 | 0.312±.031 | 0.160±.160 | 0.029±.017 | 0.000±.000 | 0.368±.102 | 0.001±.054 |
| | | Ecology | 0.011±.011 | 0.000±.000 | 0.150±.047 | 0.017±.016 | 0.000±.000 | **0.210±.120** | 0.000±.000 | 0.006±.075 |
| | | Protein-reconstructed | 0.113±.049 | 0.025±.013 | 0.000±.000 | **0.193±.097** | 0.031±.036 | 0.000±.000 | 0.096±.080 | 0.009±.075 |
| | | Healthcare | 0.091±.015 | 0.056±.011 | 0.119±.005 | 0.000±.000 | 0.000±.000 | 0.009±.005 | 0.102±.052 | **0.147±.087** |
| | | Epidemiology | 0.228±.028 | 0.087±.045 | 0.000±.000 | **0.241±.013** | 0.000±.000 | 0.127±.073 | 0.143±.067 | 0.007±.001 |

Table 16: Graph-misspecification stress test under edge reversal (mean ± standard error over 20 seeds; illustrative pilot stress test; higher is better; best mean among methods under the same perturbation setting is in bold).

| Trial limit | Metric | Dataset | BO | CBO | cCBO | DCBO | MCBO | HCBO | CoCaBO | CEO |
|---|---|---|---|---|---|---|---|---|---|---|
| 100 | GAP | ToyGraph | 0.232±.079 | 0.333±.118 | **0.826±.001** | 0.564±.060 | 0.513±.257 | 0.358±.054 | 0.279±.074 | 0.459±.084 |
| | | Synthetic | **0.651±.065** | 0.340±.084 | 0.548±.015 | 0.337±.036 | 0.456±.082 | 0.391±.088 | 0.589±.042 | 0.396±.042 |
| | | Synthetic-2 | 0.001±.004 | 0.281±.125 | 0.335±.114 | 0.000±.004 | 0.000±.000 | **0.757±.012** | 0.122±.076 | 0.526±.183 |
| | | Chain-hard | 0.787±.103 | **0.874±.051** | 0.632±.118 | 0.295±.050 | 0.409±.093 | 0.095±.062 | 0.780±.071 | 0.347±.046 |
| | | Ecology | 0.160±.128 | 0.012±.012 | **0.426±.068** | 0.255±.008 | 0.000±.000 | 0.234±.102 | 0.000±.000 | 0.228±.051 |
| | | Protein-reconstructed | 0.331±.125 | 0.363±.139 | 0.000±.000 | 0.000±.000 | **0.615±.097** | 0.288±.091 | 0.423±.096 | 0.193±.048 |
| | | Healthcare | 0.315±.054 | **0.561±.004** | 0.509±.083 | 0.341±.093 | 0.357±.029 | 0.234±.054 | 0.432±.104 | 0.483±.086 |
| | | Epidemiology | 0.506±.056 | **0.533±.134** | 0.000±.000 | 0.456±.029 | 0.000±.000 | 0.390±.077 | 0.464±.200 | 0.218±.081 |
| | PA-GAP | ToyGraph | 0.051±.039 | 0.102±.054 | **0.355±.001** | 0.095±.056 | 0.271±.141 | 0.199±.082 | 0.139±.079 | 0.203±.099 |
| | | Synthetic | 0.214±.051 | 0.141±.036 | 0.257±.045 | 0.025±.056 | 0.113±.082 | 0.075±.093 | **0.376±.027** | 0.253±.054 |
| | | Synthetic-2 | 0.005±.002 | 0.003±.002 | 0.139±.070 | 0.000±.006 | 0.000±.000 | **0.213±.043** | 0.090±.054 | 0.155±.057 |
| | | Chain-hard | **0.497±.015** | 0.434±.006 | 0.325±.099 | 0.087±.050 | 0.087±.001 | 0.066±.051 | 0.454±.043 | 0.007±.033 |
| | | Ecology | 0.031±.011 | 0.000±.005 | 0.174±.038 | 0.041±.008 | 0.000±.000 | **0.210±.120** | 0.000±.000 | 0.007±.069 |
| | | Protein-reconstructed | **0.153±.061** | 0.057±.018 | 0.000±.000 | 0.000±.000 | 0.031±.036 | 0.000±.000 | 0.018±.009 | 0.010±.072 |
| | | Healthcare | 0.120±.021 | 0.090±.014 | **0.202±.014** | 0.000±.000 | 0.071±.051 | 0.010±.006 | 0.196±.010 | 0.138±.049 |
| | | Epidemiology | 0.242±.021 | 0.173±.077 | 0.000±.000 | 0.065±.074 | 0.000±.000 | 0.138±.080 | **0.251±.094** | 0.001±.042 |
| 50 | GAP | ToyGraph | 0.202±.099 | 0.477±.097 | **0.780±.009** | 0.525±.062 | 0.411±.206 | 0.450±.093 | 0.276±.075 | 0.489±.073 |
| | | Synthetic | 0.567±.091 | 0.534±.087 | 0.425±.086 | 0.295±.050 | 0.456±.099 | 0.391±.088 | **0.569±.118** | 0.254±.057 |
| | | Synthetic-2 | 0.007±.005 | 0.310±.033 | 0.243±.127 | 0.000±.002 | 0.000±.000 | **0.536±.099** | 0.122±.076 | 0.484±.222 |
| | | Chain-hard | 0.686±.086 | **0.747±.102** | 0.411±.128 | 0.295±.050 | 0.383±.096 | 0.095±.062 | 0.744±.062 | 0.356±.101 |
| | | Ecology | 0.259±.148 | 0.000±.005 | **0.502±.105** | 0.158±.008 | 0.000±.000 | 0.234±.102 | 0.000±.000 | 0.101±.031 |
| | | Protein-reconstructed | 0.365±.061 | 0.413±.046 | 0.000±.000 | 0.000±.000 | 0.526±.097 | **0.591±.040** | 0.342±.170 | 0.264±.028 |
| | | Healthcare | 0.439±.121 | 0.522±.020 | **0.604±.007** | 0.332±.093 | 0.357±.029 | 0.234±.054 | 0.342±.077 | 0.489±.076 |
| | | Epidemiology | 0.358±.052 | 0.450±.170 | 0.000±.000 | 0.456±.039 | 0.000±.000 | 0.390±.077 | **0.624±.150** | 0.060±.050 |
| | PA-GAP | ToyGraph | 0.024±.016 | 0.089±.047 | **0.338±.006** | 0.088±.054 | 0.225±.134 | 0.199±.082 | 0.131±.073 | 0.213±.099 |
| | | Synthetic | 0.194±.054 | 0.123±.034 | 0.214±.051 | 0.023±.050 | 0.113±.082 | 0.075±.093 | **0.328±.036** | 0.246±.077 |
| | | Synthetic-2 | 0.003±.003 | 0.003±.002 | 0.110±.059 | 0.000±.005 | 0.000±.000 | 0.104±.055 | 0.090±.054 | **0.146±.056** |
| | | Chain-hard | **0.481±.029** | 0.373±.012 | 0.298±.088 | 0.087±.050 | 0.060±.010 | 0.066±.051 | 0.445±.051 | 0.001±.054 |
| | | Ecology | 0.017±.010 | 0.000±.002 | 0.166±.040 | 0.031±.010 | 0.000±.000 | **0.210±.120** | 0.000±.000 | 0.006±.075 |
| | | Protein-reconstructed | **0.137±.054** | 0.047±.017 | 0.000±.000 | 0.000±.000 | 0.031±.036 | 0.000±.000 | 0.018±.009 | 0.009±.075 |
| | | Healthcare | 0.105±.017 | 0.082±.010 | 0.178±.010 | 0.000±.000 | 0.071±.051 | 0.010±.006 | **0.180±.008** | 0.147±.087 |
| | | Epidemiology | **0.232±.025** | 0.150±.083 | 0.000±.000 | 0.065±.098 | 0.000±.000 | 0.138±.080 | 0.225±.093 | 0.008±.002 |
| 20 | GAP | ToyGraph | 0.272±.097 | 0.264±.152 | **0.639±.000** | 0.405±.074 | 0.496±.275 | 0.450±.093 | 0.167±.073 | 0.489±.073 |
| | | Synthetic | 0.390±.138 | 0.350±.208 | **0.518±.156** | 0.295±.050 | 0.456±.099 | 0.391±.088 | 0.506±.067 | 0.254±.057 |
| | | Synthetic-2 | 0.003±.002 | 0.313±.163 | 0.000±.000 | 0.507±.026 | 0.000±.000 | **0.536±.099** | 0.000±.000 | 0.484±.222 |
| | | Chain-hard | 0.761±.124 | 0.650±.090 | 0.691±.051 | 0.866±.021 | 0.249±.090 | 0.000±.000 | **0.905±.066** | 0.356±.101 |
| | | Ecology | 0.145±.145 | 0.000±.000 | **0.422±.159** | 0.221±.147 | 0.000±.000 | 0.234±.102 | 0.000±.000 | 0.101±.031 |
| | | Protein-reconstructed | 0.370±.073 | 0.194±.091 | 0.000±.000 | 0.000±.000 | 0.526±.097 | **0.556±.044** | 0.326±.163 | 0.264±.028 |
| | | Healthcare | 0.348±.119 | 0.401±.079 | 0.413±.036 | 0.332±.093 | 0.000±.000 | 0.234±.054 | 0.407±.108 | **0.489±.076** |
| | | Epidemiology | **0.694±.042** | 0.378±.226 | 0.000±.000 | 0.105±.053 | 0.000±.000 | 0.390±.077 | 0.541±.079 | 0.001±.005 |
| | PA-GAP | ToyGraph | 0.006±.004 | 0.056±.028 | **0.296±.000** | 0.068±.047 | 0.161±.157 | 0.199±.082 | 0.113±.059 | 0.213±.099 |
| | | Synthetic | 0.156±.059 | 0.088±.038 | 0.174±.061 | 0.023±.050 | 0.113±.082 | 0.000±.000 | **0.259±.047** | 0.246±.077 |
| | | Synthetic-2 | 0.004±.001 | 0.065±.036 | 0.000±.000 | **0.158±.017** | 0.000±.000 | 0.104±.055 | 0.000±.000 | 0.146±.056 |
| | | Chain-hard | **0.463±.062** | 0.218±.023 | 0.231±.057 | 0.414±.021 | 0.024±.020 | 0.000±.000 | 0.426±.069 | 0.001±.054 |
| | | Ecology | 0.011±.011 | 0.000±.000 | 0.150±.047 | 0.017±.016 | 0.000±.000 | **0.210±.120** | 0.000±.000 | 0.006±.075 |
| | | Protein-reconstructed | 0.113±.049 | 0.025±.013 | 0.000±.000 | **0.193±.097** | 0.031±.036 | 0.000±.000 | 0.017±.009 | 0.009±.075 |
| | | Healthcare | 0.091±.015 | 0.064±.006 | 0.119±.005 | 0.000±.000 | 0.000±.000 | 0.010±.006 | **0.163±.005** | 0.147±.087 |
| | | Epidemiology | **0.228±.028** | 0.139±.087 | 0.000±.000 | 0.003±.001 | 0.000±.000 | 0.138±.080 | 0.176±.082 | 0.007±.001 |

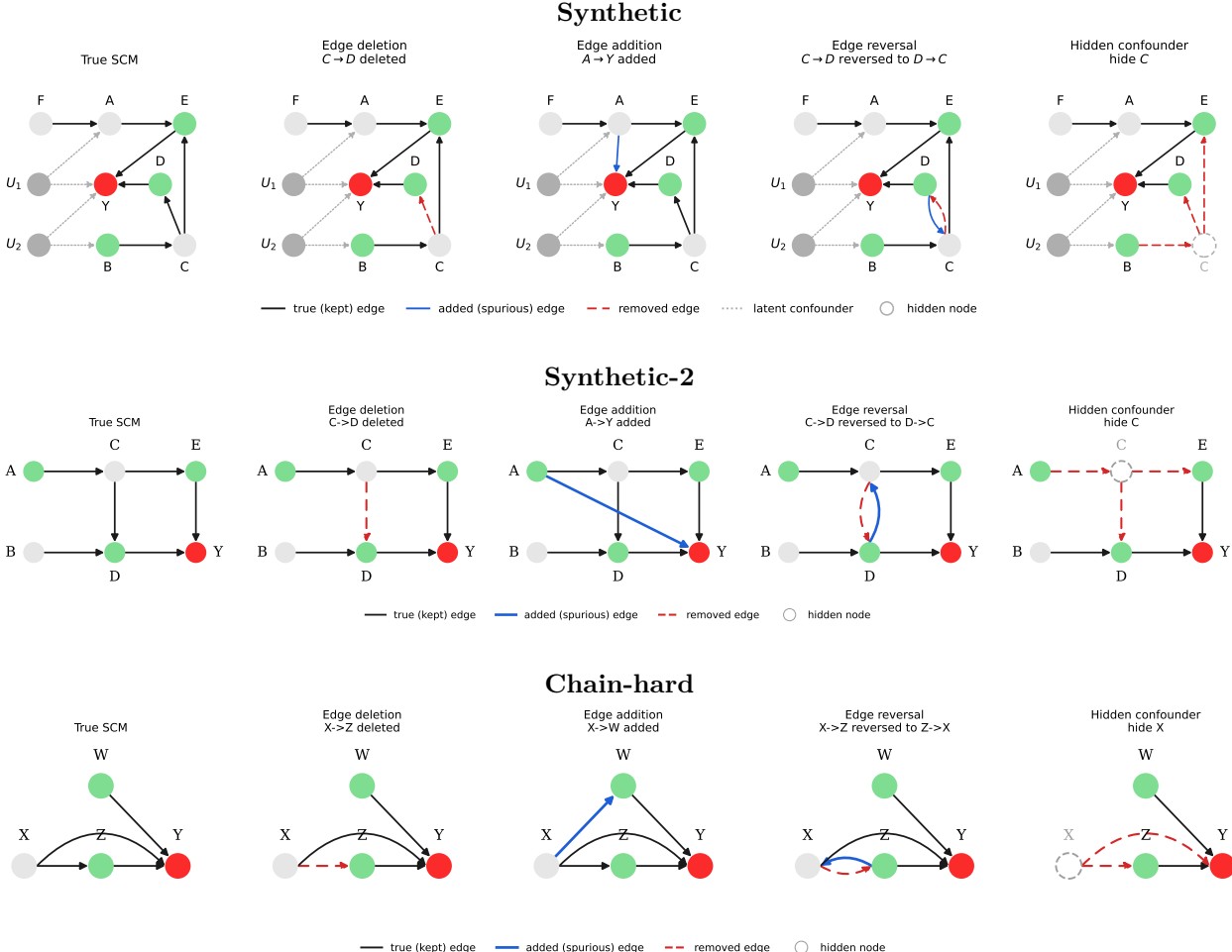

Figure 23: Graph-misspecification schematics for the domain-inspired SCMs (Protein-reconstructed, Ecology, Healthcare, and Epidemiology). Layout and visual encoding follow Figure 23. Protein-reconstructed uses the three edge perturbations PKC→PKA deletion, PKC→Erk addition, and PKC→PKA reversal. Ecology uses TA→ $\Omega_A$ deletion, Sal→NEC addition, TA→ $\Omega_A$ reversal, and hide Tem; Healthcare uses BMI→aspirin deletion, aspirin→statin addition, BMI→aspirin reversal, and hide BMI; Epidemiology uses $T \to L$ deletion, $B \to R$ addition, $T \to L$ reversal, and hide $T$. The Healthcare hide-BMI and Epidemiology hide-$T$ panels are the omitted-common-cause settings in Table 17; Ecology is illustrative. Only the learner's graph is perturbed; the true SCM and $y^\star$ are unchanged.

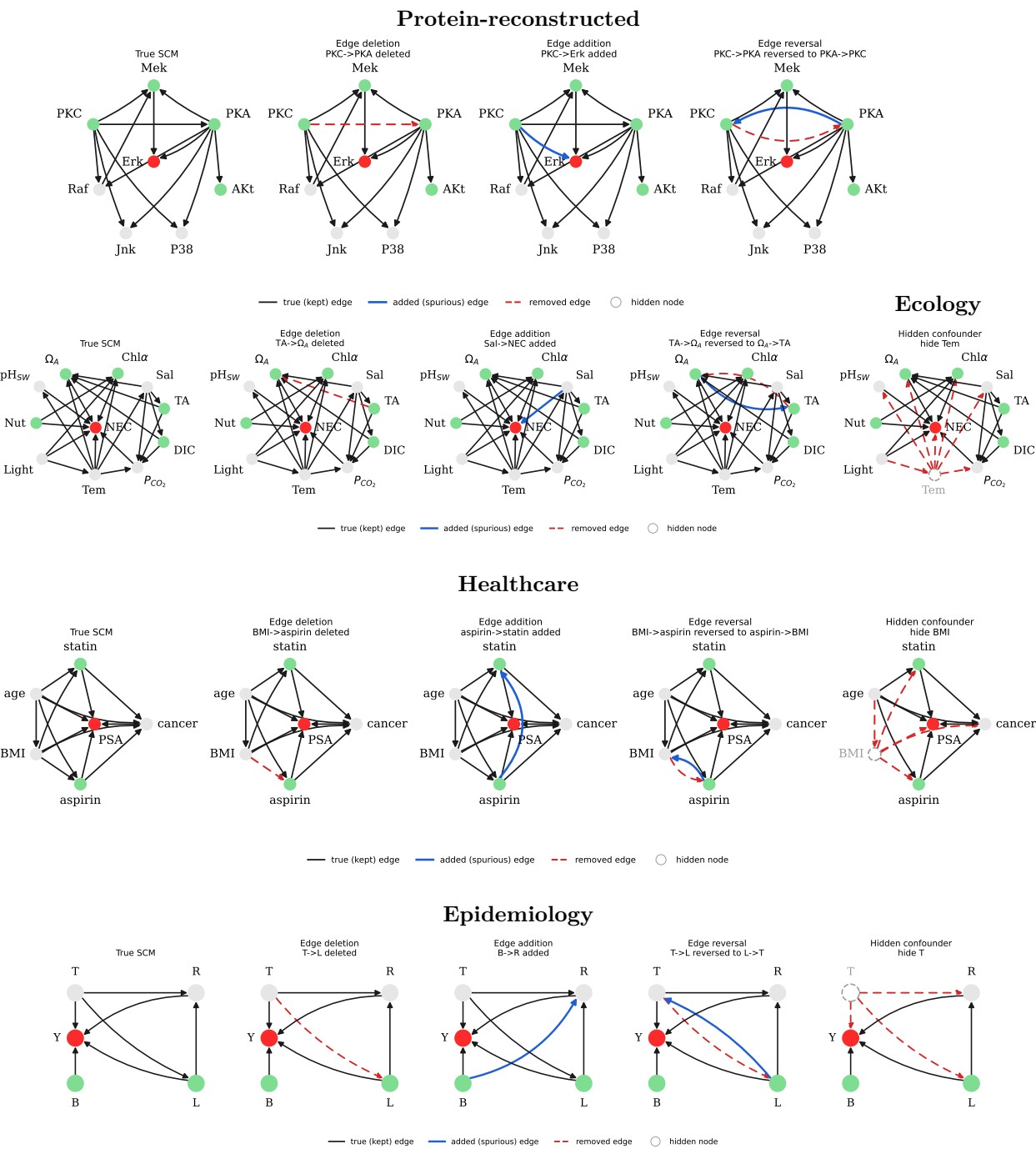

Figure 24: Graph-misspecification schematics for Protein-reconstructed and the domain-inspired benchmark SCMs (Ecology, Healthcare, and Epidemiology). Layout and visual encoding follow Figure 23. Each row shows the true SCM followed by learner-belief graphs under the misspecification perturbations shown in the panel titles. Protein-reconstructed uses only the three edge perturbations: PKC→PKA deletion, PKC→Erk addition, and PKC→PKA reversal to PKA→PKC. For Ecology, the perturbations are TA→ $\Omega_A$ deletion, Sal→NEC addition, TA→ $\Omega_A$ reversal to $\Omega_A$ →TA, and hiding Tem. For Healthcare, the perturbations are BMI→aspirin deletion, aspirin→statin addition, BMI→aspirin reversal to aspirin→BMI, and hiding BMI. For Epidemiology, the perturbations are $T \to L$ deletion, $B \to R$ addition, $T \to L$ reversal to $L \to T$, and hiding $T$. The Ecology schematic is illustrative and uses representative perturbations rather than a run-pilot setting. The Healthcare hide-BMI and Epidemiology hide-$T$ panels are the omitted common causes used in the omitted-variable stress test of Table 17. Only the learner's supplied graph is perturbed; the true SCM and $y^\star$ are unchanged.

### B.1.2 Hidden confounding (omitted-variable stress test).

Hidden confounding is a distinct assumption violation, and the graph-perturbation study above does not address it. We therefore added a companion omitted-variable stress test (Table 17). Here the learner's graph and observational view omit a genuine common cause—BMI in Healthcare, $X$ in Chain-hard, and $T$ in Epidemiology—while the true SCM and reference optimum $y^\star$ are held fixed, so the comparison isolates the effect of an unobserved confounder rather than changing the task. This omitted-variable stress test is evaluated over 20 random seeds. We currently include BO as a graph-free control and CBO/CoCaBO as belief-graph methods.

The effect is method- and dataset-dependent. Omitting the confounder degrades CBO on Chain-hard, whereas on Healthcare and some Epidemiology settings the omitted-variable runs of CBO and CoCaBO sometimes match or slightly exceed their full-information counterparts. We do not interpret these numerical increases as evidence that confounding is beneficial or harmless. Rather, they reflect finite-budget search effects: a coarser or misspecified observational view can weaken an overconfident causal prior, change candidate scopes, or accidentally induce a more favorable acquisition trajectory. The graph-free BO baseline is unchanged by construction, since it does not use the omitted causal structure. Overall, the omitted-variable stress test reinforces the same message as the graph-perturbation study: current CBO pipelines are sensitive to the causal information they are given, and robustness to hidden confounding remains an open evaluation dimension.

Table 17: Hidden-confounding / omitted-variable stress test (mean $\pm$ standard error over 20 seeds; higher is better; best mean per row in bold). The learner's graph and observational view omit a genuine common cause (Healthcare: BMI; Chain-hard: $X$; Epidemiology: $T$); the true SCM and $y^\star$ are unchanged. BO is graph-free and uses the full variable set, so its Full and Hidden rows are identical by construction.

| Trial limit | Metric | Dataset | Information | BO | CBO | CoCaBO |
|---|---|---|---|---|---|---|
| 100 | GAP | Healthcare | Full information | 0.309±.057 | **0.510±.035** | 0.226±.118 |
| | | | Hidden variable | 0.309±.057 | **0.543±.012** | 0.251±.126 |
| | | Chain-hard | Full information | 0.780±.102 | **0.814±.128** | 0.761±.129 |
| | | | Hidden variable | **0.780±.102** | 0.695±.106 | 0.770±.139 |
| | | Epidemiology | Full information | 0.508±.057 | **0.563±.038** | 0.424±.178 |
| | | | Hidden variable | 0.508±.057 | **0.546±.116** | 0.399±.094 |
| | PA-GAP | Healthcare | Full information | 0.122±.018 | 0.077±.015 | **0.125±.062** |
| | | | Hidden variable | 0.122±.018 | 0.080±.006 | **0.139±.070** |
| | | Chain-hard | Full information | **0.492±.013** | 0.437±.005 | 0.486±.008 |
| | | | Hidden variable | **0.492±.013** | 0.420±.010 | 0.479±.008 |
| | | Epidemiology | Full information | **0.244±.023** | 0.185±.059 | 0.163±.080 |
| | | | Hidden variable | **0.244±.023** | 0.182±.057 | 0.130±.091 |
| 50 | GAP | Healthcare | Full information | **0.438±.124** | 0.432±.073 | 0.225±.128 |
| | | | Hidden variable | 0.438±.124 | **0.502±.020** | 0.378±.196 |
| | | Chain-hard | Full information | 0.680±.087 | **0.889±.016** | 0.815±.146 |
| | | | Hidden variable | 0.680±.087 | 0.811±.054 | **0.873±.102** |
| | | Epidemiology | Full information | 0.357±.053 | 0.355±.021 | **0.565±.076** |
| | | | Hidden variable | 0.357±.053 | **0.590±.103** | 0.344±.188 |
| | PA-GAP | Healthcare | Full information | 0.111±.017 | 0.069±.014 | **0.114±.057** |
| | | | Hidden variable | 0.111±.017 | 0.076±.005 | **0.131±.066** |
| | | Chain-hard | Full information | **0.484±.026** | 0.379±.010 | 0.474±.016 |
| | | | Hidden variable | **0.484±.026** | 0.348±.018 | 0.468±.013 |
| | | Epidemiology | Full information | **0.230±.022** | 0.129±.061 | 0.141±.077 |
| | | | Hidden variable | **0.230±.022** | 0.162±.065 | 0.102±.084 |
| 20 | GAP | Healthcare | Full information | **0.348±.120** | 0.268±.142 | 0.328±.180 |
| | | | Hidden variable | 0.348±.120 | **0.376±.048** | 0.250±.159 |
| | | Chain-hard | Full information | 0.761±.124 | 0.718±.039 | **0.913±.047** |
| | | | Hidden variable | 0.761±.124 | 0.615±.090 | **0.938±.023** |
| | | Epidemiology | Full information | **0.694±.042** | 0.244±.122 | 0.389±.071 |
| | | | Hidden variable | **0.694±.042** | 0.429±.181 | 0.156±.156 |
| | PA-GAP | Healthcare | Full information | 0.091±.015 | 0.056±.011 | **0.100±.051** |
| | | | Hidden variable | 0.091±.015 | 0.068±.003 | **0.112±.057** |
| | | Chain-hard | Full information | **0.463±.062** | 0.228±.020 | 0.449±.038 |
| | | | Hidden variable | **0.463±.062** | 0.178±.029 | 0.437±.033 |
| | | Epidemiology | Full information | **0.228±.028** | 0.087±.045 | 0.104±.069 |
| | | | Hidden variable | **0.228±.028** | 0.128±.080 | 0.068±.068 |

Table 18: MCBO GAP and PA-GAP vs. UCB $\beta$ at $T \in \{100, 50, 20\}$, $\beta$=10 columns are the `main_hard` values (the parameter used in the paper); $\beta \in \{0.5, 1, 2, 5\}$ are 20-seed sweep re-runs (100-trial trajectory truncated to $T$). Higher is better; best score per row in bold.

| Metric | $T$ | Dataset | $\beta$=0.5 | $\beta$=1 | $\beta$=2 | $\beta$=5 | $\beta$=10 |
|---|---|---|---|---|---|---|---|
| GAP | 100 | ToyGraph | **0.748** | 0.381 | 0.746 | 0.357 | 0.547 |
| | | Synthetic | 0.659 | **0.686** | 0.684 | 0.681 | 0.652 |
| | | Synthetic-2 | 0.000 | 0.000 | 0.000 | 0.000 | **0.000** |
| | | Chain-hard | 0.516 | **0.554** | 0.435 | 0.423 | 0.502 |
| | | Ecology | 0.000 | 0.000 | 0.000 | 0.000 | **0.000** |
| | | Protein-reconstructed | 0.419 | 0.337 | 0.109 | 0.286 | **0.878** |
| | | Healthcare | 0.485 | 0.524 | **0.545** | 0.519 | 0.510 |
| | | Epidemiology | 0.000 | 0.000 | 0.000 | 0.000 | **0.000** |
| | 50 | ToyGraph | 0.545 | **0.576** | 0.492 | 0.253 | 0.513 |
| | | Synthetic | 0.605 | 0.606 | **0.651** | 0.587 | 0.630 |
| | | Synthetic-2 | 0.000 | 0.000 | 0.000 | 0.000 | **0.000** |
| | | Chain-hard | 0.392 | **0.540** | 0.381 | 0.367 | 0.215 |
| | | Ecology | 0.000 | 0.000 | 0.000 | 0.000 | **0.000** |
| | | Protein-reconstructed | 0.354 | 0.348 | 0.264 | 0.144 | **0.752** |
| | | Healthcare | 0.303 | 0.292 | 0.275 | **0.330** | 0.315 |
| | | Epidemiology | 0.000 | 0.000 | 0.000 | 0.000 | **0.000** |
| | 20 | ToyGraph | 0.333 | **0.530** | 0.282 | 0.171 | 0.406 |
| | | Synthetic | 0.523 | 0.574 | 0.573 | **0.578** | 0.558 |
| | | Synthetic-2 | 0.000 | 0.000 | 0.000 | 0.000 | **0.000** |
| | | Chain-hard | 0.420 | **0.432** | 0.422 | 0.207 | 0.360 |
| | | Ecology | 0.000 | 0.000 | 0.000 | 0.000 | **0.000** |
| | | Protein-reconstructed | 0.393 | 0.300 | 0.140 | 0.399 | **0.540** |
| | | Healthcare | 0.000 | 0.000 | 0.000 | 0.000 | **0.000** |
| | | Epidemiology | 0.000 | 0.000 | 0.000 | 0.000 | **0.000** |
| PA-GAP | 100 | ToyGraph | 0.266 | **0.295** | 0.218 | 0.127 | 0.071 |
| | | Synthetic | 0.149 | 0.154 | 0.136 | 0.125 | **0.162** |
| | | Synthetic-2 | 0.000 | 0.000 | 0.000 | 0.000 | **0.000** |
| | | Chain-hard | 0.112 | **0.174** | 0.142 | 0.067 | 0.111 |
| | | Ecology | 0.000 | 0.000 | 0.000 | 0.000 | **0.000** |
| | | Protein-reconstructed | 0.033 | 0.034 | 0.011 | **0.047** | 0.045 |
| | | Healthcare | 0.127 | 0.097 | 0.053 | **0.136** | 0.100 |
| | | Epidemiology | 0.000 | 0.000 | 0.000 | 0.000 | **0.000** |
| | 50 | ToyGraph | 0.186 | **0.261** | 0.136 | 0.089 | 0.063 |
| | | Synthetic | 0.162 | 0.165 | 0.123 | **0.186** | 0.150 |
| | | Synthetic-2 | 0.000 | 0.000 | 0.000 | 0.000 | **0.000** |
| | | Chain-hard | 0.089 | **0.152** | 0.119 | 0.035 | 0.078 |
| | | Ecology | 0.000 | 0.000 | 0.000 | 0.000 | **0.000** |
| | | Protein-reconstructed | 0.033 | 0.034 | 0.011 | **0.046** | 0.045 |
| | | Healthcare | 0.016 | 0.032 | 0.012 | **0.034** | 0.034 |
| | | Epidemiology | 0.000 | 0.000 | 0.000 | 0.000 | **0.000** |
| | 20 | ToyGraph | **0.175** | 0.171 | 0.093 | 0.028 | 0.040 |
| | | Synthetic | 0.137 | 0.137 | 0.116 | **0.141** | 0.116 |
| | | Synthetic-2 | 0.000 | 0.000 | 0.000 | 0.000 | **0.000** |
| | | Chain-hard | 0.049 | **0.116** | 0.094 | 0.016 | 0.034 |
| | | Ecology | 0.000 | 0.000 | 0.000 | 0.000 | **0.000** |
| | | Protein-reconstructed | 0.034 | 0.034 | 0.011 | 0.043 | **0.047** |
| | | Healthcare | 0.000 | 0.000 | 0.000 | 0.000 | **0.000** |
| | | Epidemiology | 0.000 | 0.000 | 0.000 | 0.000 | **0.000** |

## B.2 MCBO $\beta$ sensitivity

The main benchmark keeps the MCBO wrapper setting $\beta = 10$, matching the runs reported in Table 6. To assess how much this default matters, we reran MCBO with $\beta \in \{0.5, 1, 2, 5\}$ and compare the resulting GAP and PA-GAP values with the main $\beta = 10$ results. The best-performing $\beta$ is not consistent across datasets, budgets, or metrics: smaller values are often better on ToyGraph and Chain-hard, $\beta = 5$ is competitive on several PA-GAP rows, and $\beta = 10$ remains strongest on Protein-reconstructed. We therefore report the sensitivity sweep as an appendix diagnostic and avoid interpreting the default MCBO performance as an intrinsic property of the method.

## B.3 Ranking Statistical Testing

We assess whether the rank summaries used in the benchmark support statistically separable method differences. For each fixed budget, methods are ranked within each dataset–metric setting, where rank 1 denotes the best mean score in that setting. We then summarize the average rank across settings and report boot-

Table 19: Ranking reliability on the hard-intervention benchmark by budget. Ranks (1=best) are computed within each dataset–metric setting (8 datasets × 2 metrics, $N$=16 per budget). Entries show average rank ± s.d. and 95% bootstrap CI; lower is better. Bottom rows report the Friedman test, Nemenyi critical difference versus rank spread, and a conservative per-dataset Friedman test.

| | $T$=100 | | $T$=50 | | $T$=20 | |
|---|---|---|---|---|---|---|
| Method | Rank ± s.d. | 95% CI | Rank ± s.d. | 95% CI | Rank ± s.d. | 95% CI |
| CoCaBO | **3.38 ± 2.75** | [2.12, 4.69] | **3.44 ± 2.68** | [2.22, 4.75] | 3.88 ± 2.82 | [2.56, 5.25] |
| cCBO | 3.91 ± 2.23 | [2.88, 5.03] | 3.69 ± 2.32 | [2.66, 4.84] | 3.88 ± 2.24 | [2.84, 4.97] |
| CBO | 3.50 ± 1.59 | [2.81, 4.31] | 3.94 ± 1.18 | [3.44, 4.50] | 4.31 ± 1.35 | [3.62, 4.94] |
| BO | 4.12 ± 2.84 | [2.78, 5.53] | 3.94 ± 2.69 | [2.66, 5.22] | **3.81 ± 2.48** | [2.62, 5.00] |
| CEO | 4.78 ± 1.70 | [3.94, 5.56] | 4.88 ± 1.96 | [3.94, 5.81] | 4.25 ± 1.98 | [3.31, 5.19] |
| DCBO | 5.03 ± 2.22 | [3.94, 6.06] | 4.78 ± 2.27 | [3.75, 5.84] | 4.12 ± 2.18 | [3.09, 5.22] |
| HCBO | 5.38 ± 1.89 | [4.44, 6.25] | 5.44 ± 1.86 | [4.50, 6.25] | 5.84 ± 2.19 | [4.75, 6.84] |
| MCBO | 5.91 ± 1.89 | [4.97, 6.72] | 5.91 ± 2.19 | [4.81, 6.88] | 5.91 ± 2.01 | [4.91, 6.81] |
| Friedman $p$ | 0.0272 | | 0.0384 | | 0.0462 | |
| Nemenyi CD / spread | 2.62 / 2.53 | | 2.62 / 2.47 | | 2.62 / 2.09 | |
| Per-dataset $p$ ($N$=8) | 0.278 | | 0.342 | | 0.406 | |

Table 20: Ranking reliability on the soft-intervention benchmark by budget. Ranks (1=best) are computed within each dataset–metric setting (5 datasets × 2 metrics, $N$=10 per budget). Entries show average rank ± s.d. and 95% bootstrap CI; lower is better. Bottom rows report the Friedman test, Nemenyi critical difference versus rank spread, and a conservative per-dataset Friedman test.

| | $T$=100 | | $T$=50 | | $T$=20 | |
|---|---|---|---|---|---|---|
| Method | Rank ± s.d. | 95% CI | Rank ± s.d. | 95% CI | Rank ± s.d. | 95% CI |
| ACBO | **1.60 ± 0.84** | [1.10, 2.10] | **1.60 ± 0.70** | [1.20, 2.00] | **1.60 ± 0.66** | [1.20, 2.00] |
| BO | 1.80 ± 0.79 | [1.40, 2.30] | 1.90 ± 0.74 | [1.50, 2.30] | 2.00 ± 0.78 | [1.55, 2.45] |
| MCBO | 2.60 ± 0.52 | [2.30, 2.90] | 2.50 ± 0.85 | [2.00, 3.00] | 2.40 ± 0.77 | [1.90, 2.80] |
| Friedman $p$ | 0.0608 | | 0.1225 | | 0.1690 | |
| Nemenyi CD / spread | 1.05 / 1.00 | | 1.05 / 0.90 | | 1.05 / 0.80 | |
| Per-dataset $p$ ($N$=5) | 0.211 | | 0.311 | | 0.390 | |

strap confidence intervals over settings. We also report a Friedman omnibus test and the Nemenyi critical difference for post-hoc rank separation. Because GAP and PA-GAP are computed from the same trajectories, we additionally report a conservative per-dataset Friedman test in which the two metric ranks are collapsed within each dataset. An important caveat applies to all of these tests: the pooled settings are not mutually independent. GAP and PA-GAP share the same underlying trajectories, and the three budgets $T \in \{100, 50, 20\}$ are nested truncations of the same runs rather than independent replications. The reported $p$-values and critical differences should therefore be read as descriptive reliability checks under these dependencies, not as exact inference over independent samples; the conservative per-dataset test partially mitigates the metric-sharing dependence but not the budget nesting.

### B.3.1 Hard-intervention

The hard-intervention results show that average-rank orderings vary with budget. CoCaBO has the lowest average rank at $T = 100$ and $T = 50$, while BO has the lowest average rank at $T = 20$. However, these rank differences should be interpreted cautiously. Although the Friedman omnibus test detects rank heterogeneity at each budget, the rank spread is always smaller than the Nemenyi critical difference, so no pair of methods is statistically separable by the post-hoc test. The more conservative per-dataset Friedman test also does not reject at any budget. Thus, the ranking analysis supports the qualitative conclusion that performance is heterogeneous, but it does not support declaring a single hard-intervention method as uniformly superior.

### B.3.2 Soft-intervention

The soft-intervention results show a consistent descriptive ordering: ACBO has the lowest average rank at all three budgets, followed by BO and then MCBO. Nevertheless, the statistical tests do not support a separable winner. The Friedman omnibus test does not reject at any budget, the rank spread remains below the Nemenyi critical difference, and the conservative per-dataset test also does not reject. We therefore treat the soft-intervention ranking as descriptive evidence that ACBO is often strong in this benchmark, rather than as a statistically conclusive superiority claim.

## B.4 Average Reward Visualization

GAP, PA-GAP, and best-so-far curves summarize incumbent quality, but they do not show the quality of all evaluations made along the waqy. We therefore add average-reward trajectories as a complementary diagnostic. For each run, rewards are normalized so tqhat the reference optimum is 1 in the task's natural optimization direction. This makes average-reward curves comparable across datasets while preserving their interpretation as the mean quality of the sequence of queried interventions. These plots are not used as the primary ranking criterion, because exploratory acquisitions can intentionally query uncertain points with poor immediate reward, but they help verify that the benchmark conclusions do not depend only on final incumbent values.

Figures 25 and 26 should be read together with the best-so-far trajectories and scalar metrics in the main text. When a method has strong best-so-far performance but weaker average reward, the optimizer is finding good incumbents while spending part of its budget on exploratory or poorly calibrated trials. Conversely, strong average reward with weaker final GAP indicates reliable early sampling without necessarily discovering the best final intervention. This reinforces the paper's main evaluation message: CBO methods should be compared using multiple trajectory summaries rather than a single scalar metric.

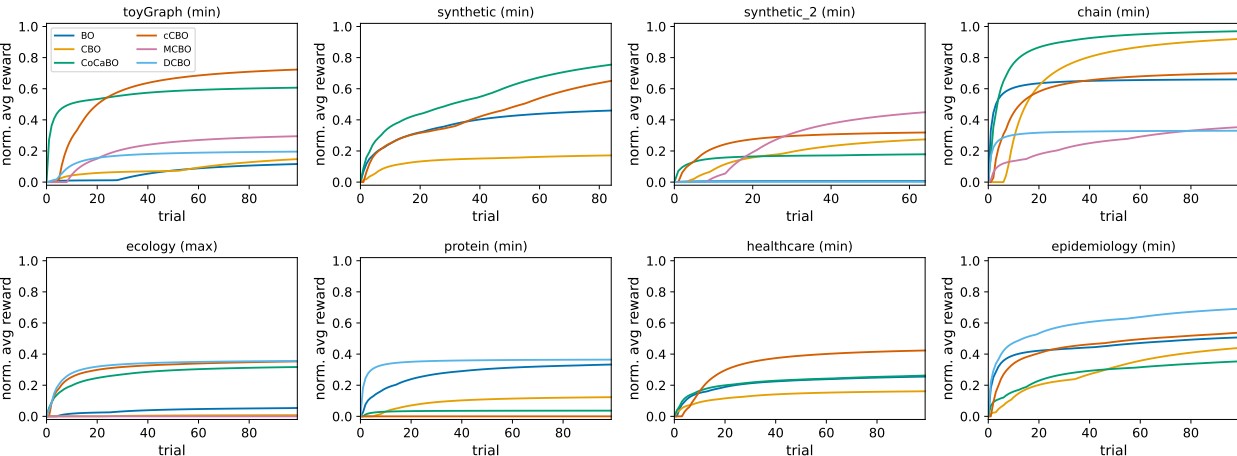

Figure 25: Average-reward trajectories for the hard-intervention benchmark.

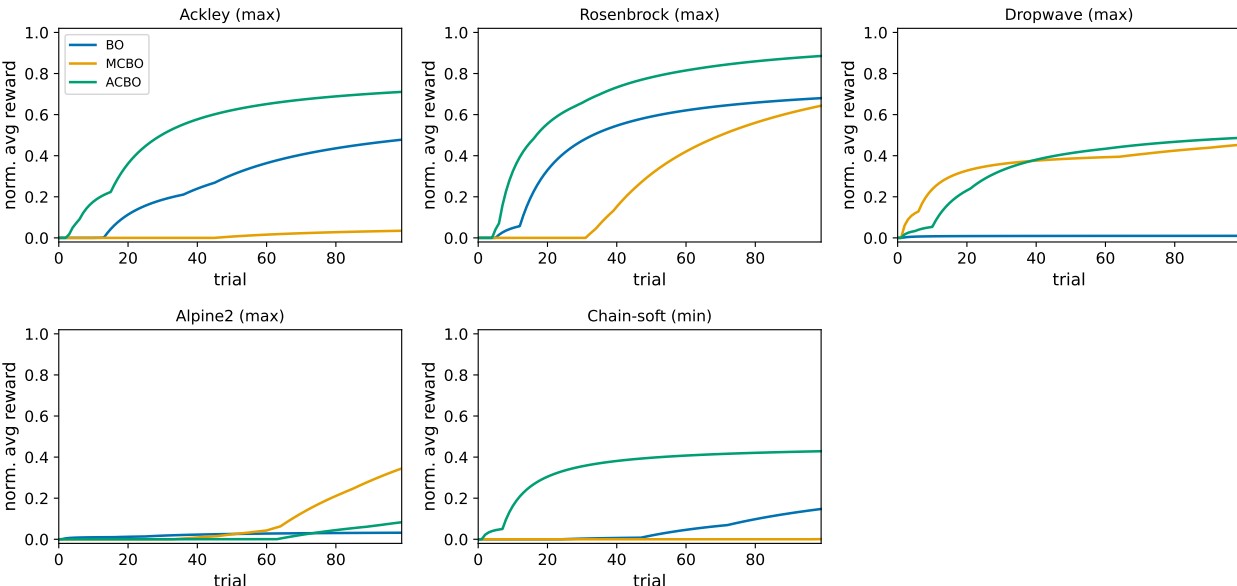

Figure 26: Average-reward trajectories for the soft-intervention/function-network benchmark.

