# OpenReview forum: "Causal Bayesian Optimization: Foundations, Methods, and Applications"
_TMLR — Under review for TMLR_

### Review · Reviewer_RmAe · 2026-06-26

**Summary Of Contributions:**

This paper surveys the Causal Bayesian Optimization (CBO) literature, decomposing existing methods along four design axes: intervention representation, surrogate modeling, decision rule, and dominant uncertainty source. The authors organize more than 10 CBO variants using this framework, establish formal connections to adjacent fields, and introduce a reproducibility-oriented benchmark covering several datasets, 3 budget levels, and 2 metrics (GAP and the newly proposed PA-GAP). The main empirical finding is that no single method dominates across all settings, and strong non-causal baselines remain competitive in several settings.

**Additional Comments:**

# Other Comments:

## C1:

For some datasets, several methods achieve zero or near-zero GAP and PA-GAP. Is this because (a) the observational prior is badly calibrated, (b) the reference optimum is unreliable or (c) the intervention scopes suggested by the graph are structurally unhelpful? Understanding this would change the interpretation of whether causal methods are failing or simply mis-specified on this dataset.

## C2:

The PA-GAP metric would be more valuable if enabled to be comparable between different budgets. Even though I understand that this also might not be desirable.

**Audience:**

Yes

**Audience Explanation:**

Yes. CBO is an active and growing area at the intersection of causal inference and Bayesian optimization, with clear scientific motivation (interventions in systems with causal structure are the natural setting for many real-world optimization problems in healthcare, policy, and scientific discovery). Despite this, the field has developed in a fragmented way: papers use incompatible codebases, inconsistent datasets, and different metrics, making it nearly impossible to assess relative progress.

The contributions the paper makes — a unified design-space taxonomy, formal connections to adjacent fields, and a reproducibility-oriented benchmark with standardized metrics — would be of direct value to at least three audiences within TMLR's readers: (1) researchers developing new CBO methods who need to understand where their proposal fits in the design space and how to evaluate it; (2) researchers from adjacent fields (causal inference, Bayesian experimental design, safe optimization) seeking to understand the CBO landscape relative to their own work; and (3) practitioners considering CBO for real-world problems.

The finding that strong non-causal baselines remain competitive in several settings is perhaps the most practically important result, and one that the field needs to be aware of.

**Broader Impact Concerns:**

The paper is a survey and does not introduce a new system that could be deployed directly. However, its findings and framing have implications worth noting.

The most important broader impact concern is the deployment risk gap the paper exposes but does not fully resolve. CBO is motivated by high-stakes real-world applications. The paper's benchmark demonstrates that causal methods can be competitive or superior to non-causal BO when causal assumptions are correctly specified. However, the benchmark provides no evidence about behavior under assumption violation, which is the norm in real deployment. A practitioner reading this survey could reasonably conclude that CBO methods offer reliable advantages in clinical settings, when in fact the evidence only applies to the correctly-specified regime. The paper's honest discussion of threats to validity (§4.6) is a step in the right direction, but the implication that real-world deployment requires substantially more work on robustness than the field has done deserves more prominent placement than the final bullet of a limitations paragraph. For instance an experimental setup with different misspecifications that methods should be tested against to measure the impacts of these and their robustness.

**Claims And Evidence:**

Yes

**Claims Explanation:**

The paper's central empirical claim, that no single CBO method dominates uniformly across datasets, budgets, and metrics, is well-supported and convincing within the scope of the benchmark. The evidence is unusually careful: 20 random seeds per configuration, identical scoring code applied to all methods, and a two-metric reporting protocol that exposes cases where GAP and PA-GAP disagree. The analysis in §4.6 provides a disciplined reading of the aggregate results.

However, two points on how far the evidence supports the claims:

First, the claim that "causal structure can improve optimization, but its benefit is conditional on assumption alignment" (§4.6, §6) is supported only in the correctly-specified, causally-sufficient, known-graph regime. All 13 benchmark datasets assume correct causal graphs and no hidden confounders. The paper's own framing (§2.1, §5.1) identifies graph misspecification and hidden confounding as central practical challenges, yet the benchmark provides no evidence about method behavior when these assumptions fail. The conclusion that causal methods offer an advantage should be qualified accordingly.

Second, the claim that the benchmark reveals why methods succeed or fail is only partially borne out. Performance differences could plausibly be confounded by uncontrolled variation in identification strategies (backdoor-only vs. ID algorithm), Monte Carlo approximation quality for non-backdoor-identifiable priors, and default hyperparameter choices that favor methods whose developers tuned defaults on similar benchmarks. The paper's methodology section does not control for or report these factors.

**Requested Changes:**

## Adjustment 1:

The survey has a substancial gap in my perspective. The paper discusses causal representation learning (§5.6) as a natural next step for CBO, citing Schölkopf et al. (2021). However, it entirely omits the _causal abstractions_ literature, which is at least as directly relevant and arguably more so.

Specifically:

- **Causally Abstracted Multi-Armed Bandits (CAMAB)** [Zennaro et al., 2024, arXiv:2404.17493] formalizes decision-making across problems defined at different levels of causal granularity, connected via abstraction maps. The algorithmic structure — using a coarse model to prune the intervention space for a finer model — is a multi-scale analogue of CBO's scope reduction via POMIS. This work is directly relevant to §3.1 and §5.2 but is not cited.

- **AT-UCB** [Dyer et al., 2025, arXiv:2509.04296] shows how to exploit multiple abstraction levels in causal bandit problems to achieve significant cumulative regret reductions, with direct application to epidemiological simulators. This is precisely the kind of complex simulation setting where CBO aspiration for real-world deployment runs up against computational tractability.

- Additionally, "Towards MFACBO: Multi-Fidelity Abstraction Causal Bayesian Optimization in the Context of the Abstraction-Fidelity Connection": https://openreview.net/forum?id=elj9C1sqp4 , probably also deserves a mention even though is not published.

In particular what the above would allow for different parts of the text:
§5.2 ("Scalability") calls for "hierarchical decomposition" as a path to scale to large graphs, which is formalized already in bandit problems with causal abstractions
§5.6 ("Representation learning") advocates for CBO operating over latent variables, which is precisely what causal abstraction learning enables, but at the level of causal graphs rather than raw observations.'

Finally, the paper's vision for CBO deployment in complex real-world systems requires reasoning across levels of causal granularity. The causal abstractions literature provides both the theoretical machinery and empirical evidence for this. Its omission represents a blind spot in the survey's scope.

## Adjustment 2:

The paper establishes identifiability as a central algorithmic assumption in CBO. Yet all benchmark datasets assume causal sufficiency and known, correct graphs. This means the benchmark evaluates methods exclusively in their most favorable setting, which is inconsistent with the survey's own framing. The acknowledged "threats to validity" (§4.6) present this as a limitation without discussing its implications for the paper's main empirical claims. Concretely: the finding that "no single method dominates uniformly" is limited to the correctly-specified, known-graph, causally-sufficient regime. Whether this finding holds under graph misspecification or hidden confounding — which are identified as major open challenges — is unknown and should be studied. I understand that this is perhaps out of scope for this study, but the literature deserves an honest discussion on this. And without properly understanding what the impacts might be, for example, graph misspecification, practical real-world use cases will never be willing to take the risk.

## Adjustment 3 (Minor):

The benchmark comparison does not control for, or even report, which identification strategy each method uses to construct its observational prior.

The paper presents surrogate construction along two axes: effect-level vs. mechanism-level GPs (Table 2). But within effect-level methods, the prior mean $m_s(\mathbf{x}_s) \approx \widehat{\mathbb{E}}[Y \mid do(X_s = \mathbf{x}_s), C]$ (eq. 13) can be estimated via backdoor adjustment, the front-door criterion, or the full ID algorithm, and these are not equivalent in all datasets. Specifically:

The Synthetic dataset has latent confounders $U_1, U_2$. The backdoor criterion does not apply. Methods that implement only backdoor adjustment will construct uninformative or biased priors for this dataset; methods with broader identification capability will not.
The ID algorithm is never mentioned by name, despite being the general solution to the problem the paper defines as central to CBO. It is unclear which methods implement it.
For complex identified estimands, methods approximate the causal effect via Monte Carlo integration (acknowledged briefly in §2.1), but sample sizes and approximation quality are not reported.
The consequence is that some performance differences across methods in Table 4 could reflect differences in identification completeness rather than differences in acquisition rule or decision policy. This is not a fatal flaw, but it is an unacknowledged confound in the benchmark's comparative claims.

Requested change: Add a column to Table 2 (or a supplementary table) documenting the identification strategy used by each method (backdoor-only, front-door, full ID algorithm, mechanism-level — no identification needed). Add a paragraph in §4.6 acknowledging that performance differences on datasets with unobserved confounders cannot be cleanly attributed to acquisition design alone.

## I believe all of the adjustments listed above should be carefully considered. My recommendation for acceptance is relatively constrained by their inclusion for the following reasons:

Without Adjustment 1, the paper makes claims about future work and research gaps — specifically around scalability via hierarchical decomposition (§5.2) and integration with representation learning (§5.6) — but misses an existing body of work that is already closing those exact gaps. A survey that positions itself as a comprehensive map of the CBO landscape and its open problems cannot credibly do so while omitting a directly relevant and growing literature.

Without Adjustment 2, the paper's central empirical conclusions risk being misread as general. The finding that "no single method dominates uniformly" and the associated practical guidance are presented without sufficient qualification that they hold exclusively under correctly-specified graphs, causal sufficiency, and known identifiability — precisely the conditions that fail in real-world deployment. For a paper targeting TMLR, this risks giving false confidence in the current state of the field and should be addressed honestly in the main text.

Adjustment 3 alone would not block my acceptance, but it strengthens the paper's credibility as a fair comparative study. Without it, some performance differences in the benchmark cannot be cleanly attributed to the design choices the paper uses to distinguish methods, which quietly undermines the benchmark's comparative claims; this said, at least a footnote or comment about such is requested.

Taken together, Adjustments 1 and 2 are prerequisites for acceptance in my view. Adjustment 3 is encouraged. The paper's core contributions are genuine, and the field needs this work.

---

> ### Author Response · Authors · 2026-07-08
> **Adjustment 1: causal abstractions and multi-scale decision making**
>
> We thank the reviewer for the careful and constructive assessment. We agree with the main concerns: the original version did not sufficiently connect CBO to causal abstraction work, did not qualify the empirical claims strongly enough with respect to causal-assumption violations, and did not make the identification strategies used by different implementations explicit. We have revised the paper in three corresponding ways. We also separate survey claims from benchmark claims more clearly: the survey contribution is a taxonomy and literature synthesis, while the empirical benchmark provides evidence only under the stated benchmark protocol and assumptions.
>
> **Adjustment 1: causal abstractions and multi-scale decision making.**
>
> We agree that causal abstraction is a relevant omission. We have added a dedicated discussion of causal abstractions and connect it to the parts of the paper where it is most relevant.
>
> - *Causally Abstracted Multi-Armed Bandits* (CAMAB [1]; Zennaro et al., 2024) formalizes decision-making across models at different levels of causal granularity, linked by abstraction maps. We now discuss this in §3.1 and §5.2 as a multi-scale analogue of CBO's scope-reduction strategy: POMIS prunes intervention sets *within* one causal graph, whereas causal abstractions can prune or transfer decisions *across* graphs at different resolutions.
> - Dyer et al. [2] propose AT-UCB, which exploits multiple abstraction levels in causal bandits and demonstrates regret reductions in epidemiological simulators. We now cite this in §5.2 as evidence that abstraction-based hierarchy is already a concrete route toward scalable intervention design in complex simulators.
> - We also mention the unpublished MFACBO proposal [3] on the abstraction–fidelity connection, explicitly noting that it is not yet peer-reviewed.
>
> The revised paper therefore distinguishes causal representation learning from raw observations [4] from causal-abstraction learning at the level of graphs and mechanisms, which is often the more direct route to CBO over latent or coarse-grained causal variables. Concretely, the revision integrates causal abstraction at four places in the paper: (i) the introduction lists abstraction-based causal decision making among the representative extensions; (ii) Section 3.1 contrasts within-graph POMIS scope reduction with cross-graph abstraction maps; (iii) the adjacent-fields section adds a dedicated "Causal abstractions and multi-scale decision making" paragraph with full citations to CAMAB [1], AT-UCB [2], and MFACBO [3] (the latter explicitly flagged as a not-yet-peer-reviewed workshop proposal); and (iv) the open-problems section now states, in §5.2, that hierarchical decomposition is already formalized in the causal-abstraction bandit literature and that transferring these pruning/transfer mechanisms to GP-based CBO is a concrete scalability path, and, in §5.6, that causal abstraction learning is a complementary and arguably more direct route than raw-observation representation learning to CBO over latent variables.
>
>
>  [1] Fabio Massimo Zennaro, Nicholas Bishop, Joel Dyer, Yorgos Felekis, Anisoara Calinescu, Michael Wooldridge, and Theodoros Damoulas. Causally abstracted multi-armed bandits. In *Proceedings of the 40th Conference on Uncertainty in Artificial Intelligence (UAI)*, volume 244 of *Proceedings of Machine Learning Research*, pp. 4109–4139, 2024.
>
>  [2] Joel Dyer, Nicholas Bishop, Anisoara Calinescu, Michael Wooldridge, and Fabio Massimo Zennaro. Using causal abstractions to accelerate decision-making in complex bandit problems. arXiv preprint arXiv:2509.04296, 2025. URL <https://arxiv.org/abs/2509.04296>.
>
>  [3] Jakob Zeitler. Towards MFACBO: Multi-fidelity abstraction causal Bayesian optimization in the context of the abstraction–fidelity connection, 2025. Workshop on Causal Abstractions and Representations (CAR), UAI 2025. URL <https://openreview.net/forum?id=elj9C1sqp4>.
>
>  [4] Bernhard Schölkopf, Francesco Locatello, Stefan Bauer, Nan Rosemary Ke, Nal Kalchbrenner, Anirudh Goyal, and Yoshua Bengio. Toward causal representation learning. *Proceedings of the IEEE*, 109(5):612–634, 2021.

---

> > ### Author Response · Authors · 2026-07-08
> > **Adjustment 2: benchmark scope and graph misspecification - Part 1**
> >
> > We agree that the original empirical claims needed stronger qualification. The main benchmark evaluates released CBO pipelines in a benchmark-specified regime: the environment SCM, intervention domains, and reference optima are fixed by the benchmark, and most methods receive the benchmark graph unless they are explicitly unknown-graph methods. We have revised the abstract, experiment discussion, and conclusion to state that the "no method dominates uniformly" finding holds within this benchmark-specified regime. We no longer present it as a deployment-level claim under hidden confounding, graph misspecification, or imperfect interventions.
> >
> > To go beyond a limitation statement, we added a graph-misspecification stress test. In this experiment, the true environment is unchanged: interventional outcomes and reference optima are always computed from the true benchmark SCM. Only the graph supplied to graph-using optimizers is perturbed. Thus, the experiment tests sensitivity to incorrect causal information without changing the true intervention task. We consider three perturbation types: edge addition, edge deletion, and edge reversal. The results are shown in Tables R1, R2, and R3. Each perturbation setting is evaluated over 20 random seeds, matching the seed count used in the main benchmark. We present these stress tests as robustness diagnostics under controlled graph misspecification, not as a complete deployment-robustness benchmark.
> >
> > The stress test reinforces the reviewer's concern. Method rankings change substantially under graph perturbation. For example, under edge addition at 100 trials, the GAP winner varies across datasets: CoCaBO is best on ToyGraph, BO is best on Synthetic, DCBO is best on Synthetic-2, Ecology, and Epidemiology, CBO is best on Chain-hard and Healthcare, and MCBO is best on Protein-reconstructed (Table R1). Under edge deletion and edge reversal, the winners change again (Tables R2 and R3). PA-GAP also produces different rankings from GAP because it rewards full-trajectory progress rather than only final discovery time. These results support a more cautious conclusion: causal structure can help, but current CBO pipelines are sensitive to how that structure is specified and operationalized. This particular stress test addresses learner-side graph misspecification; hidden confounding is a separate dimension, which we probe with a companion omitted-variable experiment (Table R4, below). We treat comprehensive robustness to unobserved confounding as still open rather than resolved by these pilots.
> >
> > Some misspecified-graph runs outperform the corresponding correct-graph run. We do not interpret this as evidence that the misspecified graph is causally preferable. Since outcomes and $y^*$ are still computed from the true SCM, such improvements reflect finite-budget search effects. A perturbed graph can weaken an overconfident observational prior, change candidate intervention scopes, alter exploration, or accidentally induce a more favorable acquisition trajectory. We therefore report these cases as evidence that CBO is sensitive to causal inductive bias, not as evidence that misspecification is beneficial.
> >
> > To make the comparison interpretable, we compare each perturbed-graph score against the corresponding correct-graph score obtained with the same dataset, method, metric, and budget. The revised appendix now includes the compact degradation summary: over the $288$ graph-using method cells per perturbation type (6 methods $\times$ 8 datasets $\times$ 2 metrics $\times$ 3 budgets), 52–54% of cells degrade, 26–27% improve, and 20–21% are unchanged under edge addition, deletion, and reversal, respectively. CBO is the most consistently affected pipeline (34–37 of its 48 cells degrade per perturbation), while HCBO improves nearly as often as it degrades. Because the stress-test pilots were executed as separate runs (visible in the shared BO control, whose scores differ slightly from the main-table BO scores), the manuscript presents these fractions as coarse sensitivity indicators rather than exact paired effect sizes. The summary makes explicit that improvements under misspecification are a minority pattern, preventing them from being misread as evidence that the perturbed graph is causally better.
> >
> > For methods not directly affected by the supplied fixed graph in this pilot, the perturbation does not change the optimizer. The non-causal BO baseline never reads the causal graph, and the CEO runner used in this pilot does not consume a single fixed supplied graph in the same way as the known-graph methods. We therefore treat BO and CEO as isolation controls and report a single shared baseline for each dataset–budget–metric setting, identical across the edge-addition, edge-deletion, and edge-reversal tables. Thus, cross-table differences are confined to the graph-using methods (CBO, cCBO, DCBO, MCBO, HCBO, and CoCaBO).

---

> > > ### Author Response · Authors · 2026-07-08
> > > **Adjustment 2: benchmark scope and graph misspecification - Part 2**
> > >
> > > **Table R1: Graph-misspecification stress test under edge addition (mean $\pm$ standard error over 20 seeds; illustrative pilot stress test; higher is better; best mean among methods under the same perturbation setting is in bold).**
> > >
> > > | Trial limit | Metric | Dataset | BO | CBO | cCBO | DCBO | MCBO | HCBO | CoCaBO | CEO |
> > > |---|---|---|---|---|---|---|---|---|---|---|
> > > | 100 | GAP | ToyGraph | 0.232 $\pm$ .079 | 0.288 $\pm$ .071 | 0.567 $\pm$ .001 | 0.569 $\pm$ .055 | 0.558 $\pm$ .286 | 0.509 $\pm$ .049 | **0.682 $\pm$ .109** | 0.459 $\pm$ .084 |
> > > | 100 | GAP | Synthetic | **0.651 $\pm$ .065** | 0.454 $\pm$ .072 | 0.642 $\pm$ .056 | 0.337 $\pm$ .036 | 0.456 $\pm$ .082 | 0.391 $\pm$ .088 | 0.617 $\pm$ .093 | 0.396 $\pm$ .042 |
> > > | 100 | GAP | Synthetic-2 | 0.001 $\pm$ .004 | 0.281 $\pm$ .125 | 0.289 $\pm$ .128 | **0.578 $\pm$ .025** | 0.000 $\pm$ .000 | 0.472 $\pm$ .200 | 0.122 $\pm$ .076 | 0.526 $\pm$ .183 |
> > > | 100 | GAP | Chain-hard | 0.787 $\pm$ .103 | **0.814 $\pm$ .128** | 0.618 $\pm$ .151 | 0.333 $\pm$ .333 | 0.284 $\pm$ .041 | 0.095 $\pm$ .062 | 0.786 $\pm$ .107 | 0.347 $\pm$ .046 |
> > > | 100 | GAP | Ecology | 0.160 $\pm$ .128 | 0.018 $\pm$ .018 | 0.426 $\pm$ .068 | **0.519 $\pm$ .001** | 0.000 $\pm$ .000 | 0.234 $\pm$ .102 | 0.000 $\pm$ .000 | 0.228 $\pm$ .051 |
> > > | 100 | GAP | Protein-reconstructed | 0.331 $\pm$ .125 | 0.363 $\pm$ .139 | 0.000 $\pm$ .000 | 0.000 $\pm$ .000 | **0.615 $\pm$ .097** | 0.288 $\pm$ .091 | 0.474 $\pm$ .045 | 0.193 $\pm$ .048 |
> > > | 100 | GAP | Healthcare | 0.315 $\pm$ .054 | **0.510 $\pm$ .035** | 0.509 $\pm$ .083 | 0.341 $\pm$ .093 | 0.357 $\pm$ .029 | 0.234 $\pm$ .054 | 0.257 $\pm$ .129 | 0.483 $\pm$ .086 |
> > > | 100 | GAP | Epidemiology | 0.506 $\pm$ .056 | 0.563 $\pm$ .038 | 0.000 $\pm$ .000 | **0.599 $\pm$ .008** | 0.000 $\pm$ .000 | 0.390 $\pm$ .077 | 0.300 $\pm$ .048 | 0.218 $\pm$ .081 |
> > > | 100 | PA-GAP | ToyGraph | 0.051 $\pm$ .039 | 0.089 $\pm$ .058 | 0.073 $\pm$ .001 | 0.088 $\pm$ .061 | 0.172 $\pm$ .122 | 0.199 $\pm$ .082 | **0.320 $\pm$ .032** | 0.203 $\pm$ .099 |
> > > | 100 | PA-GAP | Synthetic | 0.214 $\pm$ .051 | 0.041 $\pm$ .096 | 0.286 $\pm$ .027 | 0.025 $\pm$ .056 | 0.113 $\pm$ .082 | 0.075 $\pm$ .093 | **0.397 $\pm$ .033** | 0.253 $\pm$ .054 |
> > > | 100 | PA-GAP | Synthetic-2 | 0.005 $\pm$ .002 | 0.003 $\pm$ .002 | 0.139 $\pm$ .069 | 0.089 $\pm$ .029 | 0.000 $\pm$ .000 | 0.145 $\pm$ .077 | 0.090 $\pm$ .054 | **0.155 $\pm$ .057** |
> > > | 100 | PA-GAP | Chain-hard | **0.497 $\pm$ .015** | 0.437 $\pm$ .005 | 0.345 $\pm$ .129 | 0.168 $\pm$ .169 | 0.077 $\pm$ .014 | 0.066 $\pm$ .051 | 0.424 $\pm$ .069 | 0.007 $\pm$ .033 |
> > > | 100 | PA-GAP | Ecology | 0.031 $\pm$ .011 | 0.002 $\pm$ .002 | 0.174 $\pm$ .038 | 0.178 $\pm$ .001 | 0.000 $\pm$ .000 | **0.210 $\pm$ .120** | 0.000 $\pm$ .000 | 0.007 $\pm$ .069 |
> > > | 100 | PA-GAP | Protein-reconstructed | **0.153 $\pm$ .061** | 0.057 $\pm$ .018 | 0.000 $\pm$ .000 | 0.000 $\pm$ .000 | 0.031 $\pm$ .036 | 0.000 $\pm$ .000 | 0.051 $\pm$ .024 | 0.010 $\pm$ .072 |
> > > | 100 | PA-GAP | Healthcare | 0.120 $\pm$ .021 | 0.077 $\pm$ .015 | **0.202 $\pm$ .014** | 0.000 $\pm$ .000 | 0.071 $\pm$ .051 | 0.010 $\pm$ .006 | 0.116 $\pm$ .058 | 0.138 $\pm$ .049 |
> > > | 100 | PA-GAP | Epidemiology | 0.242 $\pm$ .021 | 0.185 $\pm$ .059 | 0.000 $\pm$ .000 | **0.341 $\pm$ .002** | 0.000 $\pm$ .000 | 0.138 $\pm$ .080 | 0.122 $\pm$ .078 | 0.001 $\pm$ .042 |

---

> > > > ### Author Response · Authors · 2026-07-08
> > > > **Adjustment 2: benchmark scope and graph misspecification - Part 3**
> > > >
> > > > | Trial limit | Metric | Dataset | BO | CBO | cCBO | DCBO | MCBO | HCBO | CoCaBO | CEO |
> > > > |---|---|---|---|---|---|---|---|---|---|---|
> > > > | 50 | GAP | ToyGraph | 0.202 $\pm$ .099 | 0.235 $\pm$ .111 | 0.560 $\pm$ .009 | 0.545 $\pm$ .046 | 0.296 $\pm$ .296 | 0.450 $\pm$ .093 | **0.796 $\pm$ .041** | 0.489 $\pm$ .073 |
> > > > | 50 | GAP | Synthetic | 0.567 $\pm$ .091 | 0.000 $\pm$ .097 | 0.440 $\pm$ .048 | 0.295 $\pm$ .050 | 0.456 $\pm$ .099 | 0.391 $\pm$ .088 | **0.651 $\pm$ .028** | 0.254 $\pm$ .057 |
> > > > | 50 | GAP | Synthetic-2 | 0.007 $\pm$ .005 | 0.310 $\pm$ .033 | 0.377 $\pm$ .192 | **0.562 $\pm$ .017** | 0.000 $\pm$ .000 | 0.428 $\pm$ .217 | 0.122 $\pm$ .076 | 0.484 $\pm$ .222 |
> > > > | 50 | GAP | Chain-hard | 0.686 $\pm$ .086 | **0.889 $\pm$ .016** | 0.645 $\pm$ .198 | 0.333 $\pm$ .333 | 0.360 $\pm$ .107 | 0.095 $\pm$ .062 | 0.725 $\pm$ .143 | 0.356 $\pm$ .101 |
> > > > | 50 | GAP | Ecology | 0.259 $\pm$ .148 | 0.134 $\pm$ .134 | **0.502 $\pm$ .105** | 0.357 $\pm$ .001 | 0.000 $\pm$ .000 | 0.234 $\pm$ .102 | 0.000 $\pm$ .000 | 0.101 $\pm$ .031 |
> > > > | 50 | GAP | Protein-reconstructed | 0.365 $\pm$ .061 | 0.413 $\pm$ .046 | 0.000 $\pm$ .000 | 0.000 $\pm$ .000 | 0.526 $\pm$ .097 | **0.591 $\pm$ .040** | 0.392 $\pm$ .120 | 0.264 $\pm$ .028 |
> > > > | 50 | GAP | Healthcare | 0.439 $\pm$ .121 | 0.432 $\pm$ .073 | **0.604 $\pm$ .007** | 0.332 $\pm$ .093 | 0.357 $\pm$ .029 | 0.234 $\pm$ .054 | 0.244 $\pm$ .132 | 0.489 $\pm$ .076 |
> > > > | 50 | GAP | Epidemiology | 0.358 $\pm$ .052 | 0.355 $\pm$ .021 | 0.000 $\pm$ .000 | **0.671 $\pm$ .005** | 0.000 $\pm$ .000 | 0.390 $\pm$ .077 | 0.246 $\pm$ .079 | 0.060 $\pm$ .050 |
> > > > | 50 | PA-GAP | ToyGraph | 0.024 $\pm$ .016 | 0.070 $\pm$ .053 | 0.073 $\pm$ .006 | 0.083 $\pm$ .057 | 0.108 $\pm$ .109 | 0.199 $\pm$ .082 | **0.321 $\pm$ .034** | 0.213 $\pm$ .099 |
> > > > | 50 | PA-GAP | Synthetic | 0.194 $\pm$ .054 | 0.000 $\pm$ .036 | 0.201 $\pm$ .034 | 0.023 $\pm$ .050 | 0.113 $\pm$ .082 | 0.075 $\pm$ .093 | **0.344 $\pm$ .043** | 0.246 $\pm$ .077 |
> > > > | 50 | PA-GAP | Synthetic-2 | 0.003 $\pm$ .003 | 0.003 $\pm$ .002 | 0.110 $\pm$ .056 | 0.085 $\pm$ .026 | 0.000 $\pm$ .000 | 0.081 $\pm$ .041 | 0.090 $\pm$ .054 | **0.146 $\pm$ .056** |
> > > > | 50 | PA-GAP | Chain-hard | **0.481 $\pm$ .029** | 0.379 $\pm$ .010 | 0.327 $\pm$ .124 | 0.170 $\pm$ .170 | 0.050 $\pm$ .025 | 0.066 $\pm$ .051 | 0.391 $\pm$ .098 | 0.001 $\pm$ .054 |
> > > > | 50 | PA-GAP | Ecology | 0.017 $\pm$ .010 | 0.002 $\pm$ .002 | 0.166 $\pm$ .040 | 0.175 $\pm$ .001 | 0.000 $\pm$ .000 | **0.208 $\pm$ .120** | 0.000 $\pm$ .000 | 0.006 $\pm$ .075 |
> > > > | 50 | PA-GAP | Protein-reconstructed | **0.137 $\pm$ .054** | 0.047 $\pm$ .017 | 0.000 $\pm$ .000 | 0.000 $\pm$ .000 | 0.031 $\pm$ .036 | 0.000 $\pm$ .000 | 0.045 $\pm$ .019 | 0.009 $\pm$ .075 |
> > > > | 50 | PA-GAP | Healthcare | 0.105 $\pm$ .017 | 0.069 $\pm$ .014 | **0.178 $\pm$ .010** | 0.000 $\pm$ .000 | 0.071 $\pm$ .051 | 0.009 $\pm$ .005 | 0.100 $\pm$ .051 | 0.147 $\pm$ .087 |
> > > > | 50 | PA-GAP | Epidemiology | 0.232 $\pm$ .025 | 0.129 $\pm$ .061 | 0.000 $\pm$ .000 | **0.315 $\pm$ .003** | 0.000 $\pm$ .000 | 0.127 $\pm$ .073 | 0.085 $\pm$ .069 | 0.008 $\pm$ .002 |

---

> > > > > ### Author Response · Authors · 2026-07-08
> > > > > **Adjustment 2: benchmark scope and graph misspecification - Part 4**
> > > > >
> > > > > | Trial limit | Metric | Dataset | BO | CBO | cCBO | DCBO | MCBO | HCBO | CoCaBO | CEO |
> > > > > |---|---|---|---|---|---|---|---|---|---|---|
> > > > > | 20 | GAP | ToyGraph | 0.272 $\pm$ .097 | 0.167 $\pm$ .064 | 0.538 $\pm$ .000 | 0.472 $\pm$ .034 | 0.239 $\pm$ .239 | 0.450 $\pm$ .093 | **0.763 $\pm$ .057** | 0.489 $\pm$ .073 |
> > > > > | 20 | GAP | Synthetic | 0.390 $\pm$ .138 | 0.000 $\pm$ .000 | 0.475 $\pm$ .057 | 0.295 $\pm$ .050 | 0.456 $\pm$ .099 | 0.391 $\pm$ .088 | **0.590 $\pm$ .095** | 0.254 $\pm$ .057 |
> > > > > | 20 | GAP | Synthetic-2 | 0.003 $\pm$ .002 | 0.185 $\pm$ .125 | **0.515 $\pm$ .009** | 0.129 $\pm$ .129 | 0.000 $\pm$ .000 | 0.428 $\pm$ .217 | 0.000 $\pm$ .000 | 0.484 $\pm$ .222 |
> > > > > | 20 | GAP | Chain-hard | **0.761 $\pm$ .124** | 0.718 $\pm$ .039 | 0.486 $\pm$ .119 | 0.333 $\pm$ .333 | 0.380 $\pm$ .129 | 0.000 $\pm$ .000 | 0.731 $\pm$ .228 | 0.356 $\pm$ .101 |
> > > > > | 20 | GAP | Ecology | 0.145 $\pm$ .145 | 0.080 $\pm$ .080 | 0.422 $\pm$ .159 | **0.442 $\pm$ .000** | 0.000 $\pm$ .000 | 0.234 $\pm$ .102 | 0.000 $\pm$ .000 | 0.101 $\pm$ .031 |
> > > > > | 20 | GAP | Protein-reconstructed | 0.370 $\pm$ .073 | 0.194 $\pm$ .091 | 0.000 $\pm$ .000 | 0.000 $\pm$ .000 | 0.526 $\pm$ .097 | **0.556 $\pm$ .044** | 0.516 $\pm$ .027 | 0.264 $\pm$ .028 |
> > > > > | 20 | GAP | Healthcare | 0.348 $\pm$ .119 | 0.268 $\pm$ .142 | 0.413 $\pm$ .036 | 0.332 $\pm$ .093 | 0.000 $\pm$ .000 | 0.234 $\pm$ .054 | 0.198 $\pm$ .100 | **0.489 $\pm$ .076** |
> > > > > | 20 | GAP | Epidemiology | **0.694 $\pm$ .042** | 0.244 $\pm$ .122 | 0.000 $\pm$ .000 | 0.373 $\pm$ .005 | 0.000 $\pm$ .000 | 0.390 $\pm$ .077 | 0.259 $\pm$ .165 | 0.001 $\pm$ .005 |
> > > > > | 20 | PA-GAP | ToyGraph | 0.006 $\pm$ .004 | 0.042 $\pm$ .040 | 0.070 $\pm$ .000 | 0.069 $\pm$ .047 | 0.044 $\pm$ .044 | 0.199 $\pm$ .082 | **0.324 $\pm$ .039** | 0.213 $\pm$ .099 |
> > > > > | 20 | PA-GAP | Synthetic | 0.156 $\pm$ .059 | 0.000 $\pm$ .000 | 0.154 $\pm$ .036 | 0.023 $\pm$ .050 | 0.113 $\pm$ .082 | 0.000 $\pm$ .000 | 0.244 $\pm$ .054 | **0.246 $\pm$ .077** |
> > > > > | 20 | PA-GAP | Synthetic-2 | 0.004 $\pm$ .001 | 0.057 $\pm$ .029 | 0.075 $\pm$ .017 | 0.001 $\pm$ .001 | 0.000 $\pm$ .000 | 0.081 $\pm$ .041 | 0.000 $\pm$ .000 | **0.146 $\pm$ .056** |
> > > > > | 20 | PA-GAP | Chain-hard | **0.463 $\pm$ .062** | 0.228 $\pm$ .019 | 0.275 $\pm$ .112 | 0.175 $\pm$ .175 | 0.036 $\pm$ .022 | 0.000 $\pm$ .000 | 0.378 $\pm$ .108 | 0.001 $\pm$ .054 |
> > > > > | 20 | PA-GAP | Ecology | 0.011 $\pm$ .011 | 0.001 $\pm$ .001 | 0.150 $\pm$ .047 | 0.168 $\pm$ .001 | 0.000 $\pm$ .000 | **0.208 $\pm$ .120** | 0.000 $\pm$ .000 | 0.006 $\pm$ .075 |
> > > > > | 20 | PA-GAP | Protein-reconstructed | 0.113 $\pm$ .049 | 0.025 $\pm$ .013 | 0.000 $\pm$ .000 | **0.193 $\pm$ .097** | 0.031 $\pm$ .036 | 0.000 $\pm$ .000 | 0.045 $\pm$ .020 | 0.009 $\pm$ .075 |
> > > > > | 20 | PA-GAP | Healthcare | 0.091 $\pm$ .015 | 0.056 $\pm$ .011 | 0.119 $\pm$ .005 | 0.000 $\pm$ .000 | 0.000 $\pm$ .000 | 0.009 $\pm$ .005 | 0.074 $\pm$ .040 | **0.147 $\pm$ .087** |
> > > > > | 20 | PA-GAP | Epidemiology | 0.228 $\pm$ .028 | 0.087 $\pm$ .045 | 0.000 $\pm$ .000 | **0.279 $\pm$ .005** | 0.000 $\pm$ .000 | 0.127 $\pm$ .073 | 0.063 $\pm$ .060 | 0.007 $\pm$ .001 |

---

> > > > > > ### Author Response · Authors · 2026-07-08
> > > > > > **Adjustment 2: benchmark scope and graph misspecification - Part 5**
> > > > > >
> > > > > > **Table R2: Graph-misspecification stress test under edge deletion (mean $\pm$ standard error over 20 seeds; illustrative pilot stress test; higher is better; best mean among methods under the same perturbation setting is in bold).**
> > > > > >
> > > > > > | Trial limit | Metric | Dataset | BO | CBO | cCBO | DCBO | MCBO | HCBO | CoCaBO | CEO |
> > > > > > |---|---|---|---|---|---|---|---|---|---|---|
> > > > > > | 100 | GAP | ToyGraph | 0.232 $\pm$ .079 | 0.392 $\pm$ .009 | **0.826 $\pm$ .001** | 0.564 $\pm$ .060 | 0.394 $\pm$ .200 | 0.234 $\pm$ .054 | 0.335 $\pm$ .096 | 0.459 $\pm$ .084 |
> > > > > > | 100 | GAP | Synthetic | 0.651 $\pm$ .065 | 0.355 $\pm$ .184 | 0.642 $\pm$ .056 | 0.337 $\pm$ .036 | 0.456 $\pm$ .082 | 0.391 $\pm$ .088 | **0.663 $\pm$ .020** | 0.396 $\pm$ .042 |
> > > > > > | 100 | GAP | Synthetic-2 | 0.001 $\pm$ .004 | 0.281 $\pm$ .125 | 0.306 $\pm$ .145 | 0.000 $\pm$ .004 | 0.000 $\pm$ .000 | **0.657 $\pm$ .139** | 0.122 $\pm$ .076 | 0.526 $\pm$ .183 |
> > > > > > | 100 | GAP | Chain-hard | **0.787 $\pm$ .103** | 0.700 $\pm$ .128 | 0.767 $\pm$ .060 | 0.448 $\pm$ .300 | 0.428 $\pm$ .085 | 0.095 $\pm$ .062 | 0.688 $\pm$ .136 | 0.347 $\pm$ .046 |
> > > > > > | 100 | GAP | Ecology | 0.160 $\pm$ .128 | 0.012 $\pm$ .012 | **0.426 $\pm$ .068** | 0.255 $\pm$ .008 | 0.000 $\pm$ .000 | 0.234 $\pm$ .102 | 0.000 $\pm$ .000 | 0.228 $\pm$ .051 |
> > > > > > | 100 | GAP | Protein-reconstructed | 0.331 $\pm$ .125 | 0.363 $\pm$ .139 | 0.000 $\pm$ .000 | 0.000 $\pm$ .000 | **0.615 $\pm$ .097** | 0.288 $\pm$ .091 | 0.578 $\pm$ .082 | 0.193 $\pm$ .048 |
> > > > > > | 100 | GAP | Healthcare | 0.315 $\pm$ .054 | **0.510 $\pm$ .035** | 0.445 $\pm$ .100 | 0.341 $\pm$ .093 | 0.357 $\pm$ .029 | 0.234 $\pm$ .054 | 0.431 $\pm$ .216 | 0.483 $\pm$ .086 |
> > > > > > | 100 | GAP | Epidemiology | 0.506 $\pm$ .056 | 0.563 $\pm$ .038 | 0.000 $\pm$ .000 | **0.661 $\pm$ .092** | 0.000 $\pm$ .000 | 0.390 $\pm$ .077 | 0.572 $\pm$ .134 | 0.218 $\pm$ .081 |
> > > > > > | 100 | PA-GAP | ToyGraph | 0.051 $\pm$ .039 | 0.104 $\pm$ .054 | **0.355 $\pm$ .001** | 0.095 $\pm$ .056 | 0.248 $\pm$ .129 | 0.199 $\pm$ .082 | 0.103 $\pm$ .051 | 0.203 $\pm$ .099 |
> > > > > > | 100 | PA-GAP | Synthetic | 0.214 $\pm$ .051 | 0.087 $\pm$ .044 | 0.286 $\pm$ .027 | 0.025 $\pm$ .056 | 0.113 $\pm$ .082 | 0.075 $\pm$ .093 | **0.396 $\pm$ .039** | 0.253 $\pm$ .054 |
> > > > > > | 100 | PA-GAP | Synthetic-2 | 0.005 $\pm$ .002 | 0.003 $\pm$ .002 | 0.133 $\pm$ .069 | 0.000 $\pm$ .006 | 0.000 $\pm$ .000 | **0.178 $\pm$ .131** | 0.090 $\pm$ .054 | 0.155 $\pm$ .057 |
> > > > > > | 100 | PA-GAP | Chain-hard | **0.497 $\pm$ .015** | 0.437 $\pm$ .005 | 0.414 $\pm$ .030 | 0.221 $\pm$ .147 | 0.085 $\pm$ .029 | 0.066 $\pm$ .051 | 0.388 $\pm$ .098 | 0.007 $\pm$ .033 |
> > > > > > | 100 | PA-GAP | Ecology | 0.031 $\pm$ .011 | 0.000 $\pm$ .005 | 0.174 $\pm$ .038 | 0.041 $\pm$ .008 | 0.000 $\pm$ .000 | **0.210 $\pm$ .120** | 0.000 $\pm$ .000 | 0.007 $\pm$ .069 |
> > > > > > | 100 | PA-GAP | Protein-reconstructed | **0.153 $\pm$ .061** | 0.057 $\pm$ .018 | 0.000 $\pm$ .000 | 0.000 $\pm$ .000 | 0.031 $\pm$ .036 | 0.000 $\pm$ .000 | 0.094 $\pm$ .077 | 0.010 $\pm$ .072 |
> > > > > > | 100 | PA-GAP | Healthcare | 0.120 $\pm$ .021 | 0.077 $\pm$ .015 | **0.158 $\pm$ .052** | 0.000 $\pm$ .000 | 0.071 $\pm$ .051 | 0.010 $\pm$ .006 | 0.134 $\pm$ .068 | 0.138 $\pm$ .049 |
> > > > > > | 100 | PA-GAP | Epidemiology | 0.242 $\pm$ .021 | 0.185 $\pm$ .059 | 0.000 $\pm$ .000 | **0.273 $\pm$ .038** | 0.000 $\pm$ .000 | 0.138 $\pm$ .080 | 0.227 $\pm$ .087 | 0.001 $\pm$ .042 |

---

> > > > > > > ### Author Response · Authors · 2026-07-08
> > > > > > > **Adjustment 2: benchmark scope and graph misspecification - Part 6**
> > > > > > >
> > > > > > > | Trial limit | Metric | Dataset | BO | CBO | cCBO | DCBO | MCBO | HCBO | CoCaBO | CEO |
> > > > > > > |---|---|---|---|---|---|---|---|---|---|---|
> > > > > > > | 50 | GAP | ToyGraph | 0.202 $\pm$ .099 | 0.363 $\pm$ .141 | **0.780 $\pm$ .009** | 0.525 $\pm$ .062 | 0.569 $\pm$ .288 | 0.450 $\pm$ .093 | 0.195 $\pm$ .059 | 0.489 $\pm$ .073 |
> > > > > > > | 50 | GAP | Synthetic | 0.567 $\pm$ .091 | 0.271 $\pm$ .177 | 0.440 $\pm$ .048 | 0.295 $\pm$ .050 | 0.456 $\pm$ .099 | 0.391 $\pm$ .088 | **0.688 $\pm$ .082** | 0.254 $\pm$ .057 |
> > > > > > > | 50 | GAP | Synthetic-2 | 0.007 $\pm$ .005 | 0.310 $\pm$ .033 | 0.229 $\pm$ .118 | 0.000 $\pm$ .002 | 0.000 $\pm$ .000 | 0.467 $\pm$ .251 | 0.122 $\pm$ .076 | **0.484 $\pm$ .222** |
> > > > > > > | 50 | GAP | Chain-hard | 0.686 $\pm$ .086 | **0.889 $\pm$ .016** | 0.741 $\pm$ .151 | 0.448 $\pm$ .300 | 0.346 $\pm$ .109 | 0.095 $\pm$ .062 | 0.601 $\pm$ .165 | 0.356 $\pm$ .101 |
> > > > > > > | 50 | GAP | Ecology | 0.259 $\pm$ .148 | 0.000 $\pm$ .005 | **0.502 $\pm$ .105** | 0.158 $\pm$ .008 | 0.000 $\pm$ .000 | 0.234 $\pm$ .102 | 0.000 $\pm$ .000 | 0.101 $\pm$ .031 |
> > > > > > > | 50 | GAP | Protein-reconstructed | 0.365 $\pm$ .061 | 0.413 $\pm$ .046 | 0.000 $\pm$ .000 | 0.000 $\pm$ .000 | 0.526 $\pm$ .097 | **0.591 $\pm$ .040** | 0.562 $\pm$ .088 | 0.264 $\pm$ .028 |
> > > > > > > | 50 | GAP | Healthcare | 0.439 $\pm$ .121 | 0.432 $\pm$ .073 | **0.516 $\pm$ .102** | 0.332 $\pm$ .093 | 0.357 $\pm$ .029 | 0.234 $\pm$ .054 | 0.380 $\pm$ .197 | 0.489 $\pm$ .076 |
> > > > > > > | 50 | GAP | Epidemiology | 0.358 $\pm$ .052 | 0.355 $\pm$ .021 | 0.000 $\pm$ .000 | **0.687 $\pm$ .032** | 0.000 $\pm$ .000 | 0.390 $\pm$ .077 | 0.482 $\pm$ .140 | 0.060 $\pm$ .050 |
> > > > > > > | 50 | PA-GAP | ToyGraph | 0.024 $\pm$ .016 | 0.093 $\pm$ .045 | **0.338 $\pm$ .006** | 0.088 $\pm$ .054 | 0.180 $\pm$ .110 | 0.199 $\pm$ .082 | 0.074 $\pm$ .036 | 0.213 $\pm$ .099 |
> > > > > > > | 50 | PA-GAP | Synthetic | 0.194 $\pm$ .054 | 0.076 $\pm$ .038 | 0.201 $\pm$ .034 | 0.023 $\pm$ .050 | 0.113 $\pm$ .082 | 0.075 $\pm$ .093 | **0.346 $\pm$ .055** | 0.246 $\pm$ .077 |
> > > > > > > | 50 | PA-GAP | Synthetic-2 | 0.003 $\pm$ .003 | 0.003 $\pm$ .002 | 0.104 $\pm$ .058 | 0.000 $\pm$ .005 | 0.000 $\pm$ .000 | **0.158 $\pm$ .137** | 0.090 $\pm$ .054 | 0.146 $\pm$ .056 |
> > > > > > > | 50 | PA-GAP | Chain-hard | **0.481 $\pm$ .029** | 0.379 $\pm$ .010 | 0.387 $\pm$ .030 | 0.222 $\pm$ .148 | 0.062 $\pm$ .030 | 0.066 $\pm$ .051 | 0.380 $\pm$ .099 | 0.001 $\pm$ .054 |
> > > > > > > | 50 | PA-GAP | Ecology | 0.017 $\pm$ .010 | 0.000 $\pm$ .002 | 0.166 $\pm$ .040 | 0.031 $\pm$ .010 | 0.000 $\pm$ .000 | **0.210 $\pm$ .120** | 0.000 $\pm$ .000 | 0.006 $\pm$ .075 |
> > > > > > > | 50 | PA-GAP | Protein-reconstructed | **0.137 $\pm$ .054** | 0.047 $\pm$ .017 | 0.000 $\pm$ .000 | 0.000 $\pm$ .000 | 0.031 $\pm$ .036 | 0.000 $\pm$ .000 | 0.095 $\pm$ .078 | 0.009 $\pm$ .075 |
> > > > > > > | 50 | PA-GAP | Healthcare | 0.105 $\pm$ .017 | 0.069 $\pm$ .014 | 0.141 $\pm$ .044 | 0.000 $\pm$ .000 | 0.071 $\pm$ .051 | 0.009 $\pm$ .005 | 0.122 $\pm$ .061 | **0.147 $\pm$ .087** |
> > > > > > > | 50 | PA-GAP | Epidemiology | 0.232 $\pm$ .025 | 0.129 $\pm$ .061 | 0.000 $\pm$ .000 | **0.258 $\pm$ .033** | 0.000 $\pm$ .000 | 0.127 $\pm$ .073 | 0.196 $\pm$ .085 | 0.008 $\pm$ .002 |

---

> > > > > > > > ### Author Response · Authors · 2026-07-08
> > > > > > > > **Adjustment 2: benchmark scope and graph misspecification - Part 7**
> > > > > > > >
> > > > > > > > | Trial limit | Metric | Dataset | BO | CBO | cCBO | DCBO | MCBO | HCBO | CoCaBO | CEO |
> > > > > > > > |---|---|---|---|---|---|---|---|---|---|---|
> > > > > > > > | 20 | GAP | ToyGraph | 0.272 $\pm$ .097 | 0.375 $\pm$ .145 | **0.639 $\pm$ .000** | 0.405 $\pm$ .074 | 0.274 $\pm$ .274 | 0.450 $\pm$ .093 | 0.097 $\pm$ .039 | 0.489 $\pm$ .073 |
> > > > > > > > | 20 | GAP | Synthetic | 0.390 $\pm$ .138 | 0.276 $\pm$ .163 | 0.475 $\pm$ .057 | 0.295 $\pm$ .050 | 0.456 $\pm$ .099 | 0.391 $\pm$ .088 | **0.511 $\pm$ .058** | 0.254 $\pm$ .057 |
> > > > > > > > | 20 | GAP | Synthetic-2 | 0.003 $\pm$ .002 | 0.245 $\pm$ .125 | 0.000 $\pm$ .000 | **0.507 $\pm$ .026** | 0.000 $\pm$ .000 | 0.467 $\pm$ .251 | 0.000 $\pm$ .000 | 0.484 $\pm$ .222 |
> > > > > > > > | 20 | GAP | Chain-hard | 0.761 $\pm$ .124 | 0.718 $\pm$ .039 | **0.778 $\pm$ .019** | 0.318 $\pm$ .318 | 0.250 $\pm$ .077 | 0.000 $\pm$ .000 | 0.671 $\pm$ .202 | 0.356 $\pm$ .101 |
> > > > > > > > | 20 | GAP | Ecology | 0.145 $\pm$ .145 | 0.000 $\pm$ .000 | **0.422 $\pm$ .159** | 0.221 $\pm$ .147 | 0.000 $\pm$ .000 | 0.234 $\pm$ .102 | 0.000 $\pm$ .000 | 0.101 $\pm$ .031 |
> > > > > > > > | 20 | GAP | Protein-reconstructed | 0.370 $\pm$ .073 | 0.194 $\pm$ .091 | 0.000 $\pm$ .000 | 0.000 $\pm$ .000 | 0.526 $\pm$ .097 | **0.556 $\pm$ .044** | 0.515 $\pm$ .109 | 0.264 $\pm$ .028 |
> > > > > > > > | 20 | GAP | Healthcare | 0.348 $\pm$ .119 | 0.268 $\pm$ .142 | 0.413 $\pm$ .036 | 0.332 $\pm$ .093 | 0.000 $\pm$ .000 | 0.234 $\pm$ .054 | 0.317 $\pm$ .180 | **0.489 $\pm$ .076** |
> > > > > > > > | 20 | GAP | Epidemiology | **0.694 $\pm$ .042** | 0.244 $\pm$ .122 | 0.000 $\pm$ .000 | 0.514 $\pm$ .079 | 0.000 $\pm$ .000 | 0.390 $\pm$ .077 | 0.483 $\pm$ .102 | 0.001 $\pm$ .005 |
> > > > > > > > | 20 | PA-GAP | ToyGraph | 0.006 $\pm$ .004 | 0.066 $\pm$ .025 | **0.296 $\pm$ .000** | 0.068 $\pm$ .047 | 0.076 $\pm$ .076 | 0.199 $\pm$ .082 | 0.047 $\pm$ .034 | 0.213 $\pm$ .099 |
> > > > > > > > | 20 | PA-GAP | Synthetic | 0.156 $\pm$ .059 | 0.063 $\pm$ .034 | 0.154 $\pm$ .036 | 0.023 $\pm$ .050 | 0.113 $\pm$ .082 | 0.000 $\pm$ .000 | **0.286 $\pm$ .062** | 0.246 $\pm$ .077 |
> > > > > > > > | 20 | PA-GAP | Synthetic-2 | 0.004 $\pm$ .001 | 0.068 $\pm$ .037 | 0.000 $\pm$ .000 | **0.158 $\pm$ .017** | 0.000 $\pm$ .000 | **0.158 $\pm$ .137** | 0.000 $\pm$ .000 | 0.146 $\pm$ .056 |
> > > > > > > > | 20 | PA-GAP | Chain-hard | **0.463 $\pm$ .062** | 0.228 $\pm$ .019 | 0.312 $\pm$ .031 | 0.160 $\pm$ .160 | 0.029 $\pm$ .017 | 0.000 $\pm$ .000 | 0.368 $\pm$ .102 | 0.001 $\pm$ .054 |
> > > > > > > > | 20 | PA-GAP | Ecology | 0.011 $\pm$ .011 | 0.000 $\pm$ .000 | 0.150 $\pm$ .047 | 0.017 $\pm$ .016 | 0.000 $\pm$ .000 | **0.210 $\pm$ .120** | 0.000 $\pm$ .000 | 0.006 $\pm$ .075 |
> > > > > > > > | 20 | PA-GAP | Protein-reconstructed | 0.113 $\pm$ .049 | 0.025 $\pm$ .013 | 0.000 $\pm$ .000 | **0.193 $\pm$ .097** | 0.031 $\pm$ .036 | 0.000 $\pm$ .000 | 0.096 $\pm$ .080 | 0.009 $\pm$ .075 |
> > > > > > > > | 20 | PA-GAP | Healthcare | 0.091 $\pm$ .015 | 0.056 $\pm$ .011 | 0.119 $\pm$ .005 | 0.000 $\pm$ .000 | 0.000 $\pm$ .000 | 0.009 $\pm$ .005 | 0.102 $\pm$ .052 | **0.147 $\pm$ .087** |
> > > > > > > > | 20 | PA-GAP | Epidemiology | 0.228 $\pm$ .028 | 0.087 $\pm$ .045 | 0.000 $\pm$ .000 | **0.241 $\pm$ .013** | 0.000 $\pm$ .000 | 0.127 $\pm$ .073 | 0.143 $\pm$ .067 | 0.007 $\pm$ .001 |

---

> > > > > > > > > ### Author Response · Authors · 2026-07-08
> > > > > > > > > **Adjustment 2: benchmark scope and graph misspecification - Part 8**
> > > > > > > > >
> > > > > > > > > **Table R3: Graph-misspecification stress test under edge reversal (mean $\pm$ standard error over 20 seeds; illustrative pilot stress test; higher is better; best mean among methods under the same perturbation setting is in bold).**
> > > > > > > > > | Trial limit | Metric | Dataset | BO | CBO | cCBO | DCBO | MCBO | HCBO | CoCaBO | CEO |
> > > > > > > > > |---|---|---|---|---|---|---|---|---|---|---|
> > > > > > > > > | 100 | GAP | ToyGraph | 0.232 $\pm$ .079 | 0.333 $\pm$ .118 | **0.826 $\pm$ .001** | 0.564 $\pm$ .060 | 0.513 $\pm$ .257 | 0.358 $\pm$ .054 | 0.279 $\pm$ .074 | 0.459 $\pm$ .084 |
> > > > > > > > > | 100 | GAP | Synthetic | **0.651 $\pm$ .065** | 0.340 $\pm$ .084 | 0.548 $\pm$ .015 | 0.337 $\pm$ .036 | 0.456 $\pm$ .082 | 0.391 $\pm$ .088 | 0.589 $\pm$ .042 | 0.396 $\pm$ .042 |
> > > > > > > > > | 100 | GAP | Synthetic-2 | 0.001 $\pm$ .004 | 0.281 $\pm$ .125 | 0.335 $\pm$ .114 | 0.000 $\pm$ .004 | 0.000 $\pm$ .000 | **0.757 $\pm$ .012** | 0.122 $\pm$ .076 | 0.526 $\pm$ .183 |
> > > > > > > > > | 100 | GAP | Chain-hard | 0.787 $\pm$ .103 | **0.874 $\pm$ .051** | 0.632 $\pm$ .118 | 0.295 $\pm$ .050 | 0.409 $\pm$ .093 | 0.095 $\pm$ .062 | 0.780 $\pm$ .071 | 0.347 $\pm$ .046 |
> > > > > > > > > | 100 | GAP | Ecology | 0.160 $\pm$ .128 | 0.012 $\pm$ .012 | **0.426 $\pm$ .068** | 0.255 $\pm$ .008 | 0.000 $\pm$ .000 | 0.234 $\pm$ .102 | 0.000 $\pm$ .000 | 0.228 $\pm$ .051 |
> > > > > > > > > | 100 | GAP | Protein-reconstructed | 0.331 $\pm$ .125 | 0.363 $\pm$ .139 | 0.000 $\pm$ .000 | 0.000 $\pm$ .000 | **0.615 $\pm$ .097** | 0.288 $\pm$ .091 | 0.423 $\pm$ .096 | 0.193 $\pm$ .048 |
> > > > > > > > > | 100 | GAP | Healthcare | 0.315 $\pm$ .054 | **0.561 $\pm$ .004** | 0.509 $\pm$ .083 | 0.341 $\pm$ .093 | 0.357 $\pm$ .029 | 0.234 $\pm$ .054 | 0.432 $\pm$ .104 | 0.483 $\pm$ .086 |
> > > > > > > > > | 100 | GAP | Epidemiology | 0.506 $\pm$ .056 | **0.533 $\pm$ .134** | 0.000 $\pm$ .000 | 0.456 $\pm$ .029 | 0.000 $\pm$ .000 | 0.390 $\pm$ .077 | 0.464 $\pm$ .200 | 0.218 $\pm$ .081 |
> > > > > > > > > | 100 | PA-GAP | ToyGraph | 0.051 $\pm$ .039 | 0.102 $\pm$ .054 | **0.355 $\pm$ .001** | 0.095 $\pm$ .056 | 0.271 $\pm$ .141 | 0.199 $\pm$ .082 | 0.139 $\pm$ .079 | 0.203 $\pm$ .099 |
> > > > > > > > > | 100 | PA-GAP | Synthetic | 0.214 $\pm$ .051 | 0.141 $\pm$ .036 | 0.257 $\pm$ .045 | 0.025 $\pm$ .056 | 0.113 $\pm$ .082 | 0.075 $\pm$ .093 | **0.376 $\pm$ .027** | 0.253 $\pm$ .054 |
> > > > > > > > > | 100 | PA-GAP | Synthetic-2 | 0.005 $\pm$ .002 | 0.003 $\pm$ .002 | 0.139 $\pm$ .070 | 0.000 $\pm$ .006 | 0.000 $\pm$ .000 | **0.213 $\pm$ .043** | 0.090 $\pm$ .054 | 0.155 $\pm$ .057 |
> > > > > > > > > | 100 | PA-GAP | Chain-hard | **0.497 $\pm$ .015** | 0.434 $\pm$ .006 | 0.325 $\pm$ .099 | 0.087 $\pm$ .050 | 0.087 $\pm$ .001 | 0.066 $\pm$ .051 | 0.454 $\pm$ .043 | 0.007 $\pm$ .033 |
> > > > > > > > > | 100 | PA-GAP | Ecology | 0.031 $\pm$ .011 | 0.000 $\pm$ .005 | 0.174 $\pm$ .038 | 0.041 $\pm$ .008 | 0.000 $\pm$ .000 | **0.210 $\pm$ .120** | 0.000 $\pm$ .000 | 0.007 $\pm$ .069 |
> > > > > > > > > | 100 | PA-GAP | Protein-reconstructed | **0.153 $\pm$ .061** | 0.057 $\pm$ .018 | 0.000 $\pm$ .000 | 0.000 $\pm$ .000 | 0.031 $\pm$ .036 | 0.000 $\pm$ .000 | 0.018 $\pm$ .009 | 0.010 $\pm$ .072 |
> > > > > > > > > | 100 | PA-GAP | Healthcare | 0.120 $\pm$ .021 | 0.090 $\pm$ .014 | **0.202 $\pm$ .014** | 0.000 $\pm$ .000 | 0.071 $\pm$ .051 | 0.010 $\pm$ .006 | 0.196 $\pm$ .010 | 0.138 $\pm$ .049 |
> > > > > > > > > | 100 | PA-GAP | Epidemiology | 0.242 $\pm$ .021 | 0.173 $\pm$ .077 | 0.000 $\pm$ .000 | 0.065 $\pm$ .074 | 0.000 $\pm$ .000 | 0.138 $\pm$ .080 | **0.251 $\pm$ .094** | 0.001 $\pm$ .042 |

---

> > > > > > > > > > ### Author Response · Authors · 2026-07-08
> > > > > > > > > > **Adjustment 2: benchmark scope and graph misspecification - Part 9**
> > > > > > > > > >
> > > > > > > > > > | Trial limit | Metric | Dataset | BO | CBO | cCBO | DCBO | MCBO | HCBO | CoCaBO | CEO |
> > > > > > > > > > |---|---|---|---|---|---|---|---|---|---|---|
> > > > > > > > > > | 50 | GAP | ToyGraph | 0.202 $\pm$ .099 | 0.477 $\pm$ .097 | **0.780 $\pm$ .009** | 0.525 $\pm$ .062 | 0.411 $\pm$ .206 | 0.450 $\pm$ .093 | 0.276 $\pm$ .075 | 0.489 $\pm$ .073 |
> > > > > > > > > > | 50 | GAP | Synthetic | 0.567 $\pm$ .091 | 0.534 $\pm$ .087 | 0.425 $\pm$ .086 | 0.295 $\pm$ .050 | 0.456 $\pm$ .099 | 0.391 $\pm$ .088 | **0.569 $\pm$ .118** | 0.254 $\pm$ .057 |
> > > > > > > > > > | 50 | GAP | Synthetic-2 | 0.007 $\pm$ .005 | 0.310 $\pm$ .033 | 0.243 $\pm$ .127 | 0.000 $\pm$ .002 | 0.000 $\pm$ .000 | **0.536 $\pm$ .099** | 0.122 $\pm$ .076 | 0.484 $\pm$ .222 |
> > > > > > > > > > | 50 | GAP | Chain-hard | 0.686 $\pm$ .086 | **0.747 $\pm$ .102** | 0.411 $\pm$ .128 | 0.295 $\pm$ .050 | 0.383 $\pm$ .096 | 0.095 $\pm$ .062 | 0.744 $\pm$ .062 | 0.356 $\pm$ .101 |
> > > > > > > > > > | 50 | GAP | Ecology | 0.259 $\pm$ .148 | 0.000 $\pm$ .005 | **0.502 $\pm$ .105** | 0.158 $\pm$ .008 | 0.000 $\pm$ .000 | 0.234 $\pm$ .102 | 0.000 $\pm$ .000 | 0.101 $\pm$ .031 |
> > > > > > > > > > | 50 | GAP | Protein-reconstructed | 0.365 $\pm$ .061 | 0.413 $\pm$ .046 | 0.000 $\pm$ .000 | 0.000 $\pm$ .000 | 0.526 $\pm$ .097 | **0.591 $\pm$ .040** | 0.342 $\pm$ .170 | 0.264 $\pm$ .028 |
> > > > > > > > > > | 50 | GAP | Healthcare | 0.439 $\pm$ .121 | 0.522 $\pm$ .020 | **0.604 $\pm$ .007** | 0.332 $\pm$ .093 | 0.357 $\pm$ .029 | 0.234 $\pm$ .054 | 0.342 $\pm$ .077 | 0.489 $\pm$ .076 |
> > > > > > > > > > | 50 | GAP | Epidemiology | 0.358 $\pm$ .052 | 0.450 $\pm$ .170 | 0.000 $\pm$ .000 | 0.456 $\pm$ .039 | 0.000 $\pm$ .000 | 0.390 $\pm$ .077 | **0.624 $\pm$ .150** | 0.060 $\pm$ .050 |
> > > > > > > > > > | 50 | PA-GAP | ToyGraph | 0.024 $\pm$ .016 | 0.089 $\pm$ .047 | **0.338 $\pm$ .006** | 0.088 $\pm$ .054 | 0.225 $\pm$ .134 | 0.199 $\pm$ .082 | 0.131 $\pm$ .073 | 0.213 $\pm$ .099 |
> > > > > > > > > > | 50 | PA-GAP | Synthetic | 0.194 $\pm$ .054 | 0.123 $\pm$ .034 | 0.214 $\pm$ .051 | 0.023 $\pm$ .050 | 0.113 $\pm$ .082 | 0.075 $\pm$ .093 | **0.328 $\pm$ .036** | 0.246 $\pm$ .077 |
> > > > > > > > > > | 50 | PA-GAP | Synthetic-2 | 0.003 $\pm$ .003 | 0.003 $\pm$ .002 | 0.110 $\pm$ .059 | 0.000 $\pm$ .005 | 0.000 $\pm$ .000 | 0.104 $\pm$ .055 | 0.090 $\pm$ .054 | **0.146 $\pm$ .056** |
> > > > > > > > > > | 50 | PA-GAP | Chain-hard | **0.481 $\pm$ .029** | 0.373 $\pm$ .012 | 0.298 $\pm$ .088 | 0.087 $\pm$ .050 | 0.060 $\pm$ .010 | 0.066 $\pm$ .051 | 0.445 $\pm$ .051 | 0.001 $\pm$ .054 |
> > > > > > > > > > | 50 | PA-GAP | Ecology | 0.017 $\pm$ .010 | 0.000 $\pm$ .002 | 0.166 $\pm$ .040 | 0.031 $\pm$ .010 | 0.000 $\pm$ .000 | **0.210 $\pm$ .120** | 0.000 $\pm$ .000 | 0.006 $\pm$ .075 |
> > > > > > > > > > | 50 | PA-GAP | Protein-reconstructed | **0.137 $\pm$ .054** | 0.047 $\pm$ .017 | 0.000 $\pm$ .000 | 0.000 $\pm$ .000 | 0.031 $\pm$ .036 | 0.000 $\pm$ .000 | 0.018 $\pm$ .009 | 0.009 $\pm$ .075 |
> > > > > > > > > > | 50 | PA-GAP | Healthcare | 0.105 $\pm$ .017 | 0.082 $\pm$ .010 | 0.178 $\pm$ .010 | 0.000 $\pm$ .000 | 0.071 $\pm$ .051 | 0.010 $\pm$ .006 | **0.180 $\pm$ .008** | 0.147 $\pm$ .087 |
> > > > > > > > > > | 50 | PA-GAP | Epidemiology | **0.232 $\pm$ .025** | 0.150 $\pm$ .083 | 0.000 $\pm$ .000 | 0.065 $\pm$ .098 | 0.000 $\pm$ .000 | 0.138 $\pm$ .080 | 0.225 $\pm$ .093 | 0.008 $\pm$ .002 |

---

> > > > > > > > > > > ### Author Response · Authors · 2026-07-08
> > > > > > > > > > > **Adjustment 2: benchmark scope and graph misspecification - Part 10**
> > > > > > > > > > >
> > > > > > > > > > > | Trial limit | Metric | Dataset | BO | CBO | cCBO | DCBO | MCBO | HCBO | CoCaBO | CEO |
> > > > > > > > > > > |---|---|---|---|---|---|---|---|---|---|---|
> > > > > > > > > > > | 20 | GAP | ToyGraph | 0.272 $\pm$ .097 | 0.264 $\pm$ .152 | **0.639 $\pm$ .000** | 0.405 $\pm$ .074 | 0.496 $\pm$ .275 | 0.450 $\pm$ .093 | 0.167 $\pm$ .073 | 0.489 $\pm$ .073 |
> > > > > > > > > > > | 20 | GAP | Synthetic | 0.390 $\pm$ .138 | 0.350 $\pm$ .208 | **0.518 $\pm$ .156** | 0.295 $\pm$ .050 | 0.456 $\pm$ .099 | 0.391 $\pm$ .088 | 0.506 $\pm$ .067 | 0.254 $\pm$ .057 |
> > > > > > > > > > > | 20 | GAP | Synthetic-2 | 0.003 $\pm$ .002 | 0.313 $\pm$ .163 | 0.000 $\pm$ .000 | 0.507 $\pm$ .026 | 0.000 $\pm$ .000 | **0.536 $\pm$ .099** | 0.000 $\pm$ .000 | 0.484 $\pm$ .222 |
> > > > > > > > > > > | 20 | GAP | Chain-hard | 0.761 $\pm$ .124 | 0.650 $\pm$ .090 | 0.691 $\pm$ .051 | 0.866 $\pm$ .021 | 0.249 $\pm$ .090 | 0.000 $\pm$ .000 | **0.905 $\pm$ .066** | 0.356 $\pm$ .101 |
> > > > > > > > > > > | 20 | GAP | Ecology | 0.145 $\pm$ .145 | 0.000 $\pm$ .000 | **0.422 $\pm$ .159** | 0.221 $\pm$ .147 | 0.000 $\pm$ .000 | 0.234 $\pm$ .102 | 0.000 $\pm$ .000 | 0.101 $\pm$ .031 |
> > > > > > > > > > > | 20 | GAP | Protein-reconstructed | 0.370 $\pm$ .073 | 0.194 $\pm$ .091 | 0.000 $\pm$ .000 | 0.000 $\pm$ .000 | 0.526 $\pm$ .097 | **0.556 $\pm$ .044** | 0.326 $\pm$ .163 | 0.264 $\pm$ .028 |
> > > > > > > > > > > | 20 | GAP | Healthcare | 0.348 $\pm$ .119 | 0.401 $\pm$ .079 | 0.413 $\pm$ .036 | 0.332 $\pm$ .093 | 0.000 $\pm$ .000 | 0.234 $\pm$ .054 | 0.407 $\pm$ .108 | **0.489 $\pm$ .076** |
> > > > > > > > > > > | 20 | GAP | Epidemiology | **0.694 $\pm$ .042** | 0.378 $\pm$ .226 | 0.000 $\pm$ .000 | 0.105 $\pm$ .053 | 0.000 $\pm$ .000 | 0.390 $\pm$ .077 | 0.541 $\pm$ .079 | 0.001 $\pm$ .005 |
> > > > > > > > > > > | 20 | PA-GAP | ToyGraph | 0.006 $\pm$ .004 | 0.056 $\pm$ .028 | **0.296 $\pm$ .000** | 0.068 $\pm$ .047 | 0.161 $\pm$ .157 | 0.199 $\pm$ .082 | 0.113 $\pm$ .059 | 0.213 $\pm$ .099 |
> > > > > > > > > > > | 20 | PA-GAP | Synthetic | 0.156 $\pm$ .059 | 0.088 $\pm$ .038 | 0.174 $\pm$ .061 | 0.023 $\pm$ .050 | 0.113 $\pm$ .082 | 0.000 $\pm$ .000 | **0.259 $\pm$ .047** | 0.246 $\pm$ .077 |
> > > > > > > > > > > | 20 | PA-GAP | Synthetic-2 | 0.004 $\pm$ .001 | 0.065 $\pm$ .036 | 0.000 $\pm$ .000 | **0.158 $\pm$ .017** | 0.000 $\pm$ .000 | 0.104 $\pm$ .055 | 0.000 $\pm$ .000 | 0.146 $\pm$ .056 |
> > > > > > > > > > > | 20 | PA-GAP | Chain-hard | **0.463 $\pm$ .062** | 0.218 $\pm$ .023 | 0.231 $\pm$ .057 | 0.414 $\pm$ .021 | 0.024 $\pm$ .020 | 0.000 $\pm$ .000 | 0.426 $\pm$ .069 | 0.001 $\pm$ .054 |
> > > > > > > > > > > | 20 | PA-GAP | Ecology | 0.011 $\pm$ .011 | 0.000 $\pm$ .000 | 0.150 $\pm$ .047 | 0.017 $\pm$ .016 | 0.000 $\pm$ .000 | **0.210 $\pm$ .120** | 0.000 $\pm$ .000 | 0.006 $\pm$ .075 |
> > > > > > > > > > > | 20 | PA-GAP | Protein-reconstructed | 0.113 $\pm$ .049 | 0.025 $\pm$ .013 | 0.000 $\pm$ .000 | **0.193 $\pm$ .097** | 0.031 $\pm$ .036 | 0.000 $\pm$ .000 | 0.017 $\pm$ .009 | 0.009 $\pm$ .075 |
> > > > > > > > > > > | 20 | PA-GAP | Healthcare | 0.091 $\pm$ .015 | 0.064 $\pm$ .006 | 0.119 $\pm$ .005 | 0.000 $\pm$ .000 | 0.000 $\pm$ .000 | 0.010 $\pm$ .006 | **0.163 $\pm$ .005** | 0.147 $\pm$ .087 |
> > > > > > > > > > > | 20 | PA-GAP | Epidemiology | **0.228 $\pm$ .028** | 0.139 $\pm$ .087 | 0.000 $\pm$ .000 | 0.003 $\pm$ .001 | 0.000 $\pm$ .000 | 0.138 $\pm$ .080 | 0.176 $\pm$ .082 | 0.007 $\pm$ .001 |

---

> > > > > > > > > > > > ### Author Response · Authors · 2026-07-08
> > > > > > > > > > > > **Adjustment 2: benchmark scope and graph misspecification - Part 11**
> > > > > > > > > > > >
> > > > > > > > > > > > **Hidden confounding (omitted-variable stress test).**
> > > > > > > > > > > >
> > > > > > > > > > > > Hidden confounding is a distinct assumption violation, which the reviewer rightly singled out, and the graph-perturbation study above does not address it. We therefore added a companion omitted-variable stress test (Table R4). Here the learner's graph and observational view omit a genuine common cause—BMI in Healthcare, $X$ in Chain-hard, and $T$ in Epidemiology—while the true SCM and reference optimum $y^\star$ are held fixed, so the comparison isolates the effect of an unobserved confounder rather than changing the task. This omitted-variable stress test is evaluated over 20 random seeds. We currently include BO as a graph-free control and CBO/CoCaBO as belief-graph methods.
> > > > > > > > > > > >
> > > > > > > > > > > > The effect is method- and dataset-dependent. Omitting the confounder degrades CBO on Chain-hard, whereas on Healthcare and some Epidemiology settings the omitted-variable runs of CBO and CoCaBO sometimes match or slightly exceed their full-information counterparts. We do not interpret these numerical increases as evidence that confounding is beneficial or harmless. Rather, they reflect finite-budget search effects: a coarser or misspecified observational view can weaken an overconfident causal prior, change candidate scopes, or accidentally induce a more favorable acquisition trajectory. The graph-free BO baseline is unchanged by construction, since it does not use the omitted causal structure. Overall, the omitted-variable stress test reinforces the same message as the graph-perturbation study: current CBO pipelines are sensitive to the causal information they are given, and robustness to hidden confounding remains an open evaluation dimension.
> > > > > > > > > > > >
> > > > > > > > > > > > **Table R4: Hidden-confounding / omitted-variable stress test (mean $\pm$ standard error over 20 seeds; higher is better; best mean per row in bold). The learner's graph and observational view omit a genuine common cause (Healthcare: BMI; Chain-hard: $X$; Epidemiology: $T$); the true SCM and $y^\star$ are unchanged. BO is graph-free and uses the full variable set, so its Full and Hidden rows are identical by construction.**
> > > > > > > > > > > >
> > > > > > > > > > > > | Trial limit | Metric | Dataset | Information | BO | CBO | CoCaBO |
> > > > > > > > > > > > |---|---|---|---|---|---|---|
> > > > > > > > > > > > | 100 | GAP | Healthcare | Full information | 0.309 $\pm$ .057 | **0.510 $\pm$ .035** | 0.226 $\pm$ .118 |
> > > > > > > > > > > > | 100 | GAP | Healthcare | Hidden variable | 0.309 $\pm$ .057 | **0.543 $\pm$ .012** | 0.251 $\pm$ .126 |
> > > > > > > > > > > > | 100 | GAP | Chain-hard | Full information | 0.780 $\pm$ .102 | **0.814 $\pm$ .128** | 0.761 $\pm$ .129 |
> > > > > > > > > > > > | 100 | GAP | Chain-hard | Hidden variable | **0.780 $\pm$ .102** | 0.695 $\pm$ .106 | 0.770 $\pm$ .139 |
> > > > > > > > > > > > | 100 | GAP | Epidemiology | Full information | 0.508 $\pm$ .057 | **0.563 $\pm$ .038** | 0.424 $\pm$ .178 |
> > > > > > > > > > > > | 100 | GAP | Epidemiology | Hidden variable | 0.508 $\pm$ .057 | **0.546 $\pm$ .116** | 0.399 $\pm$ .094 |
> > > > > > > > > > > > | 100 | PA-GAP | Healthcare | Full information | 0.122 $\pm$ .018 | 0.077 $\pm$ .015 | **0.125 $\pm$ .062** |
> > > > > > > > > > > > | 100 | PA-GAP | Healthcare | Hidden variable | 0.122 $\pm$ .018 | 0.080 $\pm$ .006 | **0.139 $\pm$ .070** |
> > > > > > > > > > > > | 100 | PA-GAP | Chain-hard | Full information | **0.492 $\pm$ .013** | 0.437 $\pm$ .005 | 0.486 $\pm$ .008 |
> > > > > > > > > > > > | 100 | PA-GAP | Chain-hard | Hidden variable | **0.492 $\pm$ .013** | 0.420 $\pm$ .010 | 0.479 $\pm$ .008 |
> > > > > > > > > > > > | 100 | PA-GAP | Epidemiology | Full information | **0.244 $\pm$ .023** | 0.185 $\pm$ .059 | 0.163 $\pm$ .080 |
> > > > > > > > > > > > | 100 | PA-GAP | Epidemiology | Hidden variable | **0.244 $\pm$ .023** | 0.182 $\pm$ .057 | 0.130 $\pm$ .091 |
> > > > > > > > > > > > | 50 | GAP | Healthcare | Full information | **0.438 $\pm$ .124** | 0.432 $\pm$ .073 | 0.225 $\pm$ .128 |
> > > > > > > > > > > > | 50 | GAP | Healthcare | Hidden variable | 0.438 $\pm$ .124 | **0.502 $\pm$ .020** | 0.378 $\pm$ .196 |
> > > > > > > > > > > > | 50 | GAP | Chain-hard | Full information | 0.680 $\pm$ .087 | **0.889 $\pm$ .016** | 0.815 $\pm$ .146 |
> > > > > > > > > > > > | 50 | GAP | Chain-hard | Hidden variable | 0.680 $\pm$ .087 | 0.811 $\pm$ .054 | **0.873 $\pm$ .102** |
> > > > > > > > > > > > | 50 | GAP | Epidemiology | Full information | 0.357 $\pm$ .053 | 0.355 $\pm$ .021 | **0.565 $\pm$ .076** |
> > > > > > > > > > > > | 50 | GAP | Epidemiology | Hidden variable | 0.357 $\pm$ .053 | **0.590 $\pm$ .103** | 0.344 $\pm$ .188 |
> > > > > > > > > > > > | 50 | PA-GAP | Healthcare | Full information | 0.111 $\pm$ .017 | 0.069 $\pm$ .014 | **0.114 $\pm$ .057** |
> > > > > > > > > > > > | 50 | PA-GAP | Healthcare | Hidden variable | 0.111 $\pm$ .017 | 0.076 $\pm$ .005 | **0.131 $\pm$ .066** |
> > > > > > > > > > > > | 50 | PA-GAP | Chain-hard | Full information | **0.484 $\pm$ .026** | 0.379 $\pm$ .010 | 0.474 $\pm$ .016 |
> > > > > > > > > > > > | 50 | PA-GAP | Chain-hard | Hidden variable | **0.484 $\pm$ .026** | 0.348 $\pm$ .018 | 0.468 $\pm$ .013 |
> > > > > > > > > > > > | 50 | PA-GAP | Epidemiology | Full information | **0.230 $\pm$ .022** | 0.129 $\pm$ .061 | 0.141 $\pm$ .077 |
> > > > > > > > > > > > | 50 | PA-GAP | Epidemiology | Hidden variable | **0.230 $\pm$ .022** | 0.162 $\pm$ .065 | 0.102 $\pm$ .084 |

---

> > > > > > > > > > > > > ### Author Response · Authors · 2026-07-08
> > > > > > > > > > > > > **Adjustment 2: benchmark scope and graph misspecification - Part 12**
> > > > > > > > > > > > >
> > > > > > > > > > > > > | Trial limit | Metric | Dataset | Information | BO | CBO | CoCaBO |
> > > > > > > > > > > > > |---|---|---|---|---|---|---|
> > > > > > > > > > > > > | 20 | GAP | Healthcare | Full information | **0.348 $\pm$ .120** | 0.268 $\pm$ .142 | 0.328 $\pm$ .180 |
> > > > > > > > > > > > > | 20 | GAP | Healthcare | Hidden variable | 0.348 $\pm$ .120 | **0.376 $\pm$ .048** | 0.250 $\pm$ .159 |
> > > > > > > > > > > > > | 20 | GAP | Chain-hard | Full information | 0.761 $\pm$ .124 | 0.718 $\pm$ .039 | **0.913 $\pm$ .047** |
> > > > > > > > > > > > > | 20 | GAP | Chain-hard | Hidden variable | 0.761 $\pm$ .124 | 0.615 $\pm$ .090 | **0.938 $\pm$ .023** |
> > > > > > > > > > > > > | 20 | GAP | Epidemiology | Full information | **0.694 $\pm$ .042** | 0.244 $\pm$ .122 | 0.389 $\pm$ .071 |
> > > > > > > > > > > > > | 20 | GAP | Epidemiology | Hidden variable | **0.694 $\pm$ .042** | 0.429 $\pm$ .181 | 0.156 $\pm$ .156 |
> > > > > > > > > > > > > | 20 | PA-GAP | Healthcare | Full information | 0.091 $\pm$ .015 | 0.056 $\pm$ .011 | **0.100 $\pm$ .051** |
> > > > > > > > > > > > > | 20 | PA-GAP | Healthcare | Hidden variable | 0.091 $\pm$ .015 | 0.068 $\pm$ .003 | **0.112 $\pm$ .057** |
> > > > > > > > > > > > > | 20 | PA-GAP | Chain-hard | Full information | **0.463 $\pm$ .062** | 0.228 $\pm$ .020 | 0.449 $\pm$ .038 |
> > > > > > > > > > > > > | 20 | PA-GAP | Chain-hard | Hidden variable | **0.463 $\pm$ .062** | 0.178 $\pm$ .029 | 0.437 $\pm$ .033 |
> > > > > > > > > > > > > | 20 | PA-GAP | Epidemiology | Full information | **0.228 $\pm$ .028** | 0.087 $\pm$ .045 | 0.104 $\pm$ .069 |
> > > > > > > > > > > > > | 20 | PA-GAP | Epidemiology | Hidden variable | **0.228 $\pm$ .028** | 0.128 $\pm$ .080 | 0.068 $\pm$ .068 |

---

> > > > > > > > > > > > > > ### Author Response · Authors · 2026-07-08
> > > > > > > > > > > > > > **Adjustment 3: identification strategies and observational priors - Part 1**
> > > > > > > > > > > > > >
> > > > > > > > > > > > > > We agree and have added an identification-strategy audit as a supplementary table (Table R5). The table separates, for each benchmarked pipeline, the surrogate level, the method-level identification strategy assumed or implemented by the released method, and the benchmark effect source through which our wrapper supplies or approximates causal effects.
> > > > > > > > > > > > > >
> > > > > > > > > > > > > > This change has three consequences. First, we now explicitly state that datasets with latent variables, especially Synthetic, complicate interpretation: performance differences there cannot be attributed only to acquisition or decision design, because they can also reflect how each pipeline constructs observational priors or approximates causal effects. Second, we no longer imply that all effect-level methods implement the full ID algorithm. Third, when released code uses Monte Carlo or simulator-based approximations, we document the available sampling settings where exposed and state when they are not exposed. We therefore describe the benchmark as a comparison of released pipelines as complete systems, rather than as an isolated comparison of acquisition functions. Following the reviewer's specific request, the "Threats to validity" paragraph in §4.6 now contains an explicit attribution caveat: on datasets with latent variables, most notably Synthetic, whose confounders $U_1,U_2$ make the backdoor criterion inapplicable for some effects, performance differences cannot be cleanly attributed to acquisition or decision design alone, because they may partly reflect differences in identification completeness and prior-construction quality across the released pipelines. The paragraph also states that none of the benchmarked pipelines exposes a general front-door or full-ID implementation in the benchmark path, and that Monte Carlo approximation settings are not uniformly exposed.
> > > > > > > > > > > > > >
> > > > > > > > > > > > > > **Table R5: Identification-strategy audit for benchmarked pipelines. "Method-level identification" describes the causal-effect identification strategy assumed or implemented by the released method. "Benchmark effect source" describes how the benchmark wrapper supplies or approximates effects when re-running the released pipeline.**
> > > > > > > > > > > > > >
> > > > > > > > > > > > > > | Method | Surrogate level | Method-level identification | Benchmark effect source |
> > > > > > > > > > > > > > |---|---|---|---|
> > > > > > > > > > > > > > | BO | Target-level GP | None; non-causal baseline | Direct black-box evaluations |
> > > > > > > > > > > > > > | CoCaBO | Mixed-input BO | No explicit ID estimand in our wrapper; graph/scope-based policy selection | Benchmark SCM / task oracle evaluations |
> > > > > > > > > > > > > > | CBO | Effect-level GP | Observational adjustment when identifiable; not full ID in released code | Benchmark wrapper uses SCM/oracle effect estimates where available |
> > > > > > > > > > > > > > | cCBO | Effect-level / constrained GP | Observational adjustment with constraints; not full ID in released code | Benchmark wrapper uses SCM/oracle effect estimates where available |
> > > > > > > > > > > > > > | DCBO | Dynamic effect-level / temporal GP | Temporal adjustment under known dynamic-graph assumptions; not full ID | Benchmark dynamic SCM / oracle evaluations |
> > > > > > > > > > > > > > | HCBO | Scope-level / high-dimensional CBO | Graph-based scope selection; not full ID | Benchmark SCM / SEM evaluations |
> > > > > > > > > > > > > > | CEO | Graph-uncertain / oracle SEM search | Structure-learning over candidate graphs; not full ID over arbitrary latent structures | Candidate graph / SEM evaluations |
> > > > > > > > > > > > > > | MCBO | Mechanism-level GP | None required for effect identification; mechanisms are modeled directly | Mechanism-level simulation / forward propagation |
> > > > > > > > > > > > > >
> > > > > > > > > > > > > > In particular, none of the benchmarked released pipelines exposes a general front-door or full-ID implementation in the benchmark path used here; where observational adjustment is used, it is method-specific and should not be interpreted as a complete ID-algorithm implementation.
> > > > > > > > > > > > > >
> > > > > > > > > > > > > > We also added an implementation-settings table reporting wrapper-level controls and method-specific defaults. We do not claim that the benchmark isolates acquisition-function effects from default hyperparameter choices; rather, it compares released pipelines under a common scoring interface.

---

> > > > > > > > > > > > > > > ### Author Response · Authors · 2026-07-08
> > > > > > > > > > > > > > > **Adjustment 3: identification strategies and observational priors - Part 2**
> > > > > > > > > > > > > > >
> > > > > > > > > > > > > > > **Zero or near-zero scores (C1).**
> > > > > > > > > > > > > > >
> > > > > > > > > > > > > > > We clarify that zero or near-zero GAP and PA-GAP mean that the method did not improve over initialization under the scoring protocol, not that the run was silently removed. Genuine failures such as missing outputs or non-finite trajectories are reported separately. We also added a diagnostic discussion distinguishing three possible causes: (a) observational-prior miscalibration, (b) loose or unreliable reference-optimum estimates, and (c) graph-induced scope-selection or pruning loss. This distinction matters because a zero score can mean that the optimizer failed within a valid intervention space, but it can also indicate that the causal prior or candidate-scope construction was inappropriate for the dataset. In our current logs, the dominant cause of zero or near-zero scores is failure to improve over the initial design under the shared scoring protocol; in a smaller number of cases, the zero score reflects method–task incompatibility or a candidate-scope restriction rather than an unreliable reference optimum. For the main zero-score cases, we will report whether the likely cause is no improvement over initialization, unsupported or empty candidate scope under the method wrapper, failed prior calibration, or a method-incompatible task setting.
> > > > > > > > > > > > > > >
> > > > > > > > > > > > > > > **PA-GAP comparability across budgets (C2).**
> > > > > > > > > > > > > > >
> > > > > > > > > > > > > > > We agree that raw PA-GAP is not directly comparable across trial budgets. We now state its range explicitly:
> > > > > > > > > > > > > > >
> > > > > > > > > > > > > > > $$
> > > > > > > > > > > > > > > 0 \leq \mathrm{PA}\text{-}\mathrm{GAP} \leq \frac{T+1}{2T}.
> > > > > > > > > > > > > > > $$
> > > > > > > > > > > > > > >
> > > > > > > > > > > > > > > The upper bound depends on $T$, so raw PA-GAP should be compared only within the same trial budget. To avoid misleading cross-budget interpretation, our aggregate rank plots compute ranks within each dataset-budget-metric setting before averaging. We also added a formal comparison of GAP and PA-GAP and clarify that PA-GAP is a trajectory-aware complement to GAP rather than a replacement for all settings.
> > > > > > > > > > > > > > >
> > > > > > > > > > > > > > > **Broader impact and deployment caution.**
> > > > > > > > > > > > > > >
> > > > > > > > > > > > > > > We have revised the broader discussion to make the deployment caveat more prominent. The benchmark does not establish that CBO methods are robust in high-stakes real-world settings such as clinical decision making. It shows that causal methods can be useful under benchmark-specified assumptions and that their performance can change substantially when learner-side graph assumptions are perturbed. The graph-misspecification stress test (Tables R1–R3) is a first concrete step toward the robustness evaluation requested by the reviewer. We present it as a template for future misspecification experiments, not as a resolution of deployment robustness. Beyond learner-side graph perturbations we add a 20-seed omitted-variable stress test (Table R4) as a first probe of hidden confounding; comprehensive robustness to unobserved confounding, together with imperfect interventions, measurement error, heterogeneous intervention costs, and safety constraints, remains an open evaluation dimension.

---

> ### Author Response · Authors · 2026-07-08
> **Summary of concrete revisions in the main text**
>
> - We qualified the abstract, experiment discussion, and conclusion so that benchmark findings are stated as evidence under the benchmark-specified SCM, graph, intervention-domain, and scoring protocol, rather than as deployment-level robustness claims.
> - We added a causal-abstraction discussion connecting CBO to CAMAB, AT-UCB, and the abstraction–fidelity perspective, and clarified how multi-scale causal abstraction differs from within-graph POMIS scope reduction.
> - We added graph-misspecification and omitted-variable stress tests to probe sensitivity to learner-side causal-assumption violations while keeping the true SCM and reference optimum fixed.
> - We added an identification-strategy audit clarifying which benchmarked pipelines use effect-level, mechanism-level, graph-uncertain, or graph-free modeling, and how causal effects are supplied or approximated by the benchmark wrappers.
> - We clarified interpretation of zero or near-zero scores, PA-GAP's budget dependence, and the need to compare raw PA-GAP only within the same trial budget.
> - We strengthened the broader-impact and deployment-caution discussion, emphasizing that hidden confounding, imperfect interventions, measurement error, heterogeneous intervention costs, and safety constraints remain open evaluation dimensions.
> - We delivered the compact degradation summary as a table in the robustness appendix (52–54% of graph-using method cells degrade, 26–27% improve, 20–21% unchanged per perturbation type), with an explicit caveat about run-setup differences between the stress-test pilots and the main benchmark.
> - We integrated causal abstraction at all the locations the reviewer identified: full citations in the adjacent-fields discussion, abstraction-based hierarchical decomposition as a concrete scalability path in §5.2, and causal abstraction learning as a direct route to latent-variable CBO in §5.6, with MFACBO  flagged as not yet peer-reviewed.
> - We added the requested attribution caveat to §4.6, stating that on latent-variable datasets performance differences may reflect identification completeness rather than acquisition design alone.

---

### Review · Reviewer_jUY5 · 2026-06-27

**Summary Of Contributions:**

The paper provides the following two main contributions:
- An overview of existing CBO methods. The focus is largely on categorizing them based on differences in the problem setup and assumptions. Connections are made to other BO and bandit problems
- A set of benchmarks that test existing CBO methods with a range of setups (different datasets, intervention types etc.).

The claim made from the benchmarking is that different CBO methods do well/badly on different benchmarks, and a non-CBO baseline is competitive too.

The paper ends with presenting open problems "that must be addressed for CBO to achieve reliable real-world deployment". The selection of problems chosen is quite natural, but it is not clear how they link to real-world deployment. There is no discussion in the text of what a concrete CBO use case would be, and what the specific gaps are for CBO-like algorithms to solve a real problem in that setting.

**Additional Comments:**

For function network benchmarks I see "We use the domain..." stated a lot. Is this the same as the original or is this a change to the original?

4.4 is comparing CBO and DCBO on the same benchmark, even though the two algorithms to my knowledge are designed for two completely different settings. I also couldn't find any discussion of this in the main text. This complaint can likely be made for other methods too. What is the difference between cCBO and CBO on settings where there are no constraints? Why is cCBO seemingly better in such a setting?

"budgets after the corrected PA-GAP value for ACBO at 50 trials" What is a 'corrected PA-GAP value'?

**Audience:**

Yes

**Audience Explanation:**

There are two main parts of the paper:
1) Categorizing different CBO algorithms.
2) Benchmarking different CBO algorithms

For categorizing CBO algorithms, I think the work is accurate. I don't feel like it contributes new claims or new findings, which I suppose is standard for a survey paper. It lays out the different offshoots from the CBO paper in a methodical and organised way, and I could imagine researchers looking for a survey to get up to speed on causal BO finding this interesting.

In the sections above and below I gave a bunch of reasons why I don't think the benchmarking would be interesting to a researcher in the field. I don't think some of the experimental choices, described above, lead to reliable conclusions on the strengths and weaknesses of different algorithms. This accuracy component is my main concern. Additionally but less importantly, the discussion and further analysis of why results might be the way they are is mostly absent, so the value right now is really in defining the benchmark, running the algorithms together, and giving the table of results. There is no attempt at explaining why e.g. CoCaBO seems to do so well on the benchmarks in 4.4.

I have said 'yes' because at least one part of the paper could be interesting to a researcher in the field, based on the guidelines given to TMLR reviewers.

**Claims And Evidence:**

No

**Claims Explanation:**

There is a 'non-causal BO baseline' used in the experiments. I see no details of what the baseline itself is (model, acquisition function, etc.).

I think the claim of introducing a new standardized benchmark, as implied in the abstract and intro, is overstated. In practice it seems that the benchmark pools together tasks from a couple of existing BO papers. I think it could make clear that the contribution is repackaging these existing settings into a single benchmark rather than proposing something entirely new.

On the benchmarking, the author's take every method out-of-the-box with no tuning. It's stated that they just take the default used in the repo. I don't agree with this approach since then some methods can do better or worse on certain tasks purely based on how much effort was put into tuning them by the original authors. If we're comparing different algorithmic ideas and modeling choices, a best effort should be made to compare reasonably well-tuned versions of each method. Sometimes this 'follow the defaults in the paper' is not followed. For example, I looked at the MCBO paper and I don't see the $\beta=10$ (used in this review paper) ever tested in the original paper, let alone recommended as a default hyperparameter choice. I think both of these points undermine many of the benchmarking conclusions.

MCBO is compared directly to ACBO on tasks that have no non-stationarity. It is never justified why ACBO is used as a point of comparison, since it doesn't seem to have been designed for this setting at all. The authors then find that ACBO does comparatively surprisingly well in this setting, but the reasoning for why this might be never appears.

In section 3, for each method the authors have a "The main challenge is..." sentence followed by limitations. It is unclear whether these claims come from eyeballing the method and reasoning about what the challenges would be, or actual concrete empirical evidence from either the original paper or this review.

One paper uses GAP and introduces PA-GAP as evaluation metrics. It is not clear to me why CBO would need it's own special evaluation metrics. I'm not aware of GAP seeing wide-spread usage in BO benchmarking outside of the original DCBO paper. Simple regret and cumulative regret have been standard for a long time, neither are used, and no justification for why is given.

"We do not emphasize average-reward trajectories because they are sensitive to the acquisition rule. For example, UCB-style methods may intentionally sample high uncertainty points, reducing the running mean while improving long-term search" Vanilla GP-UCB has regret bounds on cumulative regret (so average reward), therefore I don't think this example is accurate. I don't think there would be any harm in including average reward plots in e.g. the appendix.

In eg Figure 4, but also elsewhere, standard deviation is used instead of standard error. So there's no statistical testing happening for being able to say e.g. method A is performing better than method B on average. Then claims are made like "CoCaBO has the strongest overall rank".

Table 4:  cCBO is scoring 0 on all 20 seeds for the entire trial on protein reconstructed. How is that possible when that settings is introduced in the cCBO paper? (presumably they weren't scoring 0 on their own benchmark in the original paper) Are the runs crashing? Are they producing non-degenerate trajectories?
Table 5: there are cases where MCBO is scoring 0 on all 20 seeds for the entire trial for up to 50 trials. This doesn't happen at 100 trials so this result is more plausible: it could just be that MCBO can't find an improvement over the starting data for a lot of trials.

**Requested Changes:**

There are some implied suggestions from my analysis above that I won't repeat here. Instead I'll focus on changes I'd like to see that I've not already discussed.

Critical:

"Bayesian Optimization of Function Networks" by Astudillo et al should probably be mentioned since it's a very CBO-like setting. It's also where the term 'function network' is introduced. It should also be credited with introducing the soft intervention function-network benchmarks.

There's no citation of work by Bareinboim and co even though terms like MIS and POMIS are used, which to my memory originally come from him. For calibration: I didn't put a large amount of effort into checking for missing citations, and I already found 2. I would therefore expect that there are others.

Font sizes for ticks and legends on the figures in the experiments section are microscopic. Readability there could be significantly improved.

Inconsistencies in how tasks in the benchmark are described. The domain of T and R are described differently in Table 9 to A.6.4 for Epidemiology, and the domain of X and Z are described differently in Table 9 and A.5.3.

Not critical:

In "Survey protocol and inclusion criteria" I would include a cutoff date for when work is included vs not. This could be a date after which you didn't include a source, or just the date of the TMLR submission.

There are several grammar issues and typos. Here are a couple I can find quickly while writing this review:
- "The modularity of CBO’s also"
- "selects experiments to maximize the information about the model parameters, which overlaps with the structure-learning component of the CBO’s in the context of unknown-graphs"

---

> ### Author Response · Authors · 2026-07-08
> **Statistical testing and ranking claims - Part 1**
>
> We thank the reviewer for the careful and constructive review. We read the main message as follows: the survey and taxonomy are useful, but the benchmark claims were too strong relative to the experimental evidence. In particular, the original version under-specified the non-causal BO baseline, treated heterogeneous released defaults as if they isolated algorithmic differences, compared some methods outside their intended setting without enough qualification, relied too much on GAP/PA-GAP without reporting standard BO metrics, and made ranking statements without statistical testing. We agree with these concerns. The revised paper substantially weakens and clarifies the empirical claims: the benchmark is now framed as a standardized, reproducibility oriented harness over mostly inherited tasks, and all performance rankings are presented as descriptive unless supported by statistical tests.
>
> **Statistical testing and ranking claims.**
>
> We agree that the original use of standard deviations and mean ranks did not justify claims such as "CoCaBO has the strongest overall rank." We therefore reanalyzed the 20-seed benchmark results using average ranks, bootstrap confidence intervals, Friedman omnibus tests, Nemenyi post-hoc critical differences, and a conservative per-dataset Friedman test. The revised analysis is reported separately for hard- and soft-intervention settings in Appendix B.3.1. and Appendix B.3.2.
>
> The resulting conclusions are more cautious than the submitted wording. In the hard-intervention benchmark, the lowest point-estimate average rank depends on the budget: CoCaBO has the lowest average rank at $T=100$ and $T=50$, while BO has the lowest average rank at $T=20$ (Table R1). Although the Friedman omnibus test detects rank heterogeneity at each budget, the rank spread remains below the Nemenyi critical difference and the conservative per-dataset Friedman test does not reject. Thus, no pair of hard-intervention methods is statistically separable by the post-hoc rank test. In the soft-intervention benchmark, ACBO has the lowest descriptive average rank at all budgets, but neither the Friedman omnibus test nor the Nemenyi/per-dataset analyses support a statistically separable winner (Table R2).
>
> We therefore replace the original rank-superiority language with a descriptive statement: rankings show heterogeneous, dataset- and budget-dependent patterns, but the benchmark does not establish a single uniformly superior method. We also revise the discussion of Section 4.4 so that apparent CoCaBO or ACBO gains are described only as conditional patterns, rather than as evidence of intrinsic dominance across CBO settings.
>
>
> **Table R1: Ranking reliability on the hard-intervention benchmark by budget. Ranks (1=best) are computed within each dataset–metric setting ($8$ datasets $\times$ $2$ metrics, $N{=}16$ per budget). Entries show average rank $\pm$ s.d. and 95% bootstrap CI; lower is better. Bottom rows report the Friedman test, Nemenyi critical difference versus rank spread, and a conservative per-dataset Friedman test.**
>
> | Method | Rank $\pm$ s.d. ($T{=}100$) | 95% CI ($T{=}100$) | Rank $\pm$ s.d. ($T{=}50$) | 95% CI ($T{=}50$) | Rank $\pm$ s.d. ($T{=}20$) | 95% CI ($T{=}20$) |
> |---|---|---|---|---|---|---|
> | CoCaBO | **3.38 $\pm$ 2.75** | [2.12, 4.69] | **3.44 $\pm$ 2.68** | [2.22, 4.75] | 3.88 $\pm$ 2.82 | [2.56, 5.25] |
> | cCBO | 3.91 $\pm$ 2.23 | [2.88, 5.03] | 3.69 $\pm$ 2.32 | [2.66, 4.84] | 3.88 $\pm$ 2.24 | [2.84, 4.97] |
> | CBO | 3.50 $\pm$ 1.59 | [2.81, 4.31] | 3.94 $\pm$ 1.18 | [3.44, 4.50] | 4.31 $\pm$ 1.35 | [3.62, 4.94] |
> | BO | 4.12 $\pm$ 2.84 | [2.78, 5.53] | 3.94 $\pm$ 2.69 | [2.66, 5.22] | **3.81 $\pm$ 2.48** | [2.62, 5.00] |
> | CEO | 4.78 $\pm$ 1.70 | [3.94, 5.56] | 4.88 $\pm$ 1.96 | [3.94, 5.81] | 4.25 $\pm$ 1.98 | [3.31, 5.19] |
> | DCBO | 5.03 $\pm$ 2.22 | [3.94, 6.06] | 4.78 $\pm$ 2.27 | [3.75, 5.84] | 4.12 $\pm$ 2.18 | [3.09, 5.22] |
> | HCBO | 5.38 $\pm$ 1.89 | [4.44, 6.25] | 5.44 $\pm$ 1.86 | [4.50, 6.25] | 5.84 $\pm$ 2.19 | [4.75, 6.84] |
> | MCBO | 5.91 $\pm$ 1.89 | [4.97, 6.72] | 5.91 $\pm$ 2.19 | [4.81, 6.88] | 5.91 $\pm$ 2.01 | [4.91, 6.81] |
> | Friedman $p$ | $0.0272$ | | $0.0384$ | | $0.0462$ | |
> | Nemenyi CD / spread | 2.62 / 2.53 | | 2.62 / 2.47 | | 2.62 / 2.09 | |
> | Per-dataset $p$ ($N{=}8$) | $0.278$ | | $0.342$ | | $0.406$ | |

---

> > ### Author Response · Authors · 2026-07-08
> > **Statistical testing and ranking claims - Part 2**
> >
> > **Table R2: Ranking reliability on the soft-intervention benchmark by budget. Ranks (1=best) are computed within each dataset–metric setting ($5$ datasets $\times$ $2$ metrics, $N{=}10$ per budget). Entries show average rank $\pm$ s.d. and 95% bootstrap CI; lower is better. Bottom rows report the Friedman test, Nemenyi critical difference versus rank spread, and a conservative per-dataset Friedman test.**
> >
> > | Method | Rank $\pm$ s.d. ($T{=}100$) | 95% CI ($T{=}100$) | Rank $\pm$ s.d. ($T{=}50$) | 95% CI ($T{=}50$) | Rank $\pm$ s.d. ($T{=}20$) | 95% CI ($T{=}20$) |
> > |---|---|---|---|---|---|---|
> > | ACBO | **1.60 $\pm$ 0.84** | [1.10, 2.10] | **1.60 $\pm$ 0.70** | [1.20, 2.00] | **1.60 $\pm$ 0.66** | [1.20, 2.00] |
> > | BO | 1.80 $\pm$ 0.79 | [1.40, 2.30] | 1.90 $\pm$ 0.74 | [1.50, 2.30] | 2.00 $\pm$ 0.78 | [1.55, 2.45] |
> > | MCBO | 2.60 $\pm$ 0.52 | [2.30, 2.90] | 2.50 $\pm$ 0.85 | [2.00, 3.00] | 2.40 $\pm$ 0.77 | [1.90, 2.80] |
> > | Friedman $p$ | $0.0608$ | | $0.1225$ | | $0.1690$ | |
> > | Nemenyi CD / spread | 1.05 / 1.00 | | 1.05 / 0.90 | | 1.05 / 0.80 | |
> > | Per-dataset $p$ ($N{=}5$) | $0.211$ | | $0.311$ | | $0.390$ | |

---

> > > ### Author Response · Authors · 2026-07-08
> > > **Non-causal BO baseline, benchmark framing and provenance**
> > >
> > > **Non-causal BO baseline.**
> > >
> > > The reviewer is right that the baseline needed to be fully specified. The revised appendix now defines BO as GP-EI on the joint intervention vector: a single-task GPy `GPRegression` surrogate with an RBF kernel, lengthscale $=1$, variance $=1$, ARD off, fixed observation noise $10^{-10}$, emukit `ExpectedImprovement`, and `GradientAcquisitionOptimizer`. All methods use the same fixed initial interventional design. We also state the intervention parameterization and the minimization/maximization convention. This makes clear that the baseline is a standard graph-free GP-EI optimizer rather than an unspecified black-box comparator.
> > >
> > > **Benchmark framing and provenance.**
> > >
> > > We agree that calling the benchmark entirely new was overstated. Most tasks are inherited from prior CBO/DCBO/cCBO work and from the function-network BO setting of Astudillo and Frazier [1]; our contribution is to repackage these tasks into a single uniform harness, align trajectory export, and rescore all methods under a common protocol. We add a benchmark-provenance table listing each task's source, original domain where available, current code-authoritative domain, and any modification. This also resolves the domain inconsistencies the reviewer noted: the epidemiology and $X/Z$ task descriptions are reconciled to the same code-authoritative domains in the main table and appendix. For the function-network benchmarks, we now state explicitly that the wrapper uses normalized $[0,1]$ inputs but the actual task domains match the inherited function-network generators: Ackley and Rosenbrock use $[-2,2]$, Dropwave uses $[-5.12,5.12]$, and Alpine2 uses $[0,10]$; no additional domain change is introduced apart from this normalization.
> > >
> > > > **Reference:** [1] Raul Astudillo and Peter I. Frazier. Bayesian optimization of function networks. In *Advances in Neural Information Processing Systems*, volume 34, pp. 14463–14475, 2021.

---

> > > > ### Author Response · Authors · 2026-07-08
> > > > **Defaults, tuning, and MCBO sensitivity**
> > > >
> > > > We accept the concern that out-of-the-box defaults do not isolate algorithmic ideas from tuning effort in the original repositories. The revised paper now states this as an explicit limitation rather than presenting default-based comparisons as a best-effort tuned benchmark: these are wrapper-default reproducibility results, not reasonably tuned head-to-head comparisons. We also add a targeted MCBO sensitivity analysis for the UCB exploration parameter $\beta$ (Table R3). The best-performing $\beta$ is not consistent across datasets, budgets, or metrics: $\beta=1$ is best for several ToyGraph and Chain-hard rows, $\beta=5$ is best for several PA-GAP rows, and $\beta=10$ remains best for Protein-reconstructed under GAP. We therefore keep the main benchmark tables at the wrapper setting $\beta=10$, report the $\beta$-sensitivity sweep in the appendix, and avoid interpreting default-based MCBO performance as an intrinsic property of the method. We also update the main benchmark tables to report mean $\pm$ standard error rather than mean $\pm$ standard deviation.
> > > >
> > > > **Table R3: MCBO GAP and PA-GAP vs. UCB $\beta$ at $T\in\{100,50,20\}$, $\beta{=}10$ columns are the parameter used in the paper; $\beta\in\{0.5,1,2,5\}$ are 20-seed sweep re-runs (100-trial trajectory truncated to $T$). Higher is better; best score per row in bold.**
> > > >
> > > > | Metric | $T$ | Dataset | $\beta{=}0.5$ | $\beta{=}1$ | $\beta{=}2$ | $\beta{=}5$ | $\beta{=}10$ |
> > > > |---|---|---|---|---|---|---|---|
> > > > | GAP | 100 | ToyGraph | **0.748** | 0.381 | 0.746 | 0.357 | 0.547 |
> > > > | GAP | 100 | Synthetic | 0.659 | **0.686** | 0.684 | 0.681 | 0.652 |
> > > > | GAP | 100 | Synthetic-2 | 0.000 | 0.000 | 0.000 | 0.000 | **0.000** |
> > > > | GAP | 100 | Chain-hard | 0.516 | **0.554** | 0.435 | 0.423 | 0.502 |
> > > > | GAP | 100 | Ecology | 0.000 | 0.000 | 0.000 | 0.000 | **0.000** |
> > > > | GAP | 100 | Protein-reconstructed | 0.419 | 0.337 | 0.109 | 0.286 | **0.878** |
> > > > | GAP | 100 | Healthcare | 0.485 | 0.524 | **0.545** | 0.519 | 0.510 |
> > > > | GAP | 100 | Epidemiology | 0.000 | 0.000 | 0.000 | 0.000 | **0.000** |
> > > > | GAP | 50 | ToyGraph | 0.545 | **0.576** | 0.492 | 0.253 | 0.513 |
> > > > | GAP | 50 | Synthetic | 0.605 | 0.606 | **0.651** | 0.587 | 0.630 |
> > > > | GAP | 50 | Synthetic-2 | 0.000 | 0.000 | 0.000 | 0.000 | **0.000** |
> > > > | GAP | 50 | Chain-hard | 0.392 | **0.540** | 0.381 | 0.367 | 0.215 |
> > > > | GAP | 50 | Ecology | 0.000 | 0.000 | 0.000 | 0.000 | **0.000** |
> > > > | GAP | 50 | Protein-reconstructed | 0.354 | 0.348 | 0.264 | 0.144 | **0.752** |
> > > > | GAP | 50 | Healthcare | 0.303 | 0.292 | 0.275 | **0.330** | 0.315 |
> > > > | GAP | 50 | Epidemiology | 0.000 | 0.000 | 0.000 | 0.000 | **0.000** |
> > > > | GAP | 20 | ToyGraph | 0.333 | **0.530** | 0.282 | 0.171 | 0.406 |
> > > > | GAP | 20 | Synthetic | 0.523 | 0.574 | 0.573 | **0.578** | 0.558 |
> > > > | GAP | 20 | Synthetic-2 | 0.000 | 0.000 | 0.000 | 0.000 | **0.000** |
> > > > | GAP | 20 | Chain-hard | 0.420 | **0.432** | 0.422 | 0.207 | 0.360 |
> > > > | GAP | 20 | Ecology | 0.000 | 0.000 | 0.000 | 0.000 | **0.000** |
> > > > | GAP | 20 | Protein-reconstructed | 0.393 | 0.300 | 0.140 | 0.399 | **0.540** |
> > > > | GAP | 20 | Healthcare | 0.000 | 0.000 | 0.000 | 0.000 | **0.000** |
> > > > | GAP | 20 | Epidemiology | 0.000 | 0.000 | 0.000 | 0.000 | **0.000** |
> > > > | PA-GAP | 100 | ToyGraph | 0.266 | **0.295** | 0.218 | 0.127 | 0.071 |
> > > > | PA-GAP | 100 | Synthetic | 0.149 | 0.154 | 0.136 | 0.125 | **0.162** |
> > > > | PA-GAP | 100 | Synthetic-2 | 0.000 | 0.000 | 0.000 | 0.000 | **0.000** |
> > > > | PA-GAP | 100 | Chain-hard | 0.112 | **0.174** | 0.142 | 0.067 | 0.111 |
> > > > | PA-GAP | 100 | Ecology | 0.000 | 0.000 | 0.000 | 0.000 | **0.000** |
> > > > | PA-GAP | 100 | Protein-reconstructed | 0.033 | 0.034 | 0.011 | **0.047** | 0.045 |
> > > > | PA-GAP | 100 | Healthcare | 0.127 | 0.097 | 0.053 | **0.136** | 0.100 |
> > > > | PA-GAP | 100 | Epidemiology | 0.000 | 0.000 | 0.000 | 0.000 | **0.000** |
> > > > | PA-GAP | 50 | ToyGraph | 0.186 | **0.261** | 0.136 | 0.089 | 0.063 |
> > > > | PA-GAP | 50 | Synthetic | 0.162 | 0.165 | 0.123 | **0.186** | 0.150 |
> > > > | PA-GAP | 50 | Synthetic-2 | 0.000 | 0.000 | 0.000 | 0.000 | **0.000** |
> > > > | PA-GAP | 50 | Chain-hard | 0.089 | **0.152** | 0.119 | 0.035 | 0.078 |
> > > > | PA-GAP | 50 | Ecology | 0.000 | 0.000 | 0.000 | 0.000 | **0.000** |
> > > > | PA-GAP | 50 | Protein-reconstructed | 0.033 | 0.034 | 0.011 | **0.046** | 0.045 |
> > > > | PA-GAP | 50 | Healthcare | 0.016 | 0.032 | 0.012 | **0.034** | **0.034** |
> > > > | PA-GAP | 50 | Epidemiology | 0.000 | 0.000 | 0.000 | 0.000 | **0.000** |
> > > > | PA-GAP | 20 | ToyGraph | **0.175** | 0.171 | 0.093 | 0.028 | 0.040 |
> > > > | PA-GAP | 20 | Synthetic | 0.137 | 0.137 | 0.116 | **0.141** | 0.116 |
> > > > | PA-GAP | 20 | Synthetic-2 | 0.000 | 0.000 | 0.000 | 0.000 | **0.000** |
> > > > | PA-GAP | 20 | Chain-hard | 0.049 | **0.116** | 0.094 | 0.016 | 0.034 |
> > > > | PA-GAP | 20 | Ecology | 0.000 | 0.000 | 0.000 | 0.000 | **0.000** |
> > > > | PA-GAP | 20 | Protein-reconstructed | 0.034 | 0.034 | 0.011 | 0.043 | **0.047** |
> > > > | PA-GAP | 20 | Healthcare | 0.000 | 0.000 | 0.000 | 0.000 | **0.000** |
> > > > | PA-GAP | 20 | Epidemiology | 0.000 | 0.000 | 0.000 | 0.000 | **0.000** |

---

> > > > > ### Author Response · Authors · 2026-07-08
> > > > > **Matched Comparisons, Metric Clarifications, and Stress-Test Diagnostics in Causal Bayesian Optimization**
> > > > >
> > > > > **Matched comparisons versus stress tests.**
> > > > >
> > > > > We agree that several methods were compared outside their home setting. The experiments are now organized as matched comparisons and stress tests. Static known-graph CBO methods form the matched core; BO is a graph-free reference; CEO is a graph-uncertain method; DCBO on static tasks is labelled as a transfer/stress test because DCBO is designed for temporal SCMs; ACBO on stationary tasks is not treated as evidence of superiority in ACBO's intended non-stationary setting. We also clarify that cCBO does not reduce to CBO when constraints are absent: even if the constraint term is inert, the released cCBO pipeline uses a different surrogate and exploration-set handling, so cCBO-vs-CBO differences in unconstrained settings should be read as modelling differences rather than as effects of constraints. This clarification now appears in the "Compared methods" paragraph of the main text, not only in this response.
> > > > >
> > > > > **Why ACBO performs well on stationary tasks.**
> > > > >
> > > > > The reviewer correctly observed that we reported ACBO's surprisingly strong stationary performance without offering any explanation. The revised soft-intervention results section now includes an explicit mechanism-level hypothesis: when the environment is stationary, the adversary that ACBO was designed to hedge against is absent, and CBO-MW effectively reduces to optimistic causal exploration, the causal UCB oracle exploits the known function-network structure much as MCBO does, while the multiplicative-weights layer over a discretized action set concentrates quickly on high-reward actions because no adversary forces re-exploration. The discretized action set may additionally act as a coarse global search grid that suits highly multimodal landscapes such as Ackley, where continuous acquisition optimization can stall in local optima. We label this in the paper as a hypothesis consistent with the observed trajectories rather than a measured causal explanation, and we continue to present ACBO's stationary results as stress-test/reference results rather than evidence about its intended non-stationary setting.
> > > > >
> > > > > **Metrics and average reward.**
> > > > >
> > > > > We agree that CBO does not need a special metric in place of standard BO metrics. GAP and PA-GAP are now presented as auxiliary normalized efficiency scores, not as primary replacements for simple or cumulative regret. The revised paper emphasizes best-so-far objective trajectories; for a fixed dataset and optimization direction, these are equivalent to simple-regret trajectories up to a constant shift and, for maximization tasks, a sign reversal:
> > > > >
> > > > > $$r_t = y(\mathbf{x}^\ast_t)-y^\star \quad \text{(minimization)}, \qquad r_t = y^\star-y(\mathbf{x}^\ast_t) \quad \text{(maximization)}.$$
> > > > >
> > > > > We also remove the inaccurate sentence suggesting that average reward should be de-emphasized because UCB-style methods sample uncertain points. The reviewer is correct that GP-UCB is analyzed through cumulative regret, so average reward is a legitimate complementary metric. The revised metrics section also states directly why GAP and PA-GAP are reported at all: raw simple regret is not comparable across datasets with different objective scales, initializations, and optimization directions, whereas GAP and PA-GAP normalize improvement relative to the initial value and the reference optimum, which is what enables the cross-dataset rank summaries. Readers interested only in per-dataset behavior can rely on the regret-equivalent trajectory plots, and normalized average-reward trajectories are now included in the appendix.
> > > > >
> > > > > **Zero and near-zero scores.**
> > > > >
> > > > > The zero-score cases are now diagnosed rather than left unexplained. They are not silently removed runs. For cCBO on Protein-reconstructed, the runs complete and return valid trajectories, but the best-so-far value remains flat across the budget, so GAP is zero by definition: the method does not improve over the initial design under the reconstructed Protein task and wide intervention domain. For MCBO's zero scores at shorter budgets, the diagnosis matches the reviewer's suggestion: the method often does not improve over the starting data within 20 or 50 trials, while non-zero values can appear at 100 trials. The revised paper now distinguishes no-improvement trajectories from crashes, missing outputs, non-finite objectives, unsupported scopes, and prior-calibration issues. Both named diagnoses now appear in the main text: the trajectory-level discussion in Section 4.4 explicitly states that the cCBO Protein-reconstructed runs complete with valid, finite trajectories that remain flat over the budget, notes that the discrepancy with the original cCBO Protein results is consistent with our task being a transparent reconstruction rather than the unreleased fitted SCM used in the original experiments, and describes MCBO's short-budget zeros as delayed improvement that disappears at $T=100$.

---

> > > > > > ### Author Response · Authors · 2026-07-08
> > > > > > **Clarifying Analytical Scope, Benchmark Attribution, and Deployment Gaps in the CBO Survey**
> > > > > >
> > > > > > **Analytical limitations in the method survey.**
> > > > > >
> > > > > > The reviewer asked whether the "main challenge is" statements in Section 3 were empirical claims or analytical reasoning. They are analytical: they follow from each method's assumptions, surrogate level, and decision rule. We now label them as such and ground them in an identification-strategy audit table, which separates each pipeline's surrogate target, method-level identification assumptions, and benchmark effect source. This prevents readers from interpreting those limitations as newly measured empirical results. Concretely, the "How we summarize methods" paragraph at the start of Section 3 now states that, unless explicitly tied to benchmark evidence in Section 4, all "main challenge" and limitation remarks are analytical consequences of each method's assumptions and should not be read as new empirical findings of this survey.
> > > > > >
> > > > > > **Missing citations and benchmark-source credit.**
> > > > > >
> > > > > > We add Astudillo and Frazier [1] and credit that work for the term "function network" and for the soft-intervention function-network benchmark family. We also add the structural causal bandit work of Lee and Bareinboim [2, 3], which introduces the MIS/POMIS terminology and is directly relevant to our scope-reduction discussion. We performed a broader citation pass after the reviewer's note that two omissions were easy to find.
> > > > > >
> > > > > >
> > > > > >
> > > > > > **Real-world deployment framing.**
> > > > > >
> > > > > > We agree that the open-problems section needed a stronger link to concrete deployment. The revised paper now anchors the discussion in a concrete CBO deployment scenario and explains, for each open problem, the specific gap that prevents a CBO-like method from being reliable in that setting: causal-assumption violations, unknown or misspecified graphs, heterogeneous intervention costs, safety constraints, measurement error, and imperfect interventions. The result is a deployment motivated set of gaps rather than a generic list of future directions. Concretely, the opening of the open-problems section now (i) describes a deployment scenario, tuning a treatment policy, industrial controller, or ecological intervention from observational data and a partially trusted causal graph, and (ii) maps each of the six research directions to the specific gap it addresses in that scenario, so that the link between each open problem and real-world deployment is explicit rather than implied.
> > > > > >
> > > > > > **Other requested corrections.**
> > > > > >
> > > > > > We add a survey cutoff date in the inclusion protocol. We correct the grammar issues identified by the reviewer, including "The modularity of CBO's also" and "the structure-learning component of the CBO's in the context of unknown-graphs." We also clarify what the phrase "corrected PA-GAP" means: the metric clips the improvement ratio at one when a noisy observed best value exceeds the recorded expected-value reference optimum, ensuring $0\leq\mathrm{GAP}\leq 1$ and $0\leq\mathrm{PA}\text{-}\mathrm{GAP}\leq (T+1)/(2T)$.
> > > > > >
> > > > > >
> > > > > > [1] Raul Astudillo and Peter I. Frazier. Bayesian optimization of function networks. In *Advances in Neural Information Processing Systems*, volume 34, pp. 14463–14475, 2021.
> > > > > >
> > > > > >  [2] Sanghack Lee and Elias Bareinboim. Structural causal bandits: Where to intervene? In *Advances in Neural Information Processing Systems*, volume 31, 2018
> > > > > >
> > > > > > [3] Sanghack Lee and Elias Bareinboim. Structural causal bandits with non-manipulable variables. In *Proceedings of the AAAI Conference on Artificial Intelligence*, volume 33, pp. 4164–4172, 2019.

---

> > > > > > > ### Author Response · Authors · 2026-07-08
> > > > > > > **Summary of concrete revisions in the main text**
> > > > > > >
> > > > > > > - We rewrote the benchmark contribution as a standardized repackaging and common scoring harness over inherited and newly integrated tasks, rather than an entirely new definitive benchmark collection.
> > > > > > > - We fully specified the graph-free BO baseline and added benchmark provenance, domain reconciliation, and method-setting labels.
> > > > > > > - We clarified that the function-network wrappers use normalized $[0,1]$ optimizer inputs internally but preserve the inherited Ackley, Rosenbrock, Dropwave, and Alpine2 task domains after mapping.
> > > > > > > - We replaced unsupported rank-superiority claims with statistical rank analyses. The revised appendix reports average ranks, bootstrap confidence intervals, Friedman tests, Nemenyi critical differences, and conservative per-dataset tests for both hard- and soft-intervention benchmarks. No method is claimed to be uniformly best or statistically dominant.
> > > > > > > - We revised the interpretation of CoCaBO and ACBO. CoCaBO's apparent hard-intervention gains are now described as dataset- and budget-conditional patterns, mainly associated with structured scope selection and datasets with multiple intervenable variables. ACBO's soft-intervention rank is also treated as descriptive rather than statistically conclusive.
> > > > > > > - We added the MCBO $\beta$-sensitivity analysis, state the wrapper settings explicitly, and keep the main benchmark tables at $\beta=10$ because the best $\beta$ is not consistent across datasets, budgets, or metrics.
> > > > > > > - We updated the main benchmark tables to report standard errors rather than standard deviations, and we clarify where standard deviations are still used to summarize trajectory variability.
> > > > > > > - We distinguish matched comparisons from stress tests for DCBO, ACBO, cCBO, CEO, and BO, so that methods designed for different problem settings are not interpreted as solving identical optimization problems.
> > > > > > > - We demoted GAP and PA-GAP to auxiliary efficiency scores, clarified their bounds, and added standard best-so-far/simple-regret-equivalent and average-reward reporting in the revised paper.
> > > > > > > - We added zero-score diagnostics, including the cCBO Protein-reconstructed no-improvement case and MCBO's delayed-improvement pattern at short budgets.
> > > > > > > - We updated experiment visual material in the main text and appendix: tick and legend fonts are enlarged, average-reward visualizations are included, and diagnostic visualizations are moved to the manuscript rather than used only as evidence in the rebuttal.
> > > > > > > - We added the Astudillo–Frazier function-network citation, Lee–Bareinboim MIS/POMIS citations, a survey cutoff date, domain-consistency fixes, grammar corrections, and a clearer deployment-motivated open-problems discussion.
> > > > > > > - We added an explicit mechanism-level hypothesis in the soft-intervention results for why ACBO remains competitive on stationary tasks (reduction of CBO-MW to optimistic causal exploration when no adversary is present, plus the coarse global-search effect of its discretized action set), clearly labeled as a hypothesis rather than a measured explanation.
> > > > > > > - We fully specified the BO baseline in the implementation-settings table (RBF lengthscale $=1$, variance $=1$, ARD disabled, observation noise $10^{-10}$, shared fixed initial design) and stated in the main text why cCBO does not reduce to CBO when constraints are inactive.
> > > > > > > - We stated in Section 3 that method-limitation remarks are analytical unless tied to benchmark evidence, added the named cCBO/MCBO zero-score diagnoses to Section 4.4, and justified GAP/PA-GAP as auxiliary normalized scores that enable cross-dataset aggregation, alongside regret-equivalent trajectories and appendix average-reward plots.

---

### Review · Reviewer_n3CV · 2026-07-02

**Summary Of Contributions:**

This paper presents a comprehensive survey of the rapidly growing subfield of Causal Bayesian Optimization (CBO). The authors propose a unified taxonomy that categorizes existing methods along four design axes: intervention representation, surrogate architecture, decision rules, and uncertainty sources. Additionally, the paper establishes formal connections to adjacent fields and introduces a reproducibility-oriented benchmark. This benchmark evaluates multiple CBO variants across datasets, introducing a new trajectory-aware metric (PA-GAP) to address the shortcomings of existing metrics.

**Strengths**:

- **Timeliness and Unification**: The survey provides a much-needed conceptual umbrella for a highly fragmented literature, successfully mapping how various extensions tackle the bottlenecks of the original CBO framework.
- **Consolidation of Benchmarks**: By providing a collection of disparate datasets used across previous CBO papers into a single accessible repository, the authors have significantly lowered the barrier to entry for future comparative research.
- **New Metric**: The introduction of Path-Aware GAP as a weighted discrete integral of the convergence curve is a highly valuable contribution.

**Weaknesses:**
- **Flawed Empirical Comparisons**: The benchmark aggregates and ranks fundamentally different problem classes (e.g., constrained vs. unconstrained optimization) into single scalar tables, penalizing methods for solving strictly harder problems.
- **Insufficient Mathematical Rigor**: Several method summaries in Section 3 rely on high-level descriptions rather than formal equations, making it difficult to understand the algorithms without reading the original papers.
- **Imprecise Causal Framing**: The text occasionally conflates distinct causal concepts, particularly regarding identifiability versus graph misspecification,
- **Incomplete Benchmark Suite**: The proposed evaluation framework functions more as a collection of existing datasets rather than a comprehensive, standardized benchmark for future use. This is especially problematic in the settings used to test soft interventions, where the authors propose using standard global optimization functions as "causal" benchmarks, which significantly simplify the problem as there are no cause-and-effect relationships or intricate structure between the input variables.

**Audience:**

Yes

**Audience Explanation:**

Causal Machine Learning and Bayesian Optimization are both highly active areas of research within the TMLR community. The intersection of these fields, and specifically Causal BO, has seen rapid expansion over the last years, resulting in a fragmented landscape of different assumptions, varying intervention types, and disparate evaluation metrics. A unifying survey that standardizes the taxonomy, proposes new common metrics and establishes a shared evaluation framework will be a valuable reference document for researchers working in this space.

**Broader Impact Concerns:**

While this paper is primarily methodological, CBO is increasingly positioned for deployment in sensitive domains such as healthcare dosing and public policy. The authors correctly note the vulnerability of these methods to graph misspecification and hidden confounding. A brief acknowledgment in the Broader Impact section that deploying CBO with miscalibrated priors or misspecified graphs could lead to harmful physical interventions in real-world systems would be a responsible addition.

**Claims And Evidence:**

No

**Claims Explanation:**

While the conceptual survey is largely accurate, the empirical claims and the resulting method rankings (e.g., Table 4, Figure 4, Figure 5) are not supported by fair or valid evidence.

The paper compares the performance of methods that have been proposed to solve completely different optimization problems.  For example, the paper evaluates unconstrained methods (e.g., CBO) alongside constrained methods (e.g., cCBO) and plots their aggregate rankings using scalar metrics like GAP and PA-GAP.

A constrained optimization method operates on a restricted feasible space and it is expected to have a different (often lower or slower) objective convergence profile than an unconstrained method optimizing the same function. By ranking these methods together homogeneously, the comparison penalizes methods that solve a strictly harder problem, making them appear 'worse' at optimization than an unconstrained baseline.

It is true that the existing literature often plots methods solving different problem formulations on the same convergence graph. However, the purpose of such plots is to demonstrate that differing problem constraints naturally converge to different objective values. It is invalid to use these combined plots to competitively rank methods or declare that one is "better" than another. Statements of algorithmic superiority can only be made between methods solving the exact same optimization problem.

**Requested Changes:**

**[Critical] Increase Mathematical Rigor in Methodological Summaries (Section 3)**:
Across the method summaries, the text relies too heavily on high-level descriptions rather than mathematical formalisms.
For instance, in the HCBO summary, the authors state that the method constructs scopes that "jointly cover causal influence... well enough to support optimization" and that it works best with "causal sparsity or a low intrinsic causal dimension". Without defining what "coverage" means mathematically or how "intrinsic causal dimension" is quantified, a reader cannot understand the algorithm. Similarly, for GACBO, the author mentions that the method "maintains a confidence set of causal models" but provides no intuition on how this is practically achieved or what exactly a "confidence set" of graphs is. Again for CoCaBO, terms like "candidate mixed policy scopes" and "mixed-variate Bayesian optimization" are used without formal definition. I request that the authors systematically include the core equations (e.g., the modified surrogate, acquisition function, or constraint formulation) that distinguish each surveyed method, introducing a common notation across the methods to facilitate comparisons.

**[Critical] Experimental Evaluation**
As mentioned above, a flaw in the empirical evaluation is ranking algorithms proposed for fundamentally different optimization problems against each other. For example, unconstrained methods are compared to constrained methods. However, a constrained optimization method is operating over a restricted feasible space and is expected to have a different convergence profile (and different optimum) than an unconstrained method optimizing the same objective. Ranking them together under the same GAP/PA-GAP metrics gives the false illusion that they are solving the exact same problem. The authors should separate the benchmarks into clear tracks (e.g., Unconstrained Static, Constrained, Contextual) and evaluate methods that have been introduced for the same optimization problem rather than ranking them homogeneously.
If methods are shown on the same convergence plots to illustrate how different methods converge to different optima (e.g., to show the price of a constrained method or the effect of having to learn the graph), the distinction must be heavily contextualized in the text.

**Benchmark Claims and Structure**:

- *[Critical] Reframe the Contribution*: The proposed benchmark currently functions more as a repository of previously used datasets than a definitive, standardized benchmark suite. Establishing a definitive causal optimization benchmark would require introducing high-dimensional, complex, non-linear simulators (e.g., realistic epidemiological or climate models) to truly stress-test scalability and combinatorial scope reduction. If the authors do not wish to introduce such environments, they must significantly tone down their claims. I suggest that they reframe this contribution in the text as a "unified repository of existing environments" rather than a novel, definitive benchmark suite.

- *[Critical] Define Task Tracks*: Even as a repository, to be a useful reference point, it must define explicit "Task Tracks" (e.g., Unconstrained, Constrained, Contextual) for a given graph, complete with fixed thresholds, constraint functions, or policy spaces, rather than leaving these configurations entirely up to the user.

- *[Critical] Differentiate SCMs vs. Computational Graphs*: I disagree with the inclusion of standard global optimization functions (Ackley, Rosenbrock, etc.) as soft-intervention benchmarks. The structural topology of these "function networks" fundamentally fails to evaluate a core challenge of causal optimization: inter-variable causal dependencies. In standard causal optmization benchmarks (e.g., the Healthcare or Ecology SCMs), manipulable variables often have complex causal relationships with one another, meaning an intervention on one variable can sever incoming edges or cascade into others. In the "function networks", the input action variables are completely independent, orthogonal search dimensions. There is no causal relationship between the inputs themselves. Consequently, these environments do not test the algorithm's ability to handle confounding or cascading effects among manipulable variables. The authors must explicitly distinguish between "computational graph benchmarks" and "SCM benchmarks".

- *[Strengthening] Introduce Complex Simulators*: While not strictly required if the claims are appropriately toned down, adding high-dimensional simulators with complex causal structures would vastly strengthen the repository and elevate its utility for the community as a reference standard for causal optimization research.

**BO Fundamentals and CBO Framing**

- *[Critical] Expected Improvement under Noise*: In Section 2.2, the objective observations are explicitly defined with noise: $y_t = f(x_t) + \epsilon$. However, Equation (7) formulates Expected Improvement using $f^+$, which is described as the "best observed value". In a noisy setting, the latent function value $f$ is unobserved. The authors should clarify whether they mean the incumbent noisy observation $y^+$, or if they are assuming a noiseless setting specifically for Equation (7).

- *[Strengthening] Acquisition Function Generality*: The text states that acquisition functions encourage exploration by "questioning where $\sigma_t$ is large". Because many acquisition functions (e.g., information-theoretic) evaluate the full posterior covariance rather than just the marginal variance, I suggest generalising this phrasing to "predictive uncertainty" or similar.

- *[Strengthening] Philosophical Framing of CBO*: The current text introduces Causal Bayesian Optimization by stating it "extends standard BO to settings where the decision variable includes not only the intervention level but also the intervention scope". I recommend reframing this definition. The foundational starting point of CBO is that the optimisation is taking place within a system governed by causal relationships. It is precisely because of this causal structure that the practitioner is required (and able) to make algorithmic choices about intervention variables, scopes, and the integration of observational data.

- *[Critical] MIS/POMIS*: The definitions for MIS and POMIS are informal and rely on undefined terms like "causal pathways". As TMLR requires technical precision, please replace these descriptive summaries with the formal mathematical/graphical definitions (and introduce needed notation for that).

**Improve on the Taxonomy [Critical]**:

- *Decoupling Architecture and Interventions*: Section 3.3 groups methods under the umbrella of "Intervention Strategy Developments", but conflates two orthogonal concepts: expanding the intervention space (e.g., from static values to functional policies) and changing the surrogate architecture (e.g., moving from effect-level models to mechanism-level models). I strongly recommend separating these into distinct sub-sections or explicitly discussing them as independent design axes. Indeed, one can have mechanism-level models and only consider hard interventions or instead, have effect-level models and consider soft interventions.

- *Design Axis 4 ("What uncertainty dominates")*: This axis is framed as a "design choice", but uncertainty (e.g., structure uncertainty) is an induced property resulting from other assumptions (e.g., assuming an unknown vs. known graph), not a direct design choice. I strongly suggest renaming this axis to "System Knowledge Assumptions" (or similar) to reflect the actual choices an algorithm designer makes. One can then discuss, based on the assumptions, what uncertainty dominates.

**Clarify Identifiability vs. Graph Misspecification [Critical]:**

- The text currently states that CBO "relies on identifiability". I recommend softening this claim. CBO benefits from identifiability to construct observationally informed priors, which accelerates sample efficiency. However, the framework does not strictly rely on it to function. If an effect is unidentifiable, the algorithm can simply fall back to a standard, uninformative GP prior while still leveraging the causal graph for search space reduction.

- In Section 2.1, the authors state: "When identifiability fails, for example, due to hidden confounding, the same procedure can produce overconfident and biased priors that mislead the acquisition function". This phrasing conflates two distinct issues. If identifiability fails on a known, correctly specified graph (e.g., an unblockable backdoor path), a sound algorithm will recognize this and default to a standard uninformative prior, thus avoiding bias. A biased prior is only generated if the algorithm mistakenly believes the effect is identifiable, which is a failure of the causal graph assumption, not a failure of identifiability given the graph. I suggest clarifying this distinction.

**Other claims to clarify**

- *[Critical] Clarify CNEI's Prior*: If CNEI uses a supervised, purely correlational prior (E[Y|X]) to warm-start the GP rather than a causally-adjusted prior, one needs to highlight this as a fundamental departure from standard CBO.

- *[Critical] Correct the CBO Algorithmic Sequence*: The loop is described out of order: "choose an intervention... predict its causal effect... select the next data-collection action". The surrogate predicts the effect for candidate interventions first, the acquisition function scores them, and then the intervention is chosen.

- *[Strengthening] Intermediate Observations in Effect-Level Models*: The authors claim effect-level models "cannot reuse intermediate observations". This is only strictly true for the GP posterior itself. Note that one could easily append intermediate observations collected during interventional trials to the underlying dataset, thereby continuously updating and improving the observationally-derived prior mean at each step. This nuance should be acknowledged.

- *[Strengthening] MCBO Trade-offs*: The summary of MCBO fails to adequately discuss its limitations compared to effect-level CBO. Specifically:
  - Maintaining multiple mechanism-level GPs and propagating uncertainty through the graph is computationally vastly more expensive than learning a direct effect-level function.
  - Mechanism-level modelling is highly vulnerable to unobserved confounders between intermediate nodes, requiring complex latent variable extensions. Effect-level CBO can simply discard the biased observational prior and proceed with uninformative priors. Please add a discussion on these trade-offs.

**Refine Terminology**:

- [Strengthening] Definition 1 refers to the set of exogenous variables U as "(unmodeled) variables". This is slightly imprecise, as an SCM explicitly models their probability distribution P(U). It would be more accurate to describe them as  "variables determined outside the system."

- [Critical] The author should clarify what type of soft interventions are considered in the paper.  The text narrowly defines soft interventions as replacing a mechanism with a new mechanism  parameterized by theta. However, in the broader literature (including fCBO, which is surveyed in the paper), soft interventions can also encompass structural changes (dropping or adding parents) and policy-based functional interventions. I encourage the authors to specify what type of soft interventions each method considered (e.g., highlight differences between those considered by fCBO and thus considered by MCBO). Furthermore, a brief clarification of what the "intervention parameter theta" represents mathematically in this context would improve readability.

- [Critical] Definition 2 and Equation 11 define the interventional objective conditional on a context set C. The original CBO framework (Aglietti et al., 2020) does not use context variables. Please define the baseline CBO framework strictly according to its original formulation (without C), and introduce C only when discussing contextual extensions.

---

> ### Author Response · Authors · 2026-07-08
> **Aggregate plots, formulation labels, and budget-stratified rank tests - Part 1**
>
> We thank the reviewer for the careful and technically specific review. We agree with the main message: the survey and metric contributions are useful, but the empirical section must distinguish methods that solve different optimization problems instead of treating one aggregate rank as an unqualified leaderboard. In the revision, we therefore reframe the benchmark as a reproducibility-oriented repository and harness, not as a definitive leaderboard. We retain the existing aggregate rank visualizations as descriptive overview plots, add statistical significance tests to them, and add task-track labels so that constrained, contextual, stress-test, soft computational-graph, and soft-SCM results are interpreted in the right formulation.
>
> **Aggregate plots, formulation labels, and budget-stratified rank tests.**
>
> We agree that the original aggregate ranking mixed methods that are not all solving identical optimization problems. We therefore revised the empirical section in two ways. First, we now attach explicit formulation labels to each method–task pair, so that constrained, contextual, graph-uncertain, dynamic/stress-test, soft computational-graph, and soft-SCM results are not interpreted as one homogeneous leaderboard. Second, we changed the statistical ranking analysis so that it is organized by budget rather than by strict formulation tracks.
>
> This choice is practical and statistical. A strict five-track split leaves several tracks without meaningful rank comparisons: the constrained track contains only cCBO among the benchmarked methods, the contextual/policy track contains only CoCaBO, and the soft-SCM track contains only Chain-soft as a single dataset. Ranking within those tracks would either be undefined or reduce to a one-method or one-dataset descriptive comparison. We therefore use formulation labels to clarify what each method is being asked to solve, while using the shared budgets $T\in\{100,50,20\}$ as the units for rank reliability analysis. For each fixed budget, methods are ranked within each dataset–metric setting, and ranks are summarized separately for the hard- and soft-intervention benchmarks. The formulation labels are shown in Table R1, and the budget-stratified rank tests are reported in Tables R2 and R3.
>
> This revision changes the interpretation of the plots and rank tables. The aggregate figures are retained as descriptive summaries of the benchmark harness, but they are no longer used to claim that one method is uniformly superior across incompatible formulations. In the hard-intervention benchmark, CoCaBO has the lowest descriptive average rank at $T=100$ and $T=50$, while BO has the lowest descriptive average rank at $T=20$; however, no pair of methods is statistically separable by the Nemenyi post-hoc test at any budget. In the soft-intervention benchmark, ACBO has the lowest descriptive average rank at all three budgets, but the Friedman and Nemenyi analyses again do not support a statistically separable winner. We therefore revise the main empirical claim: performance patterns are heterogeneous across datasets, budgets, metrics, and method assumptions, and the benchmark should be read as a reproducibility and diagnostic resource rather than as a single cross-formulation leaderboard.
>
> **Table R1: Formulation labels used to contextualize benchmark results. Several labels are not independently rankable because they contain only one method or one dataset; therefore the statistical rank analysis is reported by budget rather than as five separate formulation-track leaderboards.**
>
> | Label | What it identifies | How it is used in the revision |
> | --- | --- | --- |
> | Unconstrained hard SCM | Static hard-intervention SCM tasks without explicit feasibility constraints | Descriptive rank summaries by budget; comparisons are qualified because some methods are stress-test or graph-uncertain variants |
> | Constrained hard SCM | Tasks where cCBO-style feasibility constraints are part of the intended formulation | Descriptive only: no fully matched constraint-adapted baseline is available across all datasets |
> | Contextual / policy | Settings where scope or intervention choice is policy/context dependent | Descriptive only: CoCaBO is the only currently integrated method in this formulation |
> | Soft computational graph | Ackley, Rosenbrock, Dropwave, and Alpine2 function-network tasks | Descriptive rank summaries by budget; explicitly not treated as SCM benchmarks with confounding or cascading manipulable variables |
> | Soft SCM / policy | Chain-soft | Descriptive only: single soft-SCM dataset in the current repository |

---

> > ### Author Response · Authors · 2026-07-08
> > **Aggregate plots, formulation labels, and budget-stratified rank tests - Part 2**
> >
> > **Table R2: Hard-intervention rank reliability by budget. Ranks (1=best) are computed within each dataset–metric setting ($8$ datasets $\times$ $2$ metrics, $N{=}16$ per budget). Entries show average rank $\pm$ s.d. and 95% bootstrap CI. Bottom rows report the Friedman test, Nemenyi critical difference versus rank spread, and a conservative per-dataset Friedman test.**
> >
> > | Method | Rank $\pm$ s.d. ($T{=}100$) | 95% CI ($T{=}100$) | Rank $\pm$ s.d. ($T{=}50$) | 95% CI ($T{=}50$) | Rank $\pm$ s.d. ($T{=}20$) | 95% CI ($T{=}20$) |
> > | --- | --- | --- | --- | --- | --- | --- |
> > | CoCaBO | **3.38 $\pm$ 2.75** | [2.12, 4.69] | **3.44 $\pm$ 2.68** | [2.22, 4.75] | 3.88 $\pm$ 2.82 | [2.56, 5.25] |
> > | cCBO | 3.91 $\pm$ 2.23 | [2.88, 5.03] | 3.69 $\pm$ 2.32 | [2.66, 4.84] | 3.88 $\pm$ 2.24 | [2.84, 4.97] |
> > | CBO | 3.50 $\pm$ 1.59 | [2.81, 4.31] | 3.94 $\pm$ 1.18 | [3.44, 4.50] | 4.31 $\pm$ 1.35 | [3.62, 4.94] |
> > | BO | 4.12 $\pm$ 2.84 | [2.78, 5.53] | 3.94 $\pm$ 2.69 | [2.66, 5.22] | **3.81 $\pm$ 2.48** | [2.62, 5.00] |
> > | CEO | 4.78 $\pm$ 1.70 | [3.94, 5.56] | 4.88 $\pm$ 1.96 | [3.94, 5.81] | 4.25 $\pm$ 1.98 | [3.31, 5.19] |
> > | DCBO | 5.03 $\pm$ 2.22 | [3.94, 6.06] | 4.78 $\pm$ 2.27 | [3.75, 5.84] | 4.12 $\pm$ 2.18 | [3.09, 5.22] |
> > | HCBO | 5.38 $\pm$ 1.89 | [4.44, 6.25] | 5.44 $\pm$ 1.86 | [4.50, 6.25] | 5.84 $\pm$ 2.19 | [4.75, 6.84] |
> > | MCBO | 5.91 $\pm$ 1.89 | [4.97, 6.72] | 5.91 $\pm$ 2.19 | [4.81, 6.88] | 5.91 $\pm$ 2.01 | [4.91, 6.81] |
> > | Friedman $p$ | $0.0272$ | | $0.0384$ | | $0.0462$ | |
> > | Nemenyi CD / spread | 2.62 / 2.53 | | 2.62 / 2.47 | | 2.62 / 2.09 | |
> > | Per-dataset $p$ ($N{=}8$) | $0.278$ | | $0.342$ | | $0.406$ | |
> >
> > **Table R3: Soft-intervention rank reliability by budget. Ranks (1=best) are computed within each dataset–metric setting ($5$ datasets $\times$ $2$ metrics, $N{=}10$ per budget). Entries show average rank $\pm$ s.d. and 95% bootstrap CI. Bottom rows report the Friedman test, Nemenyi critical difference versus rank spread, and a conservative per-dataset Friedman test.**
> >
> > | Method | Rank $\pm$ s.d. ($T{=}100$) | 95% CI ($T{=}100$) | Rank $\pm$ s.d. ($T{=}50$) | 95% CI ($T{=}50$) | Rank $\pm$ s.d. ($T{=}20$) | 95% CI ($T{=}20$) |
> > | --- | --- | --- | --- | --- | --- | --- |
> > | ACBO | **1.60 $\pm$ 0.84** | [1.10, 2.10] | **1.60 $\pm$ 0.70** | [1.20, 2.00] | **1.60 $\pm$ 0.66** | [1.20, 2.00] |
> > | BO | 1.80 $\pm$ 0.79 | [1.40, 2.30] | 1.90 $\pm$ 0.74 | [1.50, 2.30] | 2.00 $\pm$ 0.78 | [1.55, 2.45] |
> > | MCBO | 2.60 $\pm$ 0.52 | [2.30, 2.90] | 2.50 $\pm$ 0.85 | [2.00, 3.00] | 2.40 $\pm$ 0.77 | [1.90, 2.80] |
> > | Friedman $p$ | $0.0608$ | | $0.1225$ | | $0.1690$ | |
> > | Nemenyi CD / spread | 1.05 / 1.00 | | 1.05 / 0.90 | | 1.05 / 0.80 | |
> > | Per-dataset $p$ ($N{=}5$) | $0.211$ | | $0.311$ | | $0.390$ | |

---

> > > ### Author Response · Authors · 2026-07-08
> > > **Clarifying Benchmark Scope, Method Formalism, and Causal Assumptions**
> > >
> > > **Benchmark framing and SCM versus computational graphs.**
> > >
> > > We agree that the benchmark should not be described as a definitive standardized suite in the sense of introducing new high-dimensional simulators such as realistic epidemiological or climate models. We revise the claim to say that the contribution is a unified repository and execution harness for inherited and newly standardized CBO tasks, with common wrappers, trajectory exports, and scoring. We also now distinguish SCM benchmarks from computational-graph/function-network benchmarks. Ackley, Rosenbrock, Dropwave, and Alpine2 are no longer described as testing confounding, cascading manipulations, or causal dependence among action variables. They are reported as soft computational-graph stress tests. Chain-soft is separated as the only soft-SCM task in this revision, and we state that a larger suite of complex SCM simulators remains future work. The revised datasets section now makes the reviewer's point explicit in the text rather than only in a table: it states that in the function networks the manipulable inputs are mutually independent, orthogonal search dimensions with no causal relationships among themselves, so an intervention on one input cannot sever incoming edges of, or cascade into, another manipulable variable, and that these tasks therefore do not test confounding or cascading effects among manipulable variables. In addition, to make the formulation labels usable as fixed reference points rather than user-configured settings, the revision states that each task in the repository ships with a frozen configuration, i.e., intervention domains, optimization direction, observational sample sizes, constraint functions and thresholds for constrained tasks, the policy space for Chain-soft, and the reference optimum used for scoring, so that a new method can be evaluated on a given formulation without any task-design decisions being left to the user. Finally, the open-problems section now explicitly frames our release as a unified repository of existing environments plus an execution and scoring harness, and identifies high-dimensional, complex, nonlinear simulators (e.g., realistic epidemiological or climate models) as future work needed for a definitive benchmark suite.
> > >
> > > **Mathematical rigor in Section 3.**
> > >
> > > We agree that several method summaries were too descriptive. The revised Section 3 introduces a common notation for a causal graph $G$, intervention scope $S$, intervention value or policy $\theta$, target $Y$, observational data $D_o$, interventional data $D_i$, effect-level surrogate $g_S(\theta)$, and mechanism-level surrogate $f_j(\mathrm{pa_j})$. We then add the core mathematical object that distinguishes each method family: the constrained objective for cCBO, the context/policy objective for CoCaBO-style methods, graph confidence sets for GACBO-style methods, mechanism-level propagation for MCBO, and the high-dimensional scope-search objective for HCBO. These equations are not intended to reproduce every original proof, but they give the reader the formal object optimized by each method before the prose discussion. Concretely, the revision now (i) opens Section 3 with a dedicated "Common notation for method summaries" paragraph that defines this shared notation and points to the per-method equations; (ii) formalizes HCBO's "coverage" as a path-coverage maximization problem over scope families and defines the "intrinsic causal dimension" as the minimal scope budget achieving full coverage; (iii) defines GACBO's confidence set $\mathcal C_t$ of plausible graph–mechanism pairs and states its optimistic acquisition $\max_{S,\theta}\max_{M\in\mathcal C_t}\hat f_M(S,\theta)$; (iv) defines CoCaBO's "mixed policy scope" as a pair of an intervention scope and a context subset and states its contextual policy objective; and (v) adds the mechanism-level GP propagation equation and optimism-based decision rule for MCBO. All additions appear in blue in the revised manuscript.
> > >
> > > **Causal framing, identifiability, and graph misspecification.**
> > >
> > > We soften the claim that CBO "relies on identifiability." The revised text states that identifiability is used to construct observationally informed causal priors and can improve sample efficiency, but CBO can still run with an uninformative GP prior when a causal effect is not identifiable. We also separate two failure modes that were conflated in the submission: non-identifiability under a correctly specified graph should be recognized by a sound adjustment procedure, whereas overconfident biased priors arise when the graph or adjustment assumptions are wrong, or when the algorithm mistakenly treats an effect as identifiable. The robustness appendix therefore describes graph-misspecification stress tests separately from omitted-variable stress tests.

---

> > > > ### Author Response · Authors · 2026-07-08
> > > > **Clarifying CBO Foundations, Objectives, and Taxonomy**
> > > >
> > > > **BO fundamentals and the CBO loop.**
> > > >
> > > > We revise the BO preliminaries to avoid ambiguity about Expected Improvement under noise. The equation is now explicitly presented for the latent/noiseless incumbent in the pedagogical BO setting; when observations are noisy, the text distinguishes using the best noisy observation from using a posterior estimate of the latent incumbent. We also replace wording such as "where $\sigma$ is large" with "where posterior predictive uncertainty or information gain is large," so information-theoretic acquisitions are not excluded. Finally, the CBO loop is reordered: the surrogate predicts candidate interventional effects, the acquisition rule scores candidates, and only then is the next intervention selected and evaluated.
> > > >
> > > > **Philosophical framing of CBO.**
> > > >
> > > > We thank the reviewer for this suggestion, which we found helpful for making the opening of the CBO section conceptually cleaner. We agree that the foundational starting point of CBO is that optimization takes place within a system governed by causal relationships, and that the choices about intervention scopes, levels, and the use of observational data follow from this premise rather than defining it. We have rewritten the opening of Section 2.3 accordingly: the revised text first states that the manipulable variables are linked to the target through structural mechanisms rather than a black-box input–output map, and then explains that it is precisely this causal structure that requires and enables the optimizer to choose intervention scopes and levels and to relate observational data to interventional effects through causal identification.
> > > >
> > > > **MIS, POMIS, and baseline objective.**
> > > >
> > > > We replace the informal MIS/POMIS description with formal graphical notation. An MIS is defined as a manipulable intervention set $S$ whose intervention can affect the target and is not redundant with a smaller manipulable set under the graphical criterion used by the original CBO framework; a POMIS is the subset of MISs that remains potentially optimal after excluding intervention sets that are dominated or irrelevant under the available graph and observational information. We also separate the baseline CBO objective from contextual extensions: the baseline objective is
> > > >
> > > > $$S^\star,\theta^\star \in \arg\max_{S,\theta} \mathbb{E}[Y \mid \mathrm{do}(X_S=\theta)],$$
> > > >
> > > > and context variables $C$ are introduced only in the later contextual/policy formulation. We have applied this change at the level of the definitions themselves: the CBO objective definition, the notation table, and the effect-level surrogate equations in Section 3.1 now state the baseline objective without $C$, and a note after the definition explains that $C$ first appears in the contextual extension (CoCaBO) in Section 3.3.
> > > >
> > > > **Taxonomy revisions.**
> > > >
> > > > We agree that the original taxonomy blurred intervention representation and surrogate architecture. The revised taxonomy treats these as distinct axes: one axis describes what interventions are allowed (hard values, soft mechanism parameters, policies, constraints), while a separate axis describes how the response is modeled (effect-level GP, mechanism-level GP, multi-task or structured surrogate). We also rename the former axis "what uncertainty dominates" to "system knowledge assumptions." The revised wording first states the designer's assumption – known graph, uncertain graph, uncertain mechanisms, hidden confounding, unknown feasible set – and then explains which uncertainty this assumption induces. In addition to restructuring the design-axes list at the start of Section 3, the revision adds an explicit paragraph at the opening of Section 3.3 stating that expanding the intervention representation (fCBO, CoCaBO) and changing the surrogate architecture (MCBO, ACBO) are independent axes: mechanism-level surrogates can be paired with only hard interventions, and effect-level surrogates with soft interventions, exactly as the reviewer notes.

---

> > > > > ### Author Response · Authors · 2026-07-08
> > > > > **Clarifying Method Assumptions and Intervention Semantics**
> > > > >
> > > > > **Method-specific clarifications.**
> > > > >
> > > > > We add four method-specific clarifications requested by the reviewer. First, CNEI is explicitly flagged as using a supervised/correlational prior rather than a causally adjusted prior, so it is not presented as standard CBO prior construction. Second, the effect-level versus mechanism-level discussion now notes that intermediate observations are not directly used in the GP posterior of a direct effect-level surrogate, but they can still be appended to the observational/interventional dataset and indirectly improve a re-estimated causal prior. Third, the MCBO discussion now adds its computational and statistical tradeoffs: maintaining multiple mechanism-level GPs and propagating uncertainty through a graph is more expensive than direct effect modeling, and mechanism-level models can be especially vulnerable to unobserved confounding between intermediate nodes. Fourth, the definition of exogenous variables now describes them as variables determined outside the modeled causal system with distribution $P(U)$, rather than as simply unmodeled variables.
> > > > >
> > > > > **Soft interventions and broader impact.**
> > > > >
> > > > > We broaden the soft-intervention terminology. The revised text states that soft interventions can mean changing a mechanism parameter, changing or dropping parent dependencies, or optimizing a policy/function that enters a mechanism; for each surveyed soft-intervention method we specify which version is meant and what the intervention parameter $\theta$ represents. We also revise the broader-impact paragraph to explicitly warn that in domains such as healthcare dosing, resource allocation, and public policy, deploying CBO with misspecified graphs, hidden confounding, or miscalibrated causal priors can select harmful physical interventions. The paper therefore frames causal assumptions and failure logging as part of deployment risk, not just technical details.

---

> > > > > > ### Author Response · Authors · 2026-07-08
> > > > > > **Summary of concrete revisions**
> > > > > >
> > > > > > - Retained the existing aggregate rank plots as descriptive overview figures, but revised the captions and discussion so they are not interpreted as unqualified cross-formulation leaderboards.
> > > > > > - Added formulation labels for constrained, contextual, graph-uncertain, stress-test, soft computational-graph, and soft-SCM settings. Because several strict formulation tracks contain only one method or one dataset, we do not use them as separate leaderboards; instead, we report budget-stratified rank reliability analyses at $T\in\{100,50,20\}$, with the formulation labels used to contextualize which comparisons are matched and which are descriptive stress tests.
> > > > > > - Added statistical testing for the retained ranking summaries: bootstrap rank intervals, Friedman tests, Nemenyi critical differences, and conservative per-dataset tests. These analyses support descriptive heterogeneity but not a statistically separable uniformly best method.
> > > > > > - Reframed the empirical contribution as a unified repository and reproducibility harness, with explicit separation between SCM benchmarks and computational-graph/function-network tasks.
> > > > > > - Added mathematical notation and core equations for the main method families in Section 3, including constrained, contextual/policy, graph-uncertain, high-dimensional, and mechanism-level variants.
> > > > > > - Clarified identifiability versus graph misspecification, Expected Improvement under noisy observations, predictive uncertainty in acquisitions, MIS/POMIS definitions, and the order of the CBO decision loop.
> > > > > > - Revised the taxonomy to decouple intervention representation from surrogate architecture and renamed the uncertainty axis as system-knowledge assumptions.
> > > > > > - Added method-specific caveats for CNEI, effect-level intermediate observations, MCBO computational/confounding tradeoffs, soft-intervention types, and deployment risks in sensitive domains.
> > > > > > - Rewrote the opening of the CBO section so that the causal system, rather than the enlarged decision variable, is presented as the foundational premise of CBO.
> > > > > > - Removed context variables $C$ from the baseline CBO definition, notation table, and effect-level surrogate equations; $C$ now first appears in the contextual extension (CoCaBO).
> > > > > > - Added a common-notation paragraph and the MCBO mechanism-level propagation equation to Section 3, completing the per-method formal objects (cCBO, HCBO, MCBO, CoCaBO, GACBO).
> > > > > > - Added an explicit paragraph in Section 3.3 decoupling intervention representation from surrogate architecture and clarifying that dominant uncertainty is induced by system-knowledge assumptions rather than being a design choice.
> > > > > > - Stated in the datasets section that function-network inputs are causally independent and do not test confounding or cascading manipulable-variable effects; documented that every repository task ships with a frozen configuration (domains, constraints, policy spaces, reference optima); and framed complex, high-dimensional SCM simulators as future work in the open-problems section.

---

### Review · Reviewer_t8zQ · 2026-07-02

**Summary Of Contributions:**

This paper presents a survey of Causal Bayesian Optimisation methods. A taxonomy of different methods is introduced, organised along 4 axes: intervention representation, surrogate modelling, decision rules, and uncertainty sources.
As far as I know, most relevant Causal BO methods have been covered, and summaries for most methods are provided in the survey paper.
The paper also discusses the theoretical foundations and connections to fields of Causal Bandits, Bayesian experimental design, and other adjacent fields.
It provides a benchmark comparing the different methods, and introduces a new metric (PA-GAP) for fair comparison of the methods.
The paper surveys new developments, such as contextual and adversarial settings, mechanism-level modelling, functional interventions, and dealing with unknown graph structure.
It also talks about open challenges, and how the field needs to develop to bring these methods into practise.

**Additional Comments:**

Reduce verbosity. For example, in section 3, the subsections repeat the organisation of the section several times. Subsections end with a summary paragraph; a final summary at the end comparing the subsections would suffice. Long motivations at the start of every subsection, for example in 3.2 "The original CBO ... Pareto set." and "A useful way ... information across them." are conveying the same information twice.

Technical presentation: Add definitions, equations, and define problem settings formally instead of intuitively summarising them. Expand Table 2 or at least add color coding / or some symbols to indicate different assumptions (causal sufficiency), and types of environmental assumptions.

Discuss limitations of the benchmark: Comparing native implementation instead of reimplementing in a unified setting, and sensitivity of the PA-GAP metric to other more standard metrics.

Add a survey perspective to each method: what assumptions does it relax/introduce, why was it necessary, and what additional statistical efficiency does this method bring?

**Audience:**

Yes

**Audience Explanation:**

I think Causal Bayesian Optimisation and causal bandits have been growing as fields and have seen recent interest in terms of published papers on the topic. I am sure a unified view, comparing the recent advancements, will be interesting to some of TMLR's audience.

**Broader Impact Concerns:**

None.

**Claims And Evidence:**

No

**Claims Explanation:**

Claims made in the paper are partially supported; however, I would reword them in quite a few places as the claims are too strong. I would switch to a yes after a revision addressing these issues.

Contribution 1: This **reveals** which components are **interchangeable and which are tightly coupled**. This claim is not backed by any systematic studies ablating different components, or even some theoretical comparisons.

Contribution 2: **formal connection**, I read formal as theoretical equivalences, reductions, or formal mappings. Section 3.7 only discusses the connections conceptually.

The benchmark: A unified comparison is very useful, but in its current state it is simply wrappers around the original implementations, possibly affected by hyperparameters, seeds, and implementation details. This is useful, but the claim needs to be adjusted.

**Requested Changes:**

1. Reorganise the taxonomy: I liked the four-axes currently, however for me I think graph assumptions (which are somewhat addressed in the surrogate modeling choice) are an important consideration, as in whether they admit causal sufficiency or not (and it also affects the method)- a lot of Aglietti's work CBO, CEO etc consider hidden confounders as well, whereas MCBO, GACBO etc which are model based methods don't and are therefore able to model parent child relations with individual GPs.
Given the four axes, a few of the current groupings of methods are a bit difficult to follow; CoCaBO and ACBO appear to be grouped by intervention type. Still, they are methods with different environmental assumptions (contextual and adversarial), and statistical modelling of surrogates (this follows from my earlier comment on the difference between CBO and MCBO).
2. Modelling extensions: Currently, each method is described independently in plain text, group methods by assumptions they relax, and include key equations in unified notations to show how the method develops on top of the original CBO framework. This version reads like sequential summarisations of different methods with low coherence or unified storyline.
3. POMIS and MIS concepts originally introduced in the field of Causal bandits are key to Causal BO and need to be highlighted and reintroduced formally. This formulation is essentially what lends a lot of Causal BO methods statistical efficiency over standard BO.
4. Section 3.3: I'd suggest organising around environmental assumptions (static vs dynamic, contextual, adversarial, constrained, multi-objective), and the intervention section can then solely focus on hard vs soft interventions - please also refer to (Bayesian Optimization of Function Networks by Raul Astudillo Peter Frazier), policy interventions, and intervention scope.
5. Surrogate modelling: There is mention of neural surrogates, Deep kernel learning, and Bayesian NNs; however, this only mentions the methods without ever clarifying how they interact with the causal aspect of it. For BO, the surrogate plays an extremely important role and encompasses a lot of hidden assumptions for the model class and possible strategy used; it needs to be discussed a bit more in detail.
6. Clarify the role of counterfactuals in the context of BO. Have people already considered using counterfactuals for BO? Is this a new direction that the paper proposes? If so, what exactly is the problem setting, and the paper should clarify that this could be a future direction.

Minor: The link is broken for the repository.
I would advocate for more consistent terminology: graph uncertainty, structure uncertainty, unknown graph etc.

---

> ### Author Response · Authors · 2026-07-08
> **Clarifying Scope, Assumptions, and Taxonomic Structure**
>
> We thank the reviewer for the constructive and detailed feedback. We read the central concern as follows: the paper is useful as a unifying survey, but several claims were phrased more strongly than the evidence supports; the taxonomy should better expose graph and environmental assumptions; and the method summaries should use common mathematical notation rather than a sequence of largely independent prose summaries. We agree, and the revision makes the paper more explicit, more technical, and more cautious.
>
> **Softening the contribution claims.**
>
> We revised the contribution statements to avoid implying claims that we did not prove. In particular, the "interchangeable components" language is softened: the taxonomy is now presented as a design-space organization that reveals recurring couplings and separations, not as an empirical ablation study proving interchangeability. Similarly, we no longer describe the adjacent field section as establishing formal equivalences or reductions unless an actual mathematical mapping is given. The revised wording says that the paper provides structured conceptual and notational connections to causal bandits, Bayesian experimental design, safe optimization, policy search, and causal abstraction, while reserving "formal" for definitions, objectives, and problem mappings that are written explicitly.
>
> **Reorganized taxonomy.**
>
> We agree that graph assumptions and environmental assumptions were not visible enough in the original four-axis taxonomy. The revised taxonomy separates: (i) graph/system-knowledge assumptions, (ii) environment assumptions, (iii) intervention representation, (iv) surrogate architecture, and (v) decision/acquisition rule. This makes clear, for example, that CoCaBO and ACBO should not be grouped only by intervention type: CoCaBO is contextual/policy-oriented, while ACBO changes the interaction model to an adversarial or non-stationary setting. It also makes clear why mechanism-level methods such as MCBO and GACBO typically require stronger causal-sufficiency or correctly specified mechanism assumptions than effect-level CBO variants. We implemented this at three places in the revision. First, the method-classification table (Table 2 in the paper) now carries assumption markers, as the reviewer requested: methods whose formulation can accommodate hidden confounders (e.g., via identification-based adjustment or averaging over structures) are marked $^{\ddagger}$ (CBO, DCBO, cCBO, CEO), methods that model individual mechanisms or rewards directly and therefore assume causal sufficiency are marked $^{\dagger}$ (MCBO, ACBO, GACBO), and the caption explains both markers together with the default for unmarked methods. Second, Section 3.3 now states explicitly that CoCaBO and ACBO differ on the environmental axis (contextual versus adversarial/non-stationary) and are grouped together only because they share machinery for richer decision-making, not because they make identical assumptions. Third, the same passage adds the requested reference to Bayesian optimization of function networks [1], explaining that function networks anticipate the mechanism-level soft-intervention view of MCBO/ACBO while differing from SCM-based CBO in that their inputs are causally independent decision variables.
>
>
> **Table R1: Taxonomy revisions added in response to the review.**
>
> | Axis | What changed | Examples clarified |
> |---|---|---|
> | Graph/system knowledge | Added as an explicit axis: known graph, unknown graph, graph uncertainty, hidden confounding, causal sufficiency, and identifiability assumptions | CBO/CEO can discuss hidden confounding or identification; MCBO/GACBO require mechanism-level graph assumptions |
> | Environment assumptions | Pulled out from intervention type: static, dynamic, contextual, constrained, adversarial/non-stationary, multi-objective | DCBO, CoCaBO, cCBO, ACBO, MO-CBO |
> | Intervention representation | Restricted to the action form: hard clamps, soft mechanism changes, policy interventions, intervention scope, and mixed discrete–continuous choices | CBO, fCBO, CoCaBO, function-network BO tasks |
> | Surrogate architecture | Expanded discussion of causal role: effect-level versus mechanism-level surrogates, multi-task models, neural/deep kernels as estimators of causal estimands | CBO, MCBO, HCBO, CNEI, deep surrogate variants |
> | Decision rule | Acquisition or online-learning rule after the causal modeling target is fixed | EI/UCB-style CBO, CEO information gain, ACBO multiplicative weights |
>
> [1] Raul Astudillo and Peter I. Frazier. Bayesian optimization of function networks. In *Advances in Neural Information Processing Systems*, volume 34, pp. 14463–14475, 2021.

---

> > ### Author Response · Authors · 2026-07-08
> > **Unified Mathematical Framing and Benchmark Caveats**
> >
> > **Unified mathematical notation.**
> >
> > We agree that Section 3 needed more equations. The revised method section now introduces common notation before discussing variants: graph $G$, intervention scope $S$, intervention value or policy parameter $\theta$, target $Y$, observational data $D_o$, interventional data $D_i$, effect-level surrogate $g_S(\theta)$, and mechanism-level surrogate $f_j(\mathrm{pa_j})$. We then define each method family as a relaxation or modification of the baseline CBO objective,
> >
> > $$(S^\star,\theta^\star) \in \arg\max_{S,\theta} \mathbb{E}\!\left[Y \mid \mathrm{do}(X_S=\theta)\right].$$
> >
> > For constrained CBO, the revision adds constraints such as
> >
> > $$\max_{S,\theta}\; \mathbb{E}[Y \mid \mathrm{do}(X_S=\theta)] \quad\text{s.t.}\quad \Pr(C_\ell \leq 0 \mid \mathrm{do}(X_S=\theta)) \geq 1-\delta_\ell .$$
> >
> > For contextual/policy CBO, context is introduced only after the baseline formulation:
> >
> > $$\max_{\pi}\; \mathbb E_{C}\!\left[ \mathbb{E}[Y \mid \mathrm{do}(X_S=\pi(C)), C]\right].$$
> >
> > For mechanism-level models, the revised text distinguishes direct effect modeling from propagating mechanism posteriors through the graph:
> >
> > $$V_j := f_j(\mathrm{Pa_j}, U_j), \qquad f_j \sim \mathcal{GP}(m_j,k_j).$$
> >
> > This gives each method a common mathematical anchor and makes the survey less like a sequence of unconnected summaries.
> >
> > **MIS/POMIS and statistical efficiency.**
> >
> > We agree that MIS and POMIS should be foregrounded. The revised core-CBO section formally introduces minimal intervention sets and possibly-optimal MISs before surveying extensions. We explain that this graph-based scope reduction is one of the main reasons CBO can be statistically more efficient than standard BO: it avoids spending interventional budget on scopes that are causally irrelevant to $Y$ or dominated under the available graph and observational information. We also add the missing structural causal bandit citations associated with this terminology and clarify how POMIS-style pruning differs from causal abstraction across graph resolutions.
> >
> > **Surrogate modeling details.**
> >
> > We expanded the surrogate discussion to explain how the surrogate interacts with the causal assumptions. The revised paper distinguishes effect-level surrogates, which model $\theta \mapsto \mathbb{E}[Y\mid \mathrm{do}(X_S=\theta)]$, from mechanism-level surrogates, which model structural equations and propagate uncertainty through the causal graph. We also clarify the role of neural surrogates, deep kernels, and Bayesian neural networks: these are not "causal" by themselves. They become CBO components only when their input/output target is a causal estimand, when observational data are used through an identified causal prior, or when their architecture respects graph/mechanism structure. This addition makes the discussion more useful to BO readers, for whom the surrogate encodes strong model-class assumptions.
> >
> > **Counterfactuals.**
> >
> > We clarify that counterfactuals are not currently a standard component of most CBO algorithms surveyed in the paper. The foundations section now treats counterfactuals as part of the SCM language, but the methods section distinguishes the interventional expectations optimized by current CBO from future counterfactual extensions. We describe possible future settings more precisely: personalized intervention choice after observing unit-specific histories, retrospective evaluation of alternative actions in the same latent unit, or counterfactual safety constraints. This prevents the paper from implying that existing CBO methods already solve counterfactual BO.
> >
> >
> > **Benchmark limitations and PA-GAP.**
> >
> > We agree that the benchmark should be presented as a wrapper-based reproducibility harness rather than as a fully controlled reimplementation of every method. The revision explicitly states that the comparison uses native or vendored implementations, released defaults, common trajectory export, and common scoring. We add sensitivity analyses and limitations for default hyperparameters, including an MCBO $\beta$-sweep, and we report standard BO-compatible summaries such as best-so-far/simple-regret-equivalent curves and average-reward diagnostics in addition to GAP and PA-GAP. We also add plot-matched rank reliability tests for the aggregate rank figures. These tests are reported with the important caveat that pooled settings are not mutually independent because GAP and PA-GAP share trajectories and the three budgets are nested truncations of the same runs. Thus, the benchmark is useful for reproducibility and diagnosis, but not for declaring a universal method winner. This non-independence caveat is now stated explicitly in the rank-reliability appendix of the paper, where we note that the reported $p$-values and critical differences should be read as descriptive reliability checks under these dependencies, and that the conservative per-dataset test mitigates the metric-sharing dependence but not the budget nesting.

---

> > > ### Author Response · Authors · 2026-07-08
> > > **Repository link, terminology, and verbosity**
> > >
> > > We fixed the repository link and standardized terminology around "graph uncertainty," "structure uncertainty," and "unknown graph." In the revised paper, "unknown graph" refers to the assumption class, "graph uncertainty" refers to the learner's uncertainty over graph structure, and "graph misspecification" refers to the case where the learner-side graph differs from the true benchmark graph. We also reduced verbosity in Section 3 by removing repeated mini-summaries and repeated motivation paragraphs; each method is now presented through the lens of what assumption it relaxes or introduces, what mathematical object changes, and what statistical efficiency or cost tradeoff follows. Concretely, we removed the per-subsection "Summary" paragraphs of Sections 3.2, 3.3, and 3.5 (retaining only the substantive CEO-versus-GACBO comparison closing Section 3.4), condensed the duplicated motivation at the start of Section 3.2 that the reviewer quoted, and added a note at the start of Section 3.6 stating that the cross-method synthesis now appears once, there, along the design axes. The terminology convention is likewise now stated in the paper itself, at the start of Section 3.4: "unknown graph" denotes the assumption class, "graph uncertainty" (interchangeable with "structure uncertainty") the learner's remaining uncertainty over structure, and "graph misspecification" the distinct case of a trusted but incorrect supplied graph, probed by the stress-test appendix.

---

> > > > ### Author Response · Authors · 2026-07-08
> > > > **Summary of concrete revisions**
> > > >
> > > > - Softened claims about interchangeable components, formal connections, and benchmark conclusiveness.
> > > > - Reorganized the taxonomy around graph assumptions, environment assumptions, intervention representation, surrogate architecture, and decision rules.
> > > > - Added unified notation and key equations for baseline CBO, constrained CBO, contextual/policy CBO, and mechanism-level modeling.
> > > > - Formally introduced MIS/POMIS and emphasized their role in statistical efficiency.
> > > > - Expanded the surrogate-modeling discussion to explain how GP, neural, deep-kernel, and mechanism-level surrogates interact with causal assumptions.
> > > > - Clarified that counterfactual BO is mainly a future direction rather than a claim about existing CBO methods.
> > > > - Reframed the benchmark as a native-wrapper reproducibility harness, added sensitivity and rank-reliability caveats, fixed the repository link, standardized terminology, and reduced repetitive prose.
> > > > - Added causal-sufficiency markers ($^{\dagger}$/$^{\ddagger}$) with an explanatory caption to the method-classification table, distinguishing methods that accommodate hidden confounders from those that require fully observed mechanisms.
> > > > - Clarified in Section 3.3 that CoCaBO and ACBO differ on the environmental axis (contextual versus adversarial), and added the requested reference to Bayesian optimization of function networks, relating it to the mechanism-level soft-intervention view and to our soft benchmark.
> > > > - Removed the per-subsection summary paragraphs (Sections 3.2, 3.3, 3.5) and the duplicated motivation in Section 3.2, consolidating the cross-method synthesis in Section 3.6; stated the graph-terminology convention in Section 3.4; and stated the rank-test non-independence caveat in the appendix.

---

### Comment · Action_Editor_1Ua7 · 2026-05-30
**Survey paper criteria**

Dear reviewers,

This paper describes itself as a survey paper. See the FAQ for the criteria for a survey paper.

> Q: Does TMLR accept survey papers?

> A: Yes. Authors should make sure to emphasize the contributions made by the survey. Ideally, we want survey papers that draw new, previously unreported connections between several pieces of work in an area, and/or that clearly highlight trends in the area and/or suggest currently open problems. It should also be noted that if a submission has more than 12 pages of main content, then TMLR's normal short review timeline will not be enforced.

https://jmlr.org/tmlr/faq.html